# Denoising without Diffusion: Fixed-Noise Denoiser Anomaly Detection in Tabular Data

**Manuel Hirth** [1 2]   **Lukas Koberg** [3 1]   **Nasser Jazdi** [3]   **Enkelejda Kasneci** [4 2]

## Abstract

While diffusion models have advanced anomaly detection, their reliance on multi-step noise schedules introduces significant computational complexity. In this paper, we demonstrate that the generative capability of diffusion is not required for tabular anomaly detection. We revisit core principles of denoising without targeting data generation and present a deep-learning approach that streamlines these objectives into a fixed-noise formulation. Unlike denoising autoencoders that rely on reconstruction error, our method utilizes a preconditioning with an explicit linear reference channel. We train a self-supervised fixed-noise denoising predictor and derive an anomaly score from the expected deviation under repeated perturbations, yielding a stability proxy rather than merely measuring distance to the data manifold. On the well-established ADBench benchmark, our method achieves state-of-the-art performance with improvements over existing baselines of 1.22% in AUCROC and 1.13% in AUCPR, the most informative and threshold-independent metrics. Our approach emphasizes structural simplicity and efficiency, demonstrating that a single-step, stability-based objective outperforms complex generative schedules.

## 1. Introduction

Anomaly detection is a long-standing and practically important problem in machine learning, with applications ranging from fraud detection (Al-Hashedi & Magalingam, 2021)

[1]Daimler Truck AG, Stuttgart, Germany [2]Technical University of Munich, TUM School of Computation, Information and Technology, Munich, Germany [3]University of Stuttgart, Institute for Industrial Automation and Software Engineering, Stuttgart, Germany [4]Technical University of Munich, Chair for Human-Centered Technologies for Learning, Munich, Germany. Correspondence to: Manuel Hirth <manuel.hirth@daimlertruck.com>.

*Proceedings of the $43^{rd}$ International Conference on Machine Learning*, Seoul, South Korea. PMLR 306, 2026. Copyright 2026 by the author(s).

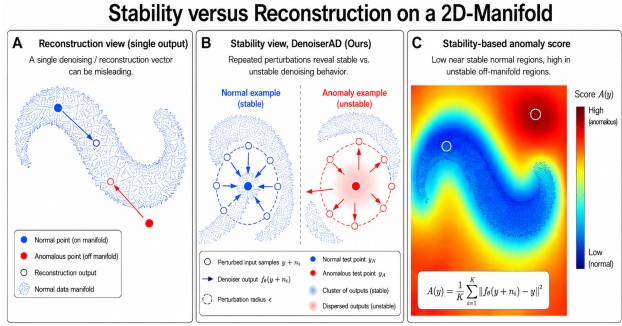

*Figure 1.* Visualizing Stability vs. Reconstruction on a 2D Manifold. A. Standard DAEs use a single reconstruction vector, which can be misleading. B. Our method probes the local vector field by perturbing the input. Normal points lie in stable regions with consistent denoising vectors (low variance), while anomalies lie in unstable regions with chaotic vectors (high variance). C. The resulting stability score $A(y)$ forms a clear "valley" along the data manifold, providing a robust anomaly signal.

and network security (Ahmad et al., 2021) to industrial monitoring (Gupta et al., 2013) and data quality assurance (Ruff et al., 2018). In many of these settings, data is naturally represented in tabular form, which remains the dominant format in enterprise and scientific applications (Xu et al., 2023b). Despite this relevance, recent advances in deep learning have largely shifted research focus toward large language models, while comparatively less attention has been paid to improving deep learning methods for tabular anomaly detection (Spinaci et al., 2025). Classical anomaly detection methods for tabular data, such as isolation-based (Liu et al., 2008; Xu et al., 2023a), density-based (Kingma, 2013; Zhou et al., 2020), or reconstruction-based (Zenati et al., 2018; Zong et al., 2018) approaches, are often limited in their ability to capture complex dependencies in high-dimensional data. Reconstruction-based methods may generalize too well and assign low anomaly scores to out-of-distribution samples, while likelihood-based models are sensitive to model misspecification and often difficult to train reliably on tabular data (Rezende & Mohamed, 2015).

Recently, diffusion models (Ho et al., 2020; Dhariwal & Nichol, 2021) and denoising-based objectives (Song & Ermon, 2019; Song et al., 2020) have emerged as powerful

tools for modeling complex data distributions, primarily in the context of generative modeling. These methods are commonly trained using multi-step noise schedules and time-dependent objectives, which are designed to support high-quality sample generation. However, the requirements of anomaly detection differ fundamentally from those of data generation: the goal is not to synthesize new samples, but to assess local manifold compliance (Wyatt et al., 2022). We argue that the iterative reverse process required for high-fidelity generation is computationally redundant for anomaly detection, where probing the denoiser at a single, informative noise scale is often sufficient.

In this work, we focus on one-class classification setting within semi-supervised anomaly detection (Ruff et al., 2021). This is closely related to LUPE, where anomaly detectors are trained on positive data only and then tested on both normal and abnormal data (Golan & El-Yaniv, 2018; Sabokrou et al., 2019; Chen et al., 2020). We revisit denoising-based training objectives from diffusion models with a focus on anomaly detection rather than generation. We deliberately avoid multi-step diffusion processes and instead adopt a single-noise formulation, in which a model is trained to recover clean samples from perturbed observations at a fixed-noise scale. Specifically, we structure the denoiser with an explicit linear reference channel. This forces the network to learn a stability correction rather than a direct reconstruction. Based on this formulation, we derive an anomaly score from the expected deviation between the denoised output and the original sample under repeated perturbations, thus measuring the functional sensitivity of the learned correction relative to the fixed linear baseline. This score can be interpreted as a stability-based energy that measures local sensitivity of the data representation to stochastic perturbations.

Viewed from a self-supervised anomaly detection perspective, DenoiserAD uses Gaussian denoising as an auxiliary training task, but deliberately decouples the inference-time anomaly score from the denoising loss itself. In contrast to many self-supervised AD methods that use the auxiliary training loss itself as the anomaly score, we use denoising only as the training task and derive a separate inference-time score that probes the local stability of the learned denoising field. Rather than using task performance directly as the anomaly signal, the trained denoiser is probed under repeated perturbations and anomalies are scored by input-referenced local instability. This places the method at the intersection of diffusion-inspired denoising and self-supervised anomaly detection, and explains why it can outperform reconstruction-style denoising scores in Table 2. Empirically, we find that this simplified formulation is sufficient for anomaly detection. In particular, our method achieves the best average performance on the ADBench benchmark (Han et al., 2022) across multiple evaluation metrics, while maintaining practical training times and fast inference.

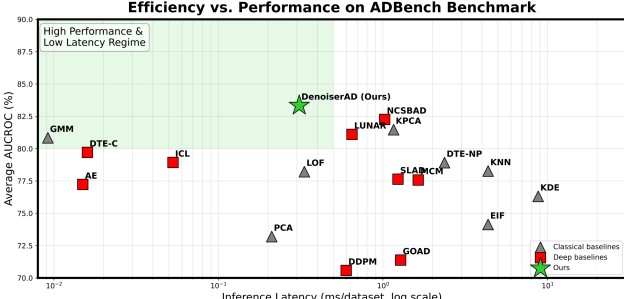

*Figure 2.* Average AUCROC versus inference time per dataset across benchmark methods. Each point corresponds to one method evaluated on the full ADBench benchmark. The proposed approach (DenoiserAD) achieves the best average detection performance while maintaining competitive inference cost.

## Summarizing our contributions

- We propose a self-supervised fixed-noise denoising framework for tabular one-class anomaly detection, in which denoising serves as the auxiliary training task while anomaly detection is performed using a separate input-referenced stability score rather than the denoising loss itself.

- The score admits a bias–stability decomposition, corresponding to a Jacobian-norm surrogate under a first-order expansion. This decomposition probes the local instability of the learned denoising field without requiring explicit likelihood estimation. Empirically, the resulting neighborhood score outperforms reconstruction-based scoring, while a direct Jacobian-only baseline is insufficient.

- We show that timestep conditioning, schedules, and multi-step reverse processes are not necessary for one-class tabular anomaly detection, and that a single fixed-noise denoiser with an appropriate score is sufficient. This leads to a substantial simplification, reducing both computational complexity and memory requirements.

- We achieve state-of-the-art performance with consistent improvements over prior diffusion-based and classical baselines on ADBench across various architectures, outperforming prior baselines across all metrics.

## 2. Related Work

Anomaly detection has been studied extensively across a wide range of application domains. In this section, we briefly review related categories. Comprehensive comparisons are presented in the following surveys, Ruff et al.

(2021); Chandola et al. (2009); Hirth et al. (2025); Pang et al. (2021); Chalapathy & Chawla (2019).

**Density Estimation and Distance-Based Methods**   Density estimation (Tang et al., 2002; Parzen, 1962) is one of the most direct approaches to anomaly detection. Classical methods aim to estimate the probability density of normal data and identify anomalies as samples with low likelihood. Both parametric and non-parametric techniques have been explored, including kernel density estimation (Latecki et al., 2007), Gaussian mixture models (Agarwal, 2007), and copula-based models such as COPOD (Li et al., 2020). Local density estimation methods, such as Local Outlier Factor (Breunig et al., 2000), assess anomalies based on neighborhood density not to global likelihood. Distance-based (Cook, 1977) and nearest-neighbor (Ramaswamy et al., 2000) are closely related. Methods such as kNN-based scoring (Hautamaki et al., 2004), clustering-based techniques (He et al., 2003; Weichert et al., 2025), and tree-based approaches like Isolation Forest (Liu et al., 2008) and its variants (Hariri et al., 2019; Bandaragoda et al., 2018; Xu et al., 2023a) measure deviations using distances, densities, or isolation depth. These techniques are often competitive on tabular benchmarks and benefit from conceptual simplicity and interpretability (Li et al., 2022b). However, their performance can degrade in high-dimensional settings or when complex feature interactions are present (Fauconnier & Haesbroeck, 2009).

**Reconstruction-Based Methods**   Reconstruction-based methods constitute a major class of deep anomaly detection approaches. These methods learn to reconstruct samples from the normal data distribution, using reconstruction errors. Approaches include PCA (Shyu et al., 2003) and autoencoders, with numerous extensions incorporating denoising objectives, variational inference, or adversarial training (Sakurada & Yairi, 2014; Aggarwal & Aggarwal, 2017; Kingma, 2013; Hawkins et al., 2002). While reconstruction-based methods can model non-linear feature dependencies, they suffer from a well-known limitation, namely sufficiently expressive models often learn an identity mapping, reconstructing anomalous samples with low error. This limitation is closely related to broader observations that deep generative or reconstruction-based models may assign high likelihood or low reconstruction error to samples outside the intended normal distribution (Nalisnick et al., 2018; Astrid et al., 2021; 2024; Merrill & Eskandarian, 2020; Cheng et al., 2021; Gong et al., 2019). These works motivate alternatives to naive reconstruction scoring, including modified autoencoder objectives, memory-augmented architectures, and mechanisms that explicitly discourage accurate reconstruction of anomalies. Our work follows the same motivation but takes a different route: rather than modifying the reconstruction objective alone, we retain a simple denoising task and change the inference-time anomaly score to measure local input-referenced stability. Therefore, standard reconstruction objectives only penalize the output value, not the local behavior of the function. Specifically, they do not explicitly measure the model's sensitivity to perturbations, which we argue is a more reliable signal for tabular anomalies, where anomalies may not correspond to easily separable patterns (Akcay et al., 2019; Aggarwal & Aggarwal, 2017; Kingma, 2013; Vietz et al., 2023; Hawkins et al., 2002). We focus on methods that can be evaluated with a single global hyperparameter setting across datasets. Methods such as ATDAD (Yang & Li, 2023), TCCM (Li et al., 2025), Disent-AD (Ye et al., 2025), and NTP-AD (Thimonier et al., 2023) rely on dataset-specific hyperparameter choices and are therefore outside the scope of our benchmark protocol.

**One-Class and Self-Supervised Methods**   One-class classification methods aim to directly characterize the support of the normal data distribution without explicitly modeling its density. Classical kernel-based approaches learn a decision boundary that encloses normal samples, while more recent methods replace kernels with deep neural networks. Tree-based and ensemble-based variants have also been widely adopted (Pevnỳ, 2016; Hendrycks et al., 2019; Li et al., 2023). Self-supervised approaches define auxiliary tasks learned from normal data alone, using task performance as an anomaly signal. Examples include transformation prediction (Golan & El-Yaniv, 2018; Schneider et al., 2022), contrastive learning (Shenkar & Wolf, 2022), and synthetic anomaly generation (Li et al., 2022a; Liu et al., 2019). These methods have shown strong empirical performance on benchmark datasets but often rely on carefully designed pretext tasks or domain-specific assumptions (Hirt et al., 2024; Dang et al., 2025; Sohn et al., 2020; Dang & Peters, 2025). Our method is also related to neural transformation learning and other self-supervised AD approaches, where normal data are transformed and the model is trained to solve an auxiliary task whose success or failure provides an anomaly signal (Qiu et al., 2021; Alvarez et al., 2022). The key difference is that DenoiserAD does not directly reuse the auxiliary denoising loss as the anomaly score. Instead, the denoiser is treated as a learned local vector field, and anomalies are scored by the stability of this field under repeated perturbations. Zero-shot methods presented in Pang et al. (2018) and Li et al. (2024) were not considered, as they perform single sample comparisons.

**Diffusion and Score-Based Methods**   Score-based generative models (Song & Ermon, 2019; Dai et al., 2025) and diffusion models (Ho et al., 2020; Wolleb et al., 2022) have recently emerged as powerful tools for modeling complex data distributions. These models learn to reverse a noise corruption process by estimating score functions or denoised samples across multiple noise scales. In image (Shin et al., 2023) and video domains (Flaborea et al., 2023), diffusion-

based approaches have been successfully applied to anomaly detection, typically in semi-supervised or reconstruction-based settings. For tabular data, existing methods often adapt full diffusion pipelines, including noise schedules, time conditioning, or iterative denoising, originally developed for generative modeling (Hirth & Kasneci, 2025; Livernoche et al., 2024). Diffusion-scheduled Denoising Autoencoders (DAEs) (Kascenas et al., 2022; Sattarov et al., 2025) are closely related to diffusion models and also rely on multi-scale denoising formulations. By stripping away the multi-step sampling and time-dependence, we isolate the core mechanism—denoising stability—resulting in a method that is both theoretically and empirically cleaner. Thus, our contribution should not be understood as proposing a stronger generative diffusion model for anomaly detection, but as identifying which part of diffusion-style denoising remains useful when the goal is self-supervised one-class anomaly scoring rather than generation.

# 3. Method

We consider the problem of anomaly detection on tabular data. Let $y \in \mathbb{R}^d$ denote a data sample drawn from an unknown distribution representing normal (non-anomalous) behavior. During training, we assume access only to samples from this normal data distribution.

## 3.1. Fixed-Noise Denoising Objective

The denoising model $f_\theta : \mathbb{R}^d \to \mathbb{R}^d$ parameterized by $\theta$ is trained using a fixed-noise objective. Given a clean training sample $y \in \mathbb{R}^d$, we generate a noisy observation $x = y + n$, where $n \sim \mathcal{N}(0, \sigma^2 I)$ and $\sigma^2$ is held constant throughout training. The model is trained to predict the clean input from the noisy observation by minimizing the expected squared error

$$L(\theta) = \mathbb{E}_{y,n} \left[ \|f_\theta(y + n) - y\|^2 \right]. \qquad (1)$$

Unlike diffusion models, we do not employ a noise schedule, do not condition the model on a timestep variable, and do not define a multi-step forward or reverse process.

In practice, we parameterize the denoiser using a simple preconditioning scheme to improve numerical stability. Specifically, we follow the normalization strategy introduced by Karras et al. (2022) for denoising networks, which rescales the input and output of the model as a function of the noise scale. In our setting, the noise scale is fixed, and the preconditioning can be interpreted as a constant reparameterization of the denoising function to unit variance. Further details are provided in Figure 3 and Appendix B. The fixed-noise formulation simplifies both training and inference. In particular, the model learns a single denoising behavior that is later investigated at inference time through repeated pertur-

bations of the same input.

## 3.2. Stability-Based Anomaly Score

At inference time, we define an anomaly score that measures the stability of the denoiser's response under repeated stochastic perturbations of the same input sample. Let $y \in \mathbb{R}^d$ denote a test sample and let $n_i \sim \mathcal{N}(0, \sigma^2 I)$ be independent noise realizations. For each perturbation, we construct a noisy input $x_i = y + n_i$ and compute the corresponding denoised output $\hat{y}_i = f_\theta(x_i)$. We define the anomaly score as

$$A(y) = \mathbb{E}_n \left[ \|\hat{y} - y\|^2 \right] \approx \frac{1}{K} \sum_{i=1}^{K} \|f_\theta(y + n_i) - y\|^2, \qquad (2)$$

where the expectation is approximated using $K$ independent perturbations. Importantly, this quantity is not directly the training objective and is not optimized explicitly. The denoising model is trained independently using the fixed-noise objective described, while the anomaly score is defined solely at inference time. While $A(y)$ is an input-referenced squared error, it functionally probes the local manifold attraction field. It integrates the denoiser's response over a local noise neighborhood and thus captures both (i) bias toward the manifold and (ii) the local sensitivity of the vector field (stability). We therefore interpret it as a local stability functional, rather than a single-shot reconstruction error. Averaging is not used to reduce stochastic reconstruction noise, but to probe the local behavior of a fixed denoising function around a given input. Formally, for a smooth denoiser and sufficiently small perturbations, a first-order expansion yields

$$f_\theta(y + n) \approx f_\theta(y) + J_f(y)\, n, \qquad (3)$$

which implies

$$A(y) \approx \|f_\theta(y) - y\|^2 + \sigma^2 \|J_f(y)\|_F^2. \qquad (4)$$

This decomposition provides an interpretive local view of the score. The anomaly score therefore decomposes into a bias term and a stability-related term. Classical denoising autoencoders implicitly minimize the first term during training, but do not expose the behavior of the denoiser under perturbations. In contrast, the proposed method explicitly probes the response of the learned function in a finite neighborhood of the input at inference time. While Eq. (3–4) reflects only the first-order approximation, in practice the aggregated score captures higher-order effects beyond infinitesimal perturbations. Normal samples tend to yield stable, consistent denoised outputs across perturbations, whereas anomalous samples often lie in regions where the learned function behaves irregularly, leading to larger anomaly scores.

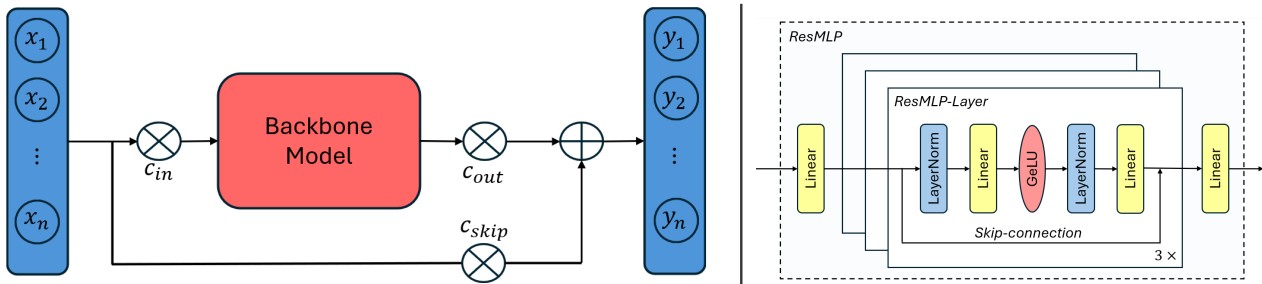

*Figure 3.* Denoiser parameterization and backbone architecture. Left: Preconditioned denoiser with an explicit linear-scaled reference channel. The input is scaled by $c_{\mathrm{in}}$, processed by a neural backbone $g_\theta$, rescaled by $c_{\mathrm{out}}$, and combined with a global linear reference channel weighted by $c_{\mathrm{skip}}$. Right: ResMLP used as the backbone $g_\theta$, consisting of stacked residual layers with linear layers, LayerNorm, and GeLU activations. Residual connections inside the backbone do not alter the global functional decomposition of the denoiser.

**Denoiser Parameterization with Explicit Reference Channel**   We parameterize the denoising function $f_\theta$ using an explicit decomposition into a linear reference channel and a learned correction term. Concretely, the denoiser is shown in Figure 3 and defined as

$$f_\theta(x) = c_{\mathrm{skip}}\, x + c_{\mathrm{out}}\, g_\theta(c_{\mathrm{in}}\, x), \qquad (5)$$

where $g_\theta : \mathbb{R}^d \to \mathbb{R}^d$ denotes a neural network backbone and $c_{\mathrm{in}}, c_{\mathrm{out}}, c_{\mathrm{skip}}$ are scalar coefficients determined by the chosen preconditioning scheme. This parameterization is employed here in a fixed-noise setting without time conditioning or noise schedules. The term $c_{\mathrm{skip}}\, x$ defines an explicit linear reference mapping, while the backbone network $g_\theta$ learns a non-linear correction to this reference at a single noise scale. The preconditioning coefficients normalize both the input to and output of the backbone network, ensuring that the learned correction operates in a well-scaled regime. This is particularly important for tabular data, where heterogeneous feature scales could otherwise dominate the magnitude and variability of the denoiser output. As a result, variability measured by the anomaly score primarily reflects functional sensitivity of the denoiser rather than numerical artifacts induced by feature scaling.

We distinguish between residual connections used inside the backbone for optimization stability and a global linear reference channel, which defines a functional reference mapping for the proposed anomaly score. The residual connections inside the backbone $g_\theta$ are used solely for optimization stability. The only functional reference mapping in the denoiser is given by the explicit linear reference channel $c_{\mathrm{skip}}x$. Crucially, this structure defines a fixed baseline behavior against which deviations induced by stochastic perturbations are measured. This imposes a structural inductive bias, as the network must actively learn a residual correction. Without this explicit reference, the anomaly score degenerates into a reconstruction-based criterion. As shown in Section 5, removing the reference channel reduces discriminative performance, showing that explicitly preserving the input as a reference can improve anomaly separability.

### 3.3. Interpretation as Local Stability

The above decomposition is the intuition for the interpretation of the proposed anomaly score. By explicitly separating a linear reference mapping from a learned correction, the denoiser's response to stochastic perturbations can be analyzed in terms of local deviations from a fixed baseline. Let $f_\theta : \mathbb{R}^d \to \mathbb{R}^d$ denote the denoising function and let $y \in \mathbb{R}^d$ be a fixed input. For sufficiently small perturbations $n \sim \mathcal{N}(0, \sigma^2 I)$, the variation of the denoiser output $f_\theta(y+n)$ reflects the local sensitivity of the function around $y$. Aggregating the squared deviation $\|f_\theta(y+n) - y\|^2$ over multiple perturbations therefore provides a measure of how stable the denoiser's response is in the neighborhood of the input. The score is not merely reconstruction error or unconditional output variance; it measures input-referenced sensitivity of the denoising map around the test point. Normal samples that lie in regions well supported by the training data tend to induce stable, consistent responses. In contrast, anomalous samples often lie in regions where the denoiser is locally unstable, small input perturbations induce greater variability in the denoised output, resulting in higher anomaly scores (Figure 1).

This decomposition is a local approximation; off-manifold anomalies may involve higher-order effects, which the multi perturbation score can still capture. Because the score requires only a differentiable denoiser $f_\theta : \mathbb{R}^d \to \mathbb{R}^d$, the method applies to standard tabular backbones; we validate this in Section 5, with additional derivations and implementation details in Appendix A, C and D.1. We provide full architectural specifications, optimization settings, and implementation details in Appendix B to support reproducibility.

## 4. Experiments

**Setting**   To ensure comparability with prior work, we use parts of the DTE benchmark setup and codebases from

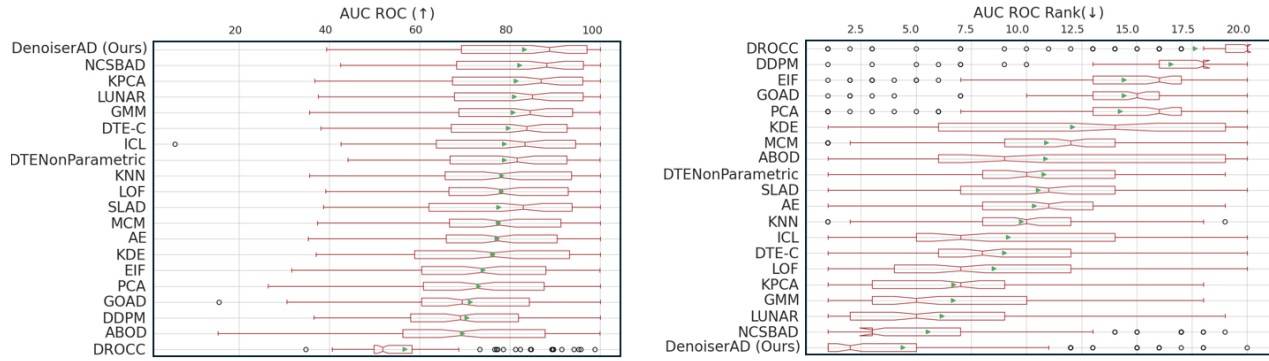

*Figure 4.* Performance box-plots across datasets. Left: Box-plots of AUCROC scores for all compared methods across the ADBench benchmark datasets. Right: Corresponding Box-plots of method ranks (lower is better) across the datasets. Each box summarizes the variability across datasets, with the central marker (green) indicating the mean performance. DenoiserAD achieves consistently high AUCROC and favorable ranks, indicating performance gains across heterogeneous tabular benchmarks.

Livernoche et al. (2024) [1] and Hirth & Kasneci (2025) [2]. In accordance with previous works (Bergman & Hoshen, 2020; Shenkar & Wolf, 2022; Livernoche et al., 2024), the training set is randomly drawn with 50% of the normal samples. The remaining 50% of normal samples and the anomalies are used for testing. Following DTE and ADBench, each dataset is constrained to a maximum of 50,000 data points; we provide more details for this in Appendix E. Results for all datasets are averaged over five different random seeds (0-4) used for the undersampling and data splitting over all datasets. Following Han et al. (2022) and Livernoche et al. (2024) we standardize each dataset using the training samples.

**Baseline Methods** We compare our approach against the best-performing and commonly used anomaly detection methods from previous work (Livernoche et al., 2024; Hirth & Kasneci, 2025; Bouman et al., 2024). This includes KPCA (Hoffmann, 2007), KDE (Latecki et al., 2007), GMM (Agarwal, 2007), kNN (Ramaswamy et al., 2000), LOF (Breunig et al., 2000), PCA (Shyu et al., 2003), EIF (Hariri et al., 2019), ABOD (Kriegel et al., 2008), LUNAR (Goodge et al., 2022), SLAD (Xu et al., 2023b), AE (Aggarwal & Aggarwal, 2017; Ramaswamy et al., 2000) from PyOD (Zhao et al., 2019) and ADBench. Additionally, we incorporate comparisons with other proposed deep learning methods such as DROCC (Goyal et al., 2020), GOAD (Bergman & Hoshen, 2020), ICL (Shenkar & Wolf, 2022), from Livernoche et al. (2024). Furthermore, we benchmark against the tabular diffusion techniques DDPM, DTE-C and DTE-NP, proposed by Livernoche et al. (2024), and NCSBAD (Hirth & Kasneci, 2025). We also add MCM from Yin et al. (2024). The baseline suite is chosen to cover three complementary comparison axes while keeping the comparison focused on

the strongest methods reported by Hirth & Kasneci (2025). First, we include the strongest classical tabular AD methods such as kNN, LOF, KPCA, GMM, EIF, and ABOD. Second, we include deep one-class and self-supervised baselines such as DROCC, GOAD, ICL, LUNAR, and SLAD, which are most directly related to our normal-only training setting. Third, we include diffusion- and score-based tabular AD methods, including DDPM, DTE-C, DTE-NP, and NCSBAD. This allows us to evaluate whether the proposed stability score improves over classical, self-supervised and diffusion-based AD baselines.

**Main Results** Figure 4 reports box plots of AUCROC and rank across ADBench. Our method achieves the best average performance across standard metrics. Due to space constraints, we summarize AUCROC in Figure 4; AUCPR, F1, adjusted variants (Campos et al., 2016), rankings for all metrics, and per-dataset result tables are provided in Appendix G. We treat threshold-free metrics (AUCROC, AUCPR, adjusted-AUCPR) as primary, since anomaly prevalence varies widely (0.03%–39.9%) and thresholded scores (F1, adjusted-F1) depend strongly on the thresholding policy; thresholded results are therefore reported as supplementary diagnostics. The higher mean/median performance and the tight rank distribution (Figure 4, right) indicate that DenoiserAD is not driven by a small subset of datasets but generalizes across diverse tabular tasks. Finally, while our approach targets the strict one-class setting with uncontaminated training data, we also evaluate fully unsupervised training with contamination (Appendix F). In this regime, a trimmed-loss variant substantially improves robustness and recovers performance, and we report these results alongside robust baselines such as GMM and EIF. We further consider a trimmed-loss variant that discards a fraction of high-loss samples during training to mitigate contamination; this yields a marked improvement over the vanilla objective and is competitive with the robust baselines.

---

[1] https://github.com/vicliv/DTE
[2] https://openreview.net/forum?id=7QDIFrtAsB

## 5. Ablation and Discussion

### 5.1. Sensitivity to the Noise Scale

We first study the sensitivity of DenoiserAD to the choice of the noise scale $\sigma^2$. The architecture (ResMLP), training procedure, and anomaly score are kept fixed. While $\sigma^2$ is fixed during training and inference, its value controls the locality of the denoising task and can strongly affect anomaly detection performance. Because the data is standardized, we perform a logarithmic sweep over $\sigma^2 \in [0.01, 1.0]$ and report the average AUCROC across all datasets. Figure 5 reports the AUCROC as a function of $\sigma^2$ over the interval $[0.01, 1.0]$ on a logarithmic scale. Three regimes can be clearly identified.

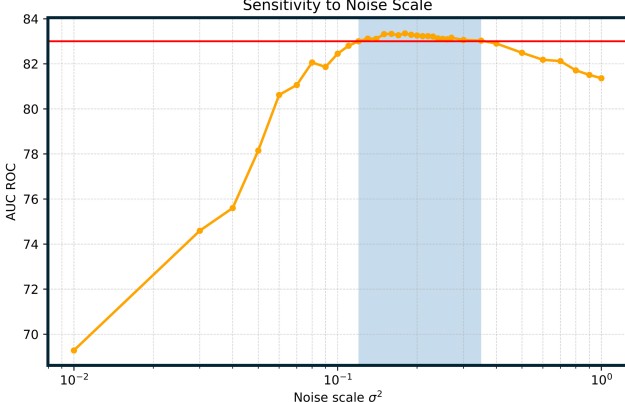

*Figure 5.* Sensitivity to the noise scale $\sigma^2$. Average AUCROC is reported over all datasets for a logarithmic sweep of $\sigma^2 \in [0.01, 1.0]$. Performance remains stable over a broad plateau between $\sigma^2 = 0.12$ and $\sigma^2 = 0.35$, indicating robustness to the choice of $\sigma^2$. Very small $\sigma^2$ values lead to a sharp performance drop, while larger $\sigma^2$ values result in a gradual degradation.

The results reveal a broad plateau region in which performance remains consistently high for intermediate noise scales spanning approximately $\sigma^2 \in [0.12, 0.35]$. Within this interval, the anomaly score is largely insensitive to the exact value of $\sigma^2$. This behavior indicates a functional regime, in which the denoiser reliably captures local stability properties of the data manifold. For very small noise scales $\sigma^2 < 0.1$, performance degrades sharply. In this regime, the injected noise is too small to meaningfully perturb the input, and the denoiser primarily learns near-identity mappings that provide limited sensitivity for anomaly detection. For larger noise scales $\sigma^2 > 0.4$, performance decreases gradually as $\sigma^2$ increases, suggesting that excessive noise increasingly obscures the local structure of the data manifold and biases the learned correction toward overly smooth responses. Here, perturbations increasingly dominate the input, and the denoiser's output becomes biased toward overly smooth projections, reducing discriminative power. We fix $\sigma^2 = 0.2$ for all experiments. Since features are standardized, this corresponds to an SNR of $1/0.2 = 5$ and yields a

moderate corruption level. This value is set before running benchmark experiments and is not tuned per dataset. In Figure 5 we additionally report a sweep over $\sigma^2$ to demonstrate robustness within $[0.12, 0.35]$.

Overall, these results demonstrate that the proposed stability-based anomaly score operates in a scale-robust regime and does not depend on a narrow optimal $\sigma^2$. It exhibits performance gains across a wide range of noise scales. This stability argues against $\sigma^2$ being a dataset-specific parameter. This results in an empirically stable regime on standardized tabular data.

### 5.2. Fixed-noise versus Noise Schedules

We next compare the fixed-noise formulation to multi-scale noise schedules, with timestep conditioning. All variants use the same ResMLP backbone and anomaly score; only the noise formulation and conditioning mechanism differ. Specifically, we evaluate the following configurations: 1. Fixed-noise scale $\sigma^2$ without timestep embedding (proposed), 2. fixed $\sigma^2$ with timestep embedding, 3. scheduled noise with timestep embedding. For the scheduled settings, we consider four noise schedules. We include the EDM noise schedule, which is also employed in TabSyn (Zhang et al., 2024) and has been shown to be effective for tabular data generation. In addition, we evaluate linear schedules over the interval $(0, 1]$, once using a single shared noise scale per batch and once sampling noise scales independently for each sample within a batch. Finally, we include a schedule restricted to the empirically stable interval $[0.12, 0.35]$, as identified in Section 5.

*Table 1.* Comparison of noise scheduling and timestep conditioning approaches. AUCROC averaged over five random seeds and all datasets.

| Method | AUCROC |
|---|---|
| EDM/Tabsyn | 69.71 |
| rand $(0, 1]$ batch | 74.09 |
| rand $(0, 1]$ sample | 72.86 |
| NCSN/NCSBAD | 81.86 |
| rand $[0.12, 0.35]$ | 81.78 |
| fixed temb | 83.11 |
| fixed (proposed) | **83.34** |

Table 1 reports the AUCROC values. Across all datasets and for five random seeds, the fixed-noise scale without timestep conditioning improves the results. Introducing timestep embeddings does not improve detection performance and slightly degrades it. Across ADBench tabular datasets under our ResMLP backbone and stability-based score, adding noise schedules and timestep conditioning reduce average AUCROC; thus a single fixed-noise level is sufficient in this experimental setting. Instead, specializing the model to a

single, informative noise scale yields a simpler and effective solution. Beyond performance, the fixed-noise scale reduces implementation complexity and removes the conditioning; in our implementation this also reduces memory usage.

### 5.3. Disentangling Stability from Reconstruction

We test whether the proposed anomaly scoring mechanism is merely denoising reconstruction with Monte-Carlo test-time augmentation (TTA), or whether performance gains require the combination of (i) an explicit reference-channel parameterization and (ii) an input-referenced stability score. To this end, we run a unified ablation that varies only the denoiser parameterization and the inference-time score, while keeping the backbone capacity, training objective, optimization, and noise scale fixed. Concretely, we consider two denoiser parameterizations: We consider DAE-style denoisers, which have been less systematically evaluated under modern tabular AD benchmark protocols, and include them as a strong baseline. We evaluate the following inference scores: $S_1(y) = \|f_\theta(y) - y\|^2$ for pure reconstruction $S_1n(y) = \|f_\theta(y+n) - y\|^2$ for a single noise draw, $S_2(y) = \mathbb{E}_n\big[\|f_\theta(y+n) - y\|^2\big]$ for input-referenced stability (ours), $S_3(y) = \mathbb{E}_n\big[\|f_\theta(y+n) - \mathbb{E}_{n'}f_\theta(y+n')\|^2\big]$ for mean-centered variance, $S_4(y) = \mathbb{E}_n\big[\|f_\theta(y+n) - (y+n)\|^2\big]$ for augmented reconstruction / DAE+TTA and proportional to the Stein score. Here $n \sim \mathcal{N}(0, \sigma^2 I)$ and the expectations are approximated using $K$ Monte-Carlo samples, matching the main setting. The precise parameterizations are given in Appendix D.1. This results in a $2 \times 4$ design (Table 2). Importantly, for a fixed denoiser parameterization, we compare different inference scores on the same trained model; separately, for a *fixed* score, we compare the two parameterizations under identical training. For better understanding and a clear illustration, Appendix D.1 provides a 2D synthetic example illustrating these effects in Figure 7. From a self-supervised anomaly detection perspective, this ablation shows that the auxiliary denoising task and the anomaly score should be separated: using the denoising/reconstruction loss itself is weaker than probing the stability of the learned denoising field at inference time.

*Table 2.* Disentangling reconstruction and stability. AUCROC averaged over five random seeds and all datasets. Definitions in Appendix D.1.

| Model | $S_1$ | $S_1n$ | $S_2$ (Ours) | $S_3$ | $S_4$ |
|-------|-------|--------|--------------|-------|-------|
| DAE   | 82.80 | 82.38  | 82.81        | 73.21 | 80.92 |
| Ours  | 82.89 | 82.68  | 83.34        | 75.88 | 81.33 |

The results in Table 2 support three conclusions. First, Monte-Carlo averaging alone is not sufficient to explain the gains: moving from single-shot reconstruction $S_1$ to input-referenced Monte-Carlo scoring $S_2$ yields only a small improvement for the Plain DAE (82.38 $\rightarrow$ 82.81, +0.43). Second, the largest improvement arises when *both* the reference-channel parameterization and the input-referenced stability score are present: DenoiserAD with $S_2$ reaches 83.34, improving by +0.53 over the Plain DAE with the same score ($S_2$), and by +0.96 over the Plain DAE baseline with $S_1$. Third, the improvement is not explained by standard denoising reconstruction under test-time augmentation: the augmented-reconstruction score $S_4 = \mathbb{E}_n\|f_\theta(y+n) - (y+n)\|^2$, which effectively measures the denoising autoencoder loss, underperforms the input-referenced stability score $S_2$ by a substantial margin (81.33 vs. 83.34). Consistent with our hypothesis, this suggests that measuring stability relative to the fixed input anchor ($y$) provides a stronger anomaly signal than measuring the ability to invert specific noise instances ($y + n$). Likewise, the mean-centered variance score $S_3$ performs poorly (75.88), indicating that output variability alone is a weak anomaly signal in this setting. Beyond the aggregate mean, DenoiserAD ($S_2$) outperforms the strongest reconstruction-style comparator in this table on 78/121 sub-datasets. The improvement is statistically significant under a Wilcoxon signed-rank test ($p < 0.05$), with moderate rank-biserial correlation $r = 0.405$. Taken together, these findings indicate that the gains are not attributable to Monte-Carlo averaging, nor to reconstruction under augmentation, but to measuring input-referenced local instability in a model that includes an explicit baseline mapping via the reference channel. We also evaluate the first-order surrogate directly via a Jacobian-norm baseline $S_{\text{Jac}}(y) = \|J_f(y)\|_F^2$. On ADBench, $S_{\text{Jac}}$ underperforms reconstruction (81.05 vs. 82.80 AUCROC) and is substantially worse than $S_2$ (83.34), indicating that infinitesimal local sensitivity alone is insufficient and that $S_2$ benefits from finite-neighborhood effects beyond the first-order approximation. Additionally, we provide a mechanistic validation of the stability decomposition in Appendix D.1 and Figure 8.

### 5.4. Architecture Independence

We investigate whether the effectiveness of the proposed stability-based formulation depends on a specific backbone architecture or diffusion type. We evaluate three representative denoising-based anomaly detection frameworks: DTE and DDPM-based AD, both introduced by Livernoche et al. (2024), and NCSBAD (Hirth & Kasneci, 2025), which uses the noise formulation from (Song & Ermon, 2019). For each framework, we consider three variants: 1. The original architecture and training procedure as proposed in the respective paper, 2. the original method with the backbone replaced by our ResMLP, 3. the original backbone using the proposed fixed-noise stability formulation, while keeping the remaining architecture. We used official implementations by the respective authors for the ADBench setting to

follow their recommended configuration. All variants using the proposed training formulation are trained with the same hyperparameter configuration as in all experiments and reported in Appendix B.

*Table 3.* Architecture independence results. AUCROC averaged over all datasets.

| Model | Org | Org ResMLP | Ours |
|---|---|---|---|
| DDPM | 70.73 | 76.82 | 77.91 |
| DTE | 79.77 | 69.39 | 80.65 |
| NCSBAD | 82.34 | 82.50 | 82.95 |
| Ours | | – | 83.34 |

Table 3 reports the AUCROC results across the methods and variants. Across all three diffusion frameworks, replacing the backbone alone yields only marginal improvements or even reduces it in the case of DTE. In contrast, applying the proposed training formulation consistently improves anomaly detection performance across architectures and diffusion models. This result shows that the performance gains do not stem only from architectural specialization, rather, they stem from the stability-based training and inference formulation. In our experiments, applying the formulation improves performance across the evaluated frameworks. These findings suggest that the proposed approach captures denoiser stability for anomaly detection and is robust across several diffusion-based backbones we tested, rather than exploiting inductive biases of a particular network design.

# 6. Limitations and Conclusion

**Limitations and Future Work** The primary limitation of our approach is its reliance on a clean training distribution. Because the method is designed to tightly model the normal data manifold, it is sensitive to contamination of the training set and can degrade substantially when anomalies are present during training. The approach is most suitable for strict one-class scenarios with an uncontaminated training set. In fully unsupervised settings with unknown contamination, performance can degrade, reflecting the design choice to model the normal manifold tightly. Improving robustness to mixed training distributions is thus a central avenue for future work. As a small first step, we report results for a simple per-batch trimmed-loss heuristic, which provides encouraging gains under contamination (Appendix F). Moreover, since denoising-based methods are sensitive to training-set contamination, extensions to weakly supervised, semi-supervised settings as well as other modifications in the loss or learning objective remain promising avenues for future work. In addition, the number of stochastic perturbations $K$ and the noise scale $\sigma^2$ remain tunable hyperparameters, especially for other data domains. While performance is stable over a broad range of $\sigma^2$, extreme values

degrade detection quality. The anomaly score requires multiple stochastic perturbations per test sample, which increases the computational cost. Although moderate in practice, this may limit scalability in large-scale settings. Finally, our approach is investigated only for tabular and feature-space anomaly detection. Although ADBench contains image- and NLP-derived datasets, these are represented as fixed numerical embeddings rather than raw images or text. Transfer to raw image, text, or time-series models may require modality-specific architectures, perturbation models, and hyperparameter choices, and remains an important direction for future work.

**Conclusion** We propose DenoiserAD, a stability-based anomaly detector built on fixed-noise denoising models. Instead of scoring a point by a single reconstruction error as in classical DAEs, which are less explored under modern large-scale tabular AD benchmarks compared to recent diffusion-based approaches, we score it by the denoiser's input-referenced consistency under repeated stochastic perturbations, which exposes "near-manifold but unsupported" anomalies that DAEs often reconstruct well. Unlike diffusion-based AD methods, DenoiserAD does not require a multi-scale noise schedule, time conditioning, likelihood estimation, or iterative multi-step denoising: a single well-chosen noise scale and multiple forward passes suffice. We further use a reference parameterization (linear baseline mapping plus a learned correction), so the anomaly score reflects local sensitivity relative to a fixed reference rather than pure reconstruction. The approach is architecture-agnostic— a residual MLP works best in our experiments, but the gains come from the training/inference scheme, not architectural specialization. Across established tabular benchmarks, local stability under perturbations provides a strong and practical signal for one-class anomaly detection.

# Reproducibility Statement

To support full transparency and reproducibility, we provide the complete experimental code in the supplementary zip-file on OpenReview, accompanied by detailed usage instructions in a README file. The experimental code submitted during review will be made publicly available through OpenReview. For the final version, we will also provide an archival code link in the OpenReview code URL field.

# Acknowledgement

The authors gratefully acknowledge the institutional support provided by Daimler Truck AG, the Technical University of Munich, and the University of Stuttgart. The research was funded by Daimler Truck AG.

## Impact Statement

This paper presents work whose goal is to advance the field of machine learning. There are many potential societal consequences of our work, none of which we feel must be specifically highlighted here.

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

# A. Formal Stability Intuition

**Setup and Notation**    Let $f_\theta : \mathbb{R}^d \to \mathbb{R}^d$ denote a trained denoising function. For a fixed input $y \in \mathbb{R}^d$, we consider perturbed inputs

$$x = y + n, \quad n \sim \mathcal{N}(0, \sigma^2 I). \tag{6}$$

The anomaly score used in this work is defined as

$$S(y) = \mathbb{E}_n[\|f_\theta(y + n) - y\|^2], \tag{7}$$

where the expectation is approximated by a finite number of perturbations at inference time. This score is not the training objective but a post-hoc functional defined on the learned denoiser.

**Local Expansion of the Denoiser**    Assume $f_\theta$ is locally differentiable in a neighborhood of $y$. A first-order Taylor expansion around $y$ yields:

$$f_\theta(y + n) \approx f_\theta(y) + J_f(y)n, \tag{8}$$

where $J_f(y) \in \mathbb{R}^{d \times d}$ is the Jacobian of $f_\theta$ at $y$. Substituting into the score:

$$S(y) \approx \mathbb{E}_n[\|f_\theta(y) - y + J_f(y)n\|^2]. \tag{9}$$

Expanding the squared norm:

$$S(y) \approx \|f_\theta(y) - y\|^2 + 2\mathbb{E}_n[(f_\theta(y) - y)^\top J_f(y)n] + \mathbb{E}_n[\|J_f(y)n\|^2]. \tag{10}$$

Since $\mathbb{E}[n] = 0$, the cross term vanishes, leaving:

$$S(y) \approx \|f_\theta(y) - y\|^2 + \sigma^2 \text{Tr}(J_f(y)J_f(y)^\top) = \|f_\theta(y) - y\|^2 + \sigma^2\|J_f(y)\|_F^2. \tag{11}$$

While the first-order expansion serves as a local approximation, the aggregated score $S(y)$ implicitly captures higher-order non-linearities in regions where the denoiser's response is highly non-smooth, further penalizing unstable extrapolations. Consistent with this being an interpretive local approximation, a direct Hutchinson/Jacobian baseline $S_{\text{Jac}}(y) = \|J_f(y)\|_F^2$ underperforms reconstruction on ADBench, so our gains come from the finite-neighborhood score $S_2$ rather than the first-order term alone.

**Interpretation**    The decomposition reveals that $S(y)$ is a composite metric. The Bias term $\|f_\theta(y) - y\|^2$ captures the systemic shift toward the manifold, while the Stability term $\sigma^2\|J_f(y)\|_F^2$ quantifies the local roughness of the denoising field. On the data manifold, we interpret the denoiser as a local contraction mapping ($\|J_f\|_F$ is small). While this first-order view suggests a stability interpretation in terms of the Jacobian, we emphasize that it is only a local approximation. In practice, we operate at moderate $\sigma^2$, and the aggregated score $S(y)$ captures higher-order non-linear effects that are not reflected by the Jacobian term alone. We validate the implied $\sigma^2$-linearity empirically and extract a slope-based stability proxy in Section D.1.

**Effect of Multiple Perturbations**    The use of multiple stochastic perturbations of the same input is essential to the proposed anomaly score. For a fixed sample $y$, each perturbation $n_i \sim \mathcal{N}(0, \sigma^2 I)$ probes the denoising function along a different random direction in input space. A single perturbation therefore yields only a direction-dependent estimate of local behavior. Thus, the anomaly score is not a Monte-Carlo reconstruction error but a functional estimator of local instability of the denoiser. Unlike augmented reconstruction $\mathbb{E}[\|f(y + n) - (y + n)\|^2]$, our score anchors to the clean reference $y$, penalizing stable-but-shifted predictions.

**Relation to Denoising Autoencoders and Score Models**    Vincent (2011) showed that denoising autoencoders implicitly estimate the score of a smoothed data distribution. Our method differs in two key aspects: we do not estimate or utilize the score explicitly, and the anomaly score is not the score but a separate stability-based functional evaluated at inference time. Our Stability Score measures the consistency of the model across perturbations, which provides information beyond the mere score direction $\nabla \log p(x)$. Thus, while related in spirit, our approach focuses on functional stability rather than score estimation or density modeling. For completeness, and following Karras et al. (2022), we provide a short derivation.

A training set of a finite number of samples $\{y_1, \ldots, y_Y\}$, can be expressed as $p_{\text{data}}(x)$ from a mixture of Dirac delta distributions:

$$p_{\text{data}}(x) = \frac{1}{Y} \sum_{i=1}^{Y} \delta(x - y_i), \tag{12}$$

with $p(x; \sigma) = p_{\text{data}} * \mathcal{N}(0, \sigma(t)^2\mathbf{I})$ follows:

$$p(x; \sigma) = p_{\text{data}} * \mathcal{N}(0, \sigma(t)^2\mathbf{I}) \tag{13}$$

$$= \int_{\mathbb{R}^d} p_{\text{data}}(x_0)\mathcal{N}(x; x_0, \sigma^2\mathbf{I})dx_0 \tag{14}$$

$$= \frac{1}{Y} \sum_{i=1}^{Y} \int_{\mathbb{R}^d} \mathcal{N}(x; x_0, \sigma^2\mathbf{I})\delta(x_0 - y_i)dx_0 \tag{15}$$

$$= \frac{1}{Y} \sum_{i=1}^{Y} \mathcal{N}(x; y_i, \sigma^2\mathbf{I}). \tag{16}$$

with the loss function:

$$\mathcal{L}(D; \sigma) = \mathbb{E}_{y \sim p_{\text{data}}} \mathbb{E}_{n \sim \mathcal{N}(0, \sigma^2\mathbf{I})} \|D(y + n; \sigma) - y\|_2^2 \tag{17}$$

$$= \mathbb{E}_{y \sim p_{\text{data}}} \mathbb{E}_{x \sim \mathcal{N}(y, \sigma^2\mathbf{I})} \|D(x; \sigma) - y\|_2^2 \tag{18}$$

$$= \frac{1}{Y} \sum_{i=1}^{Y} \int_{\mathbb{R}^d} \mathcal{N}(x; y_i, \sigma^2\mathbf{I}) \|D(x; \sigma) - y_i\|_2^2 dx \tag{19}$$

$$= \int_{\mathbb{R}^d} \frac{1}{Y} \sum_{i=1}^{Y} \mathcal{N}(x; y_i, \sigma^2\mathbf{I}) \|D(x; \sigma) - y_i\|_2^2 dx. \tag{20}$$

and minimization of $\mathcal{L}(D; \sigma)$ by minimizing $\mathcal{L}(D; x, \sigma)$ independently for each $x$: $D(x; \sigma) = \arg\min_{D(x;\sigma)} \mathcal{L}(D; x, \sigma)$

$$0 = \nabla_{D(x;\sigma)} [\mathcal{L}(D; x, \sigma)] \tag{21}$$

$$0 = \nabla_{D(x;\sigma)} \left[ \frac{1}{Y} \sum_{i=1}^{Y} \mathcal{N}(x; y_i, \sigma^2\mathbf{I}) \|D(x; \sigma) - y_i\|_2^2 \right] \tag{22}$$

$$D(x; \sigma) = \frac{\sum_i \mathcal{N}(x; y_i, \sigma^2\mathbf{I})y_i}{\sum_i \mathcal{N}(x; y_i, \sigma^2\mathbf{I})}. \tag{23}$$

After some standard algebra it is shown:

$$\nabla_x \log p(x; \sigma) = \frac{\sum_i \nabla_x \mathcal{N}(x; y_i, \sigma^2\mathbf{I})}{\sum_i \mathcal{N}(x; y_i, \sigma^2\mathbf{I})} \tag{24}$$

$$= \left( \frac{\sum_i \mathcal{N}(x; y_i, \sigma^2\mathbf{I})y_i}{\sum_i \mathcal{N}(x; y_i, \sigma^2\mathbf{I})} - x \right) / \sigma^2 \tag{25}$$

$$= (D(x; \sigma) - x)/\sigma^2. \tag{26}$$

## B. Implementation

### B.1. Architecture and Practical Considerations

In our experiments, we primarily employ a residual MLP backbone tailored to tabular inputs. It differs from the original ResMLP (Touvron et al., 2022), which was proposed for image data, and is tailored to tabular data characteristics. More

details are provided in Figure 3 and Table 4. However, the same training objective and anomaly scoring procedure can be directly applied to architectures used in prior diffusion-based or reconstruction-based anomaly detection methods, without architectural modifications. This allows for a controlled comparison that isolates the effect of the proposed formulation from architectural choices. Importantly, the observed performance gains are not only tied to a specific network design but persist across different backbones. This shows that the improvements stem from the combination of denoising-based training, fixed-scale perturbations, and stability-based inference, instead of the model capacity or architectural specialization.

The proposed formulation admits several practical design choices that affect performance but do not alter the underlying objective or interpretation. In all experiments, we employ a fixed-noise scale during training and inference, which reduces complexity and simplifies the learning problem. This avoids the need for time or noise-level conditioning. While multi-scale perturbations are common in diffusion models, we find a single, appropriately chosen noise scale to be sufficient for anomaly detection in tabular data. At inference time, anomaly scores are computed by aggregating the denoiser's response over multiple stochastic perturbations of the same input. This procedure does not correspond to an additional training objective but serves to estimate the local variability of the denoising function under noise. In practice, a small number of perturbations should be sufficient. In our default setting $K = 15$, we observe near-saturated performance (Fig. 9) at a moderate runtime overhead. We set $\sigma^2 = 0.2$ a priori because inputs are standardized and this corresponds to a moderate corruption level (SNR = 5). While the score formally scales with $K$, parallel evaluation renders inference cost effectively constant up to $K \approx 15$; a scaling becomes apparent only beyond this regime. We thus set $K = 15$, which lies in the flat region of the cost curve while already saturating performance. We then report sweeps over $\sigma^2$ and $K$ only as ablations (Sec. 5 and Sec. D.2) to show robustness; these sweeps were not used for hyperparameter selection. Finally, we employ a standard preconditioning scheme to stabilize training. This choice follows established practices in EDM (Karras et al., 2022; Bishop, 1995; Huang et al., 2023) and does not introduce additional modeling assumptions.

## B.2. Architecture and Preconditioning Details

**Preconditioned Denoiser Parameterization**    We implement the denoising function using a preconditioned parameterization inspired by the EDM formulation of Karras et al. (2022). While our method does not employ noise schedules, timestep conditioning, or multi-step sampling, the preconditioning scheme provides a numerically stable way to represent denoising functions at a fixed-noise scale. Concretely, the denoiser is parameterized as

$$f_\theta(x) = c_{\text{skip}} \cdot x + c_{\text{out}} \cdot g_\theta(c_{\text{in}} \cdot x), \tag{27}$$

where $g_\theta$ is a neural network backbone and the coefficients $c_{\text{in}}, c_{\text{out}}, c_{\text{skip}}$ depend on the noise scale $\sigma$. In contrast to diffusion models, $\sigma$ is fixed throughout training and inference. Consequently, the coefficients remain constant and can be interpreted as a static reparameterization of the denoising function rather than as part of a time-dependent process.

**Choice of Preconditioning Coefficients**    Following Karras et al. (2022), the coefficients are chosen to normalize the relative scale of the noisy input and the denoising residual. We show the short version of the derivation. For unit variance inputs of $F_\theta(\cdot)$:

$$\text{Var}_{y,n}[c_{\text{in}}(\sigma)(y + n)] = 1 \tag{28}$$

$$c_{\text{in}}(\sigma)^2(\sigma_{\text{data}}^2 + \sigma^2) = 1 \tag{29}$$

$$c_{\text{in}}(\sigma) = 1/\sqrt{\sigma^2 + \sigma_{\text{data}}^2}. \tag{30}$$

For unit variance training target $F_{\text{target}}$:

$$\text{Var}_{y,n}[F_{\text{target}}(y, n; \sigma)] = 1 \tag{31}$$

$$c_{\text{out}}(\sigma)^2 = \text{Var}_{y,n}[(1 - c_{\text{skip}}(\sigma))y + c_{\text{skip}}(\sigma)n] \tag{32}$$

$$c_{\text{out}}(\sigma)^2 = (1 - c_{\text{skip}}(\sigma))^2 \sigma_{\text{data}}^2 + c_{\text{skip}}(\sigma)^2 \sigma^2. \tag{33}$$

For the weighted linear reference channel $c_{\text{skip}}(\sigma)$ to minimize $c_{\text{out}}(\sigma)$, $c_{\text{skip}}(\sigma) = \arg\min_{c_{\text{skip}}(\sigma)} c_{\text{out}}(\sigma)$ and with $c_{\text{out}}(\sigma) \geq$

$0$, $c_{\text{skip}}(\sigma) = \arg\min_{c_{\text{skip}}(\sigma)} c_{\text{out}}(\sigma)^2$ the solution is:

$$0 = \frac{d[c_{\text{out}}(\sigma)^2]}{dc_{\text{skip}}(\sigma)} \tag{34}$$

$$0 = \frac{d[(1 - c_{\text{skip}}(\sigma))^2 \sigma_{\text{data}}^2 + c_{\text{skip}}(\sigma)^2 \sigma^2]}{dc_{\text{skip}}(\sigma)} \tag{35}$$

$$0 = (\sigma^2 + \sigma_{\text{data}}^2) c_{\text{skip}}(\sigma) - \sigma_{\text{data}}^2 \tag{36}$$

$$c_{\text{skip}}(\sigma) = \frac{\sigma_{\text{data}}^2}{\sigma^2 + \sigma_{\text{data}}^2}. \tag{37}$$

after substitution:

$$c_{\text{out}}(\sigma)^2 = (1 - [c_{\text{skip}}(\sigma)])^2 \sigma_{\text{data}}^2 + [c_{\text{skip}}(\sigma)]^2 \sigma^2 \tag{38}$$

$$c_{\text{out}}(\sigma) = \frac{\sigma \cdot \sigma_{\text{data}}}{\sqrt{\sigma^2 + \sigma_{\text{data}}^2}}. \tag{39}$$

For a fixed-noise scale $\sigma$ and standardized data, they are given by

$$c_{\text{in}} = \frac{1}{\sqrt{\sigma^2 + 1}}, \quad c_{\text{skip}} = \frac{1}{\sigma^2 + 1}, \quad c_{\text{out}} = \frac{\sigma}{\sqrt{\sigma^2 + 1}}. \tag{40}$$

This choice ensures that the backbone network operates on inputs with approximately unit variance and that the output correction has a comparable scale across different feature dimensions.

**Backbone Architecture** In our experiments, we instantiate $g_\theta$ as a residual multilayer perceptron (ResMLP). In contrast to the original ResMLP (Touvron et al., 2022) tailored to tabular data. The network consists of $L = 3$ blocks with 2 fully connected layers with hidden size $d_{hidden} = 2048$ ($¿$ input $\in \mathbb{R}^{1555}$ for InternetAds dataset) and GeLU activations and Layer Normalization in each block, along with residual connections between consecutive layers. Residual connections are used solely to facilitate optimization and stable training. They do not affect the functional decomposition of the denoiser into a reference channel and a learned correction, which is entirely determined by the outer reference channel in Eq. (A.1). The full architecture is shown in Figure 3 right. We emphasize that the proposed training objective and anomaly scoring procedure do not rely on this specific architecture. Alternative backbones can be used without modification.

**Intuition** The architecture's effectiveness is rooted in the synergy between residual learning and spectral regularization. Ideally, a denoiser should recover the identity mapping $f(y) \approx y$ for samples on the data manifold. By utilizing a residual framework, the network is explicitly optimized to learn only the deviations from this identity, which implicitly regularizes the Jacobian $\mathbf{J}_f$ toward the identity matrix (or the null matrix for the residual branch) in high-density regions. This structural bias is further reinforced by the inclusion of LayerNorm, which stabilizes internal activations and effectively constrains the spectral norm of the weight matrices. We hypothesize that these design choices encourage a flatter functional landscape on-manifold, while allowing higher sensitivity off-manifold, which can amplify the stability-based signal. The model maintains a 'flat' and stable functional landscape within the data distribution, while the GeLU nonlinearities allow for the development of steep gradients and high sensitivity in anomalous regions, thereby amplifying the stability-based detection signal.

## B.3. Algorithms

---

**Algorithm 1** Training a Fixed-Noise Preconditioned Denoiser (DenoiserAD)

---

**Require:** Normal training data $\mathcal{D}_{\text{train}} \subset \mathbb{R}^d$, noise level $\sigma > 0$, data scale $\sigma_{\text{data}} > 0$, number of Epochs $B$, batch size $b$

1: Initialize $\theta$ (backbone $g_\theta : \mathbb{R}^d \to \mathbb{R}^d$)
2: Preconditioning coefficients:

$$w(\sigma) \leftarrow \frac{\sigma^2 + \sigma_{\text{data}}^2}{(\sigma\,\sigma_{\text{data}})^2}, \quad c_{\text{skip}} \leftarrow \frac{\sigma_{\text{data}}^2}{\sigma^2 + \sigma_{\text{data}}^2}, \quad c_{\text{out}} \leftarrow \frac{\sigma\,\sigma_{\text{data}}}{\sqrt{\sigma^2 + \sigma_{\text{data}}^2}}, \quad c_{\text{in}} \leftarrow \frac{1}{\sqrt{\sigma^2 + \sigma_{\text{data}}^2}}.$$

3: **for** $t = 1$ to $E$ **do**
4:     Sample a mini-batch $\{y_i\}_{i=1}^b \sim \mathcal{D}_{\text{train}}$
5:     Sample noise $n_i \sim \mathcal{N}(0, \sigma^2 I)$ for $i = 1, \ldots, b$
6:     Corrupt inputs: $x_i \leftarrow y_i + n_i$
7:     Backbone prediction: $F_i \leftarrow g_\theta(\tilde{x}_i)$
8:     Denoiser output:
$$\hat{y}_i \leftarrow c_{\text{skip}}\, x_i + c_{\text{out}}\, F_i$$

9:     Denoising loss:
$$\mathcal{L}(\theta) \leftarrow w(\sigma)\, \|\hat{y}_i - y_i\|_2^2$$

10:     Update: $\theta \leftarrow \text{OPT}(\theta, \nabla_\theta \mathcal{L})$
11: **end for**
12: **return** $\theta$

---

---

**Algorithm 2** Inference: Anomaly Score via multiple Perturbations

---

**Require:** Trained backbone $g_\theta$, test sample $y \in \mathbb{R}^d$, noise level $\sigma > 0$, data scale $\sigma_{\text{data}} > 0$, number of perturbations $K$

1: Preconditioning coefficients

$$c_{\text{skip}} \leftarrow \frac{\sigma_{\text{data}}^2}{\sigma^2 + \sigma_{\text{data}}^2}, \quad c_{\text{out}} \leftarrow \frac{\sigma\,\sigma_{\text{data}}}{\sqrt{\sigma^2 + \sigma_{\text{data}}^2}}, \quad c_{\text{in}} \leftarrow \frac{1}{\sqrt{\sigma^2 + \sigma_{\text{data}}^2}}.$$

2: Replicate anchor: $Y \leftarrow \mathbf{1}_K \otimes y \in \mathbb{R}^{K \times d}$
3: Sample noises: $N \sim \mathcal{N}(0, \sigma^2 I) \in \mathbb{R}^{K \times d}$
4: Corrupt inputs: $X \leftarrow Y + N$
5: Batched backbone prediction: $F \leftarrow g_\theta(\tilde{X})$
6: Denoiser outputs: $\hat{Y} \leftarrow c_{\text{skip}}\, X + c_{\text{out}}\, F$
7: $A(y) \leftarrow \frac{1}{K} \sum_{k=1}^K \|\hat{Y}_{k,:} - y\|_2^2$
8: **return** $A(y)$

---

## B.4. Hyperparameter and Network Architecture

| Hyperparameter | Value |
|---|---|
| Hidden layer size | 2048 |
| Blocks | 3 |
| Activation function | GeLU |
| Optimizer | AdamW |
| Learning rate | 0.0001 |
| Batch size | 128 |
| Number of epochs | 1000 |
| $K$ | 15 |
| $\sigma^2$ | 0.2 |

*Table 4.* Hyperparameters for ResMLP and experiments

## B.5. Runtime

The experiments on the first five random seeds (0-4) of all 121 sub-datasets and baseline methods, as well as all experiments with our model, were performed on a server with 2x AMD EPYC™ GENOA 9654 processors, 2.40-3.70 GHz with 1.5TB RAM and 4x NVIDIA L40 GPUs with 48 GB GDDR6 VRAM on one GPU each.

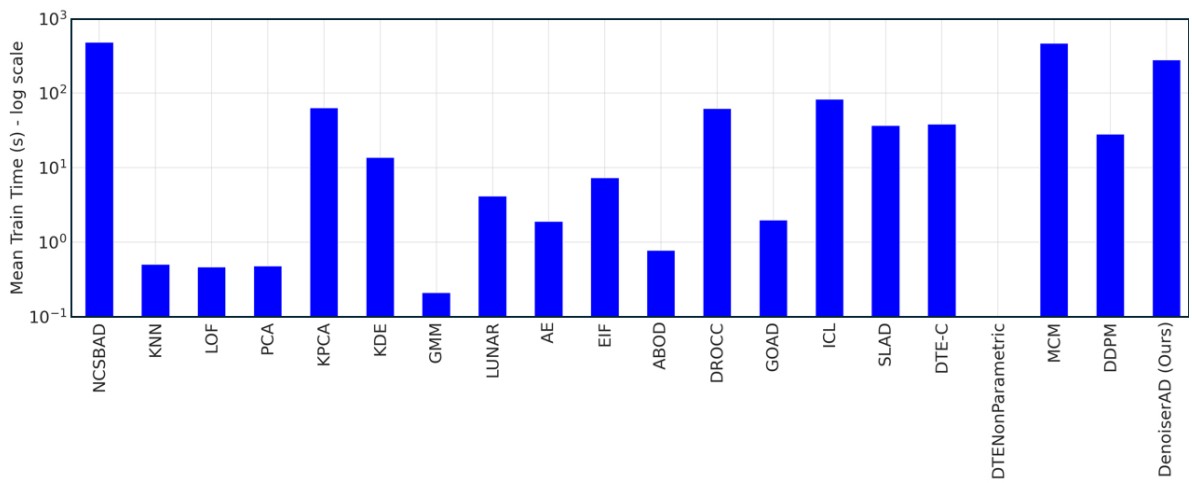

*Figure 6.* Average Training Time

## C. Local Stability Intuition of the Preconditioned Denoiser

**Functional Decomposition of the Denoiser** For the denoiser parameterization, we consider a denoising function

$$f_\theta : \mathbb{R}^d \to \mathbb{R}^d$$

parameterized using an explicit linear reference channel and a learned non-linear correction term:

$$f_\theta(x) = c_{\text{skip}} \, x + c_{\text{out}} \, g_\theta(c_{\text{in}} \, x), \tag{41}$$

where $g_\theta : \mathbb{R}^d \to \mathbb{R}^d$ denotes a neural network backbone and $c_{\text{in}}, c_{\text{out}}, c_{\text{skip}} \in \mathbb{R}$ are fixed scalar coefficients determined by the preconditioning scheme. This parameterization follows the preconditioned denoiser formulation introduced in EDM (Karras et al., 2022), but is employed here in a fixed-noise setting without time conditioning or noise schedules. Residual connections inside the backbone $g_\theta$ serve solely to facilitate optimization and do not define a functional reference mapping. The only explicit reference channel in the denoiser is given by the global linear term $c_{\text{skip}}x$.

**Local Expansion Under Input Perturbations.** Let $y \in \mathbb{R}^d$ be a fixed input and consider a small perturbation $n \sim \mathcal{N}(0, \sigma^2 I)$. Assume that $g_\theta$ is differentiable in a neighborhood of $c_{\text{in}}y$. A first-order Taylor expansion yields

$$g_\theta(c_{\text{in}}(y + n)) \approx g_\theta(c_{\text{in}}y) + J_g(c_{\text{in}}y)\, c_{\text{in}}\, n, \tag{42}$$

where $J_g(c_{\text{in}}y) \in \mathbb{R}^{d \times d}$ denotes the Jacobian of $g_\theta$ evaluated at $c_{\text{in}}y$. Substituting into Eq. (41) and collecting terms linear in $n$ gives

$$f_\theta(y + n) \approx f_\theta(y) + (c_{\text{skip}}I + c_{\text{out}}\, J_g(c_{\text{in}}y)\, c_{\text{in}})\, n, \tag{43}$$

$$f_\theta(y + n) - f_\theta(y) \approx (c_{\text{skip}}I + c_{\text{out}}\, J_g(c_{\text{in}}y)\, c_{\text{in}})\, n. \tag{44}$$

This expression shows that the local sensitivity of the denoiser decomposes into a constant isotropic contribution induced by the linear reference channel and a data-dependent contribution induced by the learned correction term.

**Connection to the Stability-Based Anomaly Score.** The anomaly score is defined as

$$A(y) = \mathbb{E}_n\left[\|f_\theta(y + n) - y\|^2\right]. \tag{45}$$

Write

$$f_\theta(y + n) - y = \underbrace{(f_\theta(y) - y)}_{=:b(y)} + \underbrace{(f_\theta(y + n) - f_\theta(y))}_{\approx M(y)n},$$

where

$$M(y) = c_{\text{skip}}I + c_{\text{out}}\, J_g(c_{\text{in}}y)\, c_{\text{in}}.$$

Then

$$\|f_\theta(y + n) - y\|^2 = \|b(y) + M(y)n\|^2 = \|b(y)\|^2 + 2\, b(y)^\top M(y)n + \|M(y)n\|^2.$$

Taking expectations over $n \sim \mathcal{N}(0, \sigma^2 I)$ yields

$$A(y) = \|f_\theta(y) - y\|^2 + \mathbb{E}_n\left[\|M(y)n\|^2\right] \tag{46}$$

$$= \|f_\theta(y) - y\|^2 + \text{Tr}\left(M(y)^\top M(y)\, \mathbb{E}[nn^\top]\right) \tag{47}$$

$$= \|f_\theta(y) - y\|^2 + \sigma^2\, \|M(y)\|_F^2. \tag{48}$$

Equation (48) decomposes the anomaly score into a bias term, measuring the deviation of the denoiser from the identity at $y$, and a stability term, measuring the local sensitivity of the denoiser around $y$. Substituting $M(y)$ yields

$$A(y) \approx \|f_\theta(y) - y\|^2 + \sigma^2\, \|c_{\text{skip}}I + c_{\text{out}}\, J_g(c_{\text{in}}y)\, c_{\text{in}}\|_F^2. \tag{49}$$

For off-manifold inputs, the correction term may be evaluated outside its well-trained region; this can increase local sensitivity and thus the stability component of the score, although higher-order effects may contribute in highly nonlinear regions. Thus, the anomaly score measures local instability of the learned correction relative to the explicit linear reference mapping, rather than reconstruction error alone.

**Interpretation for Normal and Anomalous Samples** For samples lying on or near the data manifold, the learned correction term $g_\theta$ is trained to make minimal adjustments to the reference mapping. In this regime, the Jacobian norm $\|J_g(c_{\text{in}}y)\|_F$ remains small, and the anomaly score is dominated by the constant contribution of the linear reference channel. For off-manifold inputs, the correction term may be evaluated outside its well-trained region; this can increase local sensitivity and thus the stability component of the score, although higher-order effects may contribute in highly nonlinear regions. The above analysis does not rely on specific architectural properties of the backbone $g_\theta$, beyond local differentiability. Residual connections inside the backbone influence optimization and expressivity but do not define the reference mapping used in the stability analysis. Crucially, removing the global linear reference channel would eliminate the constant baseline sensitivity term and render the interpretation of the anomaly score less well defined. Conversely, using only the reference channel without a learned correction would not allow the model to capture meaningful data-dependent variability. The anomaly signal therefore arises from the explicit decomposition into a linear reference mapping and a learned non-linear correction, rather than from residual architectures.

# D. Further Ablations and Definitions

## D.1. For Disentangling Stability from Reconstruction

**Denoiser parameterizations:**

**DenoiserAD** (Ours), reference channel + correction, $g(c_{\text{in}}x)$: ResMLP from Figure 3 with hyperparameter from Table 4:

$$f(x) = c_{\text{skip}}x + c_{\text{out}}g(c_{\text{in}}x) \tag{50}$$

**DAE**, $g(c_{\text{in}}x)$: ResMLP from Figure 3 with hyperparameter from Table 4:

$$f(x) = g(c_{\text{in}}x) \tag{51}$$

**Inference scores**, for $x_i = y + n_i$, $\hat{y}_i = f(x_i)$, $\bar{\hat{y}} = \frac{1}{K}\sum_i \hat{y}_i$:

**Pure reconstruction:**

$$S_1(y) = \|f_\theta(y) - y\|^2 \tag{52}$$

**Single noise draw:**

$$S_{1n}(y) = \|\hat{y}_1 - y\|^2, \tag{53}$$
$$\tag{54}$$

**Input-referenced stability (ours):**

$$S_2(y) = \frac{1}{K}\sum_i \|\hat{y}_i - y\|^2 \tag{55}$$

**Output variance:**

$$S_3(y) = \frac{1}{K}\sum_i \|\hat{y}_i - \bar{\hat{y}}\|^2 \tag{56}$$

**Augmented reconstruction:**

$$S_4(y) = \frac{1}{K}\sum_i \|\hat{y}_i - x_i\|^2 \tag{57}$$

**Mechanism visualization on a synthetic manifold** To illustrate what the proposed input-referenced stability score captures beyond standard reconstruction error, we visualize the score fields on a controlled 2D synthetic example (Figure 7). We generate "normal" data on a noisy sinusoidal manifold and place a critical out-of-distribution point (red $\times$) inside a data-sparse "hole" region that is geometrically close to the manifold but not supported by training samples. This setting isolates a common failure mode in reconstruction-based anomaly detection: points near the data manifold may still be reconstructed well, even when they are out of distribution, because the model can learn a projection back onto the manifold.

Figure 7 compares a plain denoising autoencoder (DAE; left) with our DenoiserAD parameterization (right) under identical noise level $\sigma$ and number of stochastic perturbations $K$.

We visualize five scoring rules evaluated on a dense grid:

$$S_1(y) = \|f_\theta(y) - y\|^2 \quad \text{(pure reconstruction)}, \tag{58}$$
$$S_{1n}(y) = \|f_\theta(y + n) - y\|^2 \quad \text{(single noise draw)}, \tag{59}$$
$$S_2(y) = \mathbb{E}_n[\|f_\theta(y + n) - y\|^2] \quad \text{(input-referenced stability; ours)}, \tag{60}$$
$$S_3(y) = \mathbb{E}_n[\|f_\theta(y + n) - \mathbb{E}_{n'}[f_\theta(y + n')]\|^2] \quad \text{(mean-centered variance)}, \text{ and} \tag{61}$$
$$S_4(y) = \mathbb{E}_n[\|f_\theta(y + n) - (y + n)\|^2] \quad \text{(augmented reconstruction / DAE+TTA)}. \tag{62}$$

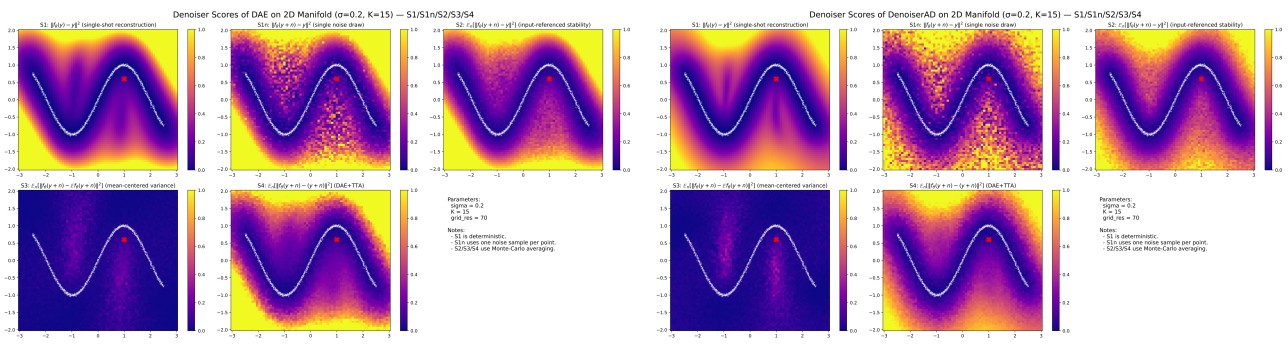

*Figure 7.* Visualization of denoiser-based anomaly scores on a 2D synthetic manifold with a critical out-of-distribution point (red ×) located near the data manifold but inside a data-sparse region ("hole"). Left: plain denoising autoencoder (DAE). Right: DenoiserAD. While single-shot reconstruction ($S_1$) remains small near the hole due to learned projection onto the manifold, the proposed input-referenced stability score ($S_2$) assigns higher scores in the data-sparse neighborhood, reflecting increased sensitivity to perturbations around $y$. Mean-centered variance ($S_3$) and augmented reconstruction ($S_4$) capture different notions of variability and do not emphasize the same failure mode.

The key observation is that $S_1$ can remain small in the hole region: even though the red × is out of distribution, the denoiser can map inputs back toward the learned manifold, leading to low reconstruction error and potentially missed anomalies. In contrast, the proposed stability score $S_2$ increases in data-sparse neighborhoods close to the manifold. Intuitively, when $y$ lies in a region not supported by training data, small perturbations $y + n$ induce larger changes in the denoiser output relative to the original reference point $y$, yielding higher $\mathbb{E}_n[\|f_\theta(y + n) - y\|^2]$. This behavior is precisely what is desirable for "near-manifold but unsupported" anomalies: it penalizes local instability rather than relying solely on pointwise reconstruction.

The remaining scores highlight why the reference to the unperturbed input matters. $S_3$ measures output variability around the mean prediction $\mathbb{E}[f_\theta(y + n)]$ and thus captures a notion of stochastic dispersion, but it discards the explicit reference to $y$; consequently, it can under-emphasize certain holes where the mean prediction is stable but shifted. $S_4$ compares $f_\theta(y + n)$ to the noisy input $(y + n)$, corresponding to augmented reconstruction / test-time augmentation and (under standard assumptions) relating to score-matching behavior; however, it is not specifically targeted to detect instabilities relative to the clean reference $y$. If the model learns to invert noise well, $S_4$ can be low even in holes; $S_2$ increases because the mapping varies with perturbation relative to $y$. Overall, Figure 7 qualitatively supports the motivation of DenoiserAD: input-referenced stability $S_2$ can expose out-of-distribution regions that appear deceptively benign under reconstruction-only scoring.

**Per-point linearity in $\sigma^2$ and stability proxy.** To assess whether our score $A_\sigma(y) = \mathbb{E}_n[\|f_\theta(y + n) - y\|^2]$ admits the first-order decomposition $A_\sigma(y) \approx b(y) + \sigma^2 s(y)$, we evaluate $A_\sigma(y)$ at multiple noise levels and fit, for each input $y$, a linear model $A_\sigma(y) = \alpha(y) + \beta(y)\sigma^2$. Across the evaluation grid, the fit is highly accurate for most points (median $R^2 = 0.98$, 75th percentile $R^2 = 0.998$), supporting the $\sigma^2$-linearity underlying our interpretation. While a minority of regions deviates from linearity (e.g., 23.8% of points have $R^2 < 0.9$), these deviations occur primarily in low-density/off-manifold areas where higher-order effects are expected. Importantly, the slope-based stability proxy $\hat{s}(y)$ is largely uncorrelated with the bias term $b(y) = \|f_\theta(y) - y\|^2$ (Spearman $\rho = -0.06$), indicating that stability captures information beyond reconstruction error.

Our analysis is based on the approximation

$$A_\sigma(y) = \mathbb{E}_n[\|f_\theta(y + n) - y\|^2] \approx \|f_\theta(y) - y\|^2 + \sigma^2 \|J_f(y)\|_F^2, \tag{63}$$

which implies that $A_\sigma(y)$ should be approximately affine in $\sigma^2$ in the regime where higher-order terms are small. We therefore compute $A_\sigma(y)$ at several noise levels $\sigma \in \{\ldots\}$ using the same Monte-Carlo directions $n = \sigma\varepsilon$ (shared $\varepsilon \sim \mathcal{N}(0, I)$) to reduce estimator variance, and fit $A_\sigma(y) = \alpha(y) + \beta(y)\sigma^2$ independently for each $y$. We report per-point $R^2$ values and summarize their distribution: median $R^2 = 0.982$, mean $R^2 = 0.885$, with the 25th percentile at $R^2 = 0.909$. These results indicate that the $\sigma^2$-linearity assumption holds strongly for the majority of points, while a smaller subset exhibits notable nonlinearity (e.g., 23.8% of points with $R^2 < 0.9$), consistent with the presence of higher-order terms away

from the locally linear regime.

We then define a stability-only proxy via a finite-difference slope between two noise levels:

$$\hat{s}(y) = \frac{A_{\sigma_2}(y) - A_{\sigma_1}(y)}{\sigma_2^2 - \sigma_1^2}, \tag{64}$$

and compare it to the bias term $b(y) = A_0(y) = \|f_\theta(y) - y\|^2$. We find that $\hat{s}(y)$ is weakly correlated with $b(y)$ (Spearman $\rho = -0.063$), suggesting that the stability component is not a trivial rescaling of reconstruction error.

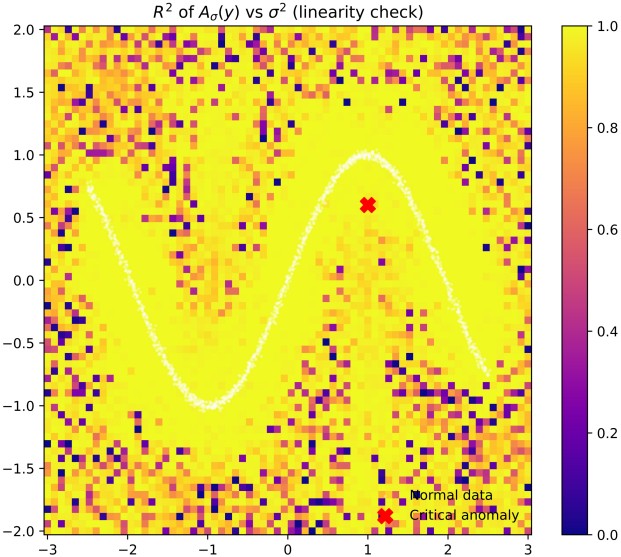

Figure 8. Per-point coefficient of determination $R^2$ of the linear fit $A_\sigma(y) = \alpha(y) + \beta(y)\sigma^2$ across the 2D evaluation grid. Most regions exhibit near-perfect $\sigma^2$-linearity (yellow), supporting the first-order decomposition used to motivate our stability term, while localized regions show deviations where higher-order effects are likely.

### D.2. Iteration Efficiency and Diminishing Returns

Figure 9 analyzes the trade-off between detection accuracy and inference cost as a function of the number of stochastic perturbations $K$ used to estimate the anomaly score.

Increasing $K$ improves AUCROC only in the low-$K$ regime, with rapid gains from $K = 1$ to $K \approx 5$. Beyond $K \approx 15$, performance saturates and additional perturbations yield marginal improvements. Although the anomaly score is defined as an average over $K$ perturbations, inference cost does not increase proportionally for small $K$ due to parallel evaluation on modern accelerators. In practice, runtime remains nearly constant up to $K \approx 15$ and only exhibits beyond this point. We therefore fix $K = 15$ in all experiments, as larger values provide diminishing returns while incurring measurable computational overhead. This creates a clear efficiency frontier: most of the achievable accuracy is attained with a small number of perturbations, while larger $K$ primarily increases computational cost. We therefore fix $K$ at a small value (marked in Figure 9), which retains near-saturated performance while reducing inference time by a factor of 3–4× compared to large-$K$ settings. These results suggests the interpretation of the proposed score as a stability estimator: only a few random probes are sufficient to reveal local instability around anomalous inputs, making the method practical for large-scale deployment.

## E. ADBench Datasets

We use all 57 ADBench datasets (which comprise 47 tabular datasets, complemented by five image and five natural language datasets embeddings in tabular shape) with 121 sub-datasets from the original ADBench benchmark (Han et al., 2022) for our experiments. Table 5 lists the names of the datasets, the number of sub-datasets, the number of samples, the number of features, the number of anomalies, the anomaly ratio, and the domain of the datasets.

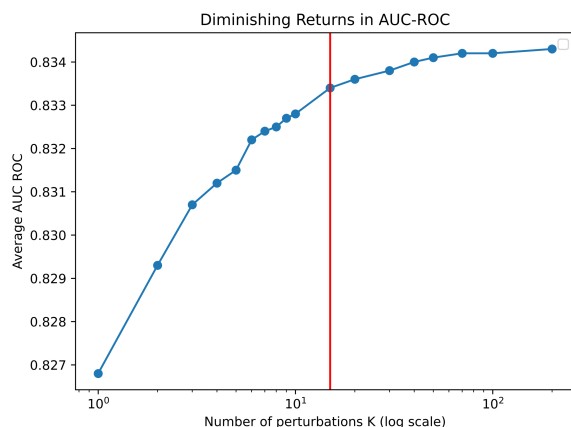 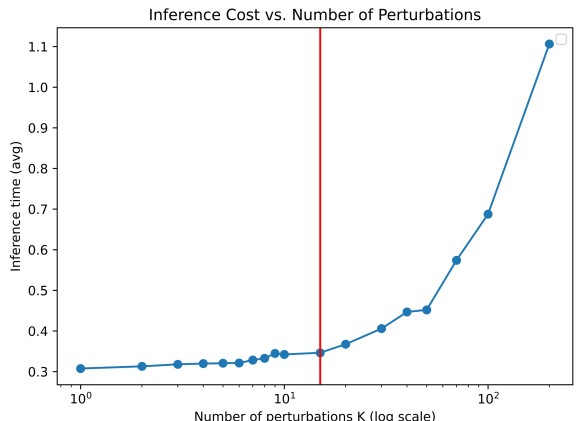

*Figure 9.* Diminishing returns of Monte-Carlo perturbations at inference. Left: Average AUCROC as a function of the number of stochastic perturbations $K$ (log scale). Performance improves rapidly for small $K$ and saturates beyond $K \approx 15$. Right: Corresponding inference time. Runtime remains nearly constant for small $K$ due to parallel evaluation, and grows only once $K$ exceeds this regime. The vertical line marks the operating point used in our experiments, illustrating that most of the achievable performance is obtained with a small number of perturbations at a fraction of the maximal cost.

| Name | #Subdatasets | #Samples | #Features | #Anomaly (%) | Category |
|---|---|---|---|---|---|
| ALOI | 1 | 49534 | 27 | 1508 (3.04) | Image |
| annthyroid | 1 | 7200 | 6 | 534 (7.42) | Healthcare |
| backdoor | 1 | 95239 | 196 | 2362 (2.48) | Network |
| breastw | 1 | 683 | 9 | 239 (34.99) | Healthcare |
| campaign | 1 | 41188 | 62 | 4640 (11.27) | Finance |
| cardio | 1 | 1831 | 21 | 176 (9.61) | Healthcare |
| Cardiotocography | 1 | 2114 | 21 | 466 (22.04) | Healthcare |
| celeba | 1 | 202599 | 39 | 4547 (2.24) | Image |
| census | 1 | 299285 | 500 | 18568 (6.20) | Sociology |
| cover | 1 | 286048 | 10 | 2747 (0.96) | Botany |
| donors | 1 | 619326 | 10 | 36710 (5.93) | Sociology |
| fault | 1 | 1941 | 27 | 673 (34.67) | Physical |
| fraud | 1 | 284807 | 29 | 492 (0.17) | Finance |
| glass | 1 | 214 | 7 | 9 (4.21) | Forensic |
| Hepatitis | 1 | 80 | 19 | 13 (16.25) | Healthcare |
| http | 1 | 567498 | 3 | 2211 (0.39) | Web |
| InternetAds | 1 | 1966 | 1555 | 368 (18.72) | Image |
| Ionosphere | 1 | 351 | 32 | 126 (35.9) | Oryctognosy |
| landstat | 1 | 6435 | 36 | 1333 (20.71) | Astronautics |
| letter | 1 | 1600 | 32 | 100 (6.25) | Image |
| Lymphography | 1 | 148 | 18 | 6 (4.05) | Healthcare |
| magic.gamma | 1 | 19020 | 10 | 6688 (35.16) | Physical |
| mammography | 1 | 11183 | 6 | 260 (2.32) | Healthcare |
| mnist | 1 | 7603 | 100 | 700 (9.21) | Image |
| musk | 1 | 3062 | 166 | 97 (3.17) | Chemistry |
| optdigits | 1 | 5216 | 64 | 150 (2.87) | Image |
| PageBlocks | 1 | 5393 | 10 | 510 (9.46) | Document |
| pendigits | 1 | 6870 | 16 | 156 (2.27) | Image |
| Pima | 1 | 768 | 8 | 268 (34.91) | Healthcare |
| satellite | 1 | 6435 | 36 | 1333 (31.64) | Astronautics |
| satimage-2 | 1 | 5803 | 36 | 71 (1.22) | Astronautics |

| Name | #Subdatasets | #Samples | #Features | #Anomaly (%) | Category |
|---|---|---|---|---|---|
| skin | 1 | 245057 | 3 | 50850 (20.75) | Image |
| smtp | 1 | 95156 | 3 | 30 (0.03) | Web |
| SpamBase | 1 | 4206 | 57 | 1678 (39.9) | Document |
| speech | 1 | 3686 | 400 | 31 (1.65) | Linguistics |
| Stamps | 1 | 340 | 9 | 31 (9.12) | Document |
| thyroid | 1 | 3772 | 6 | 93 (2.47) | Healthcare |
| vertebral | 1 | 240 | 6 | 30 (12.50) | Biology |
| vowels | 1 | 1456 | 12 | 50 (3.43) | Linguistics |
| Waveform | 1 | 3442 | 21 | 100 (2.91) | Physics |
| WBC | 1 | 223 | 9 | 10 (4.48) | Healthcare |
| WDBC | 1 | 367 | 30 | 10 (2.72) | Healthcare |
| Wilt | 1 | 4819 | 5 | 257 (5.33) | Botany |
| wine | 1 | 129 | 13 | 10 (7.75) | Chemistry |
| WPBC | 1 | 198 | 33 | 47 (23.74) | Healthcare |
| yeast | 1 | 1484 | 8 | 507 (34.16) | Biology |
| CIFAR10 | 10 | 5263 | 512 | 263 (5.00) | Image |
| FashionMNIST | 10 | 6315 | 512 | 315 (5.00) | Image |
| MNIST-C | 15 | 10000 | 512 | 500 (5.00) | Image |
| MVTec-AD | 15 | 5354 | 512 | 1258 (23.50) | Image |
| SVHN | 10 | 5208 | 512 | 260 (5.00) | Image |
| Agnews | 4 | 10000 | 768 | 500 (5.00) | NLP |
| Amazon | 1 | 10000 | 768 | 500 (5.00) | NLP |
| Imdb | 1 | 10000 | 768 | 500 (5.00) | NLP |
| Yelp | 1 | 10000 | 768 | 500 (5.00) | NLP |
| 20newsgroups | 6 | 11905 | 768 | 591 (4.96) | NLP |

*Table 5.* Summary of all multivariate datasets included in our benchmark: This table presents an overview of each dataset, including the name, the number of subdatasets, the number of samples, the number of features, the number of anomalies, and the application domain.

As mentioned in Section 4 and following previous work (Livernoche et al., 2024) and the ADBench Benchmark we subsample some datasets to 50.000 data points. For full transparency, we provide statistics for the datasets (with random seeds 0-4) where the limitation to 50,000 samples applies.

| Dataset | #Samples | #Anomalies | Anomalies % | Anomalies % full dataset |
|---|---|---|---|---|
| backdoor | 50000.00 | 1238.40 (24.39) | 2.4768 | 2.48 |
| celeba | 50000.00 | 1101.40 (23.16) | 2.2028 | 2.24 |
| census | 50000.00 | 3094.80 (55.33) | 6.1896 | 6.20 |
| cover | 50000.00 | 491.00 (17.20) | 0.9820 | 0.96 |
| donors | 50000.00 | 2965.20 (39.84) | 5.9304 | 5.93 |
| fraud | 50000.00 | 84.00 (7.62) | 0.1680 | 0.17 |
| http | 50000.00 | 186.40 (10.13) | 0.3728 | 0.39 |
| mulcross | 50000.00 | 5067.80 (32.91) | 10.1356 | 10.0 |
| skin | 50000.00 | 10393.00 (69.09) | 20.786 | 20.75 |
| smtp | 50000.00 | 17.60 (2.80) | 0.0352 | 0.03 |

*Table 6.* Statistics of subsampled datasets

**Handling of Categorical Features in DenoiserAD**     DenoiserAD is formulated to operate on real-valued tabular feature vectors with additive Gaussian perturbations. Consequently, it is directly applicable to numerical tabular data. More

generally, DenoiserAD can be applied to tabular data once all features are represented in a real-valued space. The current formulation of DenoiserAD does not include a dedicated mechanism for handling categorical features, such as special encoding or a discrete diffusion process. This approach is consistent with the ADBench benchmark, where all datasets are provided in a purely real-valued numerical representation. The handling of categorical features was performed in ADBench during its development, and thus no further explicit handling of categorical features is required within our benchmark.

# F. Fully Unsupervised Anomaly Detection

In this section, we evaluate the proposed method in the fully unsupervised anomaly detection setting, in which the training data contains an unknown fraction of anomalous samples. This regime lies outside the design scope of our method but is included for completeness and to enable a direct comparison with prior work targeting this setting. Following standard practice (Livernoche et al., 2024), the entire dataset is used for both training and evaluation, and anomaly scores are computed for all instances. Consistent with the limitations discussed in the main paper, we observe that the method is sensitive to training-set contamination and does not match the performance of specialized robust baselines under increasing contamination levels. To improve robustness under contaminated training, we additionally evaluate a trimmed-loss variant. Specifically, for each mini-batch we compute per-sample losses and update the model using only the lowest-loss 50% of samples (discarding the highest-loss half). This simple modification substantially improves performance in the fully unsupervised regime and makes the method competitive with the strongest robust baselines. We report the boxplots of results and ranks below.

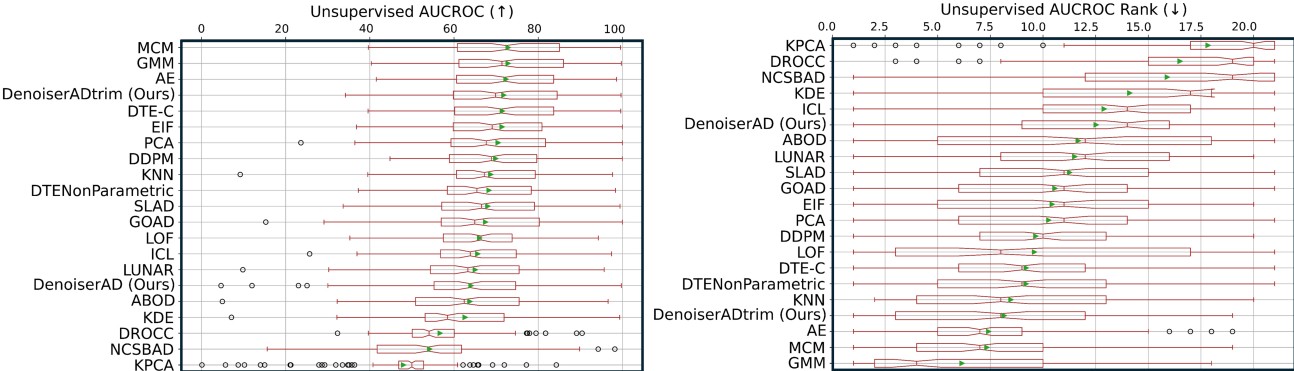

*Figure 10.* Performance box-plots across datasets in unsupervised setting. Left: Box-plots of AUCROC scores for all compared methods across the ADBench benchmark datasets. Right: Corresponding Box-plots of method ranks (lower is better) across the datasets. Each box summarizes the variability across datasets, with the central marker (green) indicating the mean performance. DenoiserADtrim corresponds to the trimmed loss version.

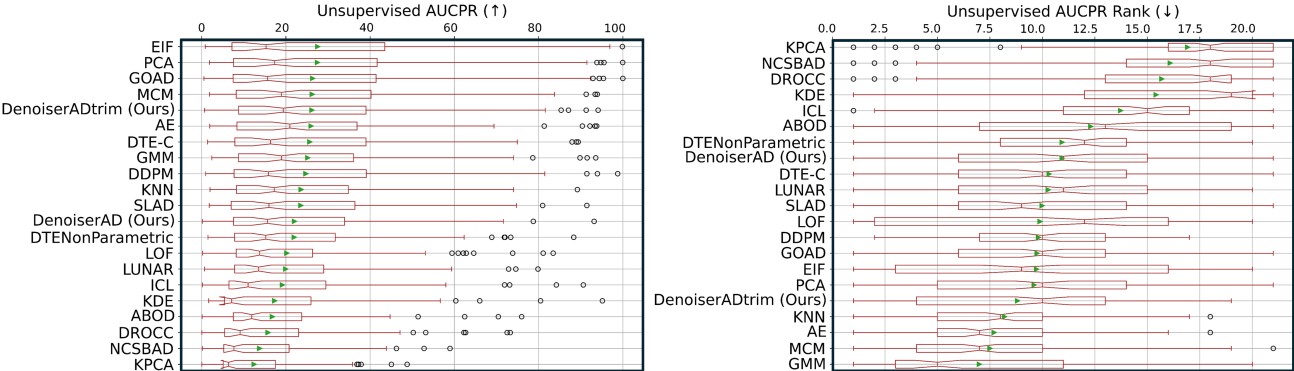

*Figure 11.* Performance box-plots across datasets in unsupervised setting. Left: Box-plots of AUCPR scores for all compared methods across the ADBench benchmark datasets. Right: Corresponding Box-plots of method ranks (lower is better) across the datasets. Each box summarizes the variability across datasets, with the central marker (green) indicating the mean performance. DenoiserADtrim corresponds to the trimmed loss version.

# G. Full Results

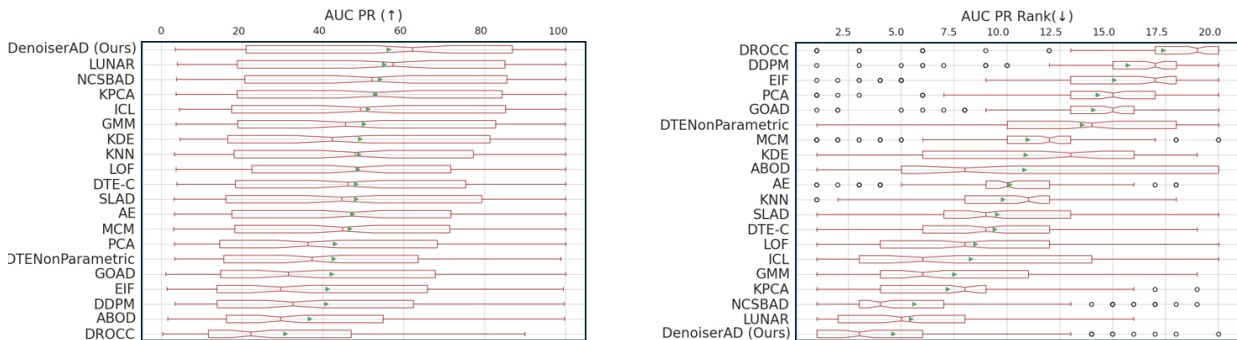

*Figure 12.* Performance box-plots across datasets. Left: Box-plots of AUCPR scores for all compared methods across the ADBench benchmark datasets. Right: Corresponding Box-plots of method ranks (lower is better) across the datasets. Each box summarizes the variability across datasets, with the central marker (green) indicating the mean performance. DenoiserAD achieves consistently high AUCPR and favorable ranks, indicating performance gains across heterogeneous tabular benchmarks.

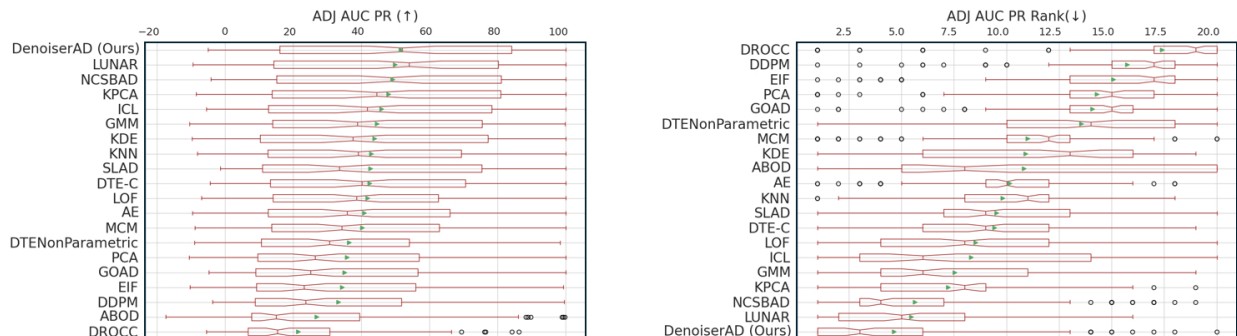

*Figure 13.* Performance box-plots across datasets. Left: Box-plots of ADJ AUCPR scores for all compared methods across the ADBench benchmark datasets. Right: Corresponding Box-plots of method ranks (lower is better) across the datasets. Each box summarizes the variability across datasets, with the central marker (green) indicating the mean performance. DenoiserAD achieves consistently high ADJ AUCPR and favorable ranks, indicating performance gains across heterogeneous tabular benchmarks.

Consistent with prior work (Bergman & Hoshen, 2020; Shenkar & Wolf, 2022; Livernoche et al., 2024; Thimonier et al., 2023), the decision threshold for all methods is determined such that the predicted number of anomalies aligns with the actual number of anomalies present in the data. For completeness, we also report thresholded metrics (F1, adjusted-F1). Following ADBench/DTE we include oracle-threshold results for comparability, which assumes knowledge of the test contamination rate. We treat oracle-F1 as benchmark-compatibility only and do not claim deployment relevance.

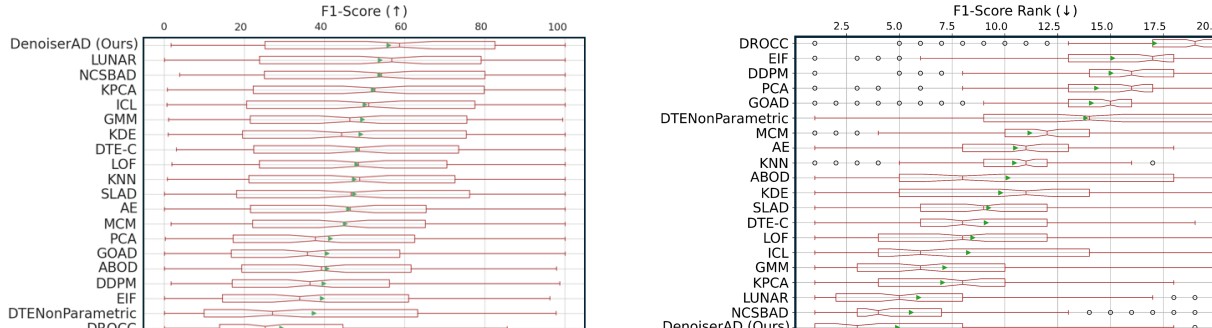

*Figure 14.* Performance box-plots across datasets. Left: Box-plots of F1-scores for all compared methods across the ADBench benchmark datasets. Right: Corresponding Box-plots of method ranks (lower is better) across the datasets. Each box summarizes the variability across datasets, with the central marker (green) indicating the mean performance. DenoiserAD achieves consistently high F1-scores and favorable ranks, indicating performance gains across heterogeneous tabular benchmarks.

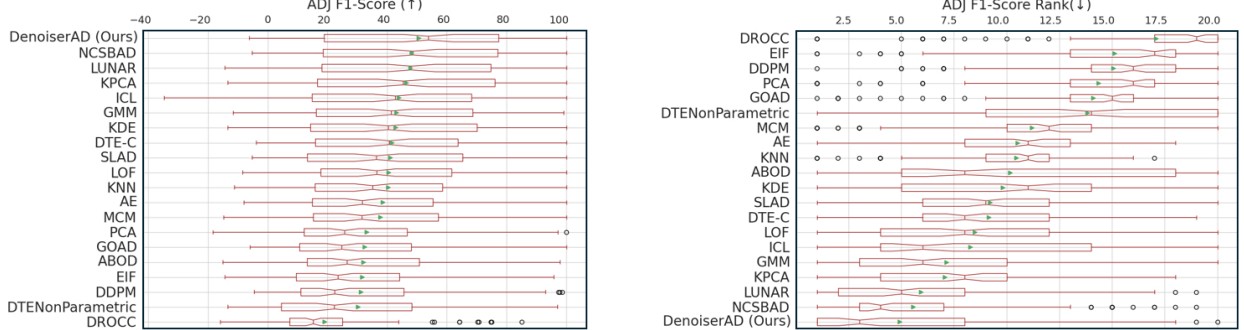

*Figure 15.* Performance box-plots across datasets. Left: Box-plots of ADJ F1-Score scores for all compared methods across the ADBench benchmark datasets. Right: Corresponding Box-plots of method ranks (lower is better) across the datasets. Each box summarizes the variability across datasets, with the central marker (green) indicating the mean performance. DenoiserAD achieves consistently high ADJ F1-Score and favorable ranks, indicating performance gains across heterogeneous tabular benchmarks.

| Model | NCSBAD | KNN | LOF | PCA | KPCA | KDE | GMM | LUNAR | AE | EIF | ABOD | DROCC | GOAD | ICL | SLAD | DTE-C | DTENonParametric | MCM | DDPM | DenoiserAD (Ours) |
|---|---|---|---|---|---|---|---|---|---|---|---|---|---|---|---|---|---|---|---|---|
| ALOI | 48.96 (0.57) [17] | 50.62 (0.70) [6] | 48.63 (0.39) [18] | 53.63 (0.98) [2] | 49.95 (0.61) [13] | 52.69 (0.84) [4] | 54.04 (0.88) [1] | 50.23 (0.66) [9] | 52.93 (0.82) [3] | 51.64 (1.28) [5] | 49.82 (0.45) [14] | 50.00 (0.00) [12] | 46.71 (2.02) [20] | 46.74 (0.55) [19] | 50.23 (0.69) [9] | 50.20 (0.69) [11] | 50.34 (0.66) [8] | 49.59 (0.60) [15] | 50.43 (0.98) [7] | 49.16 (0.05) [16] |
| annthyroid | 86.31 (0.77) [12] | 92.54 (0.32) [5] | 88.61 (0.36) [10] | 84.65 (0.62) [14] | 92.21 (0.34) [6] | 88.48 (0.52) [11] | 84.50 (0.59) [15] | 85.80 (0.59) [13] | 79.38 (0.86) [18] | 90.45 (0.31) [8] | 91.68 (0.45) [7] | 89.48 (2.97) [9] | 84.27 (3.66) [16] | 76.25 (4.79) [19] | 93.75 (0.41) [2] | 97.26 (0.15) [1] | 92.62 (0.32) [4] | 93.16 (0.39) [3] | 74.93 (2.45) [20] | 82.88 (1.16) [17] |
| backdoor | 96.24 (0.36) [3] | 93.82 (0.61) [8] | 95.42 (0.37) [5] | 64.93 (0.84) [17] | 96.99 (0.42) [1] | 90.55 (0.82) [13] | 93.00 (0.80) [10] | 95.46 (0.54) [4] | 84.58 (1.41) [14] | 78.20 (3.17) [16] | 92.11 (0.56) [11] | 94.19 (0.61) [7] | 61.64 (22.25) [18] | 93.81 (0.74) [9] | 50.00 (0.00) [20] | 91.62 (1.43) [12] | 94.45 (0.57) [6] | 83.45 (6.86) [15] | 60.35 (0.75) [19] | 96.86 (0.39) [2] |
| breastw | 98.76 (0.20) [9] | 98.81 (0.24) [8] | 90.69 (3.95) [18] | 98.98 (0.15) [7] | 99.27 (0.24) [4] | 99.27 (0.26) [4] | 98.53 (0.37) [12] | 99.00 (0.20) [6] | 98.19 (0.40) [14] | 99.28 (0.07) [3] | 42.23 (2.62) [20] | 46.89 (30.89) [19] | 98.68 (0.25) [11] | 98.40 (0.59) [13] | 99.37 (0.15) [2] | 90.76 (5.10) [17] | 94.80 (0.63) [16] | 98.74 (0.26) [10] | 96.05 (1.10) [15] | 99.40 (0.17) [1] |
| campaign | 71.65 (0.14) [15] | 78.25 (0.27) [6] | 70.53 (0.15) [17] | 76.93 (0.23) [11] | 77.37 (0.24) [8] | 76.98 (0.18) [10] | 79.70 (0.31) [4] | 71.94 (0.29) [14] | 80.09 (0.19) [3] | 76.14 (0.82) [13] | 77.14 (0.26) [9] | 50.00 (0.00) [19] | 38.26 (8.33) [20] | 78.46 (0.27) [12] | 79.60 (0.30) [5] | 78.13 (0.25) [7] | 80.71 (0.19) [2] | 71.05 (0.26) [16] | 66.35 (1.09) [18] |
| Cardiotocography | 54.73 (1.12) [17] | 61.43 (0.75) [8] | 63.96 (0.93) [5] | 78.65 (0.67) [1] | 60.52 (0.87) [10] | 61.45 (0.65) [7] | 63.77 (0.92) [6] | 60.63 (0.69) [9] | 66.63 (0.93) [4] | 73.74 (3.32) [2] | 58.47 (0.76) [12] | 41.02 (11.61) [20] | 73.45 (0.95) [3] | 56.62 (2.28) [16] | 45.80 (4.89) [19] | 58.74 (1.15) [11] | 58.21 (0.60) [14] | 57.56 (1.02) [15] | 53.18 (0.61) [18] | 58.35 (1.26) [13] |
| cardio | 89.02 (0.61) [13] | 91.89 (0.95) [8] | 92.13 (0.76) [7] | 96.36 (0.59) [1] | 91.54 (1.02) [9] | 92.54 (0.90) [6] | 90.99 (0.87) [11] | 93.08 (1.05) [5] | 89.20 (0.98) [12] | 93.55 (1.75) [3] | 91.24 (0.56) [10] | 59.14 (19.23) [20] | 95.50 (0.67) [2] | 84.04 (2.21) [17] | 83.35 (1.23) [18] | 87.24 (0.89) [16] | 88.81 (0.88) [14] | 93.39 (0.92) [4] | 78.06 (0.76) [19] | 87.88 (0.96) [15] |
| celeba | 51.35 (6.55) [18] | 72.50 (1.01) [8] | 43.05 (0.42) [19] | 80.58 (0.87) [3] | 72.48 (1.15) [9] | 68.79 (1.23) [12] | 81.09 (0.86) [2] | 63.27 (1.33) [16] | 72.25 (0.97) [10] | 75.47 (1.14) [6] | 55.83 (0.91) [17] | 68.23 (1.61) [14] | 40.90 (14.56) [20] | 67.36 (2.03) [15] | 83.50 (0.64) [1] | 70.58 (1.13) [11] | 79.12 (2.39) [4] | 68.40 (2.34) [13] | 77.21 (2.59) [5] |
| census | 70.54 (0.90) [7] | 72.18 (0.36) [2] | 58.66 (0.90) [17] | 70.47 (0.29) [9] | 72.25 (0.28) [1] | 71.20 (0.47) [6] | 70.32 (0.30) [10] | 67.44 (0.20) [13] | 71.92 (0.31) [5] | 65.05 (1.75) [14] | 54.75 (0.28) [19] | 55.20 (3.32) [18] | 40.18 (5.79) [20] | 70.54 (0.49) [7] | 62.03 (9.84) [16] | 69.66 (0.87) [12] | 71.96 (0.39) [4] | 69.87 (0.57) [11] | 63.57 (0.60) [15] | 72.02 (0.50) [3] |
| cover | 95.86 (0.82) [10] | 97.59 (0.34) [8] | 99.26 (0.12) [1] | 97.84 (0.31) [7] | 95.60 (0.41) [12] | 94.93 (0.39) [13] | 98.23 (0.52) [3] | 98.75 (0.38) [2] | 88.18 (1.61) [17] | 98.23 (0.46) [3] | 95.74 (0.64) [11] | 15.56 (4.80) [20] | 89.85 (5.98) [16] | 74.99 (7.86) [18] | 97.88 (1.51) [6] | 97.90 (0.33) [5] | 91.66 (4.71) [15] | 72.37 (1.25) [19] | 97.17 (0.18) [9] |
| donors | 88.34 (0.52) [14] | 99.49 (0.08) [6] | 97.07 (0.16) [9] | 88.29 (0.38) [15] | 99.82 (0.06) [3] | 97.42 (0.14) [8] | 92.45 (0.19) [12] | 99.98 (0.02) [1] | 95.66 (1.10) [10] | 93.80 (0.35) [11] | 15.33 (0.87) [20] | 82.32 (16.17) [17] | 40.76 (23.43) [19] | 99.86 (0.08) [2] | 88.45 (4.69) [13] | 98.45 (0.14) [7] | 99.70 (0.06) [5] | 99.77 (0.14) [4] | 80.53 (1.60) [18] | 85.79 (0.62) [16] |
| fault | 56.19 (0.91) [18] | 58.39 (0.51) [9] | 46.59 (1.21) [20] | 55.88 (1.00) [19] | 58.61 (0.86) [8] | 58.27 (0.71) [10] | 60.90 (1.00) [3] | 58.18 (0.95) [11] | 62.14 (1.05) [2] | 57.23 (1.87) [14] | 56.25 (1.09) [17] | 57.46 (3.18) [13] | 59.10 (1.24) [7] | 60.65 (1.80) [4] | 63.30 (0.71) [1] | 60.36 (1.18) [5] | 58.00 (0.68) [12] | 59.31 (1.01) [6] | 56.48 (1.67) [16] | 57.01 (0.46) [15] |
| fraud | 95.13 (1.18) [11] | 96.04 (1.18) [6] | 94.13 (2.32) [17] | 96.14 (0.68) [4] | 96.36 (1.05) [2] | 96.10 (0.95) [5] | 96.45 (0.66) [1] | 96.20 (0.83) [3] | 95.80 (1.18) [8] | 95.76 (0.58) [9] | 95.34 (1.23) [10] | 50.00 (0.00) [20] | 95.96 (25.84) [19] | 95.11 (1.41) [12] | 94.96 (1.46) [13] | 94.70 (1.11) [14] | 95.96 (0.86) [7] | 92.59 (1.86) [18] | 94.62 (1.22) [15] |
| glass | 97.63 (1.21) [14] | 92.54 (1.42) [8] | 90.13 (2.05) [9] | 75.70 (2.94) [17] | 98.80 (0.63) [3] | 88.01 (1.47) [11] | 80.20 (1.90) [15] | 96.92 (0.74) [5] | 87.90 (1.73) [12] | 84.81 (1.95) [14] | 35.47 (1.55) [20] | 77.05 (13.02) [16] | 60.47 (16.31) [19] | 99.48 (0.30) [1] | 87.08 (4.83) [13] | 92.65 (1.15) [7] | 94.68 (0.95) [6] | 88.40 (4.24) [10] | 73.69 (4.76) [18] | 99.06 (0.53) [2] |
| Hepatitis | 99.91 (0.17) [3] | 96.69 (1.16) [10] | 68.67 (6.41) [18] | 84.57 (1.97) [15] | 99.89 (0.23) [5] | 99.89 (0.23) [5] | 94.68 (1.12) [13] | 99.92 (0.17) [2] | 94.77 (1.59) [12] | 95.42 (1.11) [11] | 21.02 (6.32) [20] | 43.80 (18.58) [19] | 82.86 (2.32) [16] | 99.07 (0.26) [7] | 99.93 (0.15) [1] | 99.08 (0.62) [9] | 99.38 (0.45) [6] | 94.20 (2.04) [14] | 75.65 (4.24) [17] | 99.91 (0.17) [3] |
| http | 99.83 (0.25) [14] | 100.00 (0.00) [1] | 99.98 (0.02) [7] | 99.94 (0.01) [9] | 100.00 (0.00) [1] | 100.00 (0.00) [1] | 99.94 (0.01) [9] | 100.00 (0.00) [1] | 100.00 (0.00) [1] | 99.45 (0.13) [18] | 92.88 (0.20) [19] | 50.00 (0.00) [20] | 99.67 (0.12) [16] | 99.81 (0.37) [15] | 99.92 (0.07) [11] | 99.48 (0.08) [17] | 99.96 (0.01) [8] | 99.86 (0.16) [13] | 100.00 (0.01) [1] | 99.90 (0.21) [12] |
| InternetAds | 83.08 (0.97) [1] | 67.67 (0.86) [13] | 71.38 (0.94) [11] | 64.90 (0.89) [17] | 76.05 (1.22) [7] | 75.38 (1.25) [9] | 80.11 (0.98) [3] | 78.11 (1.31) [4] | 65.88 (0.87) [15] | 62.88 (5.56) [19] | 76.25 (1.11) [6] | 51.70 (4.90) [20] | 65.05 (0.69) [16] | 72.06 (1.18) [10] | 75.56 (0.88) [8] | 77.84 (1.51) [6] | 67.69 (0.67) [12] | 66.98 (0.80) [14] | 64.24 (1.42) [18] | 81.19 (0.23) [2] |
| Ionosphere | 99.12 (0.07) [4] | 97.57 (0.73) [10] | 94.67 (1.65) [14] | 90.15 (1.56) [18] | 99.34 (0.20) [1] | 99.18 (0.42) [3] | 97.45 (0.77) [11] | 98.80 (0.40) [6] | 98.01 (0.61) [9] | 94.91 (1.12) [13] | 74.10 (4.79) [19] | 68.67 (15.34) [20] | 93.36 (2.26) [16] | 98.95 (0.37) [5] | 98.54 (0.28) [7] | 95.90 (0.78) [12] | 98.29 (0.53) [8] | 94.38 (1.83) [15] | 93.00 (1.05) [17] | 99.27 (0.23) [2] |
| landsat | 64.18 (0.48) [9] | 67.81 (0.64) [3] | 65.99 (0.41) [5] | 43.54 (0.46) [19] | 60.64 (0.81) [11] | 63.91 (0.76) [10] | 44.95 (0.61) [18] | 68.54 (0.58) [1] | 53.69 (0.67) [15] | 57.91 (1.18) [12] | 67.19 (0.49) [4] | 53.86 (2.43) [14] | 40.11 (1.12) [20] | 64.30 (0.66) [7] | 64.59 (0.64) [6] | 47.59 (0.70) [17] | 68.48 (0.58) [2] | 57.11 (0.83) [13] | 48.57 (0.38) [16] | 64.23 (0.48) [8] |
| letter | 42.40 (1.44) [5] | 35.61 (1.16) [15] | 45.60 (1.42) [3] | 29.97 (0.95) [20] | 36.79 (1.32) [13] | 37.04 (1.15) [12] | 35.57 (1.11) [16] | 37.55 (1.00) [11] | 35.26 (1.05) [17] | 31.64 (1.47) [18] | 39.13 (1.62) [7] | 48.97 (9.31) [2] | 30.54 (1.48) [19] | 42.53 (3.19) [4] | 38.68 (1.12) [8] | 38.13 (1.39) [9] | 55.72 (0.51) [1] | 37.61 (1.53) [10] | 36.53 (1.72) [14] | 39.36 (0.43) [6] |
| Lymphography | 99.95 (0.06) [9] | 99.90 (0.09) [12] | 98.56 (0.69) [17] | 99.90 (0.04) [12] | 100.00 (0.00) [1] | 100.00 (0.00) [1] | 99.98 (0.03) [8] | 100.00 (0.00) [1] | 99.89 (0.11) [14] | 99.93 (0.11) [10] | 18.21 (6.40) [20] | 41.37 (28.53) [19] | 99.93 (0.06) [10] | 100.00 (0.00) [1] | 99.99 (0.02) [7] | 99.14 (0.60) [16] | 98.30 (0.74) [18] | 100.00 (0.00) [1] | 99.39 (0.63) [15] | 100.00 (0.00) [1] |
| magic.gamma | 87.89 (0.07) [2] | 83.41 (0.16) [10] | 83.52 (0.16) [9] | 70.76 (0.32) [18] | 83.19 (0.13) [11] | 76.27 (0.30) [16] | 80.21 (0.19) [12] | 85.76 (0.26) [4] | 84.30 (0.21) [7] | 77.53 (1.26) [14] | 84.45 (0.12) [6] | 78.63 (0.77) [13] | 68.77 (1.80) [20] | 77.40 (0.93) [15] | 72.13 (0.13) [17] | 86.56 (0.17) [3] | 83.98 (0.15) [8] | 88.77 (0.10) [1] | 70.12 (0.71) [19] | 85.67 (0.22) [5] |
| mammography | 80.61 (1.19) [15] | 87.05 (0.93) [9] | 85.09 (0.67) [11] | 89.59 (0.71) [1] | 88.50 (0.86) [2] | 87.41 (0.82) [6] | 87.66 (0.70) [4] | 87.61 (0.80) [5] | 85.85 (1.00) [10] | 87.94 (0.70) [3] | 58.88 (1.23) [20] | 81.25 (1.79) [13] | 67.79 (10.10) [19] | 76.19 (4.10) [16] | 71.68 (1.29) [18] | 87.10 (1.05) [8] | 87.16 (0.90) [7] | 80.80 (0.66) [14] | 72.68 (1.97) [17] | 84.19 (0.41) [12] |
| mnist | 92.36 (0.43) [10] | 93.86 (0.23) [4] | 92.89 (0.18) [8] | 90.26 (0.44) [14] | 94.12 (0.28) [3] | 94.45 (0.24) [1] | 92.11 (0.35) [11] | 93.18 (0.37) [7] | 87.94 (2.09) [17] | 93.80 (0.37) [6] | 84.57 (2.01) [19] | 90.94 (0.74) [12] | 90.20 (0.89) [15] | 59.85 (4.44) [16] | 85.27 (0.45) [18] | 94.22 (0.24) [2] | 92.86 (0.44) [9] | 80.82 (0.73) [20] | 90.29 (0.75) [13] |
| musk | 100.00 (0.00) [1] | 100.00 (0.00) [1] | 100.00 (0.00) [1] | 100.00 (0.00) [1] | 100.00 (0.00) [1] | 100.00 (0.00) [1] | 99.86 (0.05) [16] | 100.00 (0.00) [1] | 100.00 (0.00) [1] | 96.40 (3.09) [18] | 99.96 (0.02) [14] | 42.68 (33.82) [20] | 100.00 (0.00) [1] | 99.32 (0.27) [17] | 100.00 (0.00) [1] | 100.00 (0.00) [1] | 94.16 (0.24) [19] | 100.00 (0.00) [1] | 99.96 (0.03) [14] | 100.00 (0.00) [1] |
| optdigits | 97.39 (0.25) [2] | 93.63 (0.59) [9] | 96.40 (0.11) [4] | 57.99 (0.42) [20] | 91.18 (0.61) [11] | 96.09 (0.34) [6] | 81.18 (0.87) [16] | 99.56 (0.15) [1] | 87.61 (0.56) [13] | 79.14 (3.97) [17] | 91.61 (0.22) [10] | 84.75 (3.22) [15] | 67.84 (2.57) [18] | 96.34 (0.96) [5] | 94.97 (0.33) [7] | 86.28 (0.40) [14] | 94.96 (0.42) [8] | 89.85 (0.67) [12] | 61.44 (3.19) [19] | 96.98 (0.40) [3] |
| PageBlocks | 83.92 (0.97) [19] | 89.89 (0.29) [5] | 91.42 (0.24) [2] | 86.27 (0.33) [15] | 89.32 (0.28) [8] | 89.53 (0.29) [7] | 88.62 (0.35) [11] | 88.94 (0.57) [9] | 90.41 (0.25) [3] | 82.82 (0.94) [20] | 88.60 (0.32) [12] | 91.57 (2.40) [1] | 88.77 (1.14) [10] | 87.84 (0.36) [13] | 84.64 (0.46) [17] | 89.88 (0.44) [6] | 87.83 (0.16) [14] | 89.99 (0.36) [4] | 84.10 (0.85) [18] | 85.90 (0.31) [16] |
| pendigits | 99.83 (0.04) [4] | 99.84 (0.03) [3] | 98.99 (0.14) [9] | 93.95 (0.34) [16] | 99.65 (0.04) [7] | 99.82 (0.03) [5] | 84.77 (0.84) [18] | 99.94 (0.01) [1] | 98.28 (0.16) [10] | 97.06 (0.60) [13] | 99.46 (0.04) [8] | 76.69 (13.17) [20] | 86.65 (4.04) [17] | 96.61 (1.75) [14] | 94.42 (0.39) [15] | 97.43 (0.13) [12] | 99.88 (0.03) [2] | 98.10 (0.13) [11] | 82.83 (0.86) [19] | 99.82 (0.04) [5] |
| Pima | 78.44 (1.90) [7] | 78.10 (2.06) [8] | 70.41 (2.39) [12] | 73.75 (1.30) [11] | 86.85 (1.35) [1] | 79.90 (1.71) [6] | 74.08 (1.62) [10] | 80.35 (3.26) [5] | 70.07 (2.25) [14] | 76.70 (1.23) [9] | 69.18 (1.60) [16] | 42.40 (16.03) [20] | 68.28 (10.30) [17] | 80.68 (3.13) [3] | 61.03 (5.20) [18] | 70.13 (3.91) [13] | 80.47 (2.04) [4] | 69.82 (2.65) [15] | 58.08 (2.46) [19] | 82.94 (1.57) [2] |
| satellite | 80.61 (0.66) [9] | 82.21 (0.60) [5] | 80.30 (0.52) [10] | 66.54 (0.79) [20] | 78.98 (0.70) [11] | 81.71 (0.61) [7] | 78.44 (0.74) [12] | 82.31 (0.55) [4] | 78.23 (0.65) [13] | 77.27 (1.03) [16] | 82.00 (0.62) [6] | 73.38 (3.91) [17] | 69.60 (1.20) [19] | 84.33 (0.78) [2] | 87.50 (0.57) [1] | 78.04 (0.84) [14] | 82.32 (0.61) [3] | 70.95 (0.70) [18] | 77.45 (1.22) [15] | 81.12 (0.36) [8] |
| satimage-2 | 99.31 (0.37) [13] | 99.77 (0.15) [3] | 99.49 (0.27) [11] | 97.91 (0.52) [19] | 99.77 (0.12) [3] | 99.72 (0.16) [5] | 99.69 (0.03) [6] | 99.47 (0.28) [12] | 99.90 (0.02) [1] | 99.50 (0.28) [9] | 99.21 (0.20) [14] | 98.90 (1.08) [16] | 98.92 (0.42) [15] | 99.61 (0.32) [7] | 95.19 (2.36) [20] | 99.55 (0.14) [8] | 96.51 (0.65) [20] | 98.39 (0.87) [17] | 98.07 (0.50) [18] |
| skin | 97.79 (0.23) [6] | 99.51 (0.07) [2] | 86.37 (1.68) [14] | 59.47 (0.22) [18] | 99.33 (0.07) [3] | 88.87 (0.19) [11] | 88.76 (0.29) [12] | 98.89 (0.19) [4] | 83.94 (1.56) [16] | 90.38 (0.58) [9] | 37.31 (0.64) [19] | 89.29 (1.11) [10] | 88.38 (3.53) [17] | 5.85 (0.93) [20] | 90.97 (2.41) [8] | 91.85 (0.34) [7] | 99.63 (0.86) [1] | 88.21 (6.30) [13] | 84.29 (0.52) [15] | 98.07 (0.50) [5] |
| smtp | 96.32 (1.68) [1] | 93.51 (2.77) [6] | 95.48 (2.18) [3] | 85.52 (6.46) [16] | 91.00 (3.32) [11] | 88.95 (3.85) [13] | 82.59 (5.96) [18] | 94.25 (2.42) [5] | 87.63 (5.45) [14] | 90.34 (2.70) [12] | 92.15 (2.61) [9] | 56.80 (13.61) [20] | 83.15 (12.60) [17] | 60.32 (5.83) [19] | 95.19 (2.36) [4] | 95.68 (1.62) [2] | 92.70 (4.30) [8] | 87.30 (6.50) [15] | 91.91 (3.91) [10] | 93.44 (1.84) [7] |
| SpamBase | 84.46 (0.30) [3] | 83.19 (0.35) [8] | 73.65 (0.73) [20] | 81.26 (0.39) [16] | 84.41 (0.41) [4] | 84.00 (0.44) [5] | 80.56 (0.40) [17] | 81.52 (1.26) [13] | 82.15 (0.36) [12] | 86.81 (1.01) [1] | 83.08 (0.27) [10] | 77.29 (1.85) [19] | 81.46 (0.68) [14] | 83.35 (0.52) [6] | 84.71 (0.37) [2] | 83.09 (0.26) [9] | 82.97 (0.40) [11] | 81.42 (0.36) [15] | 78.65 (0.63) [18] | 83.27 (0.10) [7] |
| speech | 45.44 (2.51) [9] | 38.00 (2.30) [19] | 39.16 (2.26) [14] | 38.02 (2.12) [18] | 42.44 (2.03) [11] | 51.86 (3.72) [1] | 39.71 (2.12) [13] | 46.90 (4.90) [8] | 38.37 (2.08) [17] | 38.56 (5.19) [16] | 48.35 (3.64) [5] | 50.58 (5.16) [3] | 38.60 (2.78) [15] | 47.98 (4.30) [6] | 40.68 (1.87) [12] | 46.04 (6.23) [17] | 92.64 (1.96) [15] | 50.04 (0.02) [4] | 37.31 (2.12) [20] | 47.86 (2.86) [7] | 51.40 (0.73) [2] |
| Stamps | 98.36 (0.64) [14] | 96.49 (1.26) [7] | 94.58 (1.83) [12] | 94.04 (1.14) [13] | 99.25 (0.62) [1] | 98.98 (0.55) [3] | 93.73 (1.78) [14] | 98.10 (0.57) [5] | 95.07 (1.32) [10] | 94.61 (1.10) [11] | 69.10 (2.95) [19] | 42.30 (23.50) [20] | 87.94 (9.73) [16] | 95.94 (1.75) [8] | 80.44 (6.23) [17] | 92.64 (1.96) [15] | 97.82 (0.59) [6] | 95.16 (1.71) [9] | 77.73 (3.42) [18] | 98.99 (0.53) [2] |
| thyroid | 95.11 (0.69) [17] | 98.87 (0.13) [3] | 93.44 (0.92) [19] | 98.70 (0.19) [6] | 98.85 (0.13) [4] | 98.76 (0.13) [5] | 98.37 (0.29) [9] | 98.53 (0.21) [7] | 96.23 (0.36) [13] | 98.94 (0.21) [2] | 97.03 (0.28) [12] | 95.32 (1.07) [15] | 95.28 (2.01) [16] | 94.51 (2.43) [18] | 96.02 (0.93) [14] | 99.02 (0.14) [1] | 98.37 (0.17) [9] | 98.42 (0.30) [8] | 92.49 (2.92) [20] | 97.93 (0.26) [11] |
| vertebral | 74.14 (1.95) [4] | 55.99 (3.84) [16] | 60.86 (1.30) [8] | 40.95 (2.79) [18] | 93.47 (2.12) [2] | 39.61 (2.23) [19] | 46.73 (2.78) [16] | 77.13 (4.55) [3] | 51.64 (5.35) [14] | 54.60 (2.25) [11] | 46.76 (2.85) [15] | 24.27 (2.66) [20] | 43.64 (20.12) [17] | 49.94 (8.21) [12] | 73.80 (8.70) [5] | 47.27 (5.72) [14] | 62.92 (2.64) [7] | 69.64 (3.39) [6] | 58.33 (5.82) [9] | 93.97 (1.65) [1] |
| vowels | 88.66 (2.00) [1] | 82.88 (3.18) [8] | 86.64 (2.88) [3] | 54.28 (3.75) [19] | 80.35 (3.29) [14] | 79.68 (3.59) [15] | 82.41 (3.23) [10] | 83.96 (3.70) [6] | 82.10 (3.47) [11] | 65.53 (3.40) [17] | 82.44 (2.85) [9] | 45.16 (13.65) [20] | 65.11 (5.70) [18] | 85.49 (2.06) [5] | 83.94 (2.79) [7] | 88.06 (2.79) [2] | 81.44 (3.59) [12] | 80.62 (2.02) [13] | 71.75 (4.77) [16] | 86.07 (1.34) [4] |
| Waveform | 62.86 (1.68) [17] | 74.52 (1.25) [7] | 75.00 (1.57) [4] | 66.40 (1.98) [13] | 77.24 (0.84) [2] | 76.23 (1.17) [3] | 59.12 (1.77) [18] | 73.74 (1.76) [9] | 67.82 (1.36) [12] | 74.97 (2.70) [5] | 74.76 (2.11) [6] | 65.29 (4.58) [14] | 63.96 (4.37) [16] | 68.54 (5.57) [11] | 47.20 (1.23) [20] | 64.47 (1.17) [15] | 73.81 (1.47) [8] | 83.97 (0.43) [1] | 53.04 (1.67) [19] | 68.77 (1.01) [10] |
| WBC | 99.48 (0.77) [8] | 99.00 (0.47) [11] | 79.02 (3.71) [18] | 99.38 (0.23) [9] | 99.88 (0.25) [1] | 99.87 (0.19) [2] | 98.48 (0.70) [13] | 99.69 (0.28) [6] | 98.05 (0.53) [14] | 99.80 (0.13) [4] | 40.23 (3.10) [20] | 42.10 (28.27) [19] | 99.15 (0.30) [10] | 99.84 (0.20) [3] | 99.69 (0.30) [6] | 86.13 (2.21) [17] | 97.41 (1.34) [15] | 98.86 (0.69) [12] | 93.32 (1.92) [16] | 99.80 (0.27) [4] |
| WDBC | 99.99 (0.01) [1] | 99.52 (0.25) [13] | 99.84 (0.11) [7] | 99.90 (0.06) [2] | 99.88 (0.08) [3] | 99.85 (0.08) [6] | 99.72 (0.17) [9] | 99.74 (0.17) [8] | 75.58 (5.52) [19] | 34.84 (34.69) [20] | 99.55 (0.20) [12] | 99.87 (0.11) [4] | 98.78 (0.24) [15] | 98.11 (0.62) [17] | 98.76 (0.75) [16] | 90.00 (4.24) [18] | 99.70 (0.16) [10] |
| Wilt | 74.15 (1.36) [8] | 62.95 (1.09) [11] | 67.45 (1.30) [9] | 26.40 (1.86) [20] | 74.93 (0.87) [7] | 37.84 (1.35) [19] | 84.74 (0.49) [3] | 60.30 (3.09) [13] | 55.54 (0.94) [14] | 44.76 (3.03) [18] | 75.84 (0.95) [5] | 51.56 (10.21) [16] | 51.17 (3.01) [17] | 75.32 (4.60) [6] | 62.05 (1.10) [12] | 85.15 (0.65) [2] | 65.96 (1.15) [10] | 87.00 (0.82) [1] | 77.10 (0.74) [4] |
| wine | 99.95 (0.10) [4] | 99.23 (0.30) [12] | 98.44 (0.61) [14] | 93.85 (2.00) [17] | 99.97 (0.05) [3] | 99.35 (0.50) [11] | 99.94 (0.11) [5] | 99.52 (0.26) [9] | 96.98 (2.05) [15] | 22.36 (11.44) [20] | 48.16 (26.30) [19] | 94.33 (2.10) [16] | 99.92 (0.14) [6] | 100.00 (0.00) [1] | 99.86 (0.23) [7] | 99.43 (0.20) [10] | 99.14 (0.57) [13] | 86.52 (1.80) [18] | 99.85 (0.31) [8] |
| WPBC | 96.74 (1.90) [1] | 64.94 (2.47) [11] | 58.79 (1.99) [14] | 54.11 (4.18) [17] | 96.37 (2.19) [2] | 96.34 (2.02) [4] | 61.73 (2.51) [13] | 93.65 (1.96) [7] | 57.19 (2.76) [15] | 63.61 (3.01) [12] | 19.52 (4.48) [20] | 46.07 (4.33) [19] | 54.34 (6.32) [16] | 96.05 (1.62) [5] | 95.30 (2.67) [6] | 70.95 (2.58) [10] | 82.68 (0.35) [8] | 71.58 (1.98) [9] | 51.88 (1.86) [18] | 96.35 (1.38) [3] |
| yeast | 47.98 (1.07) [7] | 44.16 (0.84) [12] | 45.30 (0.74) [9] | 42.68 (1.02) [18] | 43.40 (0.68) [15] | 42.15 (0.86) [19] | 43.65 (0.61) [14] | 43.36 (0.43) [16] | 45.06 (0.39) [10] | 42.05 (0.59) [20] | 44.59 (0.70) [11] | 51.74 (4.02) [1] | 49.97 (4.80) [2] | 49.36 (2.86) [4] | 49.38 (1.22) [3] | 48.17 (0.41) [6] | 44.07 (0.62) [13] | 43.00 (0.84) [17] | 48.52 (1.31) [5] | 47.47 (0.41) [8] |
| CIFAR10 0 | 76.33 (0.81) [2] | 74.85 (0.86) [8] | 75.41 (0.91) [6] | 73.40 (0.96) [12] | 75.28 (0.89) [7] | 66.28 (1.79) [19] | 75.49 (0.78) [5] | 76.63 (0.94) [1] | 75.97 (2.09) [3] | 71.50 (1.40) [15] | 75.93 (0.63) [3] | 49.87 (5.39) [20] | 73.83 (0.86) [11] | 69.35 (1.39) [18] | 72.31 (1.13) [14] | 71.20 (1.00) [16] | 73.16 (0.83) [13] | 73.96 (0.89) [10] | 69.57 (1.92) [17] | 75.75 (0.17) [4] |
| CIFAR10 1 | 67.52 (0.95) [5] | 63.86 (1.01) [14] | 72.04 (0.81) [1] | 62.28 (1.48) [16] | 65.68 (1.12) [9] | 59.92 (1.18) [18] | 69.53 (1.03) [2] | 64.27 (1.21) [12] | 66.98 (1.26) [6] | 54.47 (1.16) [19] | 66.34 (1.16) [8] | 46.58 (4.39) [20] | 63.90 (1.74) [13] | 65.63 (1.57) [10] | 65.14 (0.73) [11] | 68.40 (0.77) [3] | 63.02 (0.92) [15] | 66.58 (1.25) [7] | 60.76 (1.76) [17] | 67.60 (0.37) [4] |
| CIFAR10 2 | 63.47 (0.79) [7] | 62.34 (0.62) [10] | 64.43 (0.89) [2] | 62.08 (0.55) [14] | 63.32 (0.65) [8] | 54.02 (0.74) [19] | 63.61 (0.62) [5] | 64.33 (0.46) [3] | 63.61 (0.66) [5] | 60.19 (0.41) [16] | 64.44 (0.71) [1] | 49.89 (3.02) [20] | 62.21 (1.15) [13] | 59.79 (1.75) [17] | 60.58 (1.16) [15] | 62.30 (0.96) [12] | 62.31 (0.45) [11] | 63.20 (0.64) [9] | 57.97 (1.50) [18] | 63.84 (0.13) [4] |
| CIFAR10 3 | 58.85 (1.84) [8] | 57.38 (2.17) [13] | 60.81 (1.94) [2] | 59.20 (2.34) [7] | 58.81 (2.10) [9] | 54.11 (1.66) [19] | 61.00 (1.97) [1] | 58.36 (2.10) [11] | 59.96 (1.98) [3] | 56.14 (1.80) [16] | 59.78 (1.97) [4] | 50.50 (2.34) [20] | 58.78 (2.07) [10] | 55.00 (1.36) [18] | 55.47 (1.75) [17] | 59.51 (1.75) [6] | 56.61 (1.83) [14] | 59.69 (2.03) [5] | 56.31 (1.37) [15] | 57.74 (0.25) [12] |
| CIFAR10 4 | 78.47 (1.04) [4] | 77.37 (1.19) [8] | 77.46 (1.20) [7] | 76.81 (1.35) [12] | 77.80 (1.21) [5] | 68.89 (1.33) [19] | 78.73 (1.14) [3] | 79.25 (1.29) [2] | 77.48 (1.23) [6] | 76.20 (1.26) [13] | 77.24 (0.75) [9] | 51.26 (3.51) [20] | 77.00 (1.51) [11] | 70.84 (1.46) [18] | 75.73 (1.06) [14] | 73.76 (1.09) [16] | 73.93 (1.08) [15] | 77.07 (1.20) [10] | 73.21 (2.30) [17] | 80.09 (0.14) [1] |
| CIFAR10 5 | 58.59 (0.15) [6] | 57.33 (0.40) [10] | 60.19 (0.43) [3] | 58.69 (0.70) [5] | 58.37 (0.45) [8] | 55.89 (1.14) [15] | 60.40 (0.30) [2] | 55.24 (0.46) [17] | 59.13 (0.48) [4] | 56.36 (0.60) [14] | 48.22 (2.27) [20] | 58.39 (1.30) [7] | 52.99 (2.86) [19] | 57.07 (0.74) [11] | 61.14 (0.48) [1] | 56.41 (0.71) [13] | 58.31 (0.51) [9] | 55.81 (2.34) [16] | 56.62 (0.39) [12] |
| CIFAR10 6 | 76.15 (0.98) [3] | 73.90 (0.94) [11] | 75.58 (1.41) [7] | 71.80 (0.86) [15] | 74.42 (0.95) [9] | 64.41 (1.63) [19] | 75.81 (1.00) [5] | 76.39 (0.83) [1] | 74.04 (1.03) [10] | 71.10 (1.08) [16] | 75.67 (0.95) [6] | 48.17 (3.48) [20] | 72.39 (0.97) [14] | 69.76 (1.79) [17] | 75.13 (1.33) [8] | 76.04 (1.35) [4] | 73.17 (0.87) [13] | 73.82 (1.08) [12] | 67.65 (1.40) [18] | 76.28 (0.15) [2] |
| CIFAR10 7 | 67.99 (0.52) [3] | 65.60 (0.64) [13] | 69.56 (0.74) [1] | 67.06 (0.54) [6] | 67.04 (0.69) [7] | 59.15 (0.86) [19] | 69.30 (0.60) [2] | 66.10 (0.31) [12] | 67.21 (0.63) [4] | 63.64 (1.16) [15] | 67.13 (0.71) [5] | 50.81 (2.68) [20] | 66.95 (0.71) [10] | 59.72 (1.57) [18] | 62.27 (1.43) [17] | 66.16 (0.77) [11] | 63.88 (0.54) [14] | 67.03 (0.61) [8] | 62.60 (1.34) [16] | 66.98 (0.28) [9] |
| CIFAR10 8 | 74.80 (0.49) [3] | 73.30 (0.56) [10] | 74.82 (0.34) [2] | 72.80 (0.66) [12] | 74.01 (0.59) [8] | 61.35 (1.56) [19] | 74.97 (0.50) [1] | 74.58 (0.53) [4] | 74.17 (0.44) [6] | 69.91 (1.45) [16] | 74.11 (0.52) [7] | 52.80 (3.23) [20] | 72.72 (0.60) [13] | 67.72 (2.08) [18] | 73.14 (0.57) [11] | 72.36 (0.59) [14] | 74.00 (0.54) [9] | 74.20 (0.09) [5] |
| CIFAR10 9 | 73.89 (0.98) [3] | 71.76 (0.87) [12] | 75.34 (1.09) [1] | 71.80 (0.72) [11] | 72.85 (0.88) [8] | 65.51 (1.15) [19] | 74.85 (0.99) [2] | 72.54 (0.86) [9] | 73.53 (0.96) [5] | 67.41 (2.17) [17] | 72.35 (0.99) [10] | 49.67 (5.59) [20] | 71.62 (0.80) [13] | 66.08 (1.34) [18] | 71.53 (1.30) [14] | 69.24 (0.97) [15] | 73.19 (0.88) [6] | 68.51 (0.88) [16] | 73.19 (0.17) [6] |
| FashionMNIST 0 | 91.69 (0.71) [3] | 90.16 (0.74) [11] | 91.47 (0.87) [5] | 91.67 (0.63) [5] | 90.43 (0.74) [8] | 75.88 (1.59) [19] | 91.53 (0.72) [4] | 92.30 (0.43) [1] | 90.24 (0.82) [10] | 90.42 (0.73) [9] | 49.79 (4.60) [20] | 88.27 (1.03) [14] | 90.87 (0.86) [6] | 90.50 (0.65) [7] | 90.01 (0.86) [12] | 86.97 (0.51) [16] | 89.60 (0.83) [13] | 86.35 (0.66) [17] | 91.98 (0.06) [2] |
| FashionMNIST 1 | 98.60 (0.18) [3] | 98.04 (0.23) [10] | 98.44 (0.22) [5] | 97.19 (0.24) [15] | 98.12 (0.21) [9] | 92.63 (0.73) [19] | 98.50 (0.22) [4] | 98.84 (0.14) [1] | 97.68 (0.25) [11] | 96.16 (0.24) [17] | 98.27 (0.24) [7] | 56.35 (16.92) [20] | 97.41 (0.28) [14] | 98.38 (0.15) [6] | 98.20 (0.20) [8] | 97.58 (0.29) [13] | 93.82 (0.39) [18] | 97.78 (0.23) [12] | 96.86 (0.47) [16] | 98.77 (0.01) [2] |
| FashionMNIST 2 | 88.98 (0.49) [3] | 86.77 (0.51) [9] | 88.61 (0.49) [4] | 83.21 (0.58) [16] | 87.09 (0.56) [7] | 76.48 (0.47) [19] | 89.15 (0.42) [2] | 87.67 (0.83) [6] | 86.05 (0.57) [12] | 78.63 (0.75) [18] | 86.94 (0.52) [8] | 53.35 (14.84) [20] | 83.89 (0.71) [15] | 88.46 (0.69) [5] | 86.63 (0.55) [11] | 86.14 (0.57) [13] | 84.57 (0.51) [14] | 86.28 (0.57) [10] | 80.94 (1.13) [17] | 89.70 (0.05) [1] |
| FashionMNIST 3 | 95.40 (0.26) [2] | 91.22 (0.43) [10] | 92.81 (0.21) [5] | 89.15 (0.57) [15] | 91.77 (0.41) [7] | 82.57 (0.93) [19] | 93.17 (0.30) [3] | 93.11 (0.38) [4] | 91.07 (0.53) [11] | 87.57 (1.19) [17] | 92.47 (0.45) [6] | 52.37 (5.29) [20] | 89.45 (0.51) [14] | 91.42 (0.65) [9] | 90.81 (0.59) [13] | 91.75 (0.32) [8] | 88.20 (0.44) [16] | 90.83 (0.55) [12] | 86.33 (1.00) [18] | 94.39 (0.07) [1] |
| FashionMNIST 4 | 88.61 (0.57) [3] | 85.78 (0.62) [10] | 87.90 (0.57) [5] | 83.23 (0.53) [16] | 86.36 (0.57) [9] | 78.51 (0.86) [19] | 88.70 (0.41) [2] | 87.61 (0.64) [6] | 85.38 (0.66) [11] | 79.00 (2.04) [18] | 86.74 (0.77) [8] | 50.77 (13.02) [20] | 83.75 (0.47) [14] | 87.10 (0.87) [6] | 84.56 (0.81) [13] | 88.79 (0.37) [1] | 83.27 (0.55) [15] | 85.22 (0.60) [12] | 81.99 (0.63) [17] | 90.07 (0.14) [1] |
| FashionMNIST 5 | 95.72 (0.60) [1] | 95.03 (0.65) [6] | 95.21 (0.67) [5] | 94.69 (0.66) [12] | 94.94 (0.61) [7] | 93.76 (0.62) [18] | 95.58 (0.55) [3] | 95.44 (0.66) [4] | 94.69 (0.65) [12] | 93.96 (0.67) [17] | 94.94 (0.72) [7] | 58.13 (21.77) [20] | 94.73 (0.67) [11] | 94.30 (0.73) [14] | 94.80 (0.61) [9] | 94.12 (0.63) [16] | 89.12 (0.66) [19] | 94.80 (0.63) [9] | 94.19 (0.69) [15] | 95.72 (0.06) [1] |
| FashionMNIST 6 | 80.99 (0.79) [4] | 77.61 (0.73) [11] | 80.32 (1.21) [6] | 72.87 (0.75) [16] | 78.34 (0.74) [9] | 72.28 (1.42) [17] | 80.50 (0.71) [5] | 81.45 (0.57) [2] | 77.30 (1.18) [11] | 68.05 (1.39) [19] | 78.44 (0.77) [8] | 40.64 (9.95) [20] | 73.89 (0.65) [15] | 81.08 (1.47) [3] | 77.68 (0.91) [10] | 79.99 (0.67) [7] | 76.14 (0.58) [14] | 76.74 (0.82) [13] | 70.54 (1.90) [18] | 83.22 (0.09) [1] |
| FashionMNIST 7 | 97.43 (0.40) [2] | 96.66 (0.40) [9] | 96.88 (0.30) [6] | 96.31 (0.49) [13] | 96.49 (0.41) [10] | 95.42 (0.54) [17] | 97.41 (0.37) [3] | 97.31 (0.43) [4] | 96.80 (0.39) [7] | 96.02 (0.79) [14] | 97.05 (0.42) [5] | 51.44 (22.31) [20] | 96.39 (0.61) [11] | 96.36 (0.53) [12] | 97.32 (0.53) [16] | 91.64 (0.51) [19] | 96.71 (0.41) [8] | 94.88 (0.43) [18] | 97.69 (0.09) [1] |
| FashionMNIST 8 | 88.11 (0.48) [3] | 81.92 (0.69) [12] | 87.83 (0.56) [5] | 76.16 (0.83) [16] | 83.22 (0.65) [10] | 69.68 (0.90) [19] | 87.92 (0.44) [4] | 82.12 (0.65) [11] | 73.60 (0.59) [17] | 84.65 (0.71) [8] | 43.14 (5.86) [20] | 77.33 (0.94) [15] | 84.54 (0.52) [9] | 86.81 (0.52) [6] | 85.32 (0.67) [7] | 81.61 (0.69) [13] | 81.30 (0.70) [14] | 72.70 (1.49) [18] | 91.39 (0.24) [1] |
| FashionMNIST 9 | 96.75 (0.41) [3] | 96.11 (0.51) [10] | 96.73 (0.37) [4] | 96.26 (0.47) [8] | 93.19 (0.95) [18] | 96.78 (0.42) [2] | 96.60 (0.59) [5] | 78.26 (0.49) [6] | 94.76 (0.76) [17] | 95.87 (0.58) [13] | 53.25 (18.39) [20] | 96.09 (0.56) [11] | 95.10 (0.72) [16] | 96.37 (0.46) [6] | 96.04 (0.57) [19] | 96.27 (0.47) [7] | 95.45 (0.58) [14] | 96.93 (0.03) [1] |
| MNIST-C brightness | 93.15 (0.27) [2] | 82.59 (0.42) [12] | 78.74 (0.35) [8] | 72.37 (0.58) [16] | 83.93 (0.39) [9] | 70.13 (1.37) [17] | 89.11 (0.29) [5] | 91.60 (0.37) [3] | 80.93 (0.36) [13] | 69.86 (1.88) [18] | 87.78 (0.33) [7] | 55.81 (5.41) [20] | 73.77 (1.06) [15] | 88.00 (0.46) [6] | 83.94 (0.28) [11] | 83.34 (0.39) [10] | 80.38 (0.38) [14] | 68.98 (1.30) [19] | 97.05 (0.10) [1] |
| MNIST-C canny edges | 94.00 (0.21) [3] | 84.33 (0.53) [12] | 92.45 (0.25) [4] | 74.10 (0.88) [16] | 85.68 (0.49) [10] | 70.13 (1.40) [19] | 90.15 (0.30) [5] | 84.27 (0.52) [13] | 72.62 (4.99) [17] | 89.44 (0.27) [7] | 57.91 (11.12) [20] | 76.47 (1.22) [15] | 88.79 (0.70) [8] | 85.30 (0.48) [11] | 75.56 (0.95) [14] | 85.30 (0.41) [9] | 70.15 (1.13) [18] | 97.19 (0.11) [1] |
| MNIST-C dotted line | 94.30 (0.17) [3] | 87.66 (0.39) [10] | 91.19 (0.27) [4] | 79.64 (0.77) [16] | 88.26 (0.40) [8] | 74.09 (0.60) [19] | 90.92 (0.30) [5] | 95.19 (0.19) [2] | 86.07 (0.49) [12] | 78.43 (1.75) [17] | 90.86 (0.27) [6] | 59.96 (4.48) [20] | 81.23 (0.86) [15] | 88.65 (0.76) [7] | 85.17 (0.15) [13] | 87.58 (0.52) [11] | 87.93 (0.41) [9] | 85.19 (0.47) [14] | 75.96 (0.85) [18] | 97.15 (0.07) [1] |
| MNIST-C fog | 98.66 (0.12) [3] | 95.24 (0.28) [11] | 97.78 (0.15) [9] | 99.77 (0.52) [16] | 95.57 (0.27) [10] | 83.44 (0.54) [19] | 97.99 (0.16) [4] | 96.28 (0.48) [8] | 98.11 (0.25) [18] | 97.47 (0.15) [6] | 44.91 (10.25) [20] | 90.71 (0.64) [15] | 96.91 (0.19) [8] | 96.73 (0.23) [5] | 97.10 (0.27) [17] | 93.51 (0.24) [14] | 94.56 (0.32) [13] | 87.70 (0.82) [17] | 99.54 (0.01) [1] |
| MNIST-C glass blur | 98.98 (0.06) [3] | 97.85 (0.08) [9] | 98.18 (0.06) [6] | 96.37 (0.10) [15] | 98.09 (0.08) [7] | 88.83 (0.78) [19] | 98.73 (0.10) [4] | 99.51 (0.06) [1] | 97.56 (0.08) [11] | 94.96 (0.95) [17] | 98.55 (0.09) [5] | 56.91 (16.07) [20] | 96.61 (0.16) [14] | 97.80 (0.36) [10] | 97.46 (0.08) [12] | 98.02 (0.14) [8] | 94.60 (0.38) [18] | 97.22 (0.08) [13] | 95.07 (0.44) [16] | 99.44 (0.01) [2] |
| MNIST-C identity | 45.01 (0.34) [17] | 48.35 (0.69) [13] | 48.60 (0.82) [11] | 48.96 (0.63) [7] | 44.92 (0.45) [18] | 44.44 (0.75) [19] | 48.40 (0.81) [12] | 48.23 (1.36) [14] | 48.98 (0.55) [6] | 48.67 (0.41) [10] | 49.91 (1.44) [1] | 49.00 (0.64) [4] | 47.08 (0.79) [16] | 49.00 (0.30) [4] | 48.82 (0.84) [9] | 47.33 (0.62) [15] | 49.22 (0.60) [2] | 49.15 (0.89) [3] | 43.95 (0.10) [20] |
| MNIST-C impulse noise | 99.98 (0.00) [2] | 99.96 (0.01) [13] | 99.94 (0.02) [15] | 99.80 (0.02) [16] | 99.97 (0.01) [6] | 99.97 (0.01) [6] | 99.97 (0.04) [6] | 99.98 (0.01) [2] | 99.63 (0.14) [17] | 99.98 (0.01) [2] | 67.53 (35.11) [20] | 99.96 (0.01) [13] | 99.99 (0.00) [1] | 95.13 (0.09) [19] | 99.96 (0.02) [13] | 99.98 (0.00) [2] | 99.97 (0.01) [6] | 99.98 (0.01) [2] |
| MNIST-C motion blur | 97.20 (0.14) [2] | 92.76 (0.34) [10] | 94.56 (0.24) [7] | 87.56 (0.52) [16] | 93.48 (0.28) [9] | 84.02 (1.21) [19] | 95.05 (0.20) [4] | 97.11 (0.18) [3] | 92.07 (0.30) [11] | 84.13 (1.32) [18] | 95.00 (0.21) [3] | 53.94 (13.37) [20] | 88.50 (0.42) [15] | 94.71 (0.39) [6] | 91.62 (0.36) [12] | 94.43 (0.31) [8] | 91.34 (0.26) [14] | 91.61 (0.35) [13] | 85.45 (0.55) [17] | 98.59 (0.05) [1] |
| MNIST-C rotate | 68.48 (0.44) [3] | 62.07 (0.64) [10] | 64.36 (0.57) [5] | 55.23 (0.50) [18] | 62.22 (0.62) [9] | 55.79 (0.95) [15] | 64.10 (0.68) [6] | 70.17 (0.95) [2] | 55.65 (0.77) [17] | 64.85 (0.37) [4] | 54.89 (1.72) [19] | 55.66 (0.50) [16] | 63.66 (1.08) [7] | 72.91 (0.99) [8] | 58.67 (0.57) [13] | 54.79 (1.12) [20] | 72.09 (0.12) [3] |
| MNIST-C scale | 88.93 (0.35) [3] | 81.45 (0.46) [11] | 67.85 (0.36) [14] | 73.23 (0.58) [16] | 81.73 (0.40) [10] | 67.90 (1.08) [19] | 86.41 (0.37) [5] | 89.40 (0.67) [2] | 79.09 (0.47) [12] | 69.21 (3.46) [18] | 82.07 (0.37) [8] | 53.67 (9.24) [20] | 74.49 (0.64) [15] | 82.99 (0.24) [7] | 78.89 (0.50) [13] | 86.40 (0.65) [6] | 82.04 (0.37) [9] | 77.71 (0.53) [14] | 69.25 (0.60) [17] | 91.81 (0.36) [1] |
| MNIST-C shear | 78.13 (1.11) [2] | 71.43 (1.32) [10] | 75.30 (1.23) [4] | 65.57 (1.27) [16] | 72.46 (1.31) [7] | 63.01 (0.55) [18] | 74.19 (1.23) [5] | 75.57 (1.40) [3] | 70.59 (1.33) [12] | 65.18 (1.70) [17] | 73.76 (1.42) [6] | 51.41 (4.50) [20] | 66.66 (1.31) [15] | 71.18 (1.09) [11] | 70.36 (1.09) [13] | 72.12 (1.39) [8] | 71.90 (1.34) [9] | 69.70 (1.36) [14] | 62.45 (1.19) [19] | 81.17 (0.18) [1] |

Continued on next page

| Model | NCSBAD | KNN | LOF | PCA | KPCA | KDE | GMM | LUNAR | AE | EIF | ABOD | DROCC | GOAD | ICL | SLAD | DTE-C | DTENonParametric | MCM | DDPM | DenoiserAD (Ours) |
|---|---|---|---|---|---|---|---|---|---|---|---|---|---|---|---|---|---|---|---|---|
| MNIST-C dot noise | 92.86 (0.47) [3] | 87.20 (0.80) [10] | 90.86 (0.56) [4] | 79.52 (1.23) [16] | 87.64 (0.79) [8] | 71.00 (0.65) [19] | 90.03 (0.64) [5] | 93.79 (0.55) [2] | 84.71 (0.97) [11] | 77.90 (1.75) [17] | 89.60 (0.74) [6] | 58.63 (6.19) [20] | 80.50 (1.42) [15] | 84.23 (0.76) [12] | 83.24 (0.67) [14] | 89.43 (0.70) [7] | 87.48 (0.65) [9] | 83.57 (1.00) [13] | 73.31 (0.90) [18] | **95.62 (0.04) [1]** |
| MNIST-C spatter | 94.77 (0.44) [3] | 91.00 (0.64) [9] | 93.46 (0.54) [5] | 85.79 (0.86) [16] | 91.49 (0.62) [7] | 76.86 (0.80) [19] | 92.90 (0.58) [6] | 95.68 (0.34) [2] | 90.43 (0.70) [11] | 85.16 (1.74) [17] | 93.84 (0.53) [4] | 53.40 (10.95) [20] | 86.67 (0.60) [15] | 90.44 (0.97) [10] | 89.53 (0.74) [13] | 91.08 (0.58) [8] | 90.06 (0.60) [12] | 89.39 (0.72) [14] | 81.86 (1.24) [18] | **96.30 (0.14) [1]** |
| MNIST-C stripe | 99.90 (0.02) [4] | 99.84 (0.01) [6] | 99.88 (0.01) [6] | 99.79 (0.03) [15] | 99.82 (0.03) [11] | 99.12 (0.25) [18] | 99.84 (0.07) [8] | **99.94 (0.07) [1]** | 99.83 (0.03) [10] | 99.41 (0.08) [17] | 99.90 (0.03) [4] | 49.24 (24.16) [20] | 99.80 (0.04) [13] | 99.92 (0.03) [2] | 99.86 (0.03) [7] | 99.82 (0.03) [9] | 99.78 (0.04) [15] | 94.97 (0.13) [19] | 99.82 (0.03) [11] | 99.92 (0.00) [2] |
| MNIST-C translate | 88.14 (0.35) [2] | 72.84 (0.64) [12] | 80.80 (0.50) [5] | 60.86 (0.76) [17] | 74.64 (0.63) [10] | 63.84 (0.51) [15] | 81.81 (0.44) [4] | 85.01 (0.57) [3] | 70.33 (0.70) [14] | 58.85 (1.55) [19] | 80.39 (0.55) [7] | 57.39 (3.90) [20] | 62.52 (1.13) [16] | 80.58 (0.91) [6] | 75.95 (0.39) [9] | 79.41 (0.56) [8] | 74.39 (0.64) [11] | 70.47 (0.68) [13] | 60.22 (1.66) [18] | **92.31 (0.07) [1]** |
| MNIST-C zigzag | 96.54 (0.20) [3] | 92.18 (0.37) [10] | 94.04 (0.24) [6] | 86.30 (0.48) [16] | 92.73 (0.33) [8] | 76.09 (0.74) [19] | 96.60 (0.41) [4] | 96.91 (0.10) [2] | 91.12 (0.45) [12] | 84.33 (1.67) [17] | 95.07 (0.21) [4] | 55.35 (6.31) [20] | 87.62 (0.45) [15] | 92.77 (0.51) [7] | 90.71 (0.62) [13] | 91.53 (0.34) [11] | 90.30 (0.48) [14] | 82.14 (1.44) [18] | 98.37 (0.02) [1] | **98.37 (0.02) [1]** |
| MVTec-AD bottle | 99.40 (0.36) [5] | 97.40 (1.16) [14] | 97.40 (1.17) [14] | 97.04 (1.15) [16] | 99.47 (0.34) [3] | 99.17 (0.38) [7] | **99.51 (0.18) [1]** | 99.18 (0.49) [6] | 97.43 (0.86) [13] | 97.51 (1.10) [12] | 59.26 (3.84) [20] | 89.71 (7.04) [19] | 96.88 (1.40) [17] | 99.14 (0.29) [8] | 99.41 (0.31) [4] | 98.12 (0.40) [9] | 97.63 (1.19) [10] | 97.56 (0.91) [11] | 95.83 (1.20) [18] | 99.50 (0.25) [2] |
| MVTec-AD cable | 92.32 (1.72) [3] | 74.97 (2.39) [11] | 74.57 (2.40) [12] | 69.46 (2.89) [17] | 92.39 (1.62) [2] | 89.80 (2.36) [7] | 92.16 (1.87) [4] | 93.46 (1.30) [2] | 83.78 (3.97) [8] | 70.12 (2.63) [13] | 68.01 (1.78) [15] | 54.13 (2.29) [19] | 51.84 (11.00) [20] | 71.41 (2.70) [15] | 91.26 (1.14) [6] | 91.38 (1.17) [5] | 87.64 (2.30) [8] | 73.71 (2.41) [14] | 64.96 (2.17) [18] | **92.40 (1.43) [1]** |
| MVTec-AD capsule | 90.32 (1.05) [7] | 72.77 (1.81) [11] | 71.58 (2.75) [12] | 65.95 (2.43) [17] | 93.34 (1.34) [3] | 92.64 (1.30) [4] | 93.46 (1.30) [2] | 83.78 (3.97) [8] | 70.12 (2.63) [13] | 68.01 (1.78) [15] | 51.97 (2.29) [20] | 55.75 (3.63) [19] | 66.19 (2.42) [16] | 92.20 (1.81) [5] | 90.45 (1.03) [6] | 78.42 (2.91) [10] | 82.21 (1.34) [9] | 69.04 (2.11) [14] | 63.01 (2.70) [18] | **94.02 (0.96) [1]** |
| MVTec-AD carpet | 90.51 (1.69) [6] | 75.36 (2.57) [11] | 73.70 (2.47) [13] | 72.74 (2.58) [16] | 91.89 (0.96) [5] | 91.62 (1.18) [5] | 91.76 (1.14) [4] | 82.22 (4.66) [10] | 73.73 (2.13) [12] | 71.85 (3.34) [17] | 57.72 (3.26) [20] | 60.37 (14.35) [19] | 73.40 (3.22) [14] | 92.13 (1.88) [2] | 90.02 (1.85) [7] | 83.70 (3.21) [8] | 82.77 (2.24) [9] | 73.31 (2.95) [15] | 68.07 (2.81) [18] | **92.58 (0.56) [1]** |
| MVTec-AD grid | 91.97 (1.42) [6] | 74.90 (1.10) [11] | 69.55 (1.35) [14] | 65.51 (0.45) [18] | 92.88 (1.59) [4] | 92.59 (1.35) [5] | 85.72 (4.93) [8] | 70.33 (0.47) [12] | 66.88 (1.47) [17] | 53.77 (4.19) [20] | 59.88 (0.32) [19] | 68.10 (0.52) [15] | 93.03 (1.46) [3] | 92.88 (1.50) [4] | 90.53 (1.46) [7] | 83.27 (1.24) [9] | 70.98 (1.86) [13] | 69.97 (0.86) [13] | 67.42 (1.72) [16] | **93.73 (0.97) [1]** |
| MVTec-AD hazelnut | 91.20 (1.04) [4] | 76.36 (2.24) [11] | 75.67 (2.46) [13] | 72.40 (2.39) [17] | 91.42 (1.16) [3] | 90.81 (0.73) [6] | 91.08 (1.23) [5] | 85.89 (3.22) [9] | 76.03 (2.58) [12] | 75.08 (2.43) [14] | 65.97 (3.71) [19] | 55.02 (10.61) [20] | 73.41 (2.29) [16] | 92.75 (1.88) [1] | 89.25 (2.73) [7] | 86.88 (2.50) [8] | 80.81 (2.03) [10] | 74.63 (2.24) [15] | 70.91 (1.65) [18] | 91.62 (1.38) [2] |
| MVTec-AD leather | **99.69 (0.18) [1]** | 99.32 (0.28) [10] | 98.69 (0.17) [16] | 99.61 (0.23) [7] | 99.55 (0.19) [8] | 99.66 (0.21) [4] | 99.67 (0.12) [2] | 99.16 (0.29) [13] | 99.14 (0.52) [15] | 78.96 (2.99) [19] | 53.28 (18.54) [20] | 99.17 (0.30) [12] | 99.37 (0.46) [9] | 99.67 (0.17) [2] | 99.63 (0.19) [5] | 98.59 (0.76) [17] | 99.63 (0.18) [5] | 99.18 (0.32) [11] | 97.74 (0.51) [18] | 99.63 (0.18) [5] |
| MVTec-AD metal nut | 93.53 (1.45) [4] | 74.07 (1.94) [11] | 73.09 (3.24) [12] | 69.57 (2.69) [17] | 93.56 (1.20) [3] | 92.78 (1.18) [6] | 93.86 (1.47) [2] | 85.91 (3.80) [9] | 71.77 (2.94) [14] | 70.16 (2.57) [16] | 51.03 (1.59) [20] | 53.34 (6.52) [19] | 70.77 (2.70) [15] | 92.99 (0.98) [5] | 90.77 (1.94) [7] | 86.64 (2.70) [8] | 81.10 (2.52) [10] | 71.80 (2.37) [13] | 67.74 (4.01) [18] | **94.07 (1.28) [1]** |
| MVTec-AD pill | 88.14 (1.86) [7] | 72.12 (2.45) [11] | 70.91 (3.21) [12] | 66.10 (1.91) [17] | 90.29 (2.64) [3] | 88.87 (2.56) [6] | 91.28 (2.26) [2] | 78.63 (4.52) [10] | 74.19 (2.22) [13] | 67.80 (1.99) [15] | 53.45 (2.22) [20] | 59.10 (6.67) [19] | 67.42 (1.68) [16] | 89.58 (0.90) [5] | 89.68 (2.20) [4] | 87.61 (2.30) [8] | 79.79 (2.04) [9] | 70.18 (2.30) [14] | 59.89 (1.82) [18] | **92.04 (0.85) [1]** |
| MVTec-AD screw | 86.94 (0.72) [6] | 67.33 (2.67) [11] | 66.37 (1.62) [12] | 60.81 (1.20) [15] | 88.60 (0.45) [4] | 89.12 (0.20) [2] | 88.28 (0.70) [5] | 79.08 (3.93) [8] | 62.15 (1.74) [14] | 59.24 (4.22) [17] | 54.59 (3.07) [20] | 58.13 (7.52) [18] | 60.41 (1.73) [16] | 89.64 (0.70) [1] | 84.45 (1.65) [7] | 75.24 (1.66) [9] | 71.98 (2.69) [10] | 62.18 (1.76) [13] | 57.86 (3.53) [19] | 88.77 (0.91) [3] |
| MVTec-AD tile | 95.88 (1.05) [3] | 85.78 (0.76) [13] | 84.88 (0.49) [14] | 83.81 (0.76) [17] | 95.77 (1.29) [4] | 94.68 (1.10) [7] | 95.41 (1.06) [5] | 91.74 (2.57) [9] | 85.93 (0.82) [11] | 84.66 (1.21) [15] | 61.89 (1.76) [19] | 53.26 (15.31) [20] | 83.84 (1.74) [16] | 95.37 (1.60) [6] | **95.98 (1.11) [1]** | 94.06 (2.03) [8] | 89.88 (0.49) [10] | 85.81 (0.53) [12] | 78.44 (0.89) [18] | 95.96 (1.07) [2] |
| MVTec-AD toothbrush | 99.94 (0.05) [3] | 96.42 (1.62) [11] | 96.64 (2.65) [12] | 96.10 (1.91) [17] | 99.93 (0.11) [4] | 99.91 (0.13) [6] | 99.92 (0.10) [5] | 99.38 (4.19) [9] | 94.94 (2.64) [12] | 22.41 (3.13) [20] | 51.56 (17.57) [19] | 81.92 (2.35) [16] | 99.91 (0.12) [6] | 99.92 (0.13) [6] | 98.74 (0.93) [9] | 99.27 (2.26) [14] | 72.82 (2.42) [18] | **99.96 (0.06) [1]** |
| MVTec-AD transistor | 93.95 (1.72) [6] | 79.67 (3.21) [11] | 78.81 (2.90) [12] | 75.66 (3.34) [17] | 94.86 (1.72) [3] | **94.90 (1.95) [1]** | 94.01 (1.59) [5] | 91.03 (3.55) [8] | 78.57 (2.99) [14] | 76.38 (6.75) [15] | 54.45 (3.19) [20] | 61.35 (13.69) [19] | 75.95 (2.88) [16] | 94.47 (1.16) [4] | 92.50 (1.89) [7] | 88.06 (2.88) [9] | 84.81 (2.50) [10] | 78.60 (2.83) [13] | 68.61 (3.64) [18] | 94.88 (1.06) [2] |
| MVTec-AD wood | 95.86 (1.13) [6] | 82.95 (1.93) [12] | 83.15 (2.26) [11] | 79.16 (2.12) [16] | 96.10 (1.91) [5] | 96.13 (1.16) [4] | 96.04 (1.07) [5] | 91.65 (1.06) [8] | 80.99 (2.24) [14] | 79.09 (2.18) [17] | 59.75 (1.22) [20] | 66.71 (11.19) [19] | 79.60 (2.05) [15] | 96.30 (1.19) [1] | 94.91 (5.18) [16] | 94.31 (1.28) [8] | 86.15 (1.89) [10] | 81.32 (2.15) [13] | 75.80 (1.69) [18] | 96.20 (1.03) [3] |
| MVTec-AD zipper | 96.90 (0.87) [2] | 87.79 (2.56) [12] | 88.66 (2.31) [11] | 82.21 (2.44) [17] | 96.52 (1.30) [4] | 95.54 (1.87) [7] | 96.11 (0.79) [6] | 93.75 (1.61) [9] | 87.18 (2.02) [14] | 82.68 (1.94) [16] | 71.43 (3.26) [18] | 58.29 (7.63) [20] | 83.17 (2.66) [15] | 96.69 (0.65) [3] | 96.38 (1.05) [5] | 94.31 (1.28) [8] | 91.16 (2.63) [10] | 87.46 (2.28) [13] | 69.95 (2.70) [19] | **97.52 (0.38) [1]** |
| SVHN 0 | 68.17 (1.86) [5] | 64.71 (1.94) [12] | 68.45 (1.76) [4] | 65.99 (1.83) [9] | 54.56 (1.61) [19] | **68.88 (1.77) [1]** | 68.64 (1.76) [2] | 64.91 (1.75) [11] | 59.14 (1.46) [18] | 67.62 (1.78) [6] | 67.62 (1.78) [6] | 51.80 (1.91) [20] | 63.35 (1.77) [14] | 66.22 (1.80) [8] | 63.18 (2.10) [16] | 67.07 (1.40) [7] | 65.25 (1.84) [10] | 64.24 (1.82) [13] | 59.34 (0.60) [17] | 68.59 (0.32) [3] |
| SVHN 1 | 67.82 (0.87) [5] | 66.97 (0.92) [9] | 65.94 (0.70) [14] | 67.06 (0.93) [8] | 57.24 (0.91) [7] | 58.84 (1.43) [19] | 68.37 (0.95) [4] | **69.87 (1.89) [1]** | 66.72 (0.93) [11] | 65.54 (1.26) [15] | 67.67 (0.82) [6] | 49.97 (2.16) [20] | 66.40 (1.62) [13] | 60.75 (0.92) [18] | 62.83 (0.75) [16] | 69.19 (0.98) [3] | 66.70 (0.90) [12] | 66.79 (0.94) [10] | 61.93 (1.02) [17] | 69.24 (0.11) [2] |
| SVHN 2 | 66.98 (0.81) [3] | 64.19 (0.70) [11] | 66.64 (0.52) [5] | 62.60 (0.51) [16] | 64.78 (0.58) [7] | 60.15 (0.59) [19] | 66.66 (0.58) [4] | 67.17 (0.78) [2] | 64.41 (0.55) [8] | 61.38 (0.69) [18] | 65.55 (0.78) [6] | 50.66 (1.50) [20] | 62.80 (0.71) [15] | 64.31 (1.25) [9] | 63.93 (1.20) [14] | 64.29 (0.52) [10] | 64.09 (0.65) [13] | 64.14 (0.48) [12] | 61.41 (1.07) [17] | **69.09 (0.13) [1]** |
| SVHN 3 | 63.82 (0.69) [5] | 64.19 (0.70) [11] | 66.03 (0.55) [16] | **64.28 (0.65) [1]** | 66.10 (1.91) [17] | 57.27 (1.28) [19] | 63.34 (0.75) [4] | 62.16 (0.41) [9] | 61.73 (0.69) [11] | 59.42 (1.17) [17] | 63.31 (0.94) [5] | 50.61 (1.78) [20] | 60.70 (0.55) [16] | 62.64 (1.21) [8] | 62.46 (1.21) [8] | 61.69 (0.45) [12] | 61.56 (0.67) [13] | 58.96 (0.87) [18] | 64.00 (0.13) [2] |
| SVHN 4 | 64.92 (0.97) [3] | 61.94 (1.00) [10] | 62.67 (1.41) [7] | 60.43 (1.08) [16] | 62.52 (1.02) [9] | 54.67 (1.29) [19] | 63.95 (1.08) [5] | 65.77 (0.80) [2] | 61.81 (1.09) [11] | 60.44 (1.45) [15] | 64.09 (1.21) [4] | 49.45 (3.07) [20] | 60.71 (1.00) [14] | 60.14 (1.59) [17] | 62.56 (0.96) [8] | 63.04 (0.80) [6] | 61.71 (1.03) [12] | 61.38 (1.13) [13] | 58.65 (1.25) [18] | **66.80 (0.10) [1]** |
| SVHN 5 | 65.51 (0.93) [3] | 63.54 (0.99) [12] | 66.03 (1.13) [2] | 62.65 (0.79) [15] | 64.13 (0.93) [9] | 56.14 (0.58) [19] | 65.33 (1.00) [4] | 64.43 (1.38) [8] | 63.76 (0.95) [10] | 60.63 (1.33) [17] | 65.20 (1.07) [6] | 48.31 (2.81) [20] | 62.93 (0.96) [14] | 61.92 (1.27) [16] | 65.25 (0.85) [5] | 63.03 (1.05) [13] | 63.64 (0.87) [11] | 60.41 (0.66) [18] | **66.97 (0.14) [1]** |
| SVHN 6 | 59.56 (0.68) [4] | 57.02 (0.72) [11] | 58.85 (0.67) [6] | 56.10 (1.05) [16] | 57.71 (0.76) [8] | 55.59 (0.94) [18] | 59.30 (0.67) [5] | 60.10 (0.50) [2] | 56.89 (0.76) [12] | 53.70 (2.27) [19] | 59.82 (0.42) [3] | 52.35 (2.57) [20] | 56.34 (0.95) [15] | 56.64 (0.63) [13] | 57.15 (0.57) [10] | 57.91 (0.80) [7] | 57.18 (0.68) [9] | 56.43 (0.72) [14] | 56.07 (1.70) [17] | **61.06 (0.16) [1]** |
| SVHN 7 | 66.55 (0.68) [12] | 66.41 (0.40) [13] | 67.63 (0.76) [4] | 67.31 (0.33) [6] | 66.94 (0.48) [10] | 64.51 (0.78) [16] | 67.45 (0.95) [5] | 67.69 (0.72) [3] | 67.03 (0.58) [8] | 64.96 (0.68) [15] | **68.37 (0.62) [1]** | 50.84 (2.19) [20] | 66.90 (0.55) [11] | 65.52 (1.71) [18] | 66.17 (1.56) [13] | 65.89 (0.94) [9] | 65.94 (0.35) [14] | 67.25 (0.49) [7] | 63.94 (1.73) [17] | 68.02 (0.24) [2] |
| SVHN 8 | 60.79 (1.55) [7] | 58.39 (1.57) [13] | 62.38 (1.61) [3] | 56.25 (1.92) [17] | 59.03 (1.55) [10] | 57.80 (0.74) [15] | 60.95 (1.76) [6] | 60.25 (1.93) [8] | 58.80 (1.86) [11] | 53.02 (1.47) [19] | 61.48 (1.71) [4] | 48.06 (1.56) [20] | 56.35 (1.59) [16] | 61.11 (0.83) [5] | 59.76 (1.15) [9] | 62.85 (1.73) [1] | 58.02 (1.39) [14] | 58.53 (1.77) [12] | 55.90 (1.26) [18] | 62.56 (0.31) [2] |
| SVHN 9 | 59.78 (1.95) [6] | 57.12 (1.91) [13] | 60.16 (1.87) [4] | 56.46 (1.94) [17] | 58.17 (1.98) [9] | 55.57 (2.02) [18] | 60.30 (2.03) [3] | 59.87 (1.95) [5] | 57.49 (2.01) [10] | 53.10 (2.44) [19] | 59.27 (1.52) [7] | 48.67 (1.40) [20] | 56.54 (1.80) [16] | 61.36 (1.97) [1] | 57.36 (1.97) [11] | 58.58 (2.11) [8] | 57.36 (1.75) [11] | 56.91 (2.12) [15] | 56.97 (1.75) [14] | 60.51 (0.29) [2] |
| agnews 0 | **71.43 (0.55) [1]** | 62.84 (0.48) [9] | 67.51 (0.69) [4] | 65.19 (0.32) [6] | 52.88 (1.58) [17] | 69.78 (0.51) [3] | 62.47 (0.49) [10] | 55.27 (0.36) [16] | 65.19 (0.19) [6] | 50.48 (0.97) [20] | 55.38 (0.91) [15] | 60.59 (0.88) [13] | 55.63 (1.31) [5] | 63.50 (0.45) [8] | 66.33 (1.22) [12] | 61.39 (0.36) [12] | 51.47 (1.78) [19] | 71.17 (0.30) [2] |
| agnews 1 | 80.87 (0.36) [1] | 66.12 (0.87) [10] | 69.10 (0.86) [3] | 58.55 (0.70) [7] | 71.63 (0.64) [5] | 58.87 (0.94) [16] | 79.15 (0.24) [4] | 69.08 (0.88) [7] | 68.06 (0.90) [8] | 70.25 (0.69) [6] | 50.00 (0.00) [20] | 60.58 (0.73) [14] | 61.25 (0.85) [13] | 54.51 (0.85) [19] | 65.17 (0.90) [11] | 67.38 (0.88) [9] | 65.03 (0.80) [12] | 55.05 (2.13) [18] | 80.25 (0.52) [2] |
| agnews 2 | **82.79 (0.63) [1]** | 74.08 (0.63) [3] | 78.28 (1.13) [4] | 62.00 (0.48) [16] | 77.10 (0.68) [5] | 60.94 (0.60) [18] | 81.96 (0.63) [3] | 75.35 (0.49) [9] | 78.73 (0.74) [8] | 64.95 (0.62) [15] | 76.02 (0.82) [6] | 48.37 (1.48) [20] | 86.03 (0.43) [13] | 65.15 (0.56) [14] | 61.47 (0.66) [17] | 74.23 (0.88) [11] | 75.81 (0.82) [7] | 58.14 (1.04) [19] | 82.62 (0.22) [2] |
| agnews 3 | 76.49 (1.11) [2] | 65.84 (1.36) [12] | 72.92 (0.90) [4] | 57.02 (0.53) [17] | 68.23 (1.36) [7] | 54.94 (0.99) [18] | 74.43 (0.94) [3] | 67.43 (1.27) [9] | 66.53 (1.20) [11] | 58.52 (0.64) [15] | 69.61 (1.37) [5] | 49.52 (0.69) [20] | 59.74 (0.85) [14] | 61.57 (1.13) [13] | 57.36 (0.95) [16] | 68.35 (0.35) [6] | 66.83 (1.35) [10] | 67.76 (1.16) [8] | 54.45 (0.76) [19] | **76.52 (0.37) [1]** |
| amazon | 62.69 (1.08) [2] | 60.92 (1.09) [7] | 57.64 (1.05) [12] | 54.91 (1.48) [15] | 61.30 (1.23) [5] | 52.37 (0.89) [18] | 61.37 (1.16) [4] | 59.97 (1.36) [9] | 59.63 (1.34) [11] | 56.45 (1.29) [13] | 61.61 (1.15) [3] | 50.00 (0.00) [20] | 56.43 (1.39) [14] | 55.51 (1.40) [16] | 60.67 (1.51) [8] | 61.13 (1.06) [6] | 59.82 (1.13) [10] | 53.58 (1.86) [17] | **62.71 (0.16) [1]** |
| imdb | 49.84 (0.63) [10] | 49.70 (0.59) [12] | 49.25 (0.73) [13] | 47.30 (0.59) [19] | 50.17 (0.42) [8] | 46.60 (0.62) [20] | 49.19 (0.47) [14] | 50.65 (0.60) [7] | 49.18 (1.06) [15] | 50.72 (0.47) [6] | 51.84 (1.80) [3] | 48.20 (1.17) [17] | 52.65 (0.47) [1] | 51.89 (0.69) [2] | 49.04 (0.82) [16] | 50.05 (0.66) [9] | 50.97 (0.54) [5] | 47.38 (1.12) [18] | 51.06 (0.25) [4] |
| yelp | 70.43 (1.19) [2] | 67.95 (1.45) [6] | 66.46 (1.02) [10] | 59.41 (1.38) [15] | 67.38 (1.23) [8] | 56.27 (0.86) [18] | 68.67 (1.00) [3] | 68.36 (1.36) [5] | 66.96 (1.11) [9] | 61.29 (1.18) [14] | 65.80 (1.19) [11] | 50.18 (1.80) [20] | 62.15 (1.58) [12] | 55.30 (0.92) [18] | 54.76 (0.55) [19] | 62.11 (0.54) [13] | 68.51 (1.42) [4] | 67.75 (1.10) [7] | 56.02 (1.91) [17] | 70.67 (0.42) [1] |
| 20news 0 | 80.24 (1.73) [2] | 72.85 (1.68) [12] | 59.80 (1.11) [1] | 76.12 (1.58) [9] | 56.23 (0.94) [19] | 78.24 (0.57) [4] | 74.55 (2.00) [10] | 78.75 (0.83) [8] | 64.77 (0.51) [15] | 76.73 (1.06) [7] | 49.89 (3.62) [20] | 64.54 (0.64) [16] | 47.23 (0.74) [18] | 61.05 (2.12) [2] | 59.22 (2.16) [4] | 55.10 (1.95) [10] | 54.43 (3.34) [12] | 56.81 (2.62) [6] | 49.42 (1.73) [16] | 80.34 (0.26) [2] |
| 20news 1 | 60.52 (2.85) [3] | 53.20 (3.47) [15] | 53.51 (3.09) [13] | 46.37 (1.01) [20] | 55.57 (3.11) [8] | 53.37 (2.24) [14] | 58.73 (2.74) [5] | 56.48 (3.12) [7] | 54.44 (3.13) [11] | 47.62 (2.30) [17] | 55.38 (3.89) [9] | 46.89 (6.55) [19] | 47.23 (0.74) [18] | 61.05 (2.12) [2] | 59.22 (2.16) [4] | 55.10 (1.95) [10] | 54.43 (3.34) [12] | 56.81 (2.62) [6] | 49.42 (1.73) [16] | **61.46 (0.06) [1]** |
| 20news 2 | 56.30 (1.61) [2] | 47.86 (1.16) [15] | 50.45 (1.98) [11] | 47.56 (2.73) [16] | 50.84 (1.56) [10] | 53.24 (2.03) [5] | 54.58 (1.80) [4] | 48.85 (0.71) [12] | 46.34 (2.65) [20] | 51.06 (1.65) [8] | 51.53 (2.22) [7] | 46.89 (2.61) [18] | 51.75 (2.55) [6] | 50.88 (1.91) [9] | 47.94 (1.51) [14] | 47.09 (2.96) [17] | **57.74 (0.06) [1]** |
| 20news 3 | 89.51 (3.34) [1] | 73.10 (8.22) [11] | 70.94 (7.14) [14] | 68.99 (5.77) [17] | 88.27 (3.87) [4] | 83.72 (3.03) [5] | 89.30 (3.95) [2] | 80.07 (5.85) [6] | 71.26 (7.81) [13] | 71.51 (9.29) [12] | 59.55 (4.66) [19] | 58.61 (8.23) [20] | 69.39 (6.92) [16] | 78.14 (1.97) [8] | 75.17 (5.19) [10] | 76.78 (4.16) [9] | 79.23 (7.55) [7] | 69.58 (7.33) [15] | 63.95 (6.16) [18] | 89.02 (2.27) [3] |
| 20news 4 | 65.33 (2.31) [3] | 49.92 (2.78) [18] | 54.61 (2.88) [11] | 51.67 (2.14) [14] | 57.22 (2.43) [7] | 48.77 (3.95) [20] | 62.28 (2.04) [4] | 49.81 (2.98) [19] | 54.57 (1.94) [8] | 54.21 (1.56) [9] | 54.73 (2.27) [10] | 50.24 (2.37) [16] | 52.50 (2.09) [13] | 56.51 (1.07) [8] | 54.17 (1.65) [12] | 47.31 (1.29) [11] | 49.96 (2.88) [17] | 57.83 (1.66) [6] | 55.55 (4.09) [9] | 65.96 (0.43) [2] |
| 20news 5 | 66.18 (2.14) [2] | 52.12 (2.53) [17] | 55.01 (1.93) [7] | 53.89 (1.93) [11] | 58.16 (2.31) [4] | 54.17 (2.05) [10] | 63.50 (2.84) [3] | 54.57 (1.94) [8] | 54.21 (1.56) [9] | 51.48 (0.95) [19] | 56.91 (3.36) [5] | 53.59 (1.88) [12] | 56.51 (1.07) [8] | 52.33 (0.94) [19] | 54.17 (1.65) [12] | 48.17 (0.61) [20] | 52.83 (2.48) [15] | 51.74 (1.69) [18] | 53.56 (1.43) [13] | 67.11 (0.83) [1] |
| Average | 82.12 (16.00) [2] | 78.36 (16.48) [9] | 78.30 (16.03) [10] | 73.32 (17.36) [16] | 81.53 (16.50) [3] | 76.40 (18.35) [14] | 80.89 (16.04) [5] | 81.16 (16.52) [4] | 77.32 (15.97) [13] | 74.25 (16.89) [15] | 69.75 (20.76) [19] | 56.70 (13.34) [20] | 71.37 (17.77) [17] | 79.00 (17.98) [7] | 77.73 (17.56) [11] | 79.77 (15.75) [6] | 79.00 (15.36) [7] | 77.65 (15.91) [12] | 70.73 (15.57) [18] | **83.34 (15.74) [1]** |

*Table 7.* Full results, including mean, standard deviation and rank AUCROC performance of all methods for each ADBench dataset and the overall average. The best performance is highlighted in **bold**, the second best highlighted with underline.

| Model | NCSBAD | KNN | LOF | PCA | KPCA | KDE | GMM | LUNAR | AE | EIF | ABOD | DROCC | GOAD | ICL | SLAD | DTE-C | DTENonParametric | MCM | DDPM | DenoiserAD (Ours) |
|---|---|---|---|---|---|---|---|---|---|---|---|---|---|---|---|---|---|---|---|---|
| ALOI | 5.96 (0.14) [12] | 6.00 (0.14) [11] | 6.46 (0.19) [4] | 6.51 (0.18) [2] | 5.89 (0.12) [15] | 6.33 (0.16) [5] | 6.49 (0.18) [3] | **6.72 (0.17) [1]** | 6.18 (0.15) [8] | 6.01 (0.18) [10] | 6.25 (0.10) [6] | 5.91 (0.00) [14] | 5.47 (0.26) [19] | 5.45 (0.09) [20] | 5.85 (0.12) [16] | 5.78 (0.13) [17] | 5.94 (0.13) [13] | 5.77 (0.10) [18] | 6.02 (0.18) [9] | 6.19 (0.05) [7] |
| annthyroid | 46.43 (0.85) [19] | 67.44 (1.24) [4] | 53.79 (1.41) [15] | 55.87 (1.55) [13] | 65.67 (1.29) [5] | 60.74 (1.53) [9] | 55.39 (1.39) [14] | 52.76 (0.99) [16] | 56.07 (1.92) [12] | 59.44 (2.35) [11] | 60.45 (1.87) [10] | 64.94 (4.91) [6] | 61.59 (5.23) [8] | 41.62 (3.93) [20] | 70.90 (1.57) [2] | **81.57 (1.00) [1]** | 63.50 (1.09) [7] | 69.50 (0.98) [3] | 47.51 (3.00) [18] | 48.29 (2.50) [17] |
| backdoor | 84.80 (1.66) [5] | 46.14 (1.79) [11] | 53.30 (1.93) [10] | 8.04 (0.21) [19] | 81.97 (2.58) [7] | 42.48 (2.56) [12] | 86.44 (1.41) [4] | 17.40 (1.51) [15] | 11.63 (2.19) [17] | 29.41 (1.19) [13] | 34.69 (1.56) [6] | 13.14 (11.43) [16] | 4.83 (0.09) [20] | 36.31 (2.17) [8] | 53.63 (1.61) [9] | 21.14 (9.13) [14] | 8.69 (0.15) [18] | 88.10 (1.92) [3] |
| breastw | 98.15 (0.71) [11] | 98.66 (0.27) [8] | 81.75 (6.88) [18] | 98.90 (0.17) [6] | 99.12 (0.37) [4] | 99.11 (0.40) [5] | 98.15 (0.49) [11] | 98.72 (0.35) [7] | 97.65 (0.50) [13] | **99.22 (0.11) [2]** | 42.71 (1.86) [20] | 62.55 (23.88) [19] | 97.42 (1.25) [14] | **99.28 (0.26) [1]** | 86.39 (3.96) [17] | 90.79 (0.91) [16] | 98.43 (0.52) [10] | 96.26 (0.90) [15] | 99.21 (0.41) [3] |
| campaign | 39.81 (0.25) [17] | 48.65 (0.38) [4] | 40.29 (0.16) [14] | 48.59 (0.18) [8] | 48.47 (0.40) [7] | 46.60 (0.42) [13] | **51.13 (0.37) [1]** | 41.17 (0.58) [15] | 48.61 (0.98) [5] | 47.12 (0.54) [11] | 20.25 (0.00) [19] | 49.12 (0.83) [3] | 48.58 (0.30) [6] | 50.34 (0.17) [2] | 44.28 (0.96) [14] | 36.05 (1.69) [18] |
| Cardiotocography | 51.21 (1.49) [15] | 56.50 (1.18) [8] | 56.57 (1.27) [7] | **69.10 (1.11) [1]** | 56.01 (1.09) [9] | 56.61 (0.87) [6] | 54.98 (0.98) [10] | 57.32 (0.77) [5] | 60.07 (1.27) [4] | 62.88 (2.73) [3] | 54.37 (0.81) [11] | 40.68 (8.50) [20] | 65.01 (0.94) [2] | 51.14 (1.80) [16] | 48.00 (1.37) [18] | 52.05 (1.71) [14] | 44.60 (0.83) [19] | 52.78 (1.45) [13] | 49.37 (1.38) [17] | 53.88 (0.99) [12] |
| cardio | 71.22 (1.86) [8] | 77.52 (1.86) [8] | 70.54 (1.91) [14] | **85.78 (1.93) [1]** | 78.03 (1.59) [7] | 79.59 (1.46) [6] | 73.22 (1.91) [11] | 80.63 (1.88) [5] | 74.39 (1.67) [10] | 81.22 (3.47) [3] | 77.29 (1.32) [9] | 50.45 (20.78) [20] | **83.88 (1.80) [2]** | 57.37 (5.98) [18] | 70.57 (2.06) [13] | 71.39 (1.86) [9] | 80.86 (2.30) [4] | 58.70 (1.43) [17] | 68.89 (1.83) [16] |
| celeba | 4.38 (0.74) [18] | 11.93 (0.71) [7] | 3.57 (0.08) [20] | **21.25 (1.54) [1]** | 11.71 (0.73) [8] | 9.45 (0.55) [13] | 16.53 (0.72) [3] | 6.34 (0.39) [16] | 11.18 (0.70) [9] | 15.01 (1.37) [5] | 4.52 (0.17) [17] | 7.41 (0.45) [15] | 4.23 (1.69) [19] | 9.22 (0.99) [14] | 15.15 (1.06) [4] | 10.44 (0.42) [11] | 17.33 (3.26) [2] | 11.17 (1.20) [10] | 13.14 (2.42) [6] |
| census | 18.65 (0.26) [11] | 21.37 (0.79) [2] | 15.68 (0.52) [18] | **21.53 (0.41) [1]** | 20.48 (0.57) [5] | 19.57 (0.41) [10] | 16.43 (0.33) [15] | 21.10 (0.38) [3] | 15.39 (0.09) [16] | 11.54 (0.18) [19] | 14.13 (1.06) [17] | 11.31 (3.27) [20] | 20.46 (0.57) [7] | 17.89 (0.45) [12] | 20.28 (0.35) [7] | 19.83 (1.01) [8] | 17.71 (0.42) [13] | 20.29 (0.25) [6] |
| cover | 82.98 (2.37) [2] | 56.49 (4.06) [9] | **83.65 (3.37) [1]** | 16.40 (0.62) [16] | 54.09 (3.51) [10] | 35.47 (2.61) [12] | 19.42 (0.99) [15] | 76.04 (2.84) [5] | 66.80 (8.57) [6] | 9.07 (1.22) [17] | 77.30 (2.84) [3] | 30.44 (4.17) [14] | 1.08 (0.06) [20] | 40.94 (19.94) [11] | 6.42 (2.82) [18] | 36.21 (7.36) [8] | 30.47 (9.84) [13] | 5.72 (0.75) [19] | 77.05 (3.03) [4] |
| donors | 40.49 (1.16) [14] | 89.24 (1.54) [6] | 64.24 (1.27) [9] | 35.70 (0.92) [17] | 95.23 (1.56) [4] | 70.65 (1.47) [8] | 43.19 (0.94) [13] | **99.37 (0.69) [1]** | 60.44 (5.57) [10] | 51.78 (1.24) [11] | 6.76 (0.07) [20] | 36.58 (3.14) [16] | 12.42 (6.55) [19] | 97.44 (1.52) [2] | 47.58 (9.42) [12] | 73.93 (1.60) [7] | 92.81 (1.38) [5] | 95.67 (2.39) [3] | 28.48 (2.78) [18] | 37.19 (1.63) [15] |
| fault | 59.31 (1.05) [16] | 61.76 (1.19) [11] | 49.82 (1.20) [20] | 60.40 (1.78) [14] | 64.22 (1.20) [4] | 62.29 (1.14) [9] | 62.33 (1.40) [8] | **66.05 (1.39) [2]** | 60.65 (1.50) [12] | 56.44 (1.18) [19] | 60.27 (3.17) [15] | 62.22 (1.47) [10] | **66.28 (0.74) [1]** | 60.52 (1.00) [13] | 58.36 (1.46) [17] | 56.26 (1.10) [18] | 61.96 (0.74) [6] |
| fraud | 52.80 (10.92) [6] | 43.07 (6.56) [11] | 61.05 (5.43) [3] | 29.55 (3.80) [15] | 36.59 (8.09) [13] | 36.32 (7.35) [14] | **61.11 (7.22) [2]** | 51.16 (6.10) [7] | 47.39 (5.52) [9] | 20.81 (4.80) [17] | 12.09 (2.86) [18] | 0.34 (0.03) [20] | 25.28 (18.30) [16] | 53.77 (10.75) [5] | 46.35 (3.63) [10] | 58.72 (5.64) [4] | 6.34 (0.92) [19] | 41.13 (10.74) [12] | **71.37 (4.05) [1]** | 49.69 (7.46) [8] |
| glass | 78.61 (7.78) [4] | 45.95 (9.69) [8] | 43.88 (10.67) [10] | 24.73 (6.35) [18] | 81.71 (9.65) [3] | 50.26 (9.92) [6] | 31.09 (8.84) [16] | 68.26 (8.35) [5] | 44.91 (7.77) [9] | 29.22 (8.52) [17] | 6.70 (1.04) [20] | 31.31 (12.63) [15] | 22.04 (8.40) [19] | **93.38 (3.83) [1]** | 41.54 (8.43) [12] | 42.69 (8.96) [12] | 46.66 (7.22) [7] | 31.64 (5.68) [14] | **83.59 (9.61) [2]** |
| Hepatitis | 99.71 (0.58) [3] | 91.01 (3.19) [10] | 45.49 (9.35) [18] | 65.20 (4.87) [15] | 99.58 (0.84) [5] | 99.58 (0.84) [5] | 88.11 (2.33) [11] | **99.74 (0.52) [2]** | 84.97 (3.08) [14] | 87.91 (3.08) [12] | 19.08 (2.22) [20] | 31.25 (10.72) [19] | 64.78 (4.89) [16] | 99.42 (1.15) [7] | **99.77 (0.46) [1]** | 96.78 (1.90) [9] | 97.85 (1.82) [8] | 86.09 (4.19) [13] | 56.43 (6.52) [17] | 99.69 (0.63) [4] |
| http | 98.54 (25.11) [15] | **100.00 (0.00) [1]** | 97.18 (3.50) [7] | 91.42 (1.24) [9] | 99.80 (0.39) [5] | 99.99 (0.01) [3] | 90.73 (1.42) [11] | **100.00 (0.01) [1]** | 99.04 (0.11) [4] | 56.28 (6.36) [18] | 9.06 (0.55) [19] | 0.74 (0.04) [20] | 67.17 (10.13) [16] | 88.38 (22.41) [13] | 89.37 (6.71) [12] | 56.73 (4.25) [17] | 91.08 (1.47) [10] | 83.25 (1.46) [14] | 99.17 (1.64) [6] | 92.81 (14.23) [3] |
| InternetAds | **69.71 (1.77) [1]** | 49.07 (1.31) [14] | 50.09 (1.15) [13] | 47.02 (1.33) [16] | 64.96 (1.73) [5] | 64.25 (2.33) [6] | 66.52 (1.90) [3] | 65.59 (4.02) [4] | 47.28 (1.23) [15] | 52.24 (7.56) [11] | 59.46 (1.39) [9] | 41.44 (3.36) [20] | 46.00 (1.24) [18] | 59.98 (0.25) [8] | 60.19 (2.55) [7] | 45.08 (4.48) [19] | 50.84 (0.84) [12] | 46.97 (1.63) [17] | 68.81 (0.72) [2] |
| Ionosphere | 90.23 (0.09) [4] | 98.06 (0.53) [11] | 94.81 (1.23) [15] | 91.88 (1.73) [18] | **99.41 (0.18) [1]** | 99.32 (0.30) [3] | 98.22 (0.42) [10] | 99.01 (0.33) [5] | 98.56 (0.32) [9] | 95.53 (1.37) [13] | 62.79 (4.85) [20] | 77.29 (12.92) [19] | 94.72 (2.11) [16] | 99.01 (0.48) [5] | 98.81 (0.25) [7] | 97.03 (0.44) [12] | 98.62 (0.41) [8] | 94.54 (2.84) [17] | 95.34 (0.46) [14] | 99.35 (0.19) [2] |
| landsat | 46.66 (0.51) [10] | 54.96 (0.05) [8] | **61.30 (0.71) [1]** | 32.68 (0.38) [17] | 46.97 (0.81) [9] | 49.02 (1.07) [7] | 31.48 (0.30) [19] | **56.61 (1.08) [2]** | 40.01 (0.30) [13] | 43.68 (2.62) [12] | 55.09 (1.25) [4] | 37.61 (1.76) [15] | 31.33 (0.46) [20] | 50.85 (1.13) [6] | 44.61 (0.68) [11] | 32.99 (0.75) [16] | 39.16 (0.74) [14] | 32.57 (0.27) [18] | 47.40 (1.08) [6] |
| letter | 9.75 (0.31) [7] | 8.78 (0.12) [15] | 11.60 (1.30) [4] | 8.06 (0.11) [20] | 8.98 (0.13) [13] | 8.98 (0.12) [13] | 8.78 (0.10) [15] | 9.00 (0.12) [12] | 8.62 (0.10) [17] | 8.24 (0.22) [18] | 10.67 (0.74) [5] | 8.14 (0.14) [19] | 12.09 (2.02) [3] | 9.27 (0.30) [9] | 9.10 (0.15) [10] | 12.90 (0.15) [2] | 9.03 (0.27) [11] | 9.24 (0.50) [9] | 10.31 (0.48) [6] |
| Lymphography | 99.38 (0.76) [9] | 98.95 (0.93) [12] | 86.76 (3.72) [17] | 98.91 (0.44) [13] | **100.00 (0.00) [1]** | **100.00 (0.00) [1]** | 99.83 (0.34) [8] | **100.00 (0.00) [1]** | 98.79 (1.18) [14] | 99.37 (0.95) [10] | 5.58 (0.52) [20] | 39.01 (28.08) [19] | 99.28 (0.63) [11] | **100.00 (0.01) [1]** | 99.86 (0.29) [7] | 90.97 (5.83) [16] | 72.32 (7.14) [18] | **100.00 (0.00) [1]** | 96.54 (3.08) [15] | **100.00 (0.00) [1]** |

| Model | NCSBAD | KNN | LOF | PCA | KPCA | KDE | GMM | LUNAR | AE | EIF | ABOD | DROCC | GOAD | ICL | SLAD | DTE-C | DTENonParametric | MCM | DDPM | DenoiserAD (Ours) |
|---|---|---|---|---|---|---|---|---|---|---|---|---|---|---|---|---|---|---|---|---|
| magic.gamma | 90.12 (0.12) [2] | 85.96 (0.17) [10] | 86.44 (0.17) [8] | 75.28 (0.28) [19] | 85.75 (0.15) [11] | 80.91 (0.25) [15] | 83.35 (0.25) [12] | 87.97 (0.10) [5] | 87.08 (0.25) [7] | 80.79 (1.16) [16] | 87.26 (0.12) [6] | 83.09 (0.62) [13] | 75.84 (1.24) [18] | 82.55 (0.48) [14] | 77.45 (0.18) [17] | 89.10 (0.18) [3] | 86.12 (0.11) [9] | 90.15 (0.12) [1] | 74.02 (0.64) [20] | 88.60 (0.18) [4] |
| mammography | 24.54 (3.29) [16] | 40.30 (3.10) [8] | 34.12 (3.34) [12] | 41.06 (1.80) [7] | 44.84 (2.92) [2] | 42.24 (2.85) [4] | 40.21 (2.06) [9] | 45.77 (1.87) [1] | 35.28 (2.04) [11] | 42.22 (2.98) [5] | 5.12 (0.13) [20] | 26.31 (2.81) [14] | 27.54 (4.39) [13] | 20.71 (2.67) [17] | 16.38 (2.14) [18] | 42.31 (2.17) [3] | 40.12 (2.42) [10] | 24.55 (2.17) [15] | 10.77 (0.93) [19] | 41.16 (0.90) [6] |
| mnist | 75.25 (1.13) [4] | 73.13 (1.42) [7] | 71.19 (1.30) [9] | 65.72 (1.68) [16] | 74.72 (1.55) [5] | 76.14 (1.00) [2] | 70.19 (1.67) [11] | 79.20 (0.84) [1] | 70.57 (1.68) [10] | 60.23 (5.93) [18] | 75.95 (1.19) [3] | 61.81 (3.33) [17] | 66.88 (2.65) [15] | 68.39 (1.42) [14] | 69.38 (1.22) [13] | 54.87 (1.88) [19] | 73.30 (1.36) [6] | 69.84 (1.62) [12] | 54.41 (1.72) [20] | 72.75 (1.70) [8] |
| musk | 100.00 (0.00) [1] | 100.00 (0.00) [1] | 100.00 (0.00) [1] | 100.00 (0.00) [1] | 100.00 (0.00) [1] | 100.00 (0.00) [1] | 98.25 (0.66) [16] | 100.00 (0.00) [1] | 100.00 (0.00) [1] | 64.24 (25.30) [18] | 99.34 (0.30) [15] | 21.28 (20.74) [20] | 100.00 (0.00) [1] | 91.38 (2.81) [17] | 100.00 (0.00) [1] | 100.00 (0.00) [1] | 35.83 (0.94) [19] | 100.00 (0.00) [1] | 99.50 (0.37) [14] | 100.00 (0.00) [1] |
| optdigits | 52.06 (2.94) [3] | 29.12 (1.90) [10] | 42.28 (0.83) [5] | 6.04 (0.05) [20] | 23.04 (1.22) [11] | 39.76 (2.06) [6] | 13.22 (0.55) [17] | 88.09 (5.55) [1] | 18.20 (0.75) [15] | 13.33 (2.72) [16] | 32.01 (0.87) [9] | 19.01 (4.55) [13] | 7.83 (0.63) [18] | 43.98 (6.12) [4] | 35.45 (1.61) [7] | 18.27 (0.53) [14] | 34.07 (1.85) [8] | 21.21 (1.18) [12] | 7.63 (0.91) [19] | 54.51 (3.39) [2] |
| PageBlocks | 58.47 (1.56) [17] | 67.93 (1.13) [5] | 73.09 (1.08) [2] | 59.91 (1.13) [16] | 64.91 (0.98) [10] | 67.98 (1.13) [4] | 62.83 (1.44) [13] | 69.12 (1.75) [3] | 67.00 (1.17) [7] | 45.63 (1.50) [20] | 67.79 (1.12) [6] | 72.59 (3.44) [1] | 64.67 (1.47) [11] | 66.05 (4.05) [8] | 57.52 (1.38) [18] | 65.26 (1.53) [9] | 51.77 (0.49) [19] | 62.00 (1.42) [14] | 60.74 (0.92) [15] | 63.77 (0.42) [12] |
| pendigits | 94.71 (1.38) [5] | 96.48 (0.64) [4] | 76.67 (2.24) [9] | 38.18 (2.97) [15] | 93.17 (0.82) [7] | 96.85 (0.63) [3] | 16.47 (1.15) [19] | 98.91 (0.18) [1] | 71.62 (1.84) [10] | 56.36 (5.51) [13] | 89.23 (1.76) [8] | 14.61 (3.08) [20] | 31.41 (6.04) [17] | 66.03 (8.54) [11] | 35.37 (2.20) [16] | 45.91 (1.51) [14] | 97.37 (0.73) [2] | 10.05 (1.25) [18] | 94.30 (1.92) [6] | |
| Pima | 78.59 (0.80) [7] | 77.16 (1.76) [8] | 68.07 (2.60) [16] | 73.06 (2.43) [11] | 85.14 (1.03) [1] | 80.84 (1.16) [3] | 73.19 (2.07) [10] | 78.75 (2.04) [6] | 70.46 (1.79) [12] | 76.32 (2.32) [9] | 64.75 (0.73) [17] | 50.65 (13.26) [20] | 69.12 (6.90) [13] | 70.83 (2.09) [4] | 63.75 (5.28) [18] | 68.25 (3.51) [15] | 78.90 (1.41) [5] | 68.75 (2.50) [14] | 60.48 (2.11) [19] | 80.86 (1.11) [2] |
| satellite | 84.80 (0.58) [10] | 85.92 (0.53) [3] | 85.82 (0.41) [4] | 77.72 (0.54) [19] | 84.25 (0.60) [12] | 85.20 (0.57) [8] | 84.31 (0.62) [11] | 85.81 (0.50) [5] | 84.02 (0.53) [14] | 82.07 (0.72) [16] | 85.57 (0.51) [6] | 77.16 (5.78) [20] | 79.30 (0.82) [17] | 86.95 (0.66) [2] | 88.56 (0.51) [1] | 84.18 (0.63) [13] | 85.53 (0.62) [7] | 77.89 (0.41) [18] | 83.00 (0.87) [15] | 85.01 (0.38) [9] |
| satimage-2 | 93.84 (0.90) [10] | 97.12 (0.69) [2] | 89.61 (1.40) [12] | 92.26 (0.67) [11] | 96.97 (0.59) [3] | 96.00 (0.75) [6] | 86.64 (1.22) [14] | 95.32 (1.14) [7] | 97.14 (0.54) [1] | 96.13 (1.12) [4] | 88.44 (1.58) [13] | 76.42 (18.17) [17] | 96.03 (0.43) [5] | 95.39 (0.90) [8] | 76.34 (2.62) [18] | 83.73 (0.99) [16] | 39.23 (2.80) [20] | 72.20 (7.89) [19] | 85.44 (2.46) [15] | |
| skin | 92.61 (1.24) [6] | 98.35 (0.36) [2] | 61.68 (1.82) [16] | 36.24 (0.22) [18] | 96.96 (0.34) [3] | 64.68 (0.31) [13] | 64.45 (0.50) [14] | 96.83 (0.76) [4] | 62.97 (4.55) [15] | 66.85 (1.36) [11] | 27.33 (0.20) [20] | 65.30 (1.95) [12] | 44.71 (3.26) [17] | 32.68 (0.83) [19] | 78.69 (7.71) [7] | 69.43 (0.70) [10] | 98.79 (0.31) [1] | 72.59 (7.83) [8] | 70.67 (0.58) [9] | 96.38 (0.79) [5] |
| smtp | 48.57 (10.51) [13] | 57.95 (9.63) [6] | 57.36 (9.87) [10] | 57.50 (9.33) [9] | 70.27 (15.22) [1] | 70.24 (15.22) [2] | 45.48 (10.48) [14] | 70.18 (15.22) [3] | 58.07 (9.61) [5] | 1.32 (0.43) [20] | 1.53 (0.23) [19] | 8.98 (17.80) [18] | 35.74 (4.38) [16] | 14.42 (16.21) [17] | 57.81 (9.67) [7] | 57.74 (9.69) [8] | 50.90 (6.86) [12] | 53.11 (22.87) [11] | 39.19 (15.85) [15] | 63.69 (12.63) [4] |
| SpamBase | 85.32 (0.49) [4] | 83.36 (0.42) [8] | 73.29 (0.61) [20] | 81.90 (0.47) [15] | 84.27 (0.46) [5] | 83.90 (0.57) [6] | 81.30 (0.49) [17] | 81.60 (1.02) [16] | 82.62 (0.40) [12] | 89.15 (0.74) [1] | 82.79 (0.42) [11] | 80.43 (1.36) [19] | 82.09 (0.63) [13] | 86.24 (0.98) [2] | 85.91 (0.47) [3] | 83.74 (0.23) [7] | 82.88 (0.30) [10] | 81.94 (0.52) [14] | 80.65 (0.52) [18] | 82.95 (0.14) [9] |
| speech | 3.70 (0.44) [6] | 3.19 (0.46) [15] | 3.68 (0.76) [7] | 3.17 (0.47) [16] | 3.58 (0.57) [9] | 4.66 (0.40) [1] | 3.59 (0.59) [8] | 3.93 (0.58) [4] | 3.17 (0.41) [16] | 3.21 (0.72) [14] | 3.33 (0.33) [12] | 4.20 (1.36) [3] | 3.08 (0.42) [18] | 4.45 (1.48) [2] | 3.05 (0.29) [19] | 3.86 (0.32) [5] | 3.29 (0.00) [13] | 2.97 (0.38) [20] | 3.42 (0.42) [10] | 3.37 (0.14) [11] |
| Stamps | 88.54 (4.84) [4] | 75.56 (8.29) [8] | 69.45 (7.92) [11] | 64.62 (6.44) [14] | 93.71 (5.16) [1] | 93.28 (4.69) [2] | 66.31 (7.80) [12] | 84.94 (5.49) [5] | 74.06 (6.18) [9] | 65.79 (5.03) [13] | 24.18 (2.00) [20] | 27.90 (23.32) [19] | 58.39 (13.94) [16] | 80.39 (5.22) [7] | 48.35 (10.78) [17] | 62.30 (8.25) [15] | 62.60 (6.01) [6] | 70.73 (7.91) [10] | 48.33 (9.33) [18] | 91.61 (5.60) [3] |
| thyroid | 69.77 (2.36) [15] | 82.77 (1.83) [2] | 61.93 (3.11) [17] | 82.69 (2.19) [3] | 80.52 (1.91) [9] | 81.41 (1.33) [6] | 81.76 (2.37) [5] | 80.69 (1.63) [8] | 60.51 (0.83) [18] | 82.33 (3.42) [4] | 44.31 (2.45) [20] | 77.08 (2.38) [12] | 81.33 (2.54) [7] | 47.23 (7.82) [19] | 77.74 (3.48) [11] | 84.52 (1.85) [1] | 66.10 (1.48) [16] | 78.73 (2.17) [10] | 72.98 (7.06) [14] | 76.63 (1.23) [13] |
| vertebral | 51.52 (2.90) [4] | 24.53 (2.98) [11] | 30.59 (2.07) [8] | 18.18 (1.70) [18] | 66.99 (6.26) [2] | 18.17 (1.92) [19] | 20.38 (1.89) [16] | 40.03 (5.69) [5] | 26.35 (2.14) [9] | 20.14 (1.78) [17] | 14.71 (1.59) [20] | 22.07 (8.34) [13] | 21.46 (2.11) [14] | 53.25 (9.63) [3] | 20.83 (3.92) [15] | 31.31 (3.64) [7] | 34.04 (3.79) [6] | 26.12 (1.58) [10] | 22.41 (2.78) [12] | 71.45 (5.56) [1] |
| vowels | 53.11 (3.22) [1] | 31.33 (2.58) [10] | 37.78 (5.60) [6] | 12.21 (2.88) [19] | 30.76 (2.50) [11] | 29.40 (2.45) [12] | 40.47 (4.52) [5] | 41.80 (2.21) [4] | 45.44 (3.34) [3] | 16.37 (2.67) [18] | 31.46 (3.09) [9] | 8.64 (5.44) [20] | 18.26 (3.30) [17] | 29.24 (6.16) [13] | 34.68 (3.43) [7] | 45.99 (1.75) [2] | 22.02 (3.06) [16] | 27.88 (2.74) [14] | 22.60 (6.09) [15] | 32.64 (2.63) [8] |
| Waveform | 9.47 (0.86) [15] | 26.15 (2.59) [5] | 29.86 (2.76) [2] | 9.26 (0.82) [16] | 20.58 (1.78) [7] | 27.55 (2.43) [3] | 8.49 (0.83) [18] | 26.78 (3.73) [4] | 13.06 (0.86) [11] | 12.89 (2.43) [12] | 16.75 (1.21) [10] | 17.91 (7.76) [9] | 9.23 (1.05) [17] | 20.35 (3.52) [8] | 5.26 (0.10) [20] | 12.78 (1.70) [13] | 23.60 (2.48) [6] | 48.66 (1.35) [1] | 7.81 (1.39) [19] | 11.71 (1.23) [14] |
| WBC | 94.84 (7.49) [8] | 90.61 (5.46) [11] | 25.72 (3.55) [19] | 94.80 (1.57) [9] | 98.85 (2.30) [1] | 98.30 (1.77) [2] | 85.80 (6.34) [13] | 97.34 (2.70) [7] | 81.95 (5.41) [14] | 98.21 (0.98) [4] | 7.95 (1.14) [20] | 26.57 (25.04) [18] | 92.04 (3.57) [10] | 98.47 (1.99) [3] | 97.66 (1.92) [6] | 31.88 (14.14) [17] | 66.89 (10.14) [16] | 89.33 (6.18) [12] | 77.34 (6.70) [15] | 98.10 (2.63) [5] |
| WDBC | 99.89 (0.15) [1] | 91.99 (3.99) [12] | 97.24 (1.77) [6] | 90.73 (3.95) [14] | 98.05 (1.27) [2] | 97.88 (1.34) [3] | 97.72 (1.55) [7] | 97.76 (1.44) [4] | 96.36 (5.51) [13] | 92.46 (6.04) [11] | 11.93 (2.42) [20] | 15.32 (21.99) [19] | 91.21 (3.75) [13] | 97.64 (2.13) [5] | 94.56 (2.12) [8] | 72.84 (4.22) [16] | 61.18 (10.69) [17] | 76.55 (14.73) [15] | 58.47 (9.51) [18] | 92.70 (3.94) [10] |
| Wilt | 20.61 (0.92) [5] | 12.08 (0.33) [12] | 15.35 (0.78) [9] | 6.48 (0.15) [20] | 16.82 (0.49) [8] | 7.48 (0.15) [19] | 27.11 (0.54) [3] | 11.48 (0.95) [13] | 10.46 (0.21) [15] | 8.36 (0.42) [18] | 18.71 (0.55) [7] | 9.93 (1.99) [17] | 11.01 (0.78) [14] | 28.44 (5.87) [2] | 12.42 (0.35) [11] | 25.11 (0.81) [4] | 13.13 (0.40) [10] | 33.83 (1.15) [1] | 10.11 (0.52) [16] | 20.10 (0.47) [6] |
| wine | 99.69 (0.62) [5] | 95.42 (2.55) [12] | 91.04 (3.74) [14] | 70.60 (8.42) [17] | 99.84 (0.31) [3] | 99.90 (0.20) [2] | 95.56 (3.15) [11] | 99.73 (0.54) [4] | 97.06 (1.70) [9] | 84.27 (12.47) [15] | 10.49 (1.08) [20] | 20.98 (11.83) [19] | 73.02 (5.95) [16] | 98.93 (2.01) [7] | 100.00 (0.00) [1] | 99.30 (1.18) [6] | 93.98 (2.55) [13] | 95.81 (2.82) [10] | 65.49 (5.11) [18] | 98.52 (2.97) [8] |
| WPBC | 89.72 (6.18) [1] | 47.86 (3.07) [13] | 43.05 (2.36) [14] | 41.92 (3.99) [16] | 89.15 (6.11) [2] | 89.11 (5.49) [3] | 49.08 (3.01) [12] | 81.46 (5.56) [7] | 49.32 (4.77) [11] | 25.30 (1.37) [20] | 37.99 (4.15) [19] | 41.66 (4.86) [17] | 58.28 (4.40) [4] | 38.73 (6.93) [6] | 63.97 (2.44) [9] | 65.02 (2.16) [8] | 60.49 (2.86) [10] | 60.49 (2.86) [10] | 40.87 (1.42) [18] | 87.43 (6.02) [5] |
| yeast | 49.88 (0.85) [4] | 47.97 (0.73) [11] | 48.71 (0.83) [9] | 46.79 (0.73) [20] | 47.76 (0.81) [12] | 47.16 (0.76) [17] | 46.87 (0.66) [19] | 47.31 (0.76) [15] | 47.24 (0.58) [16] | 46.90 (0.57) [18] | 48.70 (0.97) [10] | 52.08 (3.94) [1] | 49.58 (2.12) [6] | 49.21 (1.85) [8] | 51.17 (1.03) [2] | 49.74 (0.74) [5] | 47.54 (0.69) [14] | 47.59 (0.60) [13] | 50.07 (1.21) [3] | 49.46 (0.38) [7] |
| CIFAR10_0 | 26.15 (1.05) [3] | 25.11 (1.00) [8] | 25.70 (0.91) [6] | 24.27 (1.06) [12] | 25.99 (1.17) [4] | 22.02 (1.20) [16] | 25.76 (1.12) [5] | 26.95 (1.23) [1] | 24.96 (1.02) [9] | 21.85 (1.11) [17] | 26.28 (1.22) [2] | 12.98 (1.80) [20] | 24.36 (0.76) [11] | 19.51 (0.92) [18] | 24.21 (0.98) [13] | 22.56 (0.53) [14] | 18.58 (0.47) [19] | 24.63 (0.96) [10] | 22.51 (1.66) [15] | 25.24 (0.15) [7] |
| CIFAR10_1 | 17.04 (1.27) [6] | 14.82 (1.07) [13] | 22.35 (1.69) [1] | 14.02 (1.07) [15] | 15.75 (1.16) [11] | 13.46 (1.01) [17] | 17.90 (1.34) [2] | 15.35 (1.38) [12] | 16.79 (1.35) [7] | 10.30 (0.40) [19] | 16.54 (1.26) [8] | 10.15 (1.34) [20] | 14.73 (1.14) [14] | 17.34 (1.13) [3] | 17.17 (1.35) [5] | 17.33 (1.03) [4] | 13.09 (0.46) [18] | 16.15 (1.25) [9] | 13.75 (0.90) [16] | 16.11 (0.22) [10] |
| CIFAR10_2 | 14.44 (0.73) [4] | 13.77 (0.57) [11] | 15.27 (0.77) [1] | 13.46 (0.46) [14] | 14.08 (0.62) [9] | 11.72 (0.66) [19] | 14.94 (0.50) [2] | 14.04 (0.50) [8] | 14.39 (1.40) [4] | 12.63 (0.33) [17] | 14.41 (0.65) [5] | 10.68 (1.02) [20] | 13.63 (0.68) [13] | 13.05 (1.03) [16] | 13.73 (0.88) [12] | 14.27 (0.90) [6] | 13.43 (0.23) [15] | 13.97 (0.60) [10] | 12.48 (0.78) [18] | 14.50 (0.07) [3] |
| CIFAR10_3 | 14.02 (1.39) [7] | 13.13 (1.29) [13] | 14.89 (1.43) [1] | 13.24 (1.28) [12] | 13.82 (1.47) [8] | 12.73 (1.27) [15] | 14.51 (1.44) [2] | 14.40 (1.58) [3] | 14.39 (1.40) [4] | 11.51 (0.72) [18] | 13.76 (1.30) [9] | 11.18 (1.30) [20] | 13.25 (1.23) [11] | 12.27 (0.98) [17] | 13.34 (1.20) [10] | 14.09 (1.24) [6] | 11.41 (0.60) [19] | 14.24 (1.39) [5] | 12.57 (1.10) [16] | 13.08 (0.18) [14] |
| CIFAR10_4 | 34.35 (2.02) [4] | 32.77 (1.89) [5] | 31.90 (1.95) [12] | 31.99 (1.85) [11] | 32.44 (2.31) [8] | 28.29 (2.07) [17] | 34.46 (2.10) [3] | 34.91 (2.49) [2] | 32.72 (1.96) [6] | 29.51 (1.98) [15] | 32.10 (2.00) [10] | 15.87 (2.57) [20] | 32.23 (2.00) [9] | 27.35 (1.84) [18] | 31.83 (1.97) [13] | 29.11 (2.15) [16] | 18.91 (0.84) [19] | 32.50 (1.87) [7] | 30.40 (2.28) [14] | 35.95 (0.13) [1] |
| CIFAR10_5 | 13.61 (0.49) [7] | 12.87 (0.45) [12] | 16.16 (0.91) [1] | 13.03 (0.39) [10] | 13.56 (0.52) [9] | 12.42 (0.42) [15] | 14.31 (0.57) [3] | 12.40 (0.62) [16] | 14.08 (0.68) [4] | 10.58 (0.40) [19] | 13.61 (0.49) [7] | 9.76 (0.91) [20] | 12.92 (0.43) [11] | 11.73 (1.17) [17] | 13.72 (0.71) [5] | 14.75 (0.48) [2] | 11.42 (0.19) [18] | 13.42 (0.54) [6] | 12.71 (1.02) [13] | 12.53 (0.18) [14] |
| CIFAR10_6 | 23.70 (1.17) [1] | 20.70 (0.98) [12] | 23.32 (1.36) [3] | 19.47 (0.88) [14] | 21.47 (1.00) [9] | 18.18 (1.29) [16] | 23.10 (1.17) [4] | 22.43 (0.77) [7] | 21.06 (1.05) [10] | 17.31 (0.88) [18] | 23.04 (1.18) [5] | 12.61 (0.68) [20] | 19.65 (0.89) [13] | 16.96 (1.09) [19] | 22.30 (1.32) [8] | 22.89 (1.72) [6] | 17.83 (0.51) [17] | 20.82 (1.03) [11] | 18.85 (1.26) [15] | 23.70 (0.24) [1] |
| CIFAR10_7 | 22.43 (1.24) [3] | 21.31 (1.22) [11] | 23.79 (1.19) [1] | 21.19 (1.00) [13] | 21.52 (1.26) [9] | 17.07 (1.22) [18] | 21.90 (0.73) [6] | 22.60 (1.42) [2] | 22.02 (1.26) [5] | 19.61 (1.07) [15] | 21.53 (0.81) [8] | 13.40 (1.84) [20] | 21.34 (1.39) [10] | 17.24 (0.62) [17] | 21.24 (1.16) [12] | 19.30 (1.04) [16] | 14.06 (0.43) [19] | 21.67 (1.20) [7] | 19.84 (1.43) [14] | 22.31 (0.32) [4] |
| CIFAR10_8 | 22.01 (0.69) [4] | 20.71 (0.81) [10] | 23.62 (0.62) [1] | 20.52 (0.70) [11] | 20.81 (0.55) [9] | 16.37 (1.22) [19] | 21.78 (0.67) [6] | 22.43 (1.27) [2] | 22.03 (0.67) [3] | 17.71 (0.90) [18] | 21.89 (1.02) [5] | 13.70 (1.87) [20] | 20.43 (0.70) [12] | 19.23 (1.10) [14] | 21.53 (0.80) [8] | 19.19 (0.60) [15] | 15.18 (0.37) [17] | 21.58 (0.68) [7] | 18.67 (1.52) [16] | 20.07 (0.16) [13] |
| CIFAR10_9 | 25.86 (0.91) [4] | 23.46 (0.86) [12] | 26.74 (1.30) [1] | 23.21 (0.87) [13] | 24.22 (0.64) [8] | 19.96 (1.14) [18] | 25.96 (1.15) [3] | 26.73 (0.57) [2] | 24.29 (1.06) [7] | 20.40 (1.30) [17] | 25.11 (0.64) [5] | 14.16 (3.04) [20] | 22.93 (0.67) [14] | 21.18 (1.35) [16] | 23.83 (0.90) [11] | 23.99 (1.36) [10] | 16.44 (0.40) [19] | 24.10 (0.94) [9] | 22.22 (0.86) [15] | 25.01 (0.18) [6] |
| FashionMNIST_0 | 50.44 (3.36) [5] | 46.14 (3.38) [11] | 51.36 (3.47) [4] | 41.67 (3.13) [15] | 47.38 (3.51) [8] | 35.01 (2.54) [17] | 49.42 (3.39) [6] | 57.02 (1.70) [1] | 47.18 (3.49) [10] | 34.85 (3.22) [18] | 48.77 (3.57) [7] | 19.89 (2.86) [20] | 42.55 (3.47) [14] | 54.70 (4.71) [2] | 47.36 (3.27) [9] | 46.11 (3.14) [12] | 32.39 (0.73) [19] | 44.86 (3.31) [13] | 40.63 (2.68) [16] | 51.66 (0.47) [3] |
| FashionMNIST_1 | 87.72 (2.31) [2] | 84.87 (2.77) [8] | 86.14 (2.99) [5] | 82.10 (2.72) [15] | 84.51 (2.09) [9] | 79.81 (2.77) [16] | 83.44 (3.14) [12] | 89.67 (2.85) [1] | 84.08 (2.79) [10] | 70.22 (3.32) [18] | 86.79 (2.49) [4] | 43.49 (17.03) [20] | 82.71 (2.77) [13] | 85.33 (2.41) [7] | 85.66 (2.53) [6] | 77.88 (2.64) [17] | 46.60 (1.52) [19] | 83.73 (2.75) [11] | 82.52 (3.04) [14] | 86.80 (0.13) [3] |
| FashionMNIST_2 | 50.85 (2.82) [7] | 46.22 (3.22) [11] | 53.52 (2.81) [3] | 42.27 (3.46) [15] | 49.18 (3.48) [8] | 40.71 (3.28) [17] | 51.37 (3.08) [5] | 56.35 (2.71) [2] | 47.46 (3.35) [9] | 23.72 (0.77) [19] | 50.86 (3.10) [6] | 23.63 (12.16) [20] | 43.44 (3.32) [13] | 57.26 (2.63) [1] | 44.37 (2.82) [12] | 42.66 (2.67) [14] | 30.61 (0.68) [18] | 46.81 (3.32) [10] | 41.56 (3.95) [16] | 52.63 (0.40) [4] |
| FashionMNIST_3 | 60.51 (1.20) [3] | 54.57 (1.31) [11] | 60.38 (1.21) [4] | 50.84 (1.41) [15] | 57.34 (1.19) [8] | 47.52 (1.84) [17] | 59.60 (1.23) [6] | 61.33 (2.09) [2] | 54.92 (1.50) [10] | 38.04 (3.08) [18] | 58.89 (1.32) [7] | 25.31 (5.30) [20] | 51.66 (1.52) [14] | 59.65 (2.57) [5] | 54.39 (1.60) [12] | 56.48 (0.45) [9] | 34.03 (0.74) [19] | 53.77 (1.56) [13] | 49.58 (1.42) [16] | 63.07 (0.28) [1] |
| FashionMNIST_4 | 58.25 (1.42) [5] | 53.24 (1.38) [10] | 58.62 (1.70) [4] | 50.13 (1.27) [15] | 53.98 (1.18) [8] | 49.26 (1.06) [17] | 55.49 (2.34) [6] | 59.43 (0.42) [3] | 53.71 (1.26) [9] | 27.16 (2.12) [19] | 54.84 (1.42) [7] | 22.83 (10.21) [20] | 50.95 (1.15) [14] | 59.59 (2.27) [2] | 51.24 (1.10) [13] | 52.07 (1.93) [12] | 29.71 (0.44) [18] | 52.93 (1.25) [11] | 49.80 (1.64) [16] | 62.11 (0.17) [1] |
| FashionMNIST_5 | 86.34 (1.36) [2] | 85.96 (1.26) [6] | 85.58 (1.53) [10] | 85.64 (1.22) [9] | 80.60 (1.38) [16] | 85.28 (1.17) [13] | 85.28 (1.19) [13] | 86.20 (0.75) [4] | 85.72 (1.24) [8] | 80.11 (1.48) [17] | 86.60 (1.32) [1] | 50.26 (21.37) [19] | 85.55 (1.26) [11] | 85.10 (1.15) [15] | 85.10 (1.03) [3] | 75.90 (2.24) [18] | 37.31 (0.76) [20] | 85.91 (1.21) [7] | 85.42 (1.26) [2] | 86.14 (0.06) [5] |
| FashionMNIST_6 | 34.77 (1.63) [4] | 27.00 (1.22) [12] | 33.79 (2.29) [6] | 21.84 (0.89) [17] | 28.22 (1.16) [10] | 24.96 (1.54) [14] | 34.15 (1.48) [5] | 36.67 (1.67) [3] | 27.52 (1.35) [11] | 15.94 (1.20) [19] | 30.68 (1.62) [8] | 10.80 (4.56) [20] | 23.16 (0.54) [15] | 39.06 (2.13) [2] | 29.51 (1.51) [9] | 32.37 (1.73) [7] | 22.13 (0.73) [16] | 26.20 (1.29) [13] | 21.35 (1.18) [18] | 41.70 (0.24) [1] |
| FashionMNIST_7 | 90.13 (1.05) [3] | 89.15 (1.10) [10] | 89.23 (0.95) [7] | 88.67 (1.21) [13] | 82.23 (1.36) [17] | 88.87 (1.14) [11] | 89.21 (1.07) [8] | 89.35 (3.22) [5] | 89.29 (1.10) [6] | 86.73 (1.57) [16] | 90.34 (0.94) [2] | 46.84 (21.70) [19] | 88.76 (1.28) [12] | 88.54 (1.60) [14] | 89.19 (1.07) [9] | 77.05 (1.49) [18] | 41.68 (1.30) [20] | 89.39 (1.13) [4] | 88.15 (1.11) [15] | 91.85 (0.09) [1] |
| FashionMNIST_8 | 35.99 (1.05) [5] | 27.98 (0.96) [11] | 38.44 (1.49) [3] | 22.98 (0.70) [17] | 28.92 (1.08) [10] | 23.22 (1.14) [16] | 34.64 (0.89) [6] | 41.29 (1.98) [2] | 27.69 (0.87) [12] | 20.40 (1.81) [19] | 32.70 (1.05) [7] | 12.88 (3.08) [20] | 23.59 (0.91) [15] | 36.71 (2.23) [4] | 31.91 (0.96) [8] | 31.30 (1.43) [9] | 24.39 (0.71) [14] | 25.98 (1.03) [13] | 22.50 (0.61) [18] | 43.45 (0.63) [1] |
| FashionMNIST_9 | 85.47 (1.55) [2] | 83.84 (1.45) [8] | 84.52 (1.46) [5] | 83.36 (1.51) [10] | 81.39 (1.57) [14] | 81.27 (1.86) [15] | 82.67 (2.19) [13] | 86.14 (1.39) [7] | 77.34 (2.20) [17] | 84.90 (1.32) [4] | 41.58 (19.67) [19] | 12.88 (3.08) [20] | 23.59 (0.91) [19] | 36.71 (1.23) [16] | 84.50 (1.42) [6] | 75.66 (1.78) [18] | 39.99 (1.13) [20] | 83.62 (1.50) [9] | 84.92 (0.08) [3] | |
| MNIST-C_brightness | 48.81 (1.83) [3] | 30.26 (0.84) [11] | 38.67 (1.43) [7] | 22.34 (0.57) [18] | 32.16 (0.75) [10] | 26.17 (1.68) [15] | 38.34 (1.22) [8] | 50.57 (2.37) [2] | 28.57 (0.81) [12] | 16.79 (0.94) [20] | 39.50 (1.33) [6] | 18.79 (2.95) [19] | 23.13 (0.50) [16] | 44.39 (2.48) [4] | 32.27 (0.91) [9] | 40.00 (1.39) [5] | 27.25 (0.53) [14] | 27.54 (0.75) [13] | 22.48 (0.72) [17] | 70.16 (0.47) [1] |
| MNIST-C_canny_edges | 48.24 (0.91) [3] | 29.89 (1.01) [12] | 46.73 (1.05) [4] | 21.17 (0.94) [17] | 31.79 (1.11) [10] | 24.50 (0.95) [15] | 37.73 (0.95) [7] | 56.50 (1.45) [2] | 30.04 (1.06) [11] | 18.60 (2.88) [20] | 39.29 (1.03) [6] | 20.39 (7.33) [19] | 22.82 (1.20) [16] | 40.72 (1.57) [5] | 33.06 (0.95) [9] | 37.01 (0.82) [8] | 29.13 (0.85) [13] | 27.38 (0.99) [14] | 20.88 (0.98) [18] | 66.26 (0.86) [1] |
| MNIST-C_dotted_line | 53.17 (1.50) [3] | 38.99 (1.45) [10] | 46.79 (1.84) [5] | 30.26 (1.45) [17] | 41.26 (1.50) [8] | 31.15 (1.17) [16] | 45.05 (1.49) [7] | 62.87 (1.72) [2] | 37.70 (1.52) [12] | 25.41 (1.97) [19] | 48.04 (1.67) [4] | 22.45 (3.44) [20] | 31.88 (1.64) [15] | 45.73 (2.26) [6] | 39.13 (1.38) [9] | 38.59 (1.76) [11] | 35.72 (0.75) [13] | 35.67 (1.44) [14] | 29.09 (1.77) [18] | 68.45 (0.49) [1] |
| MNIST-C_fog | 82.86 (1.18) [3] | 68.42 (1.28) [11] | 77.90 (0.96) [5] | 58.84 (1.13) [16] | 69.03 (1.37) [10] | 58.82 (1.36) [17] | 78.50 (1.39) [4] | 89.81 (1.18) [2] | 67.77 (1.22) [12] | 38.58 (2.75) [19] | 76.98 (1.00) [6] | 23.42 (9.51) [20] | 59.89 (1.45) [14] | 75.20 (0.91) [7] | 73.10 (1.31) [8] | 72.14 (1.33) [9] | 46.11 (0.96) [18] | 67.30 (1.46) [13] | 58.87 (1.28) [15] | 90.65 (0.14) [1] |
| MNIST-C_glass_blur | 86.25 (1.42) [3] | 79.47 (1.53) [9] | 80.57 (1.78) [8] | 74.27 (1.30) [15] | 82.45 (1.00) [7] | 70.56 (0.27) [17] | 84.86 (1.04) [5] | 93.13 (1.25) [1] | 78.50 (1.45) [11] | 60.59 (5.11) [18] | 85.11 (1.26) [4] | 36.99 (17.24) [20] | 75.07 (1.41) [14] | 82.53 (1.69) [6] | 79.25 (1.16) [10] | 77.58 (1.31) [12] | 49.50 (0.40) [19] | 76.81 (1.45) [13] | 73.66 (1.42) [16] | 89.73 (0.10) [2] |
| MNIST-C_identity | 8.72 (0.10) [17] | 9.13 (0.21) [14] | 9.18 (0.22) [11] | 9.22 (0.22) [9] | 8.66 (0.13) [19] | 8.70 (0.18) [18] | 9.18 (0.17) [11] | 9.35 (0.29) [3] | 9.27 (0.22) [7] | 9.15 (0.17) [13] | 9.36 (0.11) [2] | 9.52 (0.40) [1] | 9.23 (0.20) [8] | 8.82 (0.14) [16] | 9.22 (0.14) [9] | 9.34 (0.22) [4] | 9.04 (0.19) [15] | 9.30 (0.23) [6] | 9.33 (0.19) [5] | 8.38 (0.04) [20] |
| MNIST-C_impulse_noise | 99.79 (0.05) [3] | 99.51 (0.19) [11] | 99.51 (0.19) [11] | 96.39 (0.30) [16] | 99.62 (0.18) [6] | 99.45 (0.31) [13] | 99.09 (1.11) [15] | 99.92 (0.19) [9] | 94.27 (2.73) [17] | 99.76 (0.12) [4] | 68.19 (33.26) [19] | 99.52 (0.18) [9] | 99.67 (0.35) [5] | 99.91 (0.01) [1] | 91.72 (0.30) [18] | 51.96 (0.45) [20] | 99.83 (0.04) [2] | 99.62 (0.13) [6] | 99.60 (0.01) [8] | |
| MNIST-C_motion_blur | 78.83 (1.53) [3] | 63.55 (1.87) [11] | 70.00 (1.80) [6] | 54.58 (1.83) [17] | 66.45 (1.81) [8] | 58.57 (2.23) [14] | 69.66 (1.71) [7] | 80.60 (1.49) [2] | 62.34 (1.81) [12] | 35.01 (3.08) [19] | 72.41 (1.60) [5] | 31.19 (8.98) [20] | 55.91 (1.58) [15] | 74.60 (2.19) [4] | 64.17 (1.71) [10] | 64.26 (1.62) [9] | 42.37 (0.79) [18] | 60.87 (1.85) [13] | 55.01 (1.64) [16] | 86.29 (0.32) [1] |
| MNIST-C_rotate | 16.56 (0.53) [3] | 13.46 (0.45) [10] | 14.72 (0.42) [6] | 11.45 (0.37) [19] | 13.58 (0.44) [9] | 12.13 (0.35) [16] | 14.54 (0.55) [7] | 18.21 (0.57) [2] | 13.03 (0.40) [12] | 10.90 (0.37) [20] | 14.91 (0.39) [5] | 12.16 (0.91) [15] | 11.63 (0.41) [17] | 15.28 (0.52) [4] | 12.66 (0.54) [13] | 14.09 (0.42) [8] | 12.52 (0.39) [14] | 11.52 (0.45) [18] | 19.90 (0.12) [1] | |
| MNIST-C_scale | 39.67 (1.06) [3] | 28.35 (1.01) [10] | 36.03 (1.46) [4] | 22.03 (0.81) [16] | 29.55 (0.93) [9] | 20.15 (0.70) [18] | 35.42 (1.02) [5] | 44.36 (1.47) [2] | 28.05 (0.94) [11] | 16.75 (2.68) [19] | 29.96 (1.06) [8] | 16.15 (6.16) [20] | 23.38 (1.13) [15] | 31.25 (1.13) [7] | 26.41 (0.94) [14] | 33.28 (0.93) [6] | 27.92 (0.81) [12] | 26.60 (0.88) [13] | 21.27 (0.93) [17] | 49.88 (1.05) [1] |
| MNIST-C_shear | 27.63 (1.41) [3] | 21.57 (1.14) [11] | 24.70 (1.32) [4] | 18.50 (0.95) [16] | 22.24 (1.27) [9] | 17.44 (0.49) [18] | 24.28 (1.32) [5] | 28.70 (1.95) [2] | 21.49 (1.13) [12] | 16.84 (1.42) [19] | 23.64 (1.35) [7] | 12.41 (1.88) [20] | 18.98 (1.04) [15] | 23.76 (0.88) [6] | 21.75 (0.93) [10] | 23.13 (1.12) [8] | 20.12 (1.07) [14] | 20.08 (1.17) [17] | 17.54 (0.75) [17] | 33.37 (0.19) [1] |
| MNIST-C_shot_noise | 52.49 (1.94) [3] | 38.58 (1.84) [9] | 45.77 (1.77) [4] | 29.25 (1.77) [17] | 40.22 (1.81) [8] | 29.35 (1.27) [16] | 43.75 (1.81) [6] | 61.18 (2.15) [2] | 35.34 (1.89) [12] | 24.91 (2.06) [19] | 44.99 (1.84) [5] | 22.39 (4.42) [20] | 30.35 (1.83) [15] | 37.61 (1.85) [10] | 35.87 (1.60) [11] | 41.69 (1.80) [7] | 34.30 (1.06) [13] | 33.38 (1.70) [14] | 27.94 (1.68) [18] | 65.57 (0.25) [1] |
| MNIST-C_spatter | 66.76 (1.49) [3] | 53.34 (1.33) [10] | 61.77 (0.76) [6] | 44.51 (1.54) [15] | 56.36 (1.72) [8] | 41.97 (1.28) [17] | 64.94 (1.31) [4] | 75.33 (2.02) [2] | 52.65 (1.34) [11] | 40.87 (5.74) [18] | 64.59 (1.03) [5] | 23.05 (8.01) [20] | 45.64 (1.11) [14] | 57.20 (1.48) [7] | 54.49 (1.35) [9] | 51.96 (1.29) [12] | 40.24 (1.22) [19] | 49.87 (1.68) [13] | 43.24 (1.76) [16] | 75.39 (0.60) [1] |
| MNIST-C_stripe | 98.80 (0.28) [4] | 98.34 (0.32) [7] | 98.72 (0.14) [5] | 97.99 (0.36) [12] | 97.25 (0.56) [15] | 97.15 (0.72) [16] | 97.99 (0.79) [12] | 98.28 (2.52) [8] | 98.28 (0.32) [8] | 92.32 (1.02) [17] | 98.99 (0.26) [1] | 46.92 (23.14) [20] | 98.06 (0.38) [11] | 98.87 (0.50) [3] | 98.57 (0.33) [6] | 90.44 (1.41) [18] | 51.15 (0.63) [19] | 98.24 (0.32) [10] | 97.95 (0.40) [14] | 98.95 (0.03) [2] |
| MNIST-C_translate | 34.33 (0.84) [2] | 19.29 (0.55) [12] | 26.03 (0.68) [7] | 14.30 (0.49) [19] | 20.61 (0.66) [10] | 16.79 (0.59) [15] | 26.29 (0.73) [5] | 33.24 (0.53) [3] | 18.54 (0.53) [13] | 12.39 (0.57) [20] | 26.14 (0.80) [6] | 14.90 (1.65) [16] | 14.87 (0.51) [17] | 28.78 (1.53) [4] | 21.75 (0.67) [9] | 24.55 (0.39) [8] | 19.76 (0.51) [11] | 18.17 (0.54) [14] | 14.50 (0.78) [18] | 45.39 (0.25) [1] |
| MNIST-C_zigzag | 68.80 (1.74) [3] | 52.91 (1.72) [10] | 59.48 (1.30) [7] | 42.97 (1.78) [15] | 55.52 (1.68) [8] | 39.09 (1.53) [18] | 54.79 (1.71) [9] | 73.53 (1.46) [2] | 55.11 (1.35) [11] | 33.55 (3.51) [19] | 62.75 (1.91) [4] | 24.48 (5.30) [20] | 44.83 (1.45) [14] | 60.80 (1.47) [5] | 53.10 (1.84) [9] | 51.21 (1.15) [12] | 44.75 (0.49) [15] | 49.37 (1.79) [13] | 41.72 (1.88) [17] | 80.55 (0.20) [1] |
| MVTec-AD_bottle | 98.90 (0.67) [5] | 96.98 (1.04) [12] | 96.95 (0.96) [13] | 96.37 (1.03) [15] | 99.06 (0.63) [3] | 98.53 (0.62) [7] | 99.17 (0.26) [1] | 98.77 (0.64) [6] | 97.02 (0.87) [10] | 96.84 (1.09) [14] | 37.00 (2.63) [20] | 89.96 (6.53) [19] | 96.34 (1.23) [16] | 98.46 (0.47) [8] | 90.01 (0.50) [4] | 97.58 (0.53) [9] | 94.04 (4.05) [18] | 96.97 (0.94) [11] | 95.27 (1.17) [17] | 99.11 (0.42) [2] |
| MVTec-AD_cable | 83.18 (3.61) [4] | 65.96 (4.34) [13] | 66.24 (4.14) [12] | 60.77 (4.29) [17] | 83.73 (3.83) [3] | 79.89 (4.79) [8] | 83.22 (3.91) [2] | 77.86 (4.45) [11] | 62.29 (4.84) [16] | 38.26 (2.20) [20] | 46.30 (9.59) [19] | 62.46 (3.93) [15] | 82.39 (2.74) [6] | 83.22 (2.62) [2] | 80.45 (3.59) [7] | 73.80 (6.01) [9] | 65.64 (5.22) [14] | 57.07 (4.35) [18] | 83.01 (4.12) [5] | |
| MVTec-AD_capsule | 89.36 (1.80) [5] | 73.51 (2.04) [11] | 73.40 (2.73) [12] | 67.94 (2.13) [17] | 91.10 (2.06) [3] | 89.87 (1.71) [4] | 91.30 (1.80) [2] | 82.41 (3.00) [8] | 71.76 (2.51) [13] | 69.13 (1.82) [15] | 44.79 (2.04) [20] | 59.99 (5.24) [19] | 68.53 (2.03) [16] | 88.49 (2.52) [6] | 88.46 (1.90) [7] | 80.54 (1.44) [9] | 79.04 (3.63) [10] | 71.32 (1.84) [14] | 66.16 (2.59) [18] | 91.64 (1.76) [1] |
| MVTec-AD_carpet | 85.89 (2.37) [6] | 72.34 (4.19) [11] | 71.58 (4.02) [14] | 70.64 (4.26) [16] | 86.82 (1.75) [4] | 86.91 (1.77) [3] | 78.18 (5.52) [9] | 71.93 (3.59) [12] | 69.43 (4.84) [17] | 40.83 (4.24) [20] | 58.80 (3.41) [19] | 71.34 (4.21) [15] | 87.80 (2.96) [1] | 85.58 (2.44) [5] | 80.90 (3.57) [8] | 76.41 (4.09) [10] | 66.46 (4.51) [18] | 87.74 (1.54) [2] | | |
| MVTec-AD_grid | 78.83 (3.91) [6] | 61.25 (2.29) [11] | 59.67 (2.71) [13] | 56.23 (1.88) [16] | 80.18 (4.06) [4] | 80.68 (3.98) [3] | 78.89 (4.40) [5] | 72.75 (6.02) [8] | 59.84 (1.89) [12] | 50.99 (1.47) [18] | 29.43 (2.60) [20] | 50.72 (5.64) [19] | 57.49 (1.77) [15] | 82.77 (2.63) [1] | 81.20 (2.92) [2] | 70.58 (3.18) [7] | 70.65 (3.46) [9] | 59.68 (5.40) [14] | 55.31 (2.32) [17] | 81.20 (2.92) [2] |
| MVTec-AD_hazelnut | 77.76 (2.25) [3] | 62.94 (3.03) [12] | 62.90 (3.43) [11] | 60.55 (3.26) [16] | 78.11 (3.02) [2] | 77.44 (2.55) [5] | 77.61 (2.04) [4] | 73.48 (2.32) [9] | 63.26 (3.50) [10] | 62.47 (3.52) [13] | 33.77 (4.46) [20] | 44.36 (11.23) [19] | 61.42 (3.04) [15] | 80.93 (5.41) [1] | 77.23 (4.32) [6] | 74.48 (4.47) [8] | 54.59 (3.03) [18] | 62.22 (3.61) [14] | 59.67 (4.00) [17] | 76.42 (5.11) [7] |
| MVTec-AD_leather | 99.49 (0.29) [3] | 99.03 (0.43) [9] | 98.48 (0.73) [16] | 98.88 (0.42) [12] | 99.26 (0.53) [8] | 99.31 (0.30) [7] | 99.43 (0.34) [4] | 98.84 (0.77) [14] | 56.87 (3.94) [20] | 65.19 (14.21) [19] | 98.90 (0.43) [10] | 96.56 (1.47) [15] | 99.51 (0.26) [1] | 99.30 (0.32) [5] | 99.10 (0.25) [6] | 84.98 (0.81) [18] | 97.71 (0.51) [17] | 99.38 (0.34) [6] | | |
| MVTec-AD_metal_nut | 89.88 (2.59) [4] | 71.87 (2.47) [12] | 72.89 (3.02) [11] | 68.23 (2.88) [16] | 89.67 (2.47) [5] | 88.79 (1.94) [6] | 90.21 (2.75) [3] | 82.36 (3.14) [9] | 70.90 (3.00) [14] | 64.65 (3.10) [18] | 41.61 (0.97) [20] | 54.44 (5.84) [19] | 69.24 (2.85) [15] | 90.27 (1.70) [2] | 87.86 (2.53) [7] | 85.15 (3.13) [8] | 74.11 (4.09) [10] | 71.27 (2.20) [13] | 66.98 (3.99) [17] | 90.59 (2.26) [1] |
| MVTec-AD_pill | 86.13 (3.99) [5] | 70.84 (2.38) [12] | 71.56 (3.12) [11] | 67.18 (2.00) [17] | 87.15 (4.51) [3] | 85.43 (3.75) [7] | 88.15 (4.16) [2] | 70.52 (2.14) [13] | 68.17 (1.90) [15] | 47.82 (2.65) [20] | 62.14 (5.84) [19] | 67.71 (1.78) [16] | 86.66 (4.04) [4] | 87.86 (2.53) [7] | 85.15 (3.13) [8] | 80.90 (2.24) [9] | 79.18 (2.38) [17] | 62.33 (3.10) [18] | 89.28 (2.32) [1] | |
| MVTec-AD_screw | 76.34 (2.22) [6] | 57.98 (0.89) [11] | 57.95 (1.76) [12] | 54.32 (0.73) [16] | 77.72 (1.63) [3] | 78.54 (1.38) [2] | 76.45 (2.34) [5] | 68.60 (4.22) [8] | 56.05 (1.19) [14] | 48.53 (4.02) [19] | 38.73 (2.59) [20] | 51.13 (6.80) [18] | 54.33 (1.30) [15] | 79.69 (2.15) [1] | 74.69 (1.36) [7] | 67.92 (2.12) [9] | 58.21 (2.90) [10] | 56.30 (1.67) [13] | 52.10 (2.68) [17] | 77.38 (1.75) [4] |
| MVTec-AD_tile | 92.70 (2.05) [2] | 82.51 (1.23) [13] | 81.47 (1.42) [15] | 81.64 (1.11) [14] | 91.42 (3.60) [5] | 90.44 (2.29) [6] | 92.43 (1.49) [3] | 87.61 (2.99) [9] | 82.92 (1.35) [11] | 82.90 (1.43) [12] | 42.30 (2.29) [20] | 57.73 (13.81) [18] | 81.40 (1.68) [16] | 89.78 (4.15) [7] | 92.75 (2.26) [1] | 89.69 (3.56) [8] | 81.25 (1.36) [17] | 92.05 (1.13) [4] | 77.17 (1.06) [18] | 92.06 (2.26) [4] |
| MVTec-AD_toothbrush | 99.92 (0.07) [3] | 95.36 (2.07) [11] | 92.79 (2.09) [13] | 82.57 (1.02) [17] | 99.90 (0.15) [4] | 99.87 (0.20) [6] | 99.89 (0.14) [5] | 99.01 (0.26) [8] | 89.61 (1.76) [15] | 92.92 (3.76) [12] | 31.63 (1.58) [20] | 60.36 (12.25) [19] | 84.68 (1.00) [16] | 99.93 (0.09) [2] | 99.87 (0.17) [6] | 98.39 (2.10) [9] | 92.85 (1.13) [10] | 57.77 (1.06) [18] | 99.94 (0.08) [1] | |
| MVTec-AD_transistor | 79.34 (5.67) [5] | 59.36 (5.56) [12] | 59.43 (5.12) [10] | 56.34 (5.57) [15] | 81.08 (5.75) [1] | 80.94 (6.27) [2] | 77.43 (6.23) [7] | 76.59 (6.66) [8] | 59.41 (5.04) [11] | 54.65 (8.56) [17] | 22.65 (2.12) [20] | 44.41 (15.43) [19] | 56.68 (5.19) [14] | 80.52 (3.78) [3] | 78.87 (5.27) [6] | 75.28 (5.18) [9] | 55.53 (3.63) [16] | 58.90 (5.42) [13] | 52.66 (5.54) [18] | 80.37 (4.48) [4] |
| MVTec-AD_wood | 90.97 (2.94) [5] | 76.52 (2.05) [11] | 77.09 (2.58) [10] | 73.64 (1.66) [15] | 90.68 (3.73) [6] | 91.27 (3.44) [3] | 91.04 (2.67) [4] | 84.95 (2.42) [9] | 79.03 (2.54) [8] | 70.06 (2.23) [18] | 34.53 (1.19) [20] | 64.30 (11.33) [19] | 73.87 (1.76) [14] | 91.09 (4.61) [2] | 89.28 (3.33) [7] | 85.35 (3.22) [8] | 70.31 (6.75) [17] | 72.08 (1.94) [16] | 91.09 (3.18) [2] | |
| MVTec-AD_zipper | 95.33 (1.36) [3] | 84.47 (3.58) [12] | 86.66 (3.30) [10] | 78.84 (3.64) [16] | 94.71 (1.91) [5] | 93.14 (2.71) [7] | 94.04 (1.11) [6] | 92.11 (1.91) [9] | 84.19 (3.43) [14] | 77.66 (3.29) [17] | 56.90 (4.01) [20] | 62.28 (6.70) [19] | 93.00 (1.49) [8] | 86.66 (4.73) [10] | 84.32 (3.43) [13] | 70.16 (4.39) [18] | 96.13 (0.92) [1] | | | |
| SVHN_0 | 16.93 (1.18) [7] | 15.50 (1.11) [11] | 17.54 (1.25) [3] | 14.41 (0.95) [16] | 15.64 (1.07) [9] | 13.11 (0.91) [18] | 17.05 (1.20) [5] | 18.00 (1.37) [1] | 15.43 (1.07) [12] | 12.82 (0.41) [19] | 17.00 (1.32) [6] | 11.70 (1.28) [20] | 14.62 (0.96) [15] | 17.34 (1.19) [4] | 15.57 (1.16) [10] | 16.21 (1.01) [8] | 15.03 (0.91) [14] | 15.19 (1.08) [13] | 13.53 (0.57) [17] | 17.66 (0.19) [2] |

Continued on next page

| Model | NCSBAD | KNN | LOF | PCA | KPCA | KDE | GMM | LUNAR | AE | EIF | ABOD | DROCC | GOAD | ICL | SLAD | DTE-C | DTENonParametric | MCM | DDPM | DenoiserAD (Ours) |
|---|---|---|---|---|---|---|---|---|---|---|---|---|---|---|---|---|---|---|---|---|
| SVHN 1 | 17.36 (0.93) [4] | 16.51 (0.88) [9] | 14.76 (0.66) [17] | 16.65 (0.85) [8] | 16.68 (0.87) [7] | 13.55 (0.84) [19] | 17.55 (1.01) [3] | **18.74 (0.81) [1]** | 16.32 (0.94) [12] | 16.11 (0.81) [13] | 16.73 (0.71) [6] | 11.41 (1.46) [20] | 16.46 (1.00) [10] | 13.68 (0.53) [18] | 15.08 (0.69) [15] | 17.36 (0.98) [4] | 14.91 (0.55) [16] | 16.46 (0.91) [10] | 15.54 (0.67) [14] | 18.32 (0.11) [2] |
| SVHN 2 | 19.38 (1.28) [3] | 17.97 (1.23) [9] | 18.11 (1.01) [8] | 17.32 (1.24) [15] | 18.24 (1.30) [6] | 16.58 (1.09) [18] | 18.85 (1.23) [4] | 20.69 (1.33) [2] | 17.83 (1.20) [10] | 17.06 (1.16) [16] | 18.84 (1.00) [5] | 12.25 (0.89) [20] | 17.39 (1.18) [14] | 17.45 (1.50) [13] | 18.20 (1.29) [7] | 17.67 (1.33) [11] | 15.05 (0.45) [19] | 17.50 (1.12) [12] | 16.78 (1.28) [17] | **20.92 (0.10) [1]** |
| SVHN 3 | 16.58 (0.96) [2] | 15.51 (0.79) [9] | **17.43 (1.86) [1]** | 14.68 (0.71) [14] | 15.33 (0.71) [12] | 14.02 (0.87) [17] | 16.42 (0.96) [4] | 16.02 (0.90) [8] | 15.26 (0.35) [11] | 13.11 (0.68) [19] | 16.36 (1.05) [5] | 11.42 (1.27) [20] | 14.67 (0.67) [15] | 16.53 (1.42) [3] | 15.48 (0.92) [11] | 13.94 (0.26) [18] | 13.94 (0.36) [18] | 14.32 (0.78) [16] | 16.36 (0.05) [3] | |
| SVHN 4 | 16.43 (0.42) [4] | 15.39 (0.29) [9] | 14.80 (0.60) [15] | 15.06 (0.27) [12] | 15.76 (0.35) [7] | 13.17 (0.62) [19] | 15.85 (0.43) [6] | 18.12 (0.59) [1] | 15.26 (0.35) [11] | 15.98 (0.71) [5] | 16.74 (0.62) [3] | 11.03 (1.14) [20] | 15.04 (0.30) [13] | 13.77 (0.58) [17] | 15.70 (0.40) [8] | 15.28 (0.45) [10] | 13.59 (0.42) [18] | 14.93 (0.34) [14] | 14.46 (0.46) [16] | 17.04 (0.08) [2] |
| SVHN 5 | 18.47 (0.86) [6] | 16.99 (0.78) [13] | 18.52 (1.07) [4] | 16.30 (0.67) [15] | 17.06 (0.58) [11] | 14.36 (1.16) [3] | 17.30 (0.86) [9] | 17.17 (0.86) [11] | 14.50 (0.61) [18] | 18.32 (0.74) [5] | 11.65 (0.92) [20] | 16.68 (0.81) [14] | 19.33 (1.56) [2] | 17.45 (0.94) [8] | 17.97 (0.53) [7] | 19.06 (0.74) [16] | 15.95 (0.74) [16] | 19.77 (0.05) [1] | | |
| SVHN 6 | 14.42 (0.78) [3] | 13.28 (0.76) [11] | 13.78 (0.58) [6] | 12.90 (0.80) [15] | 13.59 (0.78) [7] | 13.08 (0.81) [12] | 14.39 (0.75) [4] | 14.09 (0.72) [5] | 13.36 (0.74) [10] | 11.52 (0.55) [20] | 14.79 (0.70) [2] | 11.60 (0.92) [19] | 13.03 (0.76) [13] | 12.88 (0.52) [16] | 13.55 (0.71) [8] | 12.35 (0.42) [18] | 13.01 (0.71) [14] | 12.86 (0.08) [17] | 15.06 (0.08) [1] | |
| SVHN 7 | 17.33 (1.15) [11] | 17.31 (1.01) [12] | 17.31 (0.96) [12] | 17.58 (1.03) [6] | 17.43 (1.12) [8] | 16.70 (1.08) [16] | 17.81 (1.18) [4] | 18.70 (0.80) [2] | 17.56 (1.19) [5] | 16.73 (1.28) [15] | 17.95 (1.23) [3] | 11.56 (0.53) [20] | 17.43 (1.00) [8] | 16.92 (1.39) [14] | 19.26 (1.44) [1] | 17.73 (0.14) [5] | 17.42 (1.06) [10] | 16.66 (1.23) [17] | 17.73 (0.14) [5] | |
| SVHN 8 | 15.80 (1.05) [8] | 15.15 (0.92) [10] | 16.19 (0.93) [3] | 14.35 (0.90) [17] | 14.99 (0.87) [12] | 14.95 (0.72) [13] | 15.14 (0.97) [11] | 16.02 (1.23) [6] | 15.37 (0.96) [9] | 11.77 (0.90) [19] | 16.15 (1.11) [4] | 11.40 (1.18) [20] | 14.44 (0.89) [15] | 16.51 (0.70) [2] | 15.99 (0.73) [7] | 16.13 (1.04) [5] | 12.32 (0.69) [18] | 14.91 (0.99) [14] | 14.42 (0.76) [16] | 16.55 (0.12) [1] |
| SVHN 9 | 13.47 (1.16) [7] | 12.56 (1.08) [12] | 13.49 (1.14) [6] | 12.08 (1.05) [17] | 12.98 (1.17) [8] | 12.06 (0.96) [18] | 13.63 (1.20) [3] | **13.96 (1.35) [1]** | 12.64 (1.14) [11] | 11.20 (1.05) [19] | 13.61 (1.06) [5] | 10.50 (0.61) [20] | 12.26 (1.04) [15] | 12.89 (1.04) [10] | 12.98 (1.18) [8] | 12.27 (0.89) [14] | 12.47 (1.13) [13] | 12.26 (1.11) [15] | 13.78 (0.11) [3] | |
| agnews 0 | 19.54 (0.48) [2] | 13.66 (1.70) [5] | 10.03 (0.63) [18] | 13.52 (0.26) [11] | 14.25 (0.26) [7] | 11.25 (0.56) [15] | 17.16 (0.40) [4] | 14.14 (0.41) [3] | 10.74 (0.18) [17] | 16.01 (0.45) [5] | 9.80 (0.56) [20] | 10.82 (0.37) [16] | 13.46 (0.63) [9] | 14.99 (0.47) [6] | 13.72 (0.22) [10] | 13.19 (0.13) [14] | 10.03 (0.54) [18] | 23.61 (0.48) [1] | | |
| agnews 1 | 27.82 (0.42) [3] | 14.87 (0.39) [13] | 31.50 (1.07) [2] | 11.39 (0.27) [18] | 16.87 (0.41) [7] | 14.37 (0.61) [14] | 23.32 (0.28) [4] | 17.09 (0.41) [6] | 16.12 (0.49) [9] | 11.62 (0.49) [17] | 18.81 (0.43) [5] | 9.52 (0.00) [20] | 11.94 (0.28) [15] | 16.48 (1.08) [8] | 11.63 (0.56) [16] | 15.03 (0.35) [12] | 15.26 (0.32) [11] | 15.42 (0.61) [10] | 10.84 (0.71) [19] | 35.13 (1.08) [1] |
| agnews 2 | 33.19 (0.37) [2] | 21.86 (0.54) [12] | 29.75 (1.72) [3] | 14.26 (0.46) [17] | 23.44 (0.49) [9] | 17.72 (0.55) [13] | 28.60 (0.37) [4] | 24.20 (0.57) [6] | 15.40 (0.36) [15] | 27.15 (1.07) [5] | 10.32 (0.69) [20] | 15.70 (0.45) [14] | 15.11 (0.38) [16] | 13.08 (0.24) [19] | 27.82 (0.28) [7] | 22.63 (0.48) [10] | 23.49 (0.49) [8] | 13.52 (0.58) [18] | 42.47 (0.92) [1] | |
| agnews 3 | 31.67 (1.42) [2] | 16.87 (0.63) [12] | 24.48 (1.37) [3] | 11.40 (0.29) [18] | 18.71 (0.65) [7] | 14.32 (0.68) [14] | 23.51 (0.99) [4] | 18.71 (0.86) [7] | 17.15 (0.45) [10] | 11.90 (0.34) [17] | 22.31 (1.08) [5] | 9.73 (0.26) [20] | 12.11 (0.14) [16] | 15.46 (0.97) [13] | 12.18 (0.28) [15] | 19.11 (0.50) [6] | 17.07 (0.67) [11] | 18.14 (0.45) [9] | 11.09 (0.20) [19] | 38.86 (0.69) [1] |
| amazon | 12.35 (0.75) [4] | 11.90 (0.59) [7] | 11.01 (0.50) [14] | 10.87 (0.68) [15] | 11.59 (0.58) [9] | 11.76 (0.55) [16] | 12.00 (0.66) [6] | 11.59 (0.58) [9] | 11.16 (0.63) [13] | 12.55 (0.65) [1] | 9.52 (0.00) [20] | 11.19 (0.67) [12] | 11.90 (0.56) [7] | 11.71 (0.56) [8] | 12.33 (0.86) [3] | 10.58 (0.69) [17] | 12.53 (0.06) [2] | | | |
| imdb | 9.04 (0.20) [10] | 8.96 (0.16) [13] | 9.10 (0.13) [8] | 8.72 (0.15) [20] | 9.07 (0.15) [9] | 8.80 (0.22) [18] | 8.94 (0.19) [14] | 8.98 (0.23) [12] | 9.19 (0.17) [6] | 9.02 (0.20) [11] | 9.26 (0.15) [4] | 9.99 (0.43) [3] | 8.86 (0.22) [16] | 10.43 (0.36) [1] | 10.10 (0.30) [2] | 8.84 (0.21) [17] | 8.90 (0.13) [15] | 9.23 (0.19) [5] | 8.75 (0.24) [19] | 9.19 (0.06) [6] |
| yelp | 17.94 (1.13) [1] | 16.27 (1.13) [7] | 15.91 (0.98) [9] | 13.34 (0.85) [15] | 16.54 (1.14) [5] | 13.04 (0.48) [16] | 16.74 (1.02) [3] | 16.72 (0.96) [4] | 15.66 (0.83) [11] | 13.69 (0.82) [14] | 16.14 (0.62) [8] | 10.23 (0.81) [19] | 13.94 (0.95) [13] | 10.35 (0.29) [18] | 10.12 (0.16) [20] | 13.97 (0.61) [12] | 16.43 (1.13) [6] | 15.80 (0.88) [10] | 12.36 (0.83) [17] | 17.64 (0.45) [2] |
| 20news 0 | 30.60 (3.03) [3] | 26.96 (1.96) [10] | 40.52 (0.86) [5] | 3.18 (0.05) [20] | 20.70 (1.26) [11] | 37.92 (2.14) [6] | 10.58 (0.57) [17] | 23.98 (0.77) [15] | 16.69 (0.53) [8] | 10.69 (2.80) [16] | 29.94 (0.90) [9] | 16.54 (4.69) [13] | 5.02 (0.65) [18] | 42.27 (6.31) [4] | 33.48 (1.65) [7] | 32.06 (1.90) [8] | 18.81 (1.21) [12] | 4.82 (0.94) [19] | 53.12 (3.49) [2] | |
| 20news 1 | 15.14 (2.49) [2] | 11.15 (1.26) [13] | 10.79 (0.95) [15] | 8.96 (0.33) [20] | 12.03 (1.43) [10] | 11.59 (1.37) [11] | 12.66 (1.17) [7] | 12.60 (1.46) [8] | 12.13 (1.21) [9] | 9.23 (0.61) [18] | 11.56 (0.69) [12] | 10.09 (1.92) [16] | 9.13 (0.26) [19] | 13.92 (0.99) [3] | 12.67 (0.57) [6] | 11.12 (1.03) [14] | 13.26 (0.79) [4] | 9.49 (0.60) [17] | 16.35 (0.26) [1] | |
| 20news 2 | 11.98 (0.77) [3] | 9.19 (0.29) [14] | 10.14 (0.52) [7] | 8.98 (0.62) [17] | 9.88 (0.43) [9] | 10.31 (0.72) [5] | 11.17 (0.70) [4] | 9.46 (0.27) [12] | 8.78 (0.52) [20] | 9.80 (0.62) [10] | 10.17 (0.99) [6] | 8.93 (0.63) [19] | 10.05 (0.57) [8] | 9.60 (0.41) [11] | 12.13 (1.03) [2] | 9.18 (0.31) [15] | 9.23 (0.22) [13] | 9.04 (0.67) [16] | 12.46 (0.25) [1] | |
| 20news 3 | 38.79 (7.78) [3] | 23.51 (6.47) [9] | 20.96 (4.91) [12] | 17.62 (4.46) [17] | 39.87 (10.43) [2] | 32.62 (8.36) [5] | 41.21 (9.73) [1] | 29.39 (5.96) [7] | 21.09 (4.48) [11] | 21.30 (9.14) [10] | 10.96 (1.70) [20] | 14.93 (4.56) [18] | 18.57 (5.01) [15] | 20.34 (3.55) [13] | 19.34 (3.40) [14] | 29.74 (8.80) [6] | 17.81 (4.69) [16] | 14.50 (3.29) [19] | 36.22 (6.44) [4] | |
| 20news 4 | 16.47 (1.85) [1] | 11.20 (1.51) [18] | 12.83 (1.53) [9] | 11.72 (1.33) [15] | 13.07 (1.60) [6] | 11.52 (1.42) [17] | 15.78 (1.60) [4] | 11.61 (1.55) [16] | 12.49 (1.52) [11] | 11.94 (0.80) [13] | 10.67 (1.66) [20] | 11.75 (1.34) [14] | 16.04 (0.88) [2] | 10.86 (1.31) [19] | 12.99 (1.24) [10] | 13.01 (2.32) [7] | 16.00 (0.13) [3] | | | |
| 20news 5 | 20.14 (2.38) [2] | 11.99 (1.83) [15] | 13.72 (1.87) [7] | 11.96 (1.54) [16] | 13.33 (1.78) [8] | 13.06 (1.14) [9] | 16.06 (2.28) [3] | 13.04 (1.57) [10] | 13.00 (1.52) [11] | 11.84 (1.66) [18] | 11.30 (0.46) [19] | 12.06 (0.97) [14] | 11.86 (1.53) [17] | 14.47 (1.87) [4] | 14.04 (2.85) [5] | 10.83 (0.64) [20] | 12.74 (1.58) [12] | 12.17 (2.00) [13] | 20.36 (0.46) [1] | |
| Average | 53.98 (31.99) [3] | 49.35 (31.30) [8] | 49.01 (28.75) [9] | 43.40 (30.43) [14] | 53.23 (33.19) [4] | 49.62 (33.60) [7] | 50.54 (31.36) [6] | 55.49 (32.62) [2] | 47.67 (30.29) [12] | 41.59 (30.62) [17] | 37.29 (26.86) [19] | 30.53 (22.92) [20] | 42.44 (30.42) [16] | 51.51 (32.85) [5] | 48.56 (33.05) [11] | 48.60 (29.45) [10] | 43.14 (28.74) [15] | 47.11 (29.83) [13] | 41.28 (29.29) [18] | **56.62 (31.91) [1]** |

*Table 8.* Full results, including mean, standard deviation and rank AUCPR performance of all methods for each ADBench dataset and the overall average. The best performance is highlighted in **bold**, the second best highlighted with underline.

| Model | NCSBAD | KNN | LOF | PCA | KPCA | KDE | GMM | LUNAR | AE | EIF | ABOD | DROCC | GOAD | ICL | SLAD | DTE-C | DTENonParametric | MCM | DDPM | DenoiserAD (Ours) |
|---|---|---|---|---|---|---|---|---|---|---|---|---|---|---|---|---|---|---|---|---|
| ALOI | 2.91 (0.15) [12] | 2.95 (0.14) [11] | 3.43 (0.20) [4] | 3.48 (0.19) [2] | 2.84 (0.13) [15] | 3.30 (0.16) [5] | 3.45 (0.18) [3] | **3.70 (0.18) [1]** | 3.13 (0.15) [8] | 2.96 (0.18) [10] | 3.21 (0.10) [6] | 2.86 (0.09) [14] | 2.40 (0.20) [19] | 2.38 (0.10) [20] | 2.80 (0.12) [16] | 2.73 (0.13) [17] | 2.89 (0.14) [13] | 2.72 (0.10) [18] | 2.98 (0.19) [9] | 3.15 (0.05) [7] |
| annthyroid | 41.77 (0.92) [19] | 64.61 (1.35) [4] | 49.77 (1.53) [15] | 52.03 (1.69) [13] | 62.68 (1.40) [5] | 57.32 (1.66) [9] | 51.51 (1.51) [14] | 48.65 (1.07) [16] | 52.25 (2.09) [12] | 55.91 (2.55) [11] | 57.01 (2.03) [10] | 61.88 (5.34) [6] | 58.25 (5.68) [8] | 36.54 (4.27) [20] | 68.36 (1.71) [2] | **79.97 (1.09) [1]** | 60.32 (1.18) [7] | 66.84 (1.06) [3] | 42.94 (3.26) [18] | 43.79 (2.71) [17] |
| backdoor | 84.40 (1.70) [5] | 44.74 (1.84) [11] | 52.09 (1.97) [10] | 5.64 (0.21) [19] | 81.49 (2.64) [7] | 40.98 (2.62) [12] | 86.09 (1.45) [4] | 89.18 (2.08) [1] | 54.95 (1.34) [15] | 9.33 (2.24) [17] | 27.57 (1.22) [13] | 84.30 (3.60) [6] | 10.88 (11.73) [16] | 88.96 (1.77) [2] | 2.36 (0.09) [20] | 60.30 (2.23) [8] | 50.70 (0.23) [8] | 42.40 (1.19) [11] | 6.31 (0.15) [18] | 87.79 (1.96) [3] |
| breastw | 96.19 (1.46) [12] | 97.24 (0.55) [8] | 62.57 (14.10) [18] | 97.75 (0.34) [6] | 98.20 (0.76) [4] | 98.17 (0.81) [5] | 96.21 (1.01) [11] | 97.38 (0.72) [7] | 95.18 (1.03) [13] | 98.40 (0.22) [2] | -17.49 (3.82) [20] | 23.20 (44.87) [19] | 97.01 (0.45) [9] | 94.72 (2.56) [14] | **98.52 (0.54) [1]** | 72.10 (6.18) [17] | 81.12 (1.86) [16] | 96.78 (1.06) [10] | 92.34 (1.86) [15] | 98.37 (0.84) [3] |
| campaign | 31.06 (0.28) [17] | 41.18 (0.44) [4] | 31.61 (0.20) [16] | 41.12 (0.21) [6] | 40.97 (0.46) [7] | 38.83 (0.48) [13] | **44.03 (0.42) [1]** | 32.62 (0.66) [15] | 39.02 (0.31) [12] | 41.14 (1.13) [5] | 39.43 (0.61) [11] | 8.65 (0.00) [19] | 6.85 (5.54) [20] | 41.72 (0.94) [3] | 40.77 (0.32) [8] | 40.52 (0.23) [9] | 43.12 (0.19) [2] | 36.18 (1.09) [14] | 26.75 (1.93) [18] | |
| Cardiotocography | 31.98 (2.08) [15] | 39.35 (1.64) [8] | 39.45 (1.77) [7] | **56.92 (1.54) [1]** | 38.67 (1.53) [9] | 39.50 (1.22) [6] | 37.24 (1.37) [10] | 40.40 (1.07) [5] | 44.33 (1.77) [4] | 48.24 (3.80) [3] | 36.38 (1.14) [11] | 17.29 (11.85) [20] | 31.88 (2.51) [16] | 27.50 (1.91) [18] | 33.15 (2.36) [14] | 22.76 (1.17) [19] | 34.17 (2.02) [13] | 29.41 (1.92) [17] | 35.70 (1.38) [12] | |
| cardio | 68.78 (1.77) [12] | 74.84 (2.08) [8] | 67.04 (2.14) [14] | **84.09 (2.16) [1]** | 75.41 (1.78) [7] | 77.16 (1.65) [6] | 70.03 (2.14) [11] | 78.32 (2.10) [5] | 71.34 (1.67) [10] | 78.98 (3.88) [3] | 74.59 (1.48) [9] | 44.55 (23.25) [20] | 81.97 (2.02) [2] | 52.30 (6.76) [18] | 67.07 (2.31) [13] | 66.98 (1.99) [15] | 47.91 (1.64) [19] | 78.58 (2.57) [4] | 53.79 (1.60) [17] | 65.19 (2.04) [16] |
| celeba | 2.18 (0.75) [18] | 9.90 (0.73) [7] | 1.35 (0.09) [20] | **19.44 (1.58) [1]** | 9.67 (0.75) [8] | 7.37 (0.56) [13] | 12.44 (0.74) [3] | 4.70 (0.40) [16] | 9.14 (0.71) [9] | 13.05 (1.40) [5] | 2.32 (0.17) [17] | 5.28 (0.46) [15] | 2.02 (1.73) [19] | 8.00 (0.46) [12] | 7.13 (1.01) [14] | 13.20 (1.09) [4] | 8.38 (0.44) [11] | 15.42 (3.33) [2] | 9.12 (1.23) [10] | 11.14 (2.47) [6] |
| census | 12.90 (0.27) [11] | 15.81 (0.85) [2] | 7.50 (0.55) [18] | 14.15 (0.41) [9] | **15.99 (0.44) [1]** | 14.86 (0.61) [5] | 13.88 (0.44) [10] | 10.53 (0.35) [15] | 15.53 (0.41) [3] | 9.41 (1.06) [16] | 5.30 (0.19) [19] | 8.07 (1.14) [17] | 5.05 (3.50) [20] | 15.38 (0.46) [4] | 10.80 (4.40) [14] | 12.00 (0.49) [12] | 14.65 (0.38) [6] | 14.17 (1.08) [8] | 11.90 (0.45) [13] | 14.65 (0.26) [6] |
| cover | 82.81 (2.39) [2] | 56.05 (4.10) [9] | **83.49 (3.40) [1]** | 15.57 (0.83) [16] | 53.63 (3.55) [10] | 34.82 (2.62) [12] | 18.61 (1.00) [15] | 75.80 (2.87) [5] | 66.47 (8.65) [6] | 8.16 (1.23) [17] | 77.07 (2.87) [3] | 29.75 (4.21) [14] | 0.09 (0.06) [20] | 40.34 (20.13) [11] | 5.48 (2.85) [18] | 66.37 (8.13) [7] | 62.91 (3.69) [8] | 29.77 (9.94) [13] | 22.87 (1.19) [7] | 11.90 (0.66) [20] |
| donors | 36.48 (1.24) [14] | 88.52 (1.65) [6] | 61.83 (1.36) [9] | 31.37 (0.98) [17] | 94.91 (1.66) [4] | 68.68 (1.57) [8] | 39.37 (0.89) [13] | **99.23 (0.73) [1]** | 57.78 (5.94) [10] | 48.53 (1.32) [11] | 0.48 (0.08) [20] | 32.32 (14.03) [16] | 6.53 (9.13) [19] | 44.06 (10.06) [12] | 72.18 (1.71) [7] | 92.33 (1.48) [5] | 95.38 (2.55) [3] | 23.66 (2.97) [18] | 32.96 (1.74) [15] | |
| fault | 13.29 (2.23) [16] | 18.51 (2.53) [11] | -6.95 (2.56) [20] | 15.60 (3.80) [14] | 23.76 (2.56) [4] | 19.63 (2.42) [9] | 20.47 (2.76) [7] | 19.73 (2.99) [8] | 27.65 (2.97) [2] | 16.15 (3.19) [12] | 7.17 (2.51) [19] | 15.33 (6.75) [15] | 19.49 (3.14) [10] | 22.97 (3.69) [5] | **28.14 (1.57) [1]** | 25.96 (1.67) [3] | 15.86 (2.33) [13] | 11.27 (3.11) [17] | 11.25 (3.89) [18] | 20.61 (0.69) [6] |
| fraud | 52.72 (10.94) [6] | 42.98 (6.58) [11] | 60.98 (5.43) [3] | 29.43 (3.80) [15] | 36.48 (8.10) [13] | 36.21 (7.36) [14] | 61.05 (7.24) [2] | 51.07 (8.11) [7] | 47.30 (5.53) [9] | 20.68 (4.80) [17] | 11.94 (2.86) [18] | 0.17 (0.03) [20] | 25.16 (18.13) [16] | 53.64 (5.65) [4] | 6.18 (0.93) [19] | 41.02 (10.76) [12] | **71.32 (4.06) [1]** | 49.60 (7.47) [8] | | |
| glass | 77.65 (8.13) [4] | 43.54 (10.33) [8] | 41.39 (11.15) [10] | 21.39 (6.63) [18] | 80.90 (10.08) [3] | 48.05 (10.36) [6] | 28.03 (9.23) [16] | 65.05 (7.24) [5] | 42.47 (8.11) [9] | 26.08 (8.90) [17] | 2.55 (1.09) [20] | 28.26 (13.19) [15] | 18.58 (8.77) [19] | **93.08 (4.00) [1]** | 44.29 (8.06) [7] | 38.02 (12.74) [13] | 28.61 (5.83) [14] | 82.87 (9.31) [2] | | |
| Hepatitis | 99.64 (0.71) [3] | 88.91 (3.94) [10] | 32.76 (11.54) [18] | 57.07 (6.01) [15] | 99.48 (1.04) [5] | 99.48 (1.04) [5] | 85.34 (2.88) [11] | 99.68 (0.64) [2] | 81.47 (3.80) [14] | 85.08 (3.80) [12] | 0.18 (2.74) [20] | 15.19 (13.22) [19] | 56.56 (6.03) [16] | 99.29 (1.42) [7] | **99.72 (0.56) [1]** | 96.03 (2.34) [9] | 97.35 (2.24) [8] | 82.84 (5.17) [13] | 46.26 (8.04) [17] | 99.62 (0.77) [3] |
| http | 80.47 (25.20) [15] | **100.00 (0.00) [1]** | 97.17 (3.51) [7] | 91.94 (0.02) [10] | 99.80 (0.39) [5] | 99.99 (0.01) [3] | 90.70 (1.42) [11] | **100.00 (0.00) [1]** | 97.44 (0.11) [4] | 56.11 (6.39) [18] | 8.72 (0.55) [19] | 0.37 (0.04) [20] | 67.05 (10.17) [16] | 88.33 (22.49) [13] | 89.33 (6.74) [12] | 56.57 (24.25) [17] | 91.05 (1.48) [10] | 83.17 (16.49) [14] | 99.77 (1.65) [6] | 92.78 (14.28) [8] |
| InternetAds | 60.65 (2.30) [11] | 34.34 (1.70) [14] | 35.15 (1.49) [13] | 31.17 (1.72) [16] | 54.48 (2.24) [5] | 53.55 (3.03) [6] | 56.50 (2.48) [3] | 55.30 (5.24) [4] | 31.50 (1.58) [15] | 37.96 (9.82) [11] | 47.33 (1.80) [9] | 23.92 (4.37) [20] | 29.84 (1.61) [18] | 40.81 (0.33) [8] | 46.85 (1.70) [10] | 49.45 (3.71) [7] | 28.65 (0.63) [19] | 36.25 (0.63) [12] | 31.10 (2.12) [17] | 59.48 (0.30) [2] |
| Ionosphere | 98.25 (0.20) [4] | 95.58 (1.21) [11] | 88.21 (2.80) [15] | 81.54 (3.93) [18] | **98.67 (0.40) [1]** | 98.46 (0.67) [3] | 95.96 (0.97) [10] | 97.76 (0.75) [5] | 96.72 (0.72) [9] | 89.85 (3.10) [13] | 15.42 (11.03) [20] | 48.38 (29.38) [19] | 88.05 (5.03) [16] | 97.75 (1.08) [6] | 97.29 (0.57) [7] | 93.24 (1.01) [12] | 96.87 (0.92) [8] | 87.58 (6.46) [17] | 89.41 (1.10) [14] | 98.53 (0.43) [2] |
| landsat | 27.79 (0.69) [10] | 39.03 (1.39) [5] | **47.61 (0.96) [1]** | 8.87 (0.52) [17] | 28.22 (1.09) [9] | 30.98 (1.45) [7] | 7.24 (0.41) [19] | 41.26 (1.46) [2] | 18.79 (0.40) [13] | 23.76 (3.55) [12] | 39.21 (1.69) [4] | 15.54 (2.38) [15] | 7.04 (0.62) [20] | 33.47 (1.53) [6] | 25.03 (0.92) [11] | 30.98 (1.42) [7] | 40.77 (1.38) [3] | 17.64 (1.00) [14] | 8.72 (0.37) [18] | 28.80 (1.46) [8] |
| letter | 3.30 (0.33) [7] | 2.27 (0.13) [15] | 5.29 (1.40) [4] | 1.49 (0.12) [20] | 2.48 (0.14) [13] | 2.47 (0.13) [14] | 2.26 (0.10) [16] | 2.50 (0.12) [12] | 2.10 (0.10) [17] | 1.69 (0.24) [18] | 4.29 (0.79) [5] | **6.77 (4.84) [1]** | 1.57 (0.15) [19] | 5.81 (2.16) [3] | 2.79 (0.32) [8] | 2.60 (0.17) [10] | **6.68 (0.16) [2]** | 2.53 (0.29) [11] | 2.76 (0.53) [9] | 3.90 (0.48) [6] |
| Lymphography | 90.35 (0.79) [9] | 86.17 (3.88) [17] | 98.86 (0.45) [13] | **100.00 (0.00) [1]** | **100.00 (0.00) [1]** | **100.00 (0.00) [1]** | 99.82 (0.35) [8] | **100.00 (0.00) [1]** | 98.74 (1.23) [14] | 99.34 (0.99) [10] | 1.40 (0.54) [20] | 36.31 (29.32) [19] | 99.25 (0.66) [11] | 10.80 (0.00) [20] | 99.85 (0.30) [7] | 90.57 (6.09) [16] | 71.10 (7.45) [18] | **100.00 (0.00) [1]** | 96.39 (3.22) [15] | **100.00 (0.00) [1]** |
| magic.gamma | 78.41 (0.27) [2] | 69.32 (0.36) [10] | 70.38 (0.38) [8] | 45.98 (0.62) [19] | 68.86 (0.33) [11] | 58.28 (0.54) [15] | 63.62 (0.55) [12] | 73.71 (0.21) [5] | 71.77 (0.54) [7] | 58.04 (2.54) [16] | 72.15 (0.27) [6] | 63.05 (1.37) [13] | 47.20 (2.72) [18] | 61.87 (1.05) [14] | 50.74 (0.39) [17] | 76.18 (0.39) [3] | 69.67 (0.25) [9] | **78.49 (0.27) [1]** | 43.23 (1.40) [20] | 75.08 (0.38) [4] |
| mammography | 22.70 (3.37) [16] | 38.84 (3.17) [8] | 32.52 (3.42) [12] | 39.62 (1.85) [7] | 43.49 (2.90) [2] | 40.83 (2.91) [4] | 38.75 (2.13) [9] | **44.45 (1.91) [1]** | 32.07 (1.58) [13] | 40.80 (3.05) [5] | 2.81 (0.13) [20] | 24.52 (2.88) [14] | 25.78 (4.50) [13] | 18.78 (2.73) [17] | 14.34 (2.19) [18] | 40.90 (2.23) [3] | 38.66 (2.47) [10] | 22.71 (2.22) [15] | 8.59 (0.95) [19] | 39.73 (0.93) [6] |
| mnist | 72.46 (1.25) [14] | 70.10 (1.58) [7] | 67.94 (1.45) [9] | 61.85 (1.88) [16] | 71.87 (1.72) [5] | 73.45 (1.12) [2] | 66.83 (1.86) [11] | 76.85 (0.94) [1] | 67.25 (1.87) [10] | 55.74 (6.60) [18] | 73.23 (1.32) [3] | 57.50 (3.71) [17] | 63.15 (2.95) [15] | 64.82 (1.58) [14] | 65.93 (1.36) [13] | 49.76 (2.09) [19] | 70.29 (1.52) [6] | 66.44 (1.80) [12] | 69.67 (1.90) [8] | |
| musk | **100.00 (0.00) [1]** | **100.00 (0.00) [1]** | **100.00 (0.00) [1]** | **100.00 (0.00) [1]** | **100.00 (0.00) [1]** | **100.00 (0.00) [1]** | 98.19 (0.69) [16] | **100.00 (0.00) [1]** | **100.00 (0.00) [1]** | 63.03 (26.16) [18] | 99.32 (0.31) [15] | 18.62 (21.44) [20] | **100.00 (0.00) [1]** | 91.00 (2.90) [17] | **100.00 (0.00) [1]** | 33.66 (6.97) [19] | **100.00 (0.00) [1]** | 99.49 (0.38) [14] | **100.00 (0.00) [1]** | |
| optdigits | 50.60 (3.03) [3] | 26.96 (1.96) [10] | 40.52 (0.86) [5] | 3.18 (0.05) [20] | 20.70 (1.26) [11] | 37.92 (2.14) [6] | 10.58 (0.57) [17] | **87.72 (5.72) [1]** | 15.90 (0.77) [15] | 10.69 (2.80) [16] | 29.94 (0.90) [9] | 16.54 (4.69) [13] | 5.02 (0.65) [18] | 42.27 (6.31) [4] | 33.48 (1.65) [7] | 32.06 (1.90) [8] | 18.81 (1.21) [12] | 4.82 (0.94) [19] | 53.12 (3.49) [2] | |
| PageBlocks | 53.63 (1.74) [17] | 64.19 (1.26) [5] | 67.72 (1.20) [2] | 55.23 (1.26) [16] | 60.82 (1.10) [10] | 64.24 (1.26) [4] | 58.50 (1.60) [13] | 65.53 (1.95) [3] | 63.16 (1.30) [7] | 39.29 (1.67) [20] | 64.03 (1.25) [6] | **69.40 (3.84) [1]** | 60.55 (1.64) [11] | 62.09 (4.52) [8] | 52.57 (1.54) [18] | 61.21 (1.71) [9] | 48.04 (0.54) [19] | 57.57 (1.59) [14] | 56.16 (1.03) [15] | 59.55 (0.46) [12] |
| pendigits | 94.58 (1.41) [5] | 76.90 (0.65) [14] | 76.12 (2.29) [9] | 36.71 (3.04) [15] | 93.01 (0.85) [7] | 96.78 (0.64) [3] | 14.48 (1.18) [19] | **98.88 (0.19) [1]** | 70.94 (1.89) [10] | 55.32 (5.64) [13] | 88.97 (1.81) [8] | 12.57 (3.15) [20] | 29.78 (6.18) [17] | 65.22 (8.74) [11] | 97.31 (0.75) [2] | 12.69 (1.41) [18] | 17.12 (1.28) [18] | 94.16 (1.97) [6] | | |
| Pima | 54.22 (1.71) [7] | 51.17 (3.78) [8] | 31.72 (5.57) [16] | 42.40 (5.20) [11] | **68.22 (2.21) [1]** | 59.02 (2.47) [3] | 42.68 (4.42) [10] | 54.57 (4.36) [6] | 36.83 (3.82) [12] | 49.37 (4.97) [9] | 24.63 (1.56) [17] | -5.52 (28.35) [20] | 33.97 (14.75) [13] | 56.87 (4.46) [4] | 22.49 (1.29) [18] | 32.10 (7.50) [15] | 54.88 (3.01) [5] | 33.17 (5.36) [14] | 15.50 (4.52) [19] | 59.08 (2.37) [2] |
| satellite | 71.70 (1.07) [10] | 73.80 (0.99) [3] | 73.61 (0.76) [4] | 58.53 (1.01) [19] | 70.68 (1.11) [12] | 72.46 (1.05) [8] | 70.79 (1.15) [11] | 73.58 (0.93) [5] | 70.26 (0.99) [4] | 66.62 (1.34) [16] | 73.13 (0.94) [6] | 57.49 (10.76) [20] | 61.47 (1.55) [17] | 75.71 (1.24) [2] | **78.70 (0.95) [1]** | 70.55 (1.08) [13] | 73.54 (0.93) [5] | 68.36 (1.62) [15] | 72.10 (0.71) [9] | |
| satimage-2 | 93.76 (0.91) [10] | 97.08 (0.70) [2] | 89.48 (1.41) [12] | 92.16 (0.68) [11] | 96.94 (0.60) [3] | 95.95 (0.76) [6] | 86.47 (1.23) [14] | 95.46 (1.16) [7] | **97.11 (0.55) [1]** | 96.08 (1.14) [4] | 88.30 (1.60) [13] | 76.15 (18.40) [17] | 95.98 (0.43) [5] | 95.14 (0.93) [8] | 76.04 (2.65) [18] | 83.53 (1.00) [16] | 38.47 (2.84) [20] | 71.85 (7.99) [19] | 85.26 (2.49) [15] | |
| skin | 80.98 (1.69) [6] | 97.76 (0.48) [2] | 48.05 (2.47) [16] | 13.56 (0.30) [18] | 95.88 (0.47) [3] | 52.12 (0.43) [13] | 51.80 (0.68) [14] | 95.71 (1.03) [4] | 49.79 (6.17) [15] | 55.05 (1.85) [11] | 1.47 (0.27) [20] | 52.95 (2.64) [12] | 25.04 (4.42) [17] | 8.73 (1.12) [19] | 71.11 (0.46) [7] | 58.55 (1.06) [10] | **98.36 (0.42) [1]** | 62.84 (1.61) [6] | 60.24 (0.79) [9] | 95.09 (1.06) [5] |
| smtp | 48.55 (10.51) [13] | 57.94 (9.64) [8] | 57.35 (9.87) [10] | 57.49 (9.33) [9] | **70.26 (15.22) [1]** | 70.23 (15.22) [2] | 44.53 (9.66) [15] | 70.17 (15.23) [3] | 56.84 (9.61) [5] | 1.28 (0.43) [20] | 1.49 (0.23) [19] | 8.95 (17.81) [18] | 35.72 (4.38) [16] | 66.84 (9.45) [4] | 57.80 (9.68) [7] | 57.73 (9.69) [8] | 56.18 (2.90) [8] | 53.09 (22.87) [11] | 39.16 (15.86) [15] | 63.68 (12.63) [4] |
| SpamBase | 56.31 (1.47) [14] | 50.45 (1.26) [8] | 20.47 (1.82) [20] | 46.12 (1.41) [15] | 53.17 (1.38) [5] | 52.07 (1.71) [6] | 44.33 (1.47) [17] | 45.22 (3.04) [16] | 48.25 (1.19) [12] | **67.69 (2.20) [1]** | 48.76 (1.25) [11] | 41.72 (4.04) [19] | 46.68 (1.87) [13] | 59.03 (2.92) [2] | 58.05 (1.40) [3] | 51.58 (0.69) [7] | 49.03 (0.90) [10] | 46.23 (1.56) [14] | 42.40 (1.55) [18] | 49.24 (0.43) [9] |
| speech | 2.05 (0.45) [6] | 1.33 (0.45) [17] | 1.51 (0.48) [16] | 1.93 (0.57) [9] | 3.03 (0.41) [1] | 1.94 (0.60) [8] | 2.29 (0.59) [4] | 1.51 (0.42) [16] | 1.55 (0.74) [14] | 1.67 (0.33) [12] | 2.56 (1.38) [5] | 1.42 (0.42) [18] | 1.39 (0.30) [19] | 2.82 (1.50) [2] | 1.63 (0.00) [13] | 1.31 (0.38) [20] | 1.31 (0.30) [20] | 1.71 (0.14) [11] | | |
| Stamps | 87.20 (5.41) [4] | 72.70 (9.26) [8] | 65.86 (8.85) [11] | 60.47 (7.19) [14] | **92.97 (5.77) [1]** | 92.49 (5.24) [2] | 62.36 (8.71) [12] | 83.17 (6.13) [5] | 71.02 (6.90) [9] | 61.78 (5.62) [13] | 15.29 (2.23) [20] | 19.44 (26.05) [19] | 53.51 (15.57) [16] | 78.09 (5.83) [7] | 42.29 (12.04) [17] | 57.87 (9.22) [15] | 80.55 (6.71) [6] | 67.29 (8.84) [10] | 42.26 (10.42) [18] | 90.63 (6.26) [3] |
| thyroid | 68.98 (2.43) [15] | 82.32 (1.88) [2] | 60.94 (2.45) [13] | 82.24 (2.25) [3] | 80.02 (1.97) [9] | 80.93 (1.37) [6] | 81.29 (2.44) [5] | 80.19 (1.68) [8] | 59.49 (0.85) [18] | 81.87 (3.51) [4] | 42.86 (2.52) [20] | 76.49 (2.45) [12] | 80.85 (2.60) [7] | 45.87 (8.03) [19] | 77.16 (3.58) [11] | **84.12 (1.90) [1]** | 65.22 (2.01) [16] | 72.28 (7.24) [14] | 76.03 (1.27) [13] | |
| vertebral | 43.78 (3.36) [14] | 12.48 (3.45) [11] | 19.51 (2.40) [8] | 5.12 (1.97) [18] | 61.73 (7.26) [2] | 5.11 (2.23) [19] | 7.67 (2.19) [16] | 34.06 (6.50) [5] | 14.59 (2.40) [9] | 7.40 (2.06) [17] | 1.10 (1.84) [20] | 9.62 (9.67) [13] | 8.92 (2.44) [14] | 45.79 (11.17) [3] | 8.19 (4.54) [15] | 20.35 (4.22) [7] | 23.51 (4.39) [6] | 14.32 (1.83) [10] | 10.02 (3.22) [12] | **66.89 (6.45) [1]** |
| vowels | **51.38 (3.34) [1]** | 26.80 (2.67) [10] | 35.49 (5.81) [6] | 8.98 (2.99) [19] | 26.21 (2.60) [11] | 26.80 (2.54) [12] | 38.27 (4.68) [5] | 39.66 (2.29) [4] | 43.43 (3.47) [3] | 13.29 (2.76) [18] | 28.93 (3.21) [9] | 5.27 (5.64) [20] | 15.25 (2.44) [17] | 26.63 (6.38) [13] | 32.27 (3.56) [7] | 44.00 (1.81) [2] | 19.14 (3.17) [16] | 25.22 (2.84) [14] | 19.75 (6.31) [15] | 30.15 (2.72) [8] |
| Waveform | 6.68 (0.88) [5] | 3.83 (2.67) [5] | 27.70 (2.85) [2] | 6.46 (0.84) [16] | 18.13 (1.83) [7] | 25.32 (2.51) [5] | 5.67 (0.85) [18] | 34.24 (9.23) [16] | 10.38 (0.88) [11] | 10.20 (2.51) [12] | 14.18 (1.25) [10] | 15.37 (8.00) [9] | 6.43 (1.08) [17] | 17.90 (3.63) [8] | 2.34 (0.10) [20] | 21.25 (2.56) [6] | **47.08 (1.39) [1]** | 4.97 (1.43) [19] | 8.98 (1.26) [14] | |
| WBC | 94.58 (7.86) [8] | 90.14 (5.73) [11] | 21.99 (3.73) [19] | 94.54 (1.65) [9] | **98.79 (2.41) [1]** | 98.74 (1.86) [2] | 95.03 (6.66) [13] | 97.21 (2.50) [7] | 81.05 (5.68) [14] | 98.12 (1.03) [4] | 3.33 (1.20) [20] | 22.89 (26.29) [18] | 91.64 (3.75) [10] | 98.39 (2.09) [3] | 97.54 (2.01) [6] | 28.67 (4.35) [17] | 65.23 (10.64) [16] | 88.79 (6.49) [12] | 76.21 (7.03) [15] | 98.00 (2.77) [5] |
| WDBC | **99.88 (0.15) [1]** | 91.75 (4.11) [12] | 97.16 (1.82) [6] | 90.45 (4.06) [14] | 97.99 (1.31) [2] | 97.82 (1.38) [3] | 97.14 (1.60) [7] | 97.69 (1.48) [4] | 94.19 (4.15) [9] | 92.23 (6.22) [11] | 9.30 (2.49) [20] | 12.79 (22.64) [19] | 90.95 (3.86) [13] | 97.57 (2.19) [5] | 94.40 (2.18) [8] | 72.03 (4.35) [16] | 60.02 (11.01) [17] | 75.85 (15.17) [15] | 57.23 (9.80) [18] | 92.48 (4.06) [10] |

*Continued on next page*

| Model | NCSBAD | KNN | LOF | PCA | KPCA | KDE | GMM | LUNAR | AE | EIF | ABOD | DROCC | GOAD | ICL | SLAD | DTE-C | DTENonParametric | MCM | DDPM | DenoiserAD (Ours) |
|---|---|---|---|---|---|---|---|---|---|---|---|---|---|---|---|---|---|---|---|---|
| Wilt | 15.87 (0.97) [5] | 6.83 (0.35) [12] | 10.30 (0.82) [9] | 0.90 (0.16) [20] | 11.85 (0.52) [8] | 1.95 (0.16) [19] | 22.76 (0.57) [3] | 6.20 (1.00) [13] | 5.11 (0.23) [15] | 2.89 (0.45) [18] | 13.86 (0.59) [7] | 4.55 (2.10) [17] | 5.70 (0.83) [14] | 24.17 (6.22) [2] | 7.19 (0.37) [11] | 20.64 (0.85) [4] | 7.94 (0.43) [10] | 29.88 (1.22) [1] | 4.75 (0.55) [16] | 15.33 (0.50) [6] |
| wine | 99.66 (0.68) [5] | 94.98 (2.79) [12] | 90.18 (4.10) [14] | 67.77 (9.23) [17] | 99.83 (0.35) [3] | 99.99 (0.22) [2] | 95.13 (3.46) [11] | 99.70 (0.59) [4] | 96.78 (1.87) [9] | 82.75 (13.67) [15] | 1.86 (-1.19) [20] | 13.36 (12.97) [19] | 70.42 (6.52) [16] | 98.83 (2.20) [7] | 100.00 (0.00) [1] | 99.23 (1.30) [6] | 93.40 (2.79) [13] | 95.40 (3.09) [10] | 62.17 (5.60) [18] | 98.38 (3.25) [8] |
| WPBC | 85.27 (8.85) [1] | 25.29 (4.40) [13] | 18.40 (3.37) [14] | 16.78 (5.72) [16] | 84.46 (8.61) [2] | 84.40 (7.87) [3] | 27.04 (4.52) [12] | 73.44 (7.97) [7] | 18.12 (4.00) [15] | 27.38 (6.84) [11] | -7.04 (1.96) [20] | 11.15 (5.95) [19] | 16.40 (6.96) [17] | 83.21 (6.30) [4] | 48.37 (3.50) [9] | 49.88 (3.10) [8] | 43.39 (4.09) [10] | 15.28 (2.04) [18] | 81.99 (8.63) [5] | 98.38 (3.25) [8] |
| yeast | -4.19 (1.77) [4] | -8.16 (1.52) [11] | -6.61 (1.73) [9] | -10.62 (1.52) [20] | -8.58 (1.69) [12] | -9.84 (1.61) [17] | -10.46 (1.36) [19] | -9.52 (1.57) [15] | -9.67 (1.21) [16] | -10.39 (1.18) [18] | -6.63 (2.01) [10] | 0.39 (0.20) [1] | -4.82 (4.41) [6] | -5.57 (3.84) [8] | -1.49 (2.13) [2] | -4.47 (1.54) [5] | -9.04 (1.42) [14] | -8.94 (1.24) [13] | -3.79 (2.51) [3] | -5.07 (0.80) [7] |
| CIFAR10 0 | 22.05 (1.11) [3] | 20.95 (1.05) [8] | 21.57 (0.96) [6] | 20.07 (1.12) [12] | 21.88 (1.23) [4] | 17.69 (1.27) [16] | 21.64 (1.18) [5] | 22.89 (1.30) [1] | 20.80 (1.08) [9] | 17.52 (1.17) [17] | 22.18 (1.28) [2] | 8.14 (1.90) [20] | 20.16 (0.80) [11] | 15.04 (0.98) [18] | 20.00 (1.03) [13] | 18.26 (0.56) [14] | 14.06 (0.50) [19] | 20.45 (1.02) [10] | 18.21 (1.76) [15] | 21.09 (0.16) [7] |
| CIFAR10 1 | 12.43 (1.35) [6] | 10.09 (1.13) [13] | 18.05 (1.78) [1] | 9.24 (1.13) [15] | 11.08 (1.22) [11] | 8.65 (1.06) [17] | 13.34 (1.42) [2] | 10.65 (1.45) [12] | 12.17 (1.43) [7] | 5.32 (0.42) [19] | 11.90 (1.35) [8] | 5.17 (1.42) [20] | 9.99 (1.20) [14] | 12.75 (1.19) [3] | 12.57 (1.40) [5] | 12.74 (1.09) [4] | 8.26 (0.48) [18] | 11.50 (1.32) [9] | 8.96 (0.95) [16] | 11.45 (0.23) [10] |
| CIFAR10 2 | 9.69 (0.77) [4] | 8.98 (0.60) [11] | 10.56 (0.81) [1] | 8.66 (0.48) [14] | 9.31 (0.65) [9] | 6.82 (0.70) [19] | 9.53 (0.67) [7] | 10.21 (0.52) [2] | 9.55 (0.65) [6] | 7.77 (0.35) [17] | 9.66 (0.68) [5] | 5.72 (1.08) [20] | 8.83 (0.72) [13] | 8.22 (1.09) [16] | 9.52 (0.95) [8] | 8.62 (0.25) [15] | 8.94 (0.93) [12] | 7.62 (0.82) [18] | 9.75 (0.07) [3] |
| CIFAR10 3 | 9.24 (1.47) [7] | 8.30 (1.36) [13] | 10.17 (1.50) [1] | 8.42 (1.35) [12] | 9.04 (1.55) [8] | 7.89 (1.34) [15] | 9.76 (1.52) [2] | 9.65 (1.67) [3] | 9.64 (1.48) [4] | 6.60 (0.76) [18] | 8.97 (1.38) [9] | 6.25 (1.37) [20] | 8.44 (1.30) [11] | 7.40 (1.03) [17] | 8.52 (1.27) [10] | 9.32 (1.31) [6] | 6.49 (0.63) [19] | 9.48 (1.47) [5] | 7.72 (1.16) [16] | 8.25 (0.18) [14] |
| CIFAR10 4 | 30.70 (2.13) [4] | 29.04 (2.00) [5] | 28.12 (2.05) [12] | 28.22 (1.96) [11] | 28.69 (2.44) [8] | 24.31 (2.19) [17] | 30.82 (2.22) [3] | 31.29 (2.63) [2] | 28.99 (2.07) [6] | 25.60 (2.09) [15] | 28.33 (2.11) [10] | 11.19 (2.71) [20] | 28.47 (2.19) [9] | 23.32 (1.94) [18] | 28.05 (2.08) [13] | 25.18 (2.27) [16] | 14.40 (0.86) [19] | 28.75 (1.98) [7] | 26.53 (2.40) [14] | 32.39 (0.13) [1] |
| CIFAR10 5 | 8.82 (0.52) [7] | 8.03 (0.48) [12] | 11.50 (0.97) [1] | 8.20 (0.41) [10] | 8.77 (0.55) [9] | 7.56 (0.44) [15] | 9.55 (0.60) [3] | 7.53 (0.66) [16] | 9.31 (0.72) [4] | 5.61 (0.42) [19] | 8.81 (0.51) [8] | 4.75 (0.96) [20] | 8.08 (0.46) [11] | 6.83 (1.23) [17] | 8.93 (0.75) [5] | 10.01 (0.51) [2] | 6.50 (0.20) [18] | 8.88 (0.57) [6] | 7.86 (1.07) [13] | 7.67 (0.19) [14] |
| CIFAR10 6 | 19.47 (1.23) [1] | 16.30 (1.04) [12] | 19.06 (1.44) [3] | 15.00 (0.93) [14] | 17.11 (1.06) [9] | 13.64 (1.36) [16] | 18.83 (1.24) [4] | 18.12 (0.82) [7] | 16.68 (1.11) [10] | 12.72 (0.93) [18] | 18.77 (1.25) [5] | 7.76 (0.72) [20] | 15.18 (0.94) [13] | 12.35 (1.15) [19] | 17.99 (1.39) [8] | 18.61 (1.81) [6] | 13.26 (0.53) [17] | 16.42 (1.09) [11] | 14.34 (1.36) [15] | 19.46 (0.25) [2] |
| CIFAR10 7 | 18.12 (1.31) [3] | 16.94 (1.29) [11] | 19.55 (1.25) [1] | 16.81 (1.15) [13] | 17.16 (1.35) [9] | 12.47 (1.29) [18] | 17.57 (0.77) [6] | 18.30 (1.50) [2] | 15.14 (1.13) [15] | 17.18 (0.86) [8] | 8.60 (1.94) [20] | 16.97 (1.47) [10] | 12.65 (0.66) [17] | 16.86 (1.23) [12] | 14.83 (1.09) [16] | 9.28 (0.45) [19] | 17.32 (1.27) [7] | 15.39 (1.51) [14] | 17.99 (0.34) [4] |
| CIFAR10 8 | 17.68 (0.73) [4] | 16.31 (0.85) [10] | 19.37 (0.65) [1] | 16.10 (0.74) [11] | 16.42 (0.58) [9] | 11.73 (1.29) [19] | 17.44 (0.71) [6] | 19.17 (1.34) [2] | 17.70 (0.71) [3] | 13.14 (0.95) [18] | 17.55 (1.08) [5] | 8.91 (1.97) [20] | 16.01 (0.73) [12] | 14.74 (1.16) [14] | 17.17 (0.85) [8] | 14.70 (0.63) [15] | 13.61 (0.39) [17] | 17.23 (0.72) [7] | 14.15 (1.61) [16] | 15.64 (0.17) [13] |
| CIFAR10 9 | 21.74 (0.96) [4] | 19.21 (0.91) [12] | 22.67 (1.37) [1] | 18.94 (0.93) [13] | 20.01 (0.67) [8] | 15.51 (1.21) [18] | 21.85 (1.21) [3] | 22.66 (0.60) [2] | 20.09 (1.12) [7] | 15.98 (1.37) [17] | 20.96 (0.68) [5] | 9.39 (3.21) [20] | 18.65 (0.71) [14] | 16.80 (1.43) [16] | 19.59 (0.95) [11] | 19.77 (1.44) [10] | 19.88 (0.99) [9] | 17.90 (0.91) [15] | 20.85 (0.19) [6] |
| FashionMNIST 0 | 47.70 (3.54) [5] | 43.15 (3.57) [11] | 48.67 (3.67) [4] | 38.44 (3.30) [15] | 44.46 (3.71) [8] | 31.41 (2.68) [17] | 46.62 (3.58) [6] | 54.64 (1.79) [1] | 44.25 (3.68) [10] | 31.24 (3.40) [18] | 45.93 (3.77) [7] | 15.46 (3.01) [20] | 39.36 (3.66) [14] | 52.19 (4.97) [2] | 44.44 (3.45) [9] | 43.13 (3.31) [12] | 28.64 (0.77) [19] | 41.80 (3.49) [13] | 37.34 (2.83) [16] | 48.98 (0.49) [3] |
| FashionMNIST 1 | 87.04 (2.43) [2] | 84.03 (2.93) [9] | 85.37 (3.15) [5] | 81.11 (2.87) [15] | 83.65 (2.21) [9] | 78.69 (2.92) [16] | 82.53 (3.31) [12] | 89.10 (3.01) [1] | 83.20 (2.95) [10] | 68.57 (3.50) [18] | 86.06 (2.62) [4] | 40.36 (17.97) [20] | 81.75 (2.92) [13] | 84.51 (2.54) [7] | 84.87 (2.67) [6] | 76.65 (2.78) [17] | 43.64 (1.60) [19] | 82.83 (2.91) [11] | 81.56 (3.21) [14] | 86.07 (0.14) [3] |
| FashionMNIST 2 | 48.13 (2.98) [7] | 43.24 (3.40) [11] | 50.95 (2.96) [3] | 39.08 (3.65) [15] | 46.36 (3.67) [8] | 37.42 (3.46) [17] | 48.67 (3.26) [5] | 53.93 (2.86) [2] | 19.50 (0.81) [19] | 48.14 (3.27) [6] | 19.40 (12.83) [20] | 40.31 (3.51) [13] | 54.89 (2.78) [1] | 51.29 (2.98) [12] | 39.48 (2.82) [14] | 26.77 (0.72) [18] | 49.34 (3.51) [10] | 38.32 (4.17) [16] | 50.21 (0.43) [4] |
| FashionMNIST 3 | 58.32 (1.26) [3] | 52.05 (1.38) [11] | 58.19 (1.28) [4] | 48.12 (1.48) [15] | 54.98 (1.25) [8] | 44.61 (1.94) [17] | 57.36 (1.30) [6] | 59.18 (2.20) [2] | 52.42 (1.59) [10] | 34.61 (3.25) [18] | 56.61 (1.39) [7] | 21.17 (5.59) [20] | 48.98 (1.60) [14] | 57.41 (2.71) [5] | 51.87 (1.69) [12] | 54.07 (0.48) [9] | 30.37 (0.79) [19] | 51.21 (1.63) [13] | 46.79 (1.50) [16] | 61.63 (0.29) [1] |
| FashionMNIST 4 | 55.93 (1.49) [5] | 50.65 (1.45) [10] | 56.33 (1.79) [4] | 47.37 (1.34) [15] | 51.44 (1.25) [8] | 46.45 (1.12) [17] | 53.03 (2.47) [6] | 57.18 (0.44) [3] | 51.14 (1.33) [9] | 23.12 (2.23) [19] | 52.34 (1.51) [7] | 18.55 (10.77) [20] | 48.23 (1.22) [14] | 57.35 (2.39) [2] | 48.54 (1.16) [13] | 49.42 (2.03) [12] | 50.32 (1.32) [11] | 60.18 (1.21) [1] | 47.02 (1.73) [16] | 60.01 (0.18) [1] |
| FashionMNIST 5 | 85.58 (1.44) [2] | 85.18 (1.34) [6] | 84.78 (1.62) [10] | 84.84 (1.29) [9] | 79.52 (1.46) [16] | 84.46 (1.23) [14] | 84.47 (1.25) [13] | 85.44 (0.70) [4] | 84.93 (1.31) [8] | 79.01 (1.56) [17] | 85.86 (1.39) [1] | 47.51 (22.55) [19] | 84.75 (1.33) [11] | 84.28 (1.22) [15] | 85.55 (1.08) [3] | 74.56 (2.36) [18] | 33.73 (0.81) [20] | 85.13 (1.26) [7] | 84.61 (1.33) [12] | 85.37 (0.07) [5] |
| FashionMNIST 6 | 31.15 (1.72) [4] | 22.96 (1.29) [12] | 30.12 (2.42) [6] | 17.51 (0.94) [17] | 24.24 (1.22) [10] | 20.80 (1.63) [14] | 30.50 (1.57) [5] | 33.16 (1.76) [3] | 23.50 (1.42) [11] | 11.28 (1.27) [19] | 26.84 (1.72) [8] | 5.85 (4.81) [20] | 18.90 (0.57) [15] | 35.68 (2.24) [2] | 25.60 (1.59) [9] | 28.62 (1.83) [7] | 17.81 (0.39) [16] | 22.12 (1.35) [13] | 16.99 (1.25) [18] | 38.47 (0.25) [1] |
| FashionMNIST 7 | 89.59 (1.11) [3] | 88.54 (1.17) [10] | 88.64 (1.00) [7] | 88.04 (1.28) [13] | 81.25 (1.43) [17] | 88.25 (1.20) [11] | 88.61 (1.12) [8] | 88.76 (3.40) [5] | 88.69 (1.16) [6] | 86.00 (1.66) [16] | 89.80 (0.99) [2] | 43.90 (22.90) [19] | 88.13 (1.35) [12] | 87.91 (1.69) [14] | 88.59 (1.13) [9] | 75.78 (1.57) [18] | 38.45 (1.37) [20] | 88.80 (1.19) [4] | 87.49 (1.17) [15] | 90.55 (0.10) [1] |
| FashionMNIST 8 | 32.45 (1.09) [5] | 23.98 (1.01) [11] | 35.03 (1.57) [3] | 18.71 (0.74) [17] | 24.98 (1.14) [10] | 18.96 (1.20) [16] | 31.02 (0.94) [6] | 38.04 (2.09) [2] | 23.68 (0.92) [12] | 15.99 (1.91) [19] | 28.97 (1.10) [7] | 8.05 (3.25) [20] | 19.35 (0.96) [15] | 33.20 (2.35) [4] | 28.14 (1.03) [8] | 27.50 (1.51) [9] | 20.20 (0.75) [14] | 21.88 (1.09) [13] | 18.20 (0.65) [18] | 40.32 (0.66) [1] |
| FashionMNIST 9 | 84.67 (1.64) [2] | 82.95 (1.53) [8] | 83.66 (1.54) [5] | 82.44 (1.59) [10] | 80.36 (1.65) [14] | 80.23 (1.96) [15] | 81.71 (2.31) [13] | 84.87 (3.92) [1] | 83.29 (1.47) [7] | 76.08 (2.33) [17] | 84.06 (1.40) [4] | 38.34 (20.76) [19] | 82.41 (1.72) [11] | 80.15 (3.77) [16] | 83.64 (1.49) [6] | 74.31 (1.88) [18] | 36.56 (1.19) [20] | 82.71 (1.58) [9] | 82.30 (1.82) [12] | 84.09 (0.08) [3] |
| MNIST-C brightness | 45.97 (1.93) [3] | 26.38 (0.89) [11] | 35.26 (1.51) [7] | 18.03 (0.60) [18] | 28.39 (0.80) [10] | 22.07 (1.78) [15] | 34.91 (1.29) [8] | 47.82 (2.50) [2] | 24.60 (0.85) [12] | 12.17 (0.99) [20] | 36.13 (1.40) [6] | 14.28 (3.12) [19] | 18.86 (0.53) [16] | 41.31 (2.62) [4] | 28.51 (0.96) [9] | 36.67 (1.47) [5] | 23.21 (0.56) [14] | 23.51 (0.79) [13] | 18.18 (0.76) [17] | 68.50 (0.50) [1] |
| MNIST-C canny edges | 45.36 (0.97) [3] | 26.00 (1.07) [12] | 43.77 (1.11) [4] | 16.79 (0.99) [17] | 28.00 (1.17) [10] | 20.31 (1.00) [15] | 34.27 (1.00) [7] | 54.09 (1.53) [2] | 26.16 (1.13) [11] | 14.07 (3.04) [20] | 35.92 (1.09) [6] | 15.96 (7.74) [19] | 18.53 (1.27) [16] | 37.43 (1.66) [5] | 29.34 (1.00) [9] | 33.50 (0.86) [8] | 25.19 (0.90) [13] | 23.35 (1.05) [14] | 16.48 (1.04) [18] | 64.39 (0.91) [1] |
| MNIST-C dotted line | 52.68 (1.99) [3] | 35.60 (1.53) [10] | 43.84 (1.94) [5] | 26.39 (1.53) [17] | 38.00 (1.58) [8] | 27.33 (1.24) [16] | 41.99 (1.57) [7] | 60.81 (1.82) [2] | 34.24 (1.61) [12] | 21.26 (2.08) [19] | 45.15 (1.76) [4] | 18.14 (3.63) [20] | 28.10 (1.72) [15] | 42.72 (2.41) [6] | 35.75 (1.46) [9] | 35.18 (1.36) [11] | 32.55 (0.79) [13] | 32.10 (1.52) [14] | 25.15 (1.88) [18] | 66.70 (0.51) [1] |
| MNIST-C fog | 81.90 (1.25) [3] | 66.66 (1.35) [11] | 76.67 (1.01) [5] | 56.56 (1.20) [16] | 67.31 (1.45) [10] | 56.54 (1.43) [17] | 77.31 (1.46) [4] | 89.25 (1.25) [2] | 65.97 (1.28) [12] | 35.17 (2.90) [19] | 75.71 (1.06) [6] | 19.17 (10.04) [20] | 57.66 (1.54) [14] | 73.82 (0.96) [7] | 71.61 (1.38) [8] | 70.59 (1.41) [9] | 43.11 (1.01) [18] | 65.48 (1.54) [13] | 56.58 (1.35) [15] | 90.13 (0.15) [1] |
| MNIST-C glass blur | 85.49 (1.50) [3] | 78.33 (1.62) [9] | 79.49 (1.88) [8] | 72.84 (1.37) [15] | 81.48 (1.05) [7] | 68.93 (0.28) [17] | 84.01 (1.10) [5] | 92.75 (1.32) [1] | 77.31 (1.54) [11] | 58.40 (5.40) [18] | 84.28 (1.33) [4] | 33.49 (18.20) [20] | 73.68 (1.49) [14] | 81.57 (1.79) [6] | 76.33 (1.38) [12] | 46.70 (0.42) [19] | 75.52 (1.54) [13] | 72.20 (1.50) [16] | 89.16 (0.21) [2] |
| MNIST-C identity | 3.65 (0.10) [17] | 4.09 (0.22) [14] | 4.13 (0.23) [11] | 4.18 (0.23) [9] | 3.58 (0.14) [19] | 3.63 (0.19) [18] | 4.13 (0.18) [11] | 4.32 (0.30) [2] | 4.23 (0.24) [7] | 4.11 (0.18) [13] | 4.32 (0.11) [2] | 4.50 (0.42) [1] | 4.19 (0.21) [8] | 3.75 (0.14) [16] | 4.18 (0.14) [9] | 4.30 (0.24) [4] | 3.99 (0.19) [15] | 4.26 (0.24) [6] | 4.29 (0.20) [5] | 3.20 (0.04) [20] |
| MNIST-C impulse noise | 99.78 (0.06) [3] | 99.48 (0.20) [11] | 99.12 (0.31) [14] | 99.48 (0.20) [11] | 96.19 (0.32) [16] | 99.59 (0.19) [7] | 99.42 (0.33) [13] | 99.05 (1.17) [15] | 99.90 (0.30) [9] | 93.95 (2.88) [17] | 99.74 (0.12) [4] | 66.42 (35.11) [19] | 99.50 (0.19) [9] | 99.65 (0.37) [5] | 99.91 (0.01) [1] | 91.26 (0.32) [18] | 49.29 (0.47) [20] | 99.82 (0.04) [2] | 99.60 (0.14) [6] | 99.57 (0.01) [8] |
| MNIST-C motion blur | 77.65 (1.61) [3] | 61.53 (1.97) [11] | 68.34 (1.91) [6] | 52.06 (1.94) [17] | 64.59 (1.91) [8] | 56.27 (2.35) [14] | 67.97 (1.81) [7] | 79.52 (1.57) [2] | 60.24 (1.91) [12] | 31.40 (3.25) [19] | 70.88 (1.69) [5] | 27.37 (9.48) [20] | 53.46 (1.67) [15] | 73.19 (2.31) [4] | 62.18 (1.80) [10] | 62.28 (1.71) [9] | 39.38 (0.84) [18] | 58.69 (1.96) [13] | 52.51 (1.72) [16] | 85.53 (0.34) [1] |
| MNIST-C rotate | 11.92 (0.56) [3] | 6.85 (0.47) [10] | 9.98 (0.44) [6] | 6.53 (0.39) [19] | 8.78 (0.47) [9] | 7.25 (0.37) [16] | 9.79 (0.58) [7] | 13.66 (0.61) [2] | 8.19 (0.43) [12] | 5.95 (0.39) [20] | 10.18 (0.41) [5] | 7.28 (0.96) [15] | 6.72 (0.43) [17] | 10.58 (0.55) [4] | 7.81 (0.57) [13] | 9.32 (0.45) [8] | 8.38 (0.28) [11] | 7.66 (0.41) [14] | 6.61 (0.48) [18] | 15.45 (0.13) [1] |
| MNIST-C scale | 36.32 (1.11) [3] | 24.37 (1.07) [10] | 32.48 (1.54) [4] | 17.70 (0.86) [16] | 25.64 (0.98) [9] | 15.71 (0.83) [18] | 31.83 (1.07) [5] | 41.27 (1.55) [2] | 24.05 (0.99) [11] | 12.12 (2.83) [19] | 26.06 (1.11) [8] | 11.49 (6.50) [20] | 19.13 (1.19) [15] | 27.43 (1.19) [7] | 22.32 (0.99) [14] | 29.57 (0.98) [6] | 23.92 (0.85) [12] | 22.52 (0.93) [13] | 16.89 (0.98) [17] | 47.10 (1.11) [1] |
| MNIST-C shear | 23.61 (1.49) [3] | 17.21 (1.21) [11] | 20.52 (1.39) [4] | 13.97 (1.00) [16] | 17.92 (1.33) [9] | 12.85 (0.52) [18] | 20.07 (1.40) [5] | 24.74 (2.06) [2] | 17.13 (1.20) [12] | 12.22 (1.50) [19] | 19.40 (1.43) [7] | 7.54 (1.99) [20] | 14.48 (1.10) [15] | 19.53 (0.93) [6] | 17.41 (0.98) [10] | 18.87 (1.18) [8] | 13.48 (1.03) [14] | 16.08 (1.13) [13] | 12.96 (0.80) [17] | 29.67 (0.19) [1] |
| MNIST-C shot noise | 49.86 (2.05) [3] | 35.17 (1.94) [9] | 42.76 (1.87) [4] | 25.32 (1.86) [17] | 36.90 (1.91) [8] | 25.43 (1.34) [16] | 40.62 (1.91) [6] | 59.03 (2.27) [2] | 31.75 (1.99) [12] | 20.74 (2.17) [19] | 41.93 (1.94) [5] | 18.07 (4.66) [20] | 26.48 (1.93) [15] | 34.14 (1.95) [10] | 32.30 (1.69) [11] | 38.45 (1.90) [7] | 30.65 (1.12) [13] | 29.67 (1.89) [14] | 23.94 (1.77) [18] | 63.66 (0.26) [1] |
| MNIST-C spatter | 64.91 (1.57) [3] | 50.75 (1.40) [10] | 59.65 (0.80) [6] | 41.42 (1.63) [15] | 53.94 (1.82) [8] | 38.75 (1.35) [17] | 62.99 (1.38) [4] | 73.96 (2.13) [2] | 50.02 (1.42) [11] | 37.58 (6.06) [18] | 62.63 (1.09) [5] | 18.78 (8.46) [20] | 42.63 (1.17) [14] | 54.82 (1.56) [7] | 51.96 (1.43) [9] | 49.29 (1.37) [12] | 36.92 (1.29) [19] | 47.04 (1.61) [13] | 40.09 (1.86) [16] | 74.02 (0.64) [1] |
| MNIST-C stripe | 98.73 (0.29) [4] | 98.25 (0.34) [7] | 98.64 (0.14) [5] | 97.88 (0.37) [12] | 97.09 (0.61) [15] | 96.99 (0.77) [16] | 97.88 (0.83) [12] | 98.19 (2.66) [8] | 98.19 (0.35) [6] | 91.90 (1.08) [17] | 98.94 (0.27) [1] | 43.97 (24.43) [20] | 97.95 (0.40) [11] | 98.81 (0.53) [3] | 98.49 (0.35) [6] | 89.91 (1.49) [18] | 48.44 (0.67) [19] | 98.14 (0.33) [10] | 97.84 (0.43) [14] | 98.89 (0.03) [2] |
| MNIST-C translate | 30.68 (0.88) [2] | 14.81 (0.58) [12] | 21.92 (0.72) [7] | 9.55 (0.52) [19] | 16.20 (0.69) [10] | 12.17 (0.63) [15] | 22.19 (0.77) [5] | 29.53 (0.55) [3] | 14.01 (0.56) [13] | 7.52 (0.60) [20] | 22.03 (0.84) [6] | 10.17 (1.74) [16] | 10.14 (0.54) [17] | 24.82 (1.62) [4] | 17.40 (0.71) [9] | 20.36 (0.42) [8] | 15.30 (0.54) [11] | 13.63 (0.57) [14] | 9.75 (0.82) [18] | 46.58 (0.27) [1] |
| MNIST-C zigzag | 67.07 (1.84) [3] | 50.30 (1.81) [10] | 57.23 (1.37) [7] | 39.80 (1.87) [15] | 53.05 (1.78) [8] | 35.70 (1.62) [18] | 57.51 (1.89) [6] | 72.06 (1.57) [2] | 48.87 (1.82) [11] | 29.86 (3.71) [19] | 60.68 (2.02) [4] | 20.28 (5.60) [20] | 41.76 (1.53) [14] | 58.62 (1.55) [5] | 50.49 (1.94) [9] | 48.50 (1.21) [12] | 46.56 (1.89) [13] | 38.48 (1.99) [17] | 79.47 (0.21) [1] |
| MVTec-AD bottle | 98.49 (0.91) [5] | 95.84 (1.42) [12] | 95.82 (1.31) [13] | 95.03 (1.41) [15] | 98.71 (0.86) [3] | 97.99 (0.85) [7] | 98.32 (0.87) [6] | 95.92 (1.18) [10] | 95.67 (1.48) [14] | 13.85 (3.59) [20] | 86.27 (8.93) [19] | 95.00 (1.68) [16] | 97.89 (0.63) [8] | 98.65 (0.68) [4] | 96.70 (0.73) [9] | 95.86 (1.29) [11] | 93.53 (1.60) [17] | 98.70 (0.57) [2] |
| MVTec-AD cable | 75.37 (5.28) [4] | 50.15 (6.36) [13] | 50.56 (6.07) [12] | 42.55 (6.28) [17] | 76.17 (4.87) [1] | 70.55 (7.01) [8] | 75.42 (5.72) [3] | 67.58 (9.51) [9] | 50.71 (6.52) [11] | 44.77 (7.08) [16] | 9.38 (3.22) [20] | 21.35 (14.04) [19] | 45.02 (5.76) [15] | 74.21 (4.01) [6] | 75.43 (3.83) [2] | 71.36 (5.25) [7] | 51.68 (1.13) [14] | 50.10 (6.67) [14] | 37.12 (6.36) [18] | 75.11 (6.03) [5] |
| MVTec-AD capsule | 81.04 (3.21) [5] | 52.79 (3.63) [11] | 52.60 (4.86) [12] | 42.86 (3.79) [17] | 84.15 (3.68) [3] | 81.95 (3.04) [4] | 84.50 (3.21) [2] | 68.66 (5.35) [8] | 49.69 (4.47) [13] | 45.00 (3.24) [15] | 1.62 (3.63) [20] | 28.71 (9.33) [19] | 43.92 (3.63) [16] | 80.91 (4.50) [6] | 79.43 (3.39) [7] | 65.33 (2.57) [9] | 62.65 (2.90) [10] | 48.89 (3.29) [14] | 39.70 (4.62) [18] | 85.10 (3.13) [1] |
| MVTec-AD carpet | 79.71 (3.41) [6] | 60.20 (6.03) [11] | 59.12 (5.78) [14] | 57.77 (6.12) [16] | 81.04 (2.52) [4] | 80.34 (2.41) [5] | 81.17 (2.55) [3] | 68.61 (7.94) [9] | 59.62 (5.16) [12] | 56.03 (6.96) [17] | 14.88 (6.10) [20] | 40.74 (22.16) [19] | 58.77 (6.05) [15] | 82.45 (4.26) [1] | 79.26 (2.51) [7] | 72.52 (5.14) [8] | 64.70 (6.93) [10] | 59.14 (5.87) [13] | 51.75 (6.48) [18] | 82.36 (2.22) [2] |
| MVTec-AD grid | 73.28 (4.93) [6] | 51.11 (2.88) [11] | 49.12 (3.41) [13] | 44.77 (2.37) [16] | 74.99 (5.12) [4] | 75.63 (5.03) [3] | 73.37 (5.55) [5] | 65.61 (7.60) [8] | 49.33 (2.38) [12] | 38.17 (1.86) [18] | 10.95 (3.28) [20] | 37.82 (10.90) [19] | 46.37 (2.23) [15] | 78.27 (3.32) [1] | 73.23 (4.01) [7] | 62.97 (4.37) [9] | 38.53 (3.35) [10] | 48.54 (3.83) [14] | 43.62 (2.93) [17] | 76.28 (3.62) [2] |
| MVTec-AD hazelnut | 73.44 (2.69) [3] | 55.73 (3.62) [12] | 55.80 (4.09) [11] | 52.88 (3.90) [16] | 73.85 (3.60) [2] | 73.05 (3.05) [5] | 73.26 (2.44) [4] | 68.35 (2.78) [9] | 56.12 (4.18) [10] | 55.17 (4.20) [13] | 20.89 (5.33) [20] | 33.54 (13.41) [19] | 53.92 (3.63) [15] | 77.22 (6.46) [1] | 72.81 (5.16) [6] | 69.52 (5.34) [8] | 45.76 (3.62) [18] | 54.88 (4.31) [14] | 51.83 (4.85) [17] | 71.84 (6.10) [7] |
| MVTec-AD leather | 90.21 (0.44) [3] | 98.51 (0.66) [9] | 97.67 (0.51) [16] | 98.29 (0.65) [12] | 98.87 (0.81) [8] | 98.93 (0.47) [7] | 99.13 (0.52) [4] | 99.24 (0.28) [1] | 98.30 (0.64) [11] | 98.22 (1.19) [14] | 33.80 (6.05) [20] | 46.58 (21.81) [19] | 98.31 (0.66) [10] | 97.78 (2.26) [15] | 99.06 (0.49) [5] | 94.34 (3.88) [18] | 98.26 (0.73) [13] | 96.49 (0.79) [17] | 99.05 (0.53) [6] |
| MVTec-AD metal nut | 83.32 (4.27) [4] | 53.65 (4.07) [12] | 55.32 (4.98) [11] | 47.64 (4.74) [16] | 82.97 (4.07) [5] | 81.53 (3.20) [6] | 83.86 (4.53) [3] | 70.94 (5.18) [9] | 52.05 (4.94) [14] | 41.76 (5.11) [18] | 3.79 (1.59) [20] | 24.92 (9.62) [19] | 49.32 (4.69) [15] | 83.97 (2.80) [2] | 80.01 (4.17) [7] | 75.53 (5.16) [8] | 57.34 (6.74) [10] | 52.65 (3.62) [13] | 45.59 (6.58) [17] | 84.50 (3.73) [1] |
| MVTec-AD pill | 73.40 (6.89) [5] | 44.09 (4.55) [12] | 45.48 (5.99) [11] | 37.08 (3.82) [17] | 75.36 (8.45) [3] | 72.07 (7.18) [7] | 77.29 (7.98) [2] | 55.67 (10.30) [9] | 43.48 (4.11) [13] | 38.98 (3.65) [15] | -0.05 (5.08) [20] | 27.41 (11.19) [19] | 38.10 (3.42) [16] | 72.56 (4.14) [6] | 74.43 (7.74) [4] | 62.41 (8.39) [8] | 34.05 (7.05) [10] | 43.09 (4.07) [14] | 27.72 (2.11) [18] | 79.44 (4.46) [1] |
| MVTec-AD screw | 65.41 (3.25) [6] | 38.59 (1.30) [11] | 38.54 (2.57) [12] | 33.23 (1.06) [16] | 67.44 (2.38) [3] | 68.64 (2.02) [2] | 65.59 (3.41) [5] | 54.11 (6.17) [8] | 35.76 (1.74) [14] | 24.77 (5.88) [19] | 10.45 (3.79) [20] | 28.58 (9.94) [18] | 33.25 (1.90) [15] | 70.32 (3.14) [1] | 63.11 (1.90) [7] | 53.13 (3.09) [9] | 38.35 (4.25) [10] | 36.13 (2.44) [13] | 29.99 (3.92) [17] | 66.93 (2.57) [4] |
| MVTec-AD tile | 89.37 (2.98) [2] | 74.53 (1.79) [13] | 73.02 (2.07) [15] | 73.26 (1.62) [14] | 87.51 (5.25) [5] | 86.07 (3.34) [6] | 88.97 (2.17) [3] | 82.02 (3.18) [9] | 75.12 (1.97) [11] | 75.10 (2.09) [12] | 15.96 (3.34) [20] | 38.44 (20.11) [19] | 72.91 (2.45) [16] | 85.12 (6.04) [7] | 89.44 (3.29) [1] | 84.98 (5.19) [8] | 72.69 (3.46) [17] | 75.16 (1.65) [10] | 66.75 (1.54) [18] | 88.44 (3.29) [4] |
| MVTec-AD toothbrush | 99.87 (0.11) [3] | 92.38 (3.40) [11] | 88.18 (3.42) [13] | 71.43 (1.67) [17] | 99.84 (0.25) [4] | 99.70 (0.32) [6] | 99.69 (0.43) [8] | 82.96 (2.89) [15] | 88.39 (6.17) [12] | 33.65 (0.48) [20] | -12.09 (2.59) [20] | 35.01 (20.07) [19] | 73.89 (1.63) [16] | 99.98 (0.14) [2] | 99.70 (0.28) [6] | 91.37 (2.11) [10] | 85.00 (3.44) [14] | 58.90 (2.85) [18] | 99.90 (0.13) [1] |
| MVTec-AD transistor | 75.91 (6.61) [5] | 52.61 (6.48) [12] | 52.69 (5.98) [10] | 49.08 (6.50) [15] | 77.94 (6.71) [1] | 77.77 (7.31) [2] | 73.68 (7.27) [7] | 72.60 (7.77) [8] | 52.66 (5.88) [11] | 47.11 (9.99) [17] | 9.78 (2.47) [20] | 35.17 (18.00) [19] | 49.48 (6.05) [14] | 77.28 (4.41) [3] | 75.35 (6.15) [6] | 71.17 (6.04) [9] | 48.14 (4.23) [16] | 52.06 (5.97) [13] | 44.79 (6.47) [18] | 77.10 (5.23) [4] |
| MVTec-AD wood | 88.21 (3.83) [5] | 68.22 (2.68) [11] | 70.08 (3.37) [10] | 65.58 (2.17) [15] | 87.84 (4.87) [6] | 88.60 (4.49) [3] | 88.30 (3.49) [4] | 80.34 (3.17) [9] | 68.04 (2.60) [13] | 60.90 (2.91) [18] | 14.50 (1.56) [20] | 53.37 (14.80) [19] | 65.87 (2.30) [14] | 86.00 (4.35) [7] | 80.86 (4.20) [8] | 61.79 (5.01) [17] | 68.17 (2.28) [12] | 63.54 (2.53) [16] | 88.57 (4.15) [2] |
| MVTec-AD zipper | 91.41 (2.50) [3] | 75.48 (6.06) [10] | 75.43 (8.06) [11] | 61.10 (6.70) [16] | 90.28 (3.51) [5] | 87.39 (4.97) [7] | 89.04 (2.04) [6] | 85.49 (3.51) [9] | 70.94 (6.31) [14] | 58.93 (6.05) [17] | 20.76 (7.37) [20] | 30.66 (12.32) [19] | 63.19 (6.97) [15] | 92.40 (1.62) [2] | 90.58 (2.82) [4] | 87.29 (2.75) [8] | 75.47 (6.41) [11] | 71.17 (6.31) [13] | 45.14 (8.08) [18] | 92.89 (1.69) [1] |
| SVHN 0 | 12.32 (1.24) [7] | 10.82 (1.17) [11] | 12.96 (1.32) [3] | 9.66 (1.00) [16] | 10.96 (1.13) [9] | 8.28 (0.96) [18] | 12.44 (1.26) [5] | 13.45 (1.45) [1] | 10.74 (1.15) [10] | 7.98 (0.43) [19] | 12.40 (1.40) [6] | 6.80 (1.35) [20] | 9.88 (1.01) [15] | 12.76 (1.26) [4] | 10.89 (1.23) [10] | 11.56 (1.06) [8] | 10.32 (0.96) [14] | 10.49 (1.14) [13] | 8.74 (0.60) [17] | 13.09 (0.20) [2] |
| SVHN 1 | 12.76 (0.98) [5] | 11.87 (0.93) [9] | 10.02 (0.70) [17] | 12.02 (0.90) [8] | 12.05 (0.92) [7] | 8.75 (0.89) [19] | 12.97 (1.07) [3] | 14.23 (0.85) [1] | 11.67 (0.99) [12] | 11.45 (0.85) [13] | 12.10 (0.75) [6] | 6.49 (1.55) [20] | 11.82 (1.15) [10] | 8.88 (0.56) [18] | 10.36 (0.73) [15] | 12.37 (1.04) [4] | 10.18 (0.58) [16] | 11.81 (0.96) [11] | 10.85 (0.71) [14] | 13.78 (0.11) [2] |
| SVHN 2 | 14.90 (1.35) [3] | 13.42 (1.30) [9] | 13.56 (1.07) [8] | 12.73 (1.31) [15] | 13.70 (1.38) [6] | 11.94 (1.15) [18] | 14.35 (1.30) [4] | 16.28 (1.40) [2] | 13.26 (1.27) [10] | 12.46 (1.23) [16] | 14.33 (1.05) [5] | 7.37 (0.94) [20] | 12.80 (1.25) [14] | 12.86 (1.58) [13] | 13.65 (1.36) [7] | 13.09 (1.40) [11] | 10.33 (0.47) [19] | 12.92 (1.19) [12] | 12.16 (1.36) [17] | 16.53 (0.10) [1] |
| SVHN 3 | 11.95 (1.01) [2] | 10.82 (0.64) [9] | 12.85 (1.12) [1] | 9.94 (0.75) [14] | 10.63 (0.75) [12] | 9.24 (0.92) [17] | 11.79 (1.01) [4] | 11.36 (0.95) [8] | 10.80 (0.86) [10] | 8.28 (0.72) [19] | 11.71 (1.11) [5] | 6.50 (1.35) [20] | 9.93 (0.71) [15] | 11.89 (1.49) [3] | 11.45 (0.84) [7] | 10.78 (0.98) [11] | 9.16 (0.30) [18] | 10.51 (0.79) [13] | 9.56 (0.82) [16] | 11.71 (0.06) [5] |
| SVHN 4 | 11.80 (0.44) [4] | 10.69 (0.31) [9] | 10.08 (0.63) [15] | 10.34 (0.29) [12] | 11.08 (0.37) [7] | 8.35 (0.66) [19] | 11.18 (0.46) [6] | 13.57 (0.62) [1] | 10.56 (0.37) [11] | 11.32 (0.75) [5] | 12.11 (0.65) [3] | 6.10 (1.20) [20] | 10.32 (0.32) [13] | 8.98 (0.61) [17] | 11.03 (0.42) [8] | 10.58 (0.47) [10] | 8.80 (0.44) [18] | 10.21 (0.36) [14] | 9.71 (0.49) [16] | 12.44 (0.08) [2] |
| SVHN 5 | 13.63 (0.91) [6] | 12.85 (1.23) [9] | 13.99 (1.13) [4] | 11.65 (0.71) [15] | 12.46 (0.61) [11] | 10.19 (0.87) [17] | 14.04 (1.23) [3] | 12.71 (0.91) [9] | 12.98 (0.68) [8] | 9.75 (0.64) [18] | 13.79 (0.78) [5] | 6.74 (0.97) [20] | 11.85 (0.85) [14] | 14.85 (1.65) [2] | 12.87 (0.99) [8] | 13.42 (0.56) [7] | 9.29 (0.65) [19] | 12.40 (0.76) [12] | 11.29 (0.78) [16] | 15.31 (0.85) [1] |
| SVHN 6 | 9.67 (0.82) [3] | 8.47 (0.80) [11] | 9.00 (0.62) [6] | 8.07 (0.84) [15] | 8.80 (0.82) [7] | 8.25 (0.85) [12] | 9.64 (0.80) [4] | 9.33 (0.76) [5] | 8.56 (0.78) [10] | 6.61 (0.58) [20] | 10.07 (0.74) [2] | 6.70 (0.97) [19] | 8.20 (0.81) [13] | 8.05 (0.55) [16] | 8.75 (0.76) [8] | 8.69 (0.55) [9] | 7.47 (0.45) [18] | 8.18 (0.75) [14] | 8.03 (1.03) [17] | 10.34 (0.89) [1] |
| SVHN 7 | 12.75 (1.22) [11] | 12.72 (1.07) [13] | 13.01 (1.09) [6] | 12.85 (1.19) [9] | 12.08 (1.14) [16] | 12.49 (1.10) [14] | 12.99 (1.15) [7] | 12.12 (1.36) [15] | 13.40 (1.29) [3] | 6.66 (0.57) [20] | 12.86 (1.06) [8] | 13.11 (0.73) [5] | 11.87 (1.52) [1] | 10.72 (0.32) [19] | 12.84 (1.20) [10] | 13.04 (1.29) [17] | 13.16 (0.15) [5] |
| SVHN 8 | 11.14 (1.11) [8] | 10.45 (0.97) [10] | 11.55 (0.98) [3] | 9.60 (0.95) [17] | 10.27 (0.91) [12] | 10.23 (0.76) [13] | 10.44 (1.02) [11] | 11.36 (1.30) [6] | 10.68 (1.01) [9] | 6.88 (0.96) [19] | 11.50 (1.17) [4] | 6.48 (1.24) [20] | 9.70 (0.94) [15] | 11.48 (0.73) [2] | 11.33 (0.77) [7] | 11.48 (1.09) [5] | 7.46 (0.73) [18] | 10.19 (1.04) [14] | 9.68 (0.80) [16] | 11.93 (0.13) [1] |
| SVHN 9 | 8.67 (1.23) [7] | 7.71 (1.14) [12] | 8.69 (1.21) [6] | 7.20 (1.11) [17] | 8.15 (1.24) [8] | 7.18 (1.01) [18] | 8.84 (1.27) [4] | 9.18 (1.43) [1] | 7.79 (1.20) [11] | 6.27 (1.11) [19] | 8.81 (1.12) [5] | 5.53 (0.64) [20] | 7.39 (1.09) [15] | 9.18 (1.12) [1] | 8.05 (1.10) [10] | 8.15 (1.24) [8] | 7.40 (0.94) [14] | 7.61 (1.20) [13] | 7.30 (1.07) [16] | 8.99 (0.11) [3] |
| agnews 0 | 15.07 (0.50) [2] | 8.72 (0.28) [11] | 13.57 (0.73) [3] | 5.04 (0.36) [18] | 9.49 (0.30) [7] | 6.32 (0.59) [15] | 12.56 (0.42) [4] | 9.37 (0.43) [8] | 8.51 (0.26) [12] | 5.78 (0.19) [17] | 11.34 (0.47) [5] | 4.79 (0.60) [20] | 5.86 (0.39) [16] | 8.99 (0.67) [9] | 7.10 (0.49) [14] | 11.01 (0.87) [6] | 8.92 (0.23) [10] | 5.93 (0.57) [19] | 19.37 (0.51) [1] |
| agnews 1 | 23.81 (0.41) [3] | 10.14 (0.41) [13] | 27.70 (1.12) [2] | 6.47 (0.29) [18] | 12.26 (0.43) [7] | 9.62 (0.64) [14] | 19.06 (0.30) [4] | 12.48 (0.43) [6] | 11.46 (0.52) [9] | 6.71 (0.51) [17] | 14.30 (0.46) [5] | 4.49 (0.00) [20] | 7.05 (0.29) [15] | 11.84 (1.14) [8] | 6.72 (0.60) [16] | 10.31 (0.37) [12] | 10.55 (0.33) [11] | 10.72 (0.64) [10] | 5.88 (0.75) [19] | 31.53 (1.14) [1] |
| agnews 2 | 29.48 (0.39) [2] | 17.51 (0.57) [12] | 28.93 (0.33) [3] | 9.50 (0.48) [17] | 19.19 (0.52) [9] | 13.15 (0.58) [13] | 24.64 (0.39) [4] | 19.99 (0.61) [6] | 19.28 (0.66) [8] | 10.70 (0.38) [15] | 23.10 (1.13) [5] | 5.34 (0.73) [20] | 11.02 (0.47) [14] | 10.39 (0.40) [16] | 8.25 (0.25) [19] | 17.95 (0.53) [11] | 19.76 (0.45) [7] | 8.72 (0.61) [18] | 39.27 (0.98) [1] |
| agnews 3 | 27.88 (1.49) [2] | 12.25 (0.66) [12] | 20.28 (1.44) [3] | 6.48 (0.31) [18] | 14.19 (0.69) [8] | 9.56 (0.72) [14] | 19.26 (1.05) [4] | 14.20 (0.91) [7] | 12.54 (0.48) [10] | 7.00 (0.35) [17] | 17.99 (1.13) [5] | 4.71 (0.26) [20] | 7.23 (0.15) [16] | 16.70 (1.13) [13] | 7.30 (0.30) [15] | 14.62 (0.52) [6] | 12.46 (0.70) [11] | 13.59 (0.47) [9] | 6.15 (0.21) [19] | 35.46 (0.73) [1] |
| amazon | 7.48 (0.79) [4] | 7.01 (0.62) [5] | 6.06 (0.53) [14] | 5.92 (0.71) [15] | 7.15 (0.71) [5] | 5.80 (0.58) [16] | 7.10 (0.70) [6] | 6.68 (0.64) [9] | 6.59 (0.66) [10] | 6.23 (0.67) [13] | 7.70 (0.69) [1] | 4.49 (0.00) [20] | 6.26 (0.71) [12] | 5.20 (0.31) [18] | 4.86 (0.59) [7] | 7.01 (0.59) [7] | 6.56 (0.73) [11] | 7.67 (0.06) [2] |
| imdb | 3.99 (0.21) [10] | 3.91 (0.17) [13] | 4.05 (0.14) [8] | 3.64 (0.16) [20] | 4.02 (0.16) [9] | 3.73 (0.23) [18] | 3.88 (0.20) [14] | 3.92 (0.24) [12] | 4.15 (0.19) [6] | 3.97 (0.21) [11] | 4.22 (0.16) [4] | 4.99 (0.45) [3] | 3.79 (0.23) [16] | 5.45 (0.36) [1] | 5.11 (0.31) [2] | 3.77 (0.22) [17] | 3.84 (0.15) [15] | 4.18 (0.20) [5] | 3.68 (0.25) [19] | 4.14 (0.07) [7] |
| yelp | 13.99 (1.20) [1] | 11.62 (1.20) [7] | 10.82 (1.00) [9] | 8.53 (0.89) [15] | 11.90 (1.21) [5] | 8.22 (0.51) [16] | 12.11 (1.08) [3] | 12.09 (1.01) [4] | 10.97 (0.88) [11] | 8.89 (0.87) [14] | 11.48 (0.66) [8] | 5.24 (0.86) [19] | 9.16 (1.00) [13] | 5.37 (0.31) [18] | 5.12 (0.17) [20] | 9.19 (0.64) [12] | 11.79 (1.20) [6] | 11.40 (0.78) [9] | 13.06 (0.48) [2] |
| 20news 0 | 26.15 (1.31) [2] | 15.46 (0.92) [12] | 23.13 (1.98) [3] | 7.80 (0.82) [19] | 18.11 (0.93) [9] | 10.56 (1.37) [15] | 21.16 (0.72) [4] | 19.04 (1.31) [6] | 19.67 (1.62) [5] | 9.14 (0.75) [16] | 17.08 (0.92) [10] | 8.34 (3.34) [18] | 8.79 (0.64) [17] | 13.23 (1.11) [13] | 11.29 (1.12) [14] | 18.50 (1.67) [8] | 16.70 (1.03) [11] | 18.40 (1.25) [7] | 7.02 (0.70) [20] | 27.56 (0.41) [1] |
| 20news 1 | 10.45 (2.62) [2] | 6.25 (1.33) [13] | 5.86 (1.00) [15] | 3.93 (0.34) [20] | 7.18 (1.51) [10] | 6.71 (1.45) [11] | 7.84 (1.24) [7] | 7.77 (1.54) [8] | 7.27 (1.28) [9] | 4.22 (0.64) [18] | 6.67 (0.73) [12] | 5.12 (2.02) [16] | 4.11 (0.28) [19] | 9.17 (1.04) [3] | 7.85 (0.61) [6] | 8.20 (1.37) [5] | 6.21 (1.09) [14] | 8.47 (0.84) [4] | 4.49 (0.64) [17] | 11.73 (0.27) [1] |
| 20news 2 | 7.12 (0.81) [3] | 4.18 (0.31) [14] | 5.19 (0.57) [9] | 3.96 (0.66) [17] | 4.91 (0.46) [9] | 5.89 (0.76) [5] | 6.24 (0.89) [4] | 4.47 (0.28) [12] | 3.85 (0.55) [20] | 3.76 (0.55) [20] | 4.82 (0.65) [10] | 5.22 (1.04) [8] | 3.91 (0.66) [19] | 7.98 (1.48) [2] | 7.28 (1.09) [2] | 4.23 (0.44) [13] | 4.03 (0.71) [16] | 7.63 (0.27) [1] |
| 20news 3 | 35.82 (8.16) [3] | 19.80 (6.79) [9] | 17.12 (5.15) [12] | 13.62 (4.67) [17] | 36.95 (0.93) [2] | 29.35 (8.79) [5] | 38.36 (10.20) [1] | 25.96 (6.25) [7] | 17.26 (4.70) [11] | 17.48 (9.59) [10] | 6.64 (1.78) [20] | 10.81 (4.78) [18] | 14.62 (5.25) [15] | 16.48 (3.73) [13] | 15.42 (3.56) [14] | 20.00 (4.03) [8] | 26.34 (9.22) [6] | 13.83 (4.91) [16] | 10.36 (3.45) [19] | 33.12 (6.76) [4] |
| 20news 4 | 11.89 (1.95) [1] | 6.32 (1.59) [18] | 8.04 (1.61) [9] | 6.87 (1.40) [15] | 8.29 (1.69) [6] | 6.66 (1.50) [17] | 11.15 (1.69) [4] | 6.75 (1.63) [16] | 8.20 (1.49) [8] | 7.68 (1.61) [11] | 7.10 (0.84) [13] | 5.76 (1.20) [20] | 6.90 (1.24) [14] | 8.63 (1.02) [5] | 7.63 (0.79) [12] | 11.42 (0.93) [2] | 5.97 (1.38) [19] | 7.79 (1.31) [10] | 8.23 (2.44) [7] | 11.38 (0.12) [3] |

| Model | NCSBAD | KNN | LOF | PCA | KPCA | KDE | GMM | LUNAR | AE | EIF | ABOD | DROCC | GOAD | ICL | SLAD | DTE-C | DTENonParametric | MCM | DDPM | DenoiserAD (Ours) |
|---|---|---|---|---|---|---|---|---|---|---|---|---|---|---|---|---|---|---|---|---|
| 20news 5 | 15.74 (2.51) [2] | 7.15 (1.93) [15] | 8.97 (1.97) [7] | 7.11 (1.63) [16] | 8.55 (1.87) [8] | 8.28 (1.20) [9] | 11.43 (1.43) [3] | 8.25 (1.65) [10] | 8.21 (1.60) [11] | 6.98 (1.75) [18] | 6.41 (0.48) [19] | 7.21 (1.03) [14] | 7.01 (1.67) [17] | 9.76 (1.97) [4] | 9.31 (3.01) [5] | 9.13 (1.63) [6] | 5.92 (0.67) [20] | 7.94 (1.66) [12] | 7.34 (2.11) [13] | 15.97 (0.48) [1] |
| Average | 48.42 (33.31) [3] | 43.32 (32.21) [8] | 42.27 (29.76) [11] | 36.29 (30.73) [15] | 48.40 (34.42) [4] | 44.33 (34.66) [7] | 45.07 (32.52) [6] | 50.33 (34.02) [2] | 41.30 (30.95) [12] | 34.85 (30.55) [17] | 27.45 (28.73) [19] | 21.36 (19.81) [20] | 35.38 (30.49) [16] | 46.38 (33.99) [5] | 43.16 (33.96) [9] | 42.80 (29.95) [10] | 36.86 (28.57) [14] | 40.70 (30.40) [13] | 33.77 (29.30) [18] | 51.95 (33.37) [1] |

*Table 9.* Full results, including mean, standard deviation and rank ADJ AUCPR performance of all methods for each ADBench dataset and the overall average. The best performance is highlighted in **bold**, the second best highlighted with underline.

| Model | NCSBAD | KNN | LOF | PCA | KPCA | KDE | GMM | LUNAR | AE | EIF | ABOD | DROCC | GOAD | ICL | SLAD | DTE-C | DTENonParametric | MCM | DDPM | DenoiserAD (Ours) |
|---|---|---|---|---|---|---|---|---|---|---|---|---|---|---|---|---|---|---|---|---|
| ALOI | 5.50 (0.26) [13] | 5.79 (0.48) [10] | 7.87 (0.74) [2] | 7.65 (0.48) [3] | 5.79 (0.59) [10] | 5.79 (0.41) [10] | 6.81 (0.40) [7] | 8.31 (0.37) [1] | 4.80 (0.39) [16] | 5.92 (0.58) [8] | 6.96 (0.44) [5] | 0.00 (0.00) [20] | 4.51 (0.39) [18] | 4.82 (0.52) [15] | 5.44 (0.46) [14] | 4.20 (0.45) [19] | 5.88 (0.43) [9] | 4.71 (0.27) [17] | 6.94 (0.57) [6] | 7.44 (0.11) [4] |
| annthyroid | 44.25 (1.49) [19] | 61.81 (1.30) [5] | 49.88 (1.49) [14] | 49.75 (1.44) [15] | 58.56 (0.90) [8] | 54.25 (1.36) [11] | 50.06 (1.13) [13] | 47.25 (0.98) [17] | 52.12 (2.40) [12] | 54.37 (1.30) [10] | 58.25 (1.35) [9] | 59.38 (4.50) [6] | 58.75 (6.44) [7] | 44.81 (4.00) [18] | 66.12 (1.33) [2] | 76.44 (1.26) [1] | 62.19 (0.93) [4] | 65.13 (0.92) [3] | 43.50 (2.94) [20] | 48.39 (1.66) [16] |
| backdoor | 88.43 (1.05) [1] | 52.33 (2.05) [12] | 71.94 (1.85) [10] | 8.43 (0.98) [18] | 87.19 (1.28) [4] | 45.19 (1.84) [13] | 85.61 (0.94) [6] | 88.25 (1.14) [2] | 13.16 (0.81) [15] | 8.11 (3.86) [19] | 82.61 (0.74) [8] | 85.55 (0.89) [7] | 12.83 (13.90) [16] | 87.44 (1.30) [3] | 0.00 (0.00) [20] | 79.67 (2.04) [9] | 62.30 (1.58) [11] | 17.32 (8.47) [14] | 11.94 (0.89) [17] | 86.95 (0.86) [5] |
| breastw | 95.96 (0.73) [9] | 95.59 (0.50) [10] | 86.47 (3.07) [17] | 95.39 (0.46) [11] | 96.94 (0.52) [4] | 97.12 (0.60) [3] | 95.30 (0.60) [13] | 96.17 (0.58) [6] | 94.58 (0.30) [15] | 96.28 (0.71) [5] | 84.62 (2.39) [19] | 47.26 (28.25) [20] | 94.98 (0.37) [14] | 96.03 (0.62) [8] | 96.06 (0.65) [7] | 86.39 (3.28) [18] | 97.82 (2.88) [1] | 95.38 (0.62) [12] | 90.47 (2.31) [16] | 97.28 (0.18) [2] |
| campaign | 41.23 (0.39) [17] | 50.17 (0.28) [5] | 42.16 (0.32) [16] | 48.56 (0.28) [10] | 49.94 (0.32) [6] | 47.84 (0.29) [12] | 53.55 (0.35) [1] | 45.14 (0.53) [14] | 49.02 (0.28) [9] | 46.20 (0.66) [13] | 48.25 (0.48) [11] | 0.00 (0.00) [20] | 17.00 (7.13) [19] | 51.03 (0.62) [4] | 49.56 (0.32) [7] | 53.03 (0.59) [2] | 52.62 (0.19) [3] | 44.68 (0.64) [15] | 38.67 (1.26) [18] | |
| Cardiotocography | 39.79 (1.47) [17] | 45.36 (1.08) [8] | 47.57 (1.45) [5] | 61.71 (0.83) [1] | 42.86 (1.04) [10] | 42.86 (0.93) [10] | 45.43 (0.86) [7] | 46.43 (0.78) [6] | 48.29 (0.99) [4] | 56.71 (3.38) [3] | 41.70 (0.93) [13] | 30.14 (8.21) [20] | 57.86 (0.78) [2] | 42.86 (4.48) [10] | 32.43 (1.45) [19] | 36.07 (1.37) [18] | 43.07 (0.95) [9] | 40.14 (0.92) [15] | 40.14 (0.77) [15] | 40.64 (2.11) [14] |
| cardio | 66.23 (1.39) [7] | 61.89 (2.71) [13] | 62.26 (1.98) [11] | 75.09 (2.64) [1] | 64.34 (1.62) [9] | 68.68 (1.10) [5] | 61.89 (2.57) [13] | 70.75 (2.07) [4] | 66.79 (3.50) [6] | 71.32 (0.75) [3] | 47.92 (20.98) [20] | 73.21 (1.94) [2] | 55.47 (3.65) [18] | 62.26 (3.48) [11] | 60.00 (2.57) [16] | 56.04 (3.51) [17] | 65.85 (2.56) [8] | 53.58 (4.60) [19] | 61.48 (0.43) [15] | |
| celeba | 3.84 (1.24) [18] | 17.17 (1.01) [6] | 1.79 (0.26) [19] | 26.95 (0.99) [1] | 15.86 (1.15) [10] | 14.24 (1.27) [12] | 21.39 (1.13) [3] | 9.47 (0.97) [15] | 15.54 (0.89) [11] | 20.58 (1.48) [4] | 1.04 (1.31) [20] | 8.03 (0.90) [16] | 4.31 (4.27) [17] | 13.67 (0.48) [13] | 13.63 (1.37) [14] | 18.71 (2.38) [5] | 16.06 (1.01) [9] | 23.83 (4.19) [2] | 16.90 (1.24) [7] | 16.35 (4.47) [8] |
| census | 19.45 (0.79) [11] | 22.54 (0.69) [3] | 12.90 (0.80) [19] | 20.74 (0.68) [8] | 22.16 (0.59) [5] | 21.30 (0.90) [7] | 20.24 (0.81) [9] | 14.95 (0.44) [16] | 22.82 (0.77) [2] | 13.22 (1.51) [17] | 17.80 (0.75) [14] | 15.48 (1.40) [15] | 9.51 (4.67) [20] | 23.91 (0.89) [1] | 13.18 (10.77) [18] | 17.81 (0.66) [13] | 22.37 (0.54) [4] | 19.86 (1.55) [10] | 19.44 (0.84) [12] | 21.49 (0.63) [6] |
| cover | 80.25 (1.61) [2] | 66.29 (3.08) [9] | 82.45 (2.90) [1] | 16.56 (1.77) [16] | 63.86 (2.85) [10] | 43.18 (2.82) [12] | 18.41 (0.56) [15] | 77.86 (1.95) [4] | 69.61 (6.78) [8] | 77.88 (1.59) [3] | 75.58 (1.25) [5] | 40.86 (5.13) [13] | 0.00 (0.00) [20] | 43.23 (16.17) [11] | 9.83 (4.41) [19] | 73.43 (5.16) [6] | 69.97 (3.45) [7] | 38.63 (10.18) [14] | 10.14 (1.48) [18] | 78.70 (1.89) [3] |
| donors | 44.54 (1.26) [14] | 94.83 (0.85) [6] | 75.15 (1.45) [8] | 37.73 (1.15) [18] | 98.17 (0.45) [2] | 73.02 (0.69) [9] | 44.22 (0.43) [15] | 99.58 (0.16) [1] | 64.30 (4.54) [11] | 48.79 (1.96) [13] | 71.75 (1.54) [10] | 38.45 (20.04) [17] | 10.26 (17.03) [20] | 96.80 (1.45) [4] | 56.26 (8.26) [12] | 86.32 (1.20) [7] | 97.16 (0.53) [3] | 95.61 (2.97) [5] | 31.05 (2.56) [19] | 41.08 (1.22) [16] |
| fault | 54.95 (0.44) [14] | 54.90 (0.53) [15] | 50.20 (0.57) [20] | 55.30 (0.97) [13] | 55.59 (0.87) [11] | 57.87 (0.53) [4] | 54.55 (0.76) [17] | 58.71 (1.03) [3] | 54.75 (1.48) [16] | 55.54 (1.11) [12] | 57.52 (2.84) [5] | 56.09 (1.02) [7] | 57.43 (2.09) [6] | 55.79 (0.92) [10] | 55.84 (0.78) [9] | 58.96 (0.80) [2] | 56.09 (1.47) [7] | 53.40 (0.24) [19] | | |
| fraud | 57.74 (12.37) [5] | 48.68 (4.90) [11] | 63.13 (5.61) [2] | 36.60 (4.45) [15] | 46.82 (11.04) [13] | 43.95 (8.78) [14] | 62.55 (7.89) [3] | 56.81 (8.61) [7] | 52.80 (4.90) [9] | 30.78 (5.30) [17] | 19.56 (9.75) [18] | 0.00 (0.00) [19] | 31.40 (20.25) [16] | 57.11 (9.42) [6] | 50.33 (1.97) [10] | 62.32 (4.49) [4] | 0.00 (0.00) [19] | 47.74 (7.69) [12] | 75.06 (4.46) [1] | 53.80 (6.51) [8] |
| glass | 70.99 (3.68) [3] | 32.63 (15.13) [11] | 27.67 (11.11) [12] | 18.48 (8.78) [19] | 77.06 (9.19) [4] | 43.94 (12.04) [6] | 19.48 (8.01) [18] | 57.59 (13.17) [5] | 40.07 (11.07) [8] | 20.73 (10.37) [16] | 0.00 (0.00) [20] | 25.14 (16.45) [15] | 20.48 (12.35) [17] | 85.85 (8.72) [1] | 34.43 (10.76) [10] | 38.50 (9.03) [9] | 41.27 (11.25) [7] | 27.67 (17.68) [12] | 25.61 (5.86) [14] | 84.63 (7.89) [2] |
| Hepatitis | 99.60 (0.79) [2] | 81.57 (2.85) [11] | 39.86 (8.17) [19] | 59.85 (8.51) [16] | 99.60 (0.79) [2] | 99.60 (0.79) [2] | 78.10 (4.19) [13] | 99.60 (0.79) [2] | 80.10 (4.01) [12] | 82.15 (3.20) [10] | 67.81 (8.35) [15] | 25.37 (16.47) [20] | 58.34 (7.32) [17] | 99.60 (0.79) [2] | 99.60 (0.79) [2] | 94.27 (3.79) [8] | 93.83 (3.45) [9] | 74.09 (6.05) [14] | 53.09 (7.18) [18] | 99.64 (0.71) [1] |
| http | 82.65 (22.66) [13] | 100.00 (0.00) [1] | 96.35 (2.42) [7] | 92.85 (0.98) [9] | 99.44 (0.46) [5] | 99.64 (0.45) [3] | 92.34 (1.08) [10] | 99.03 (0.34) [2] | 99.45 (0.75) [4] | 26.22 (16.42) [18] | 0.00 (0.00) [19] | 0.00 (0.00) [19] | 55.09 (18.06) [15] | 80.41 (38.32) [14] | 89.32 (8.31) [12] | 28.94 (11.35) [17] | 90.54 (2.08) [11] | 47.30 (38.79) [16] | 98.71 (1.76) [6] | 93.37 (12.17) [8] |
| InternetAds | 68.87 (0.72) [1] | 50.86 (0.97) [12] | 54.75 (1.51) [11] | 48.54 (0.97) [13] | 61.09 (1.51) [5] | 60.18 (1.25) [7] | 65.25 (1.58) [3] | 64.43 (1.73) [4] | 45.16 (5.87) [18] | 59.73 (1.28) [8] | 36.92 (3.47) [20] | 59.11 (4.69) [18] | 55.84 (1.78) [10] | 57.47 (0.99) [9] | 64.61 (1.83) [?] | 47.42 (5.78) [14] | 48.05 (0.72) [13] | 44.25 (1.29) [19] | 68.21 (0.82) [2] | |
| Ionosphere | 94.23 (0.47) [5] | 91.27 (2.12) [11] | 87.83 (2.27) [14] | 80.03 (2.64) [19] | 95.52 (0.88) [1] | 95.35 (1.50) [2] | 93.19 (1.29) [9] | 93.76 (1.90) [6] | 93.67 (1.02) [7] | 88.27 (0.96) [13] | 80.48 (2.87) [18] | 66.34 (12.13) [20] | 86.27 (4.56) [17] | 94.39 (0.90) [4] | 93.57 (0.62) [8] | 90.42 (1.17) [12] | 92.08 (1.23) [10] | 86.58 (3.92) [16] | 86.81 (1.40) [15] | 94.92 (1.18) [3] |
| landsat | 49.00 (0.51) [7] | 50.92 (0.82) [5] | 33.60 (0.53) [18] | 53.08 (0.79) [1] | 47.95 (0.94) [8] | 42.65 (1.03) [12] | 30.33 (0.35) [20] | 51.52 (0.78) [4] | 39.98 (0.27) [15] | 43.35 (0.79) [11] | 50.65 (0.69) [6] | 40.05 (3.62) [14] | 33.70 (1.11) [16] | 52.98 (1.54) [2] | 46.77 (0.50) [10] | 33.20 (0.96) [18] | 51.67 (0.86) [3] | 41.23 (0.42) [13] | 33.20 (0.84) [18] | 46.81 (0.63) [9] |
| letter | 4.33 (2.26) [6] | 6.07 (0.82) [15] | 10.33 (1.94) [2] | 0.67 (0.82) [15] | 0.67 (0.82) [15] | 1.67 (1.49) [11] | 1.00 (0.82) [13] | 0.00 (0.00) [18] | 2.33 (1.33) [10] | 5.33 (2.87) [4] | 11.00 (6.80) [1] | 1.00 (0.82) [13] | 6.00 (1.33) [3] | 1.67 (1.49) [11] | 3.00 (1.25) [9] | 0.00 (0.00) [18] | 3.67 (1.94) [8] | 4.00 (1.33) [7] | 4.60 (0.81) [5] | |
| Lymphography | 96.66 (4.10) [9] | 94.25 (5.39) [12] | 76.44 (4.79) [17] | 93.54 (2.94) [13] | 100.00 (0.00) [1] | 100.00 (0.00) [1] | 99.09 (1.82) [8] | 100.00 (0.00) [1] | 95.95 (5.04) [10] | 0.00 (0.00) [20] | 33.82 (27.83) [19] | 95.91 (4.22) [11] | 99.13 (1.74) [7] | 82.41 (9.67) [16] | 70.15 (12.13) [18] | 100.00 (0.00) [1] | 90.11 (4.53) [15] | 100.00 (0.00) [1] | | |
| magic.gamma | 80.58 (0.17) [2] | 76.25 (0.12) [9] | 76.06 (0.15) [10] | 65.24 (0.43) [19] | 75.79 (0.15) [11] | 68.86 (0.23) [16] | 73.13 (0.36) [12] | 78.33 (0.27) [5] | 76.93 (0.24) [7] | 70.37 (1.27) [15] | 77.43 (0.20) [6] | 72.59 (0.69) [13] | 62.65 (1.20) [20] | 70.97 (0.74) [14] | 66.01 (0.22) [17] | 79.92 (0.16) [3] | 81.76 (0.23) [1] | 65.47 (0.79) [18] | 78.82 (0.20) [4] | |
| mammography | 27.44 (4.00) [16] | 39.62 (2.54) [8] | 38.46 (2.60) [11] | 43.33 (2.38) [4] | 44.87 (3.42) [2] | 43.21 (2.28) [5] | 43.59 (1.67) [3] | 45.51 (1.22) [1] | 36.67 (1.24) [13] | 41.92 (3.36) [6] | 39.23 (2.24) [9] | 30.51 (2.94) [15] | 33.97 (4.48) [14] | 20.90 (2.77) [18] | 18.97 (2.45) [19] | 38.33 (2.04) [12] | 40.64 (2.05) [7] | 24.87 (2.04) [17] | 14.62 (1.88) [20] | 39.08 (1.23) [10] |
| mnist | 71.10 (1.79) [8] | 72.52 (0.68) [4] | 71.86 (1.24) [6] | 64.57 (1.05) [16] | 71.52 (0.84) [2] | 74.33 (0.73) [1] | 69.14 (0.98) [10] | 71.95 (1.16) [5] | 69.86 (1.18) [9] | 56.95 (4.69) [18] | 71.48 (0.59) [7] | 59.62 (3.22) [17] | 65.81 (2.63) [14] | 68.72 (0.95) [11] | 73.29 (0.73) [3] | 64.98 (1.31) [15] | 51.90 (0.85) [20] | 68.77 (2.01) [12] | | |
| musk | 100.00 (0.00) [1] | 100.00 (0.00) [1] | 100.00 (0.00) [1] | 100.00 (0.00) [1] | 100.00 (0.00) [1] | 100.00 (0.00) [1] | 91.72 (3.34) [16] | 100.00 (0.00) [1] | 100.00 (0.00) [1] | 58.62 (20.46) [18] | 96.21 (1.29) [15] | 15.86 (17.63) [19] | 100.00 (0.00) [1] | 82.07 (4.31) [17] | 100.00 (0.00) [1] | 100.00 (0.00) [1] | 10.00 (7.89) [20] | 100.00 (0.00) [1] | 97.93 (1.69) [14] | 100.00 (0.00) [1] |
| optdigits | 58.22 (3.49) [2] | 20.22 (5.98) [10] | 50.67 (1.13) [4] | 0.22 (0.44) [20] | 9.11 (2.57) [14] | 41.11 (3.98) [7] | 2.44 (1.30) [18] | 85.56 (2.22) [1] | 9.11 (2.37) [14] | 7.33 (3.03) [16] | 43.11 (1.63) [6] | 20.00 (0.00) [11] | 0.44 (0.54) [19] | 49.56 (4.39) [5] | 39.78 (3.61) [8] | 11.78 (2.29) [12] | 31.56 (4.19) [9] | 9.78 (2.76) [13] | 6.22 (2.18) [17] | 56.00 (2.80) [3] |
| PageBlocks | 61.57 (1.64) [5] | 59.54 (1.00) [11] | 65.56 (0.70) [2] | 46.54 (1.22) [19] | 57.32 (1.11) [13] | 58.63 (0.94) [12] | 56.21 (1.15) [14] | 59.93 (1.03) [8] | 59.93 (1.03) [8] | 44.44 (1.19) [20] | 61.57 (1.03) [5] | 67.91 (2.48) [1] | 51.44 (1.00) [17] | 63.66 (1.02) [3] | 52.16 (1.00) [16] | 61.31 (1.25) [7] | 47.71 (5.01) [18] | 62.16 (1.19) [4] | 53.73 (0.76) [15] | 59.65 (1.78) [10] |
| pendigits | 91.06 (1.73) [5] | 90.64 (1.43) [6] | 74.89 (1.97) [19] | 42.98 (2.74) [15] | 86.17 (1.17) [7] | 92.34 (1.70) [3] | 18.09 (2.93) [19] | 95.53 (0.43) [1] | 65.32 (1.73) [10] | 57.02 (5.89) [13] | 80.85 (1.50) [8] | 17.23 (6.84) [20] | 38.94 (6.04) [17] | 64.04 (7.47) [11] | 55.32 (3.23) [14] | 93.19 (0.85) [2] | 79.57 (3.08) [12] | 23.19 (2.27) [18] | 91.54 (0.75) [4] | |
| Pima | 72.90 (2.20) [5] | 71.58 (2.22) [8] | 66.54 (2.88) [14] | 70.66 (1.15) [9] | 77.85 (1.86) [1] | 71.87 (0.50) [7] | 70.40 (2.38) [10] | 74.05 (2.08) [2] | 66.54 (2.56) [14] | 70.23 (1.94) [11] | 67.07 (0.78) [12] | 45.78 (12.78) [20] | 65.76 (8.91) [16] | 72.90 (2.05) [5] | 59.70 (3.70) [18] | 66.63 (3.11) [13] | 73.01 (1.78) [4] | 65.63 (2.09) [17] | 58.02 (2.76) [19] | 73.98 (2.67) [3] |
| satellite | 72.34 (0.97) [5] | 71.70 (0.57) [10] | 72.68 (0.60) [4] | 62.47 (0.83) [20] | 69.53 (0.84) [15] | 69.79 (0.67) [14] | 71.23 (0.74) [12] | 72.05 (0.73) [7] | 72.03 (0.68) [8] | 66.58 (1.22) [18] | 73.03 (0.60) [3] | 67.97 (3.34) [16] | 64.06 (1.10) [19] | 73.78 (0.46) [2] | 78.15 (0.82) [1] | 71.34 (0.70) [11] | 72.16 (0.76) [6] | 66.64 (0.90) [17] | 69.98 (1.00) [13] | 71.73 (0.45) [9] |
| satimage-2 | 89.77 (1.86) [7] | 91.16 (0.93) [4] | 83.26 (1.54) [18] | 88.37 (0.00) [10] | 91.63 (1.14) [2] | 90.23 (1.74) [5] | 80.00 (2.79) [16] | 89.30 (1.14) [8] | 80.67 (2.71) [15] | 92.56 (0.93) [1] | 82.33 (2.37) [15] | 75.35 (17.00) [17] | 91.63 (1.14) [2] | 89.30 (2.79) [8] | 87.91 (3.09) [14] | 74.88 (1.74) [18] | 95.13 (4.20) [?] | 44.65 (7.41) [20] | 83.66 (1.44) [13] | |
| skin | 90.84 (0.61) [6] | 96.74 (0.45) [2] | 70.51 (1.93) [15] | 37.70 (0.49) [19] | 96.27 (0.27) [3] | 75.32 (0.36) [12] | 78.06 (0.40) [11] | 93.54 (0.48) [4] | 65.06 (3.13) [17] | 79.15 (0.72) [9] | 81.93 (0.73) [7] | 78.07 (1.49) [10] | 53.62 (3.17) [18] | 0.58 (1.11) [20] | 74.42 (2.41) [13] | 81.93 (0.51) [7] | 97.06 (0.45) [1] | 74.28 (0.78) [14] | 67.97 (0.82) [16] | 90.13 (1.47) [6] |
| smtp | 91.49 (15.19) [10] | 74.18 (8.36) [19] | 76.18 (8.32) [1] | 76.18 (8.32) [1] | 76.18 (8.32) [1] | 76.18 (8.32) [1] | 42.56 (9.30) [16] | 76.18 (8.32) [1] | 76.18 (8.32) [1] | 0.00 (0.00) [19] | 15.00 (30.00) [18] | 50.06 (6.93) [14] | 15.71 (16.39) [17] | 76.18 (8.32) [1] | 66.51 (6.34) [13] | 76.18 (8.32) [1] | 80.36 (0.44) [1] | 80.79 (0.39) [8] | 55.68 (24.43) [12] | 69.59 (3.95) [10] |
| SpamBase | 80.85 (0.12) [6] | 80.54 (0.51) [9] | 74.34 (0.53) [20] | 78.53 (0.40) [15] | 81.63 (0.36) [2] | 81.55 (0.54) [3] | 77.86 (0.31) [17] | 79.05 (1.15) [13] | 79.52 (0.45) [11] | 81.21 (1.15) [4] | 80.95 (0.10) [5] | 75.35 (1.42) [19] | 78.75 (0.66) [14] | 79.27 (0.61) [12] | 81.89 (0.44) [1] | 80.36 (0.44) [10] | 80.79 (0.39) [8] | 78.53 (0.37) [15] | 76.15 (0.32) [18] | 80.83 (0.22) [7] |
| speech | 4.32 (1.32) [6] | 2.16 (1.08) [14] | 4.32 (1.32) [6] | 4.32 (1.32) [6] | 8.65 (2.02) [1] | 8.11 (2.42) [2] | 6.49 (2.16) [3] | 1.62 (1.32) [16] | 2.70 (3.42) [13] | 0.54 (1.08) [19] | 4.32 (4.39) [6] | 3.24 (2.65) [12] | 4.32 (1.76) [6] | 5.41 (1.71) [4] | 5.41 (1.71) [4] | 1.62 (1.32) [16] | 1.62 (1.32) [16] | 1.64 (1.47) [15] | | |
| Stamps | 85.71 (3.22) [5] | 79.45 (8.07) [7] | 68.05 (10.26) [11] | 64.72 (6.44) [14] | 94.58 (3.43) [1] | 91.15 (5.26) [2] | 66.63 (9.98) [13] | 80.09 (4.56) [4] | 77.57 (3.07) [8] | 67.85 (5.77) [12] | 8.60 (5.65) [20] | 25.29 (24.99) [19] | 58.49 (15.18) [16] | 71.93 (5.00) [9] | 46.60 (9.04) [18] | 64.72 (6.24) [14] | 85.52 (2.99) [6] | 70.38 (11.44) [10] | 49.67 (5.54) [17] | 91.10 (2.43) [3] |
| thyroid | 65.36 (0.87) [15] | 76.07 (2.67) [5] | 53.21 (4.29) [20] | 78.36 (1.36) [4] | 76.79 (1.96) [3] | 75.36 (2.62) [6] | 73.57 (3.27) [10] | 72.50 (3.68) [11] | 55.71 (3.07) [18] | 78.93 (3.81) [1] | 59.29 (2.86) [17] | 71.43 (2.99) [12] | 75.36 (2.08) [6] | 54.29 (6.25) [19] | 76.79 (2.99) [3] | 78.93 (4.08) [1] | 63.21 (4.87) [16] | 75.36 (0.71) [6] | 70.36 (6.83) [13] | 70.32 (0.36) [14] |
| vertebral | 47.60 (2.31) [14] | 20.45 (4.69) [11] | 29.26 (3.10) [8] | 12.46 (2.37) [17] | 69.99 (6.90) [2] | 9.63 (3.20) [19] | 10.47 (4.46) [18] | 40.20 (10.72) [5] | 25.32 (2.91) [10] | 0.72 (1.45) [20] | 14.62 (17.99) [16] | 15.58 (3.19) [14] | 56.38 (8.57) [3] | 17.08 (8.69) [13] | 34.18 (5.07) [7] | 35.41 (7.21) [6] | 25.30 (2.74) [12] | 71.10 (5.19) [1] | | |
| vowels | 42.67 (3.89) [3] | 31.33 (1.63) [12] | 38.00 (7.48) [6] | 14.67 (1.63) [18] | 33.33 (2.11) [9] | 30.00 (0.00) [13] | 45.33 (4.52) [2] | 44.00 (2.49) [2] | 42.00 (1.63) [5] | 17.33 (5.73) [17] | 32.00 (2.67) [10] | 6.00 (7.42) [19] | 20.67 (6.46) [16] | 32.00 (6.53) [10] | 34.67 (3.40) [8] | 42.67 (4.27) [3] | 6.00 (8.00) [19] | 29.33 (2.49) [14] | 25.33 (4.99) [15] | 36.80 (2.71) [7] |
| Waveform | 11.33 (2.45) [15] | 26.33 (2.45) [4] | 27.67 (2.49) [2] | 10.00 (1.49) [16] | 23.33 (2.79) [9] | 25.33 (2.45) [6] | 9.67 (1.25) [17] | 27.67 (2.00) [2] | 14.00 (0.82) [12] | 11.67 (4.08) [14] | 19.00 (1.33) [10] | 23.67 (7.48) [8] | 9.33 (2.71) [19] | 26.33 (3.71) [4] | 3.33 (1.05) [20] | 13.00 (2.45) [13] | 46.67 (1.49) [1] | 9.67 (3.06) [17] | 14.40 (1.15) [11] | |
| WBC | 91.46 (9.21) [6] | 86.43 (4.15) [11] | 19.47 (8.21) [19] | 89.90 (1.66) [9] | 97.50 (5.00) [1] | 94.72 (5.22) [2] | 78.61 (6.68) [13] | 90.39 (7.00) [7] | 74.10 (7.96) [14] | 91.96 (2.44) [5] | 0.00 (0.00) [20] | 28.55 (26.50) [18] | 89.90 (1.66) [9] | 93.79 (5.94) [4] | 90.05 (5.97) [8] | 33.26 (3.49) [17] | 61.43 (15.28) [16] | 83.87 (6.81) [12] | 71.49 (7.42) [15] | 93.94 (4.17) [3] |
| WDBC | 97.84 (2.70) [1] | 85.74 (4.71) [12] | 91.14 (6.35) [8] | 84.49 (6.30) [14] | 93.64 (3.97) [3] | 91.49 (5.19) [5] | 91.31 (3.92) [6] | 89.89 (7.46) [9] | 88.54 (6.22) [11] | 91.22 (4.82) [7] | 0.00 (0.00) [20] | 10.00 (20.00) [19] | 85.74 (4.71) [12] | 92.81 (4.64) [3] | 88.81 (5.31) [10] | 70.66 (8.12) [16] | 72.93 (14.29) [15] | 92.12 (4.95) [4] | | |
| Wilt | 26.49 (1.45) [4] | 2.47 (0.64) [15] | 16.36 (1.61) [7] | 1.82 (0.64) [17] | 3.90 (0.41) [12] | 0.91 (0.32) [20] | 30.91 (0.88) [3] | 2.08 (1.04) [16] | 3.38 (0.49) [13] | 1.04 (0.32) [19] | 18.18 (1.69) [6] | 1.17 (1.19) [18] | 13.64 (2.69) [9] | 33.77 (5.56) [2] | 7.40 (1.20) [11] | 13.77 (1.90) [8] | 3.25 (0.58) [14] | 43.51 (0.92) [1] | 8.05 (2.77) [10] | 19.14 (1.54) [5] |
| wine | 98.87 (2.26) [5] | 87.11 (2.49) [13] | 83.06 (7.21) [14] | 66.45 (10.85) [19] | 99.62 (0.75) [2] | 99.62 (0.75) [2] | 90.61 (7.04) [11] | 98.11 (3.77) [7] | 90.76 (3.69) [10] | 79.84 (4.46) [15] | 1.21 (1.56) [20] | 15.27 (12.54) [19] | 65.71 (10.47) [17] | 99.21 (0.97) [4] | 100.00 (0.00) [1] | 96.47 (5.14) [8] | 92.45 (2.49) [9] | 87.34 (7.10) [12] | 57.94 (4.76) [18] | 98.18 (3.64) [6] |
| WPBC | 89.50 (3.25) [5] | 50.39 (2.34) [11] | 61.48 (3.88) [14] | 35.14 (3.90) [18] | 90.39 (3.33) [1] | 90.24 (4.11) [2] | 45.03 (3.76) [13] | 84.52 (3.16) [7] | 90.54 (5.45) [15] | 46.02 (5.79) [12] | 21.64 (7.84) [20] | 33.82 (4.14) [19] | 36.88 (4.83) [17] | 87.62 (4.31) [6] | 61.49 (2.91) [9] | 66.50 (3.46) [8] | 56.73 (2.57) [16] | 89.77 (2.75) [4] | | |
| yeast | 49.34 (1.68) [7] | 46.45 (0.73) [11] | 47.83 (0.79) [10] | 43.03 (0.32) [20] | 45.39 (1.28) [14] | 45.39 (1.46) [14] | 46.25 (0.57) [12] | 44.93 (0.34) [17] | 48.03 (0.55) [9] | 44.93 (0.85) [17] | 45.53 (0.82) [13] | 51.51 (3.16) [1] | 51.25 (4.22) [2] | 50.99 (1.75) [3] | 49.41 (1.09) [6] | 50.33 (0.47) [4] | 45.39 (0.91) [14] | 44.74 (0.66) [19] | 49.67 (1.73) [5] | 48.88 (0.40) [8] |
| CIFAR10 0 | 30.89 (2.13) [4] | 29.07 (1.70) [13] | 32.03 (1.73) [1] | 29.49 (1.73) [8] | 30.89 (2.02) [4] | 26.20 (1.46) [16] | 31.39 (2.14) [2] | 29.49 (1.48) [8] | 25.95 (1.44) [17] | 31.39 (1.68) [2] | 15.05 (1.76) [19] | 29.11 (1.79) [13] | 23.42 (1.33) [18] | 30.00 (1.53) [6] | 28.99 (1.67) [14] | 5.95 (1.58) [20] | 29.24 (1.89) [10] | 29.13 (0.78) [12] | | |
| CIFAR10 1 | 21.14 (2.45) [9] | 18.61 (2.49) [14] | 29.75 (2.30) [1] | 18.48 (2.67) [15] | 20.76 (2.45) [10] | 16.71 (2.06) [16] | 23.67 (2.55) [3] | 18.86 (2.06) [13] | 23.02 (2.02) [2] | 9.11 (2.06) [19] | 21.52 (2.89) [8] | 11.77 (2.61) [18] | 19.49 (2.58) [12] | 23.04 (2.39) [5] | 23.54 (1.89) [4] | 22.28 (1.29) [7] | 0.76 (1.52) [20] | 22.40 (2.36) [6] | 15.44 (1.90) [17] | 19.85 (0.65) [11] |
| CIFAR10 2 | 17.59 (2.06) [7] | 15.44 (2.14) [16] | 20.13 (2.66) [1] | 15.19 (2.50) [17] | 16.33 (2.51) [12] | 16.46 (1.65) [10] | 17.85 (1.85) [5] | 18.48 (1.85) [3] | 14.18 (2.10) [18] | 17.09 (1.06) [8] | 12.91 (2.49) [19] | 15.82 (2.47) [14] | 16.33 (2.61) [13] | 17.72 (2.53) [6] | 18.48 (2.70) [3] | 5.43 (1.65) [20] | 16.46 (2.83) [10] | 17.68 (1.40) [4] | 18.56 (0.63) [2] | |
| CIFAR10 3 | 16.58 (2.10) [10] | 19.37 (1.73) [4] | 16.71 (2.29) [8] | 17.22 (2.21) [7] | 14.94 (1.68) [16] | 18.10 (2.42) [5] | 17.97 (2.18) [3] | 17.59 (2.26) [5] | 11.65 (0.76) [19] | 16.58 (1.85) [10] | 12.15 (2.31) [18] | 16.33 (2.61) [13] | 14.18 (1.32) [17] | 17.72 (2.12) [6] | 16.58 (1.85) [10] | 7.22 (6.74) [20] | 17.99 (1.42) [5] | 15.06 (2.02) [15] | 18.56 (0.63) [2] | |
| CIFAR10 4 | 39.24 (1.92) [1] | 34.94 (1.62) [5] | 35.32 (1.57) [9] | 35.82 (1.86) [7] | 33.42 (2.21) [17] | 37.22 (2.02) [3] | 37.09 (1.42) [4] | 34.03 (1.74) [14] | 33.92 (1.17) [14] | 36.08 (2.56) [5] | 18.35 (4.20) [19] | 35.70 (1.53) [8] | 30.13 (2.42) [18] | 35.95 (1.47) [6] | 33.92 (1.42) [14] | 5.57 (10.83) [20] | 35.06 (1.29) [10] | 38.94 (0.78) [2] | | |
| CIFAR10 5 | 15.70 (1.47) [6] | 13.67 (1.77) [15] | 19.62 (1.65) [1] | 14.05 (1.52) [14] | 15.32 (1.29) [9] | 13.54 (1.24) [17] | 16.96 (2.02) [3] | 15.44 (1.03) [7] | 16.58 (2.13) [4] | 10.00 (0.74) [18] | 15.44 (1.24) [7] | 9.87 (1.90) [19] | 14.30 (0.65) [13] | 13.67 (2.85) [15] | 16.20 (2.14) [5] | 18.35 (1.20) [2] | 5.73 (6.11) [20] | 14.14 (1.29) [12] | 14.94 (2.52) [10] | 14.90 (0.44) [11] |
| CIFAR10 6 | 26.96 (1.53) [2] | 25.06 (1.03) [11] | 27.85 (2.74) [1] | 23.29 (1.01) [15] | 26.20 (1.73) [6] | 22.15 (2.30) [16] | 26.58 (1.33) [5] | 26.08 (1.16) [8] | 24.68 (1.06) [12] | 20.38 (1.09) [17] | 26.20 (1.30) [6] | 17.34 (1.73) [19] | 24.05 (1.44) [13] | 19.24 (1.99) [18] | 26.84 (1.30) [4] | 26.96 (2.14) [2] | 2.15 (3.68) [20] | 25.70 (1.24) [10] | 23.92 (1.35) [14] | 26.01 (0.71) [9] |
| CIFAR10 7 | 24.94 (2.18) [7] | 22.66 (0.25) [11] | 28.61 (2.17) [1] | 25.44 (1.62) [6] | 24.68 (1.92) [10] | 18.73 (1.17) [18] | 26.20 (1.66) [3] | 24.94 (2.73) [7] | 26.46 (1.67) [2] | 21.14 (1.86) [16] | 22.91 (1.01) [15] | 15.32 (3.48) [19] | 25.82 (1.57) [5] | 19.11 (1.41) [17] | 24.30 (1.36) [11] | 21.27 (1.07) [?] | 5.57 (7.01) [20] | 24.56 (0.25) [5] | 22.28 (1.57) [12] | 22.05 (0.72) [13] |
| CIFAR10 8 | 24.68 (1.06) [4] | 22.66 (0.25) [11] | 28.48 (0.69) [1] | 20.76 (0.47) [17] | 23.54 (0.47) [9] | 21.39 (1.67) [14] | 24.30 (0.51) [6] | 27.09 (1.89) [2] | 24.18 (0.47) [7] | 18.86 (1.76) [18] | 23.80 (1.17) [8] | 17.85 (3.26) [19] | 21.27 (1.77) [15] | 22.78 (1.39) [10] | 25.19 (1.29) [3] | 21.27 (1.03) [15] | 5.57 (7.01) [20] | 24.56 (0.25) [5] | 22.28 (1.57) [12] | 22.05 (0.72) [13] |
| CIFAR10 9 | 28.86 (1.36) [7] | 27.72 (1.76) [9] | 32.28 (1.88) [1] | 25.70 (1.63) [15] | 27.72 (1.29) [9] | 23.42 (2.00) [17] | 30.76 (1.10) [2] | 28.99 (1.47) [6] | 27.34 (1.25) [12] | 21.77 (1.36) [18] | 29.11 (0.90) [5] | 25.44 (0.74) [16] | 25.19 (1.01) [15] | 28.48 (1.46) [8] | 30.76 (1.53) [2] | 5.44 (10.89) [20] | 27.34 (1.23) [12] | 30.42 (0.42) [4] | | |
| FashionMNIST 0 | 59.47 (2.69) [2] | 53.86 (3.27) [10] | 48.04 (2.92) [17] | 54.60 (2.51) [7] | 48.47 (2.28) [15] | 58.52 (3.25) [4] | 63.07 (1.31) [1] | 54.07 (3.20) [9] | 38.41 (2.63) [18] | 53.44 (2.61) [11] | 26.24 (3.79) [19] | 49.31 (2.83) [14] | 56.51 (3.58) [6] | 50.69 (2.49) [13] | 8.04 (11.72) [20] | 51.43 (3.13) [12] | 48.36 (2.33) [16] | 57.52 (0.73) [3] | | |
| FashionMNIST 1 | 80.00 (1.69) [3] | 76.72 (1.74) [9] | 77.57 (1.59) [8] | 73.76 (1.40) [16] | 77.67 (1.76) [7] | 75.87 (1.63) [12] | 79.37 (1.98) [5] | 82.22 (0.72) [2] | 76.30 (1.81) [10] | 69.31 (2.12) [18] | 79.79 (1.47) [4] | 44.55 (15.11) [19] | 74.39 (1.88) [15] | 78.41 (0.62) [6] | 76.30 (1.75) [10] | 73.54 (2.04) [17] | 23.07 (9.71) [20] | 55.56 (1.47) [13] | 74.12 (1.36) [14] | 82.35 (0.43) [1] |
| FashionMNIST 2 | 54.29 (1.79) [5] | 49.31 (1.69) [12] | 54.71 (2.26) [3] | 45.50 (2.39) [16] | 50.58 (2.36) [8] | 44.23 (1.82) [17] | 53.76 (1.91) [6] | 54.39 (1.65) [4] | 49.42 (1.79) [11] | 29.52 (1.88) [18] | 50.48 (2.23) [9] | 28.25 (12.42) [19] | 46.24 (2.38) [14] | 56.19 (2.26) [2] | 50.05 (1.73) [10] | 51.96 (2.28) [7] | 5.40 (6.55) [20] | 48.61 (1.50) [13] | 44.64 (1.26) [14] | 58.73 (0.57) [1] |
| FashionMNIST 3 | 60.95 (0.40) [4] | 56.51 (0.70) [10] | 61.16 (1.40) [3] | 54.81 (0.72) [14] | 57.88 (0.72) [8] | 52.40 (0.40) [17] | 60.11 (0.42) [5] | 62.96 (1.30) [2] | 56.51 (0.91) [10] | 45.40 (4.12) [18] | 58.84 (1.58) [7] | 30.69 (4.55) [19] | 54.39 (0.70) [15] | 59.79 (1.21) [6] | 56.72 (1.03) [9] | 55.15 (1.18) [13] | 10.37 (14.31) [20] | 56.30 (1.23) [12] | 53.54 (1.22) [16] | 63.24 (0.55) [1] |
| FashionMNIST 4 | 55.87 (1.26) [4] | 51.53 (1.09) [11] | 47.94 (1.09) [17] | 53.12 (1.36) [9] | 49.31 (0.78) [14] | 55.87 (1.44) [4] | 55.34 (2.05) [6] | 51.85 (1.54) [10] | 33.76 (2.66) [18] | 53.33 (0.91) [8] | 24.87 (10.13) [20] | 48.04 (1.08) [16] | 53.72 (1.57) [7] | 56.30 (1.23) [12] | 55.18 (1.83) [5] | 61.02 (0.81) [1] | | | | |
| FashionMNIST 5 | 79.79 (1.22) [9] | 79.89 (1.21) [7] | 79.05 (1.48) [14] | 79.79 (1.40) [9] | 80.42 (1.11) [3] | 80.85 (1.13) [2] | 79.37 (1.30) [13] | 80.21 (1.25) [5] | 80.11 (1.04) [6] | 74.81 (1.23) [18] | 81.06 (1.88) [1] | 46.77 (20.16) [19] | 79.47 (1.22) [12] | 78.52 (1.52) [17] | 79.89 (1.11) [7] | 76.93 (1.52) [17] | 19.26 (17.06) [20] | 80.32 (1.08) [4] | 79.58 (1.19) [11] | 78.29 (0.38) [16] |
| FashionMNIST 6 | 47.94 (1.91) [2] | 38.41 (1.85) [11] | 45.93 (2.25) [6] | 29.84 (1.23) [16] | 40.42 (1.19) [10] | 36.40 (2.10) [14] | 47.72 (1.71) [3] | 47.62 (0.95) [4] | 38.20 (1.96) [12] | 17.46 (3.86) [18] | 40.53 (1.63) [9] | 13.76 (7.51) [19] | 32.91 (1.62) [15] | 47.62 (1.86) [4] | 41.69 (1.69) [8] | 43.70 (2.43) [7] | 6.14 (6.17) [20] | 37.04 (1.34) [13] | 29.52 (1.22) [17] | 53.40 (0.62) [1] |

*Continued on next page*

| Model | NCSBAD | KNN | LOF | PCA | KPCA | KDE | GMM | LUNAR | AE | EIF | ABOD | DROCC | GOAD | ICL | SLAD | DTE-C | DTENonParametric | MCM | DDPM | DenoiserAD (Ours) |
|---|---|---|---|---|---|---|---|---|---|---|---|---|---|---|---|---|---|---|---|---|
| FashionMNIST 7 | 83.39 (1.56) [9] | 83.17 (1.62) [10] | 82.96 (1.36) [14] | 82.86 (1.59) [15] | 83.49 (1.55) [8] | 83.60 (1.74) [6] | 83.92 (1.79) [3] | 84.23 (1.75) [2] | 83.60 (1.50) [6] | 82.22 (1.56) [17] | 84.34 (1.59) [1] | 44.23 (21.33) [19] | 82.86 (1.59) [15] | 83.81 (1.40) [4] | 83.70 (1.51) [5] | 81.80 (1.59) [18] | 23.60 (16.05) [20] | 83.17 (1.72) [10] | 83.07 (1.74) [13] | 83.17 (0.00) [10] |
| FashionMNIST 8 | 40.32 (1.69) [4] | 30.69 (1.74) [11] | 40.53 (1.28) [3] | 23.70 (1.96) [18] | 31.96 (2.16) [10] | 29.64 (1.04) [12] | 38.52 (1.22) [6] | 44.66 (1.44) [2] | 29.31 (1.14) [13] | 24.02 (3.31) [17] | 35.13 (1.66) [7] | 19.05 (2.63) [19] | 25.08 (2.00) [16] | 39.89 (2.31) [5] | 33.65 (1.52) [9] | 34.07 (1.97) [8] | 17.78 (10.46) [20] | 27.30 (1.91) [14] | 25.40 (1.46) [15] | 46.98 (0.83) [1] |
| FashionMNIST 9 | 78.20 (1.08) [3] | 76.83 (1.08) [10] | 78.10 (1.36) [4] | 75.87 (1.66) [15] | 77.67 (1.44) [7] | 76.83 (1.69) [10] | 77.14 (1.44) [9] | 81.06 (2.35) [1] | 77.57 (1.14) [8] | 71.85 (2.20) [18] | 78.73 (1.03) [2] | 41.90 (18.05) [19] | 75.66 (1.30) [16] | 77.99 (1.09) [6] | 76.40 (1.28) [13] | 74.50 (1.40) [17] | 23.49 (15.40) [20] | 76.61 (1.27) [12] | 76.30 (1.31) [14] | 78.10 (0.53) [4] |
| MNIST-C brightness | 50.00 (2.11) [3] | 30.27 (0.77) [12] | 39.40 (1.72) [7] | 23.73 (0.68) [18] | 32.47 (0.96) [11] | 32.53 (1.82) [10] | 39.60 (1.67) [6] | 53.20 (1.07) [2] | 28.20 (0.96) [13] | 19.13 (1.54) [20] | 41.67 (1.21) [5] | 22.60 (2.48) [19] | 24.20 (0.34) [17] | 47.20 (1.50) [4] | 35.13 (1.34) [9] | 26.00 (3.99) [15] | 26.93 (0.53) [14] | 25.80 (1.41) [16] | 72.08 (0.83) [1] |
| MNIST-C canny edges | 51.93 (1.36) [3] | 31.47 (1.33) [12] | 50.80 (0.98) [4] | 23.73 (0.98) [18] | 32.27 (1.53) [10] | 32.13 (1.72) [11] | 39.40 (1.51) [8] | 56.73 (1.37) [2] | 31.40 (1.70) [13] | 21.07 (3.65) [19] | 42.27 (0.39) [6] | 27.93 (9.17) [15] | 43.87 (1.96) [5] | 36.80 (1.42) [9] | 44.07 (1.64) [7] | 20.80 (10.17) [20] | 28.87 (1.56) [14] | 24.27 (1.84) [17] | 69.72 (0.84) [1] |
| MNIST-C dotted line | 59.80 (1.31) [3] | 44.60 (0.93) [10] | 52.13 (1.24) [5] | 37.27 (1.10) [16] | 46.67 (1.19) [8] | 39.13 (1.02) [14] | 50.73 (1.48) [6] | 64.47 (1.13) [2] | 43.47 (1.42) [12] | 30.87 (3.30) [18] | 52.93 (1.48) [4] | 28.87 (4.21) [19] | 38.87 (1.67) [15] | 48.87 (1.45) [7] | 45.20 (1.29) [9] | 44.47 (1.64) [11] | 21.60 (11.75) [20] | 41.00 (1.19) [13] | 36.27 (2.07) [17] | 73.48 (0.20) [1] |
| MNIST-C fog | 79.20 (1.65) [3] | 60.53 (1.56) [11] | 72.53 (1.24) [5] | 61.47 (1.53) [10] | 58.53 (1.75) [14] | 73.60 (1.58) [4] | 86.60 (0.65) [2] | 89.20 (1.07) [2] | 59.93 (1.51) [12] | 41.60 (2.82) [18] | 71.93 (1.12) [6] | 27.27 (9.36) [20] | 55.13 (1.44) [16] | 69.00 (0.89) [7] | 30.13 (15.60) [19] | 59.33 (1.21) [13] | 55.53 (1.28) [15] | 92.60 (0.46) [1] |
| MNIST-C glass blur | 84.67 (0.76) [3] | 74.27 (0.80) [10] | 78.40 (0.90) [6] | 69.47 (1.29) [17] | 76.13 (1.13) [8] | 69.73 (1.10) [16] | 81.20 (1.07) [4] | 89.20 (1.07) [2] | 73.67 (0.92) [12] | 61.47 (4.70) [18] | 80.27 (0.83) [5] | 39.67 (15.32) [19] | 70.20 (1.15) [15] | 77.20 (0.65) [7] | 74.20 (0.69) [11] | 74.73 (1.18) [9] | 35.20 (12.55) [20] | 71.80 (1.20) [13] | 70.33 (1.25) [14] | 90.28 (0.20) [1] |
| MNIST-C identity | 8.33 (0.56) [15] | 9.00 (0.60) [10] | 8.60 (0.77) [11] | 8.47 (1.56) [14] | 8.60 (0.74) [11] | 8.53 (0.34) [13] | 9.60 (0.88) [2] | 9.20 (0.50) [6] | 9.47 (0.96) [3] | 7.53 (0.62) [18] | 9.87 (0.72) [1] | 9.20 (1.50) [6] | 8.33 (1.14) [15] | 7.40 (1.27) [19] | 9.07 (1.16) [9] | 9.13 (0.81) [8] | 8.20 (1.65) [17] | 9.47 (1.00) [3] | 9.33 (0.73) [5] | 7.16 (0.50) [20] |
| MNIST-C impulse noise | 98.07 (0.39) [5] | 97.27 (0.25) [13] | 97.20 (0.34) [15] | 97.27 (0.25) [13] | 96.27 (0.33) [18] | 98.20 (0.27) [4] | 97.93 (0.33) [6] | 99.33 (0.30) [1] | 97.33 (0.21) [11] | 92.00 (2.12) [18] | 97.93 (0.25) [6] | 66.07 (34.15) [19] | 97.33 (0.21) [11] | 98.67 (0.23) [3] | 98.80 (0.27) [2] | 92.40 (0.49) [17] | 37.47 (13.22) [20] | 97.87 (0.27) [9] | 97.60 (0.44) [10] | 97.88 (0.10) [8] |
| MNIST-C motion blur | 73.07 (1.51) [3] | 56.53 (1.09) [12] | 65.07 (1.10) [6] | 50.07 (1.82) [17] | 58.53 (0.88) [9] | 58.00 (1.14) [10] | 62.87 (1.07) [7] | 76.00 (0.79) [2] | 55.20 (1.53) [13] | 38.73 (2.45) [18] | 67.20 (0.98) [5] | 32.73 (10.06) [19] | 50.73 (1.12) [16] | 69.60 (1.55) [4] | 57.60 (1.24) [11] | 61.13 (1.44) [8] | 26.60 (8.70) [20] | 53.93 (1.83) [14] | 51.07 (1.29) [15] | 82.52 (0.45) [1] |
| MNIST-C rotate | 19.67 (0.99) [3] | 13.87 (0.58) [14] | 15.20 (0.88) [8] | 11.93 (0.74) [18] | 14.33 (0.70) [10] | 14.20 (0.86) [11] | 16.73 (0.57) [5] | 21.73 (1.00) [2] | 13.93 (0.57) [13] | 11.00 (1.53) [19] | 15.93 (0.57) [6] | 14.20 (1.38) [11] | 12.40 (0.93) [17] | 18.00 (0.84) [4] | 14.67 (0.99) [9] | 15.33 (0.92) [7] | 9.47 (2.90) [20] | 13.07 (0.49) [15] | 12.53 (0.69) [16] | 25.08 (0.41) [1] |
| MNIST-C scale | 45.53 (1.20) [3] | 33.27 (1.47) [12] | 40.87 (1.61) [5] | 27.27 (0.98) [17] | 35.47 (1.57) [9] | 28.60 (1.29) [15] | 41.80 (1.22) [4] | 48.73 (1.06) [2] | 33.73 (1.51) [11] | 18.67 (4.27) [20] | 36.27 (1.90) [7] | 21.20 (0.96) [19] | 28.67 (1.32) [14] | 36.20 (1.02) [8] | 34.40 (1.47) [10] | 38.93 (0.85) [6] | 24.13 (7.62) [18] | 32.93 (1.47) [13] | 28.47 (1.50) [16] | 54.48 (1.32) [1] |
| MNIST-C shear | 34.40 (1.50) [3] | 26.20 (1.13) [12] | 29.07 (1.54) [6] | 22.73 (1.14) [17] | 27.07 (1.04) [10] | 23.33 (1.05) [14] | 30.00 (0.79) [5] | 35.13 (1.92) [2] | 26.33 (1.44) [11] | 21.20 (1.71) [18] | 28.53 (1.67) [7] | 23.33 (0.92) [8] | 22.87 (1.59) [16] | 28.33 (0.93) [8] | 27.73 (1.24) [9] | 31.33 (1.56) [4] | 14.80 (5.37) [20] | 29.20 (1.20) [6] | 25.08 (0.41) [13] | 38.92 (0.41) [1] |
| MNIST-C shot noise | 53.00 (2.14) [3] | 40.13 (2.35) [9] | 46.93 (1.53) [4] | 31.73 (1.69) [17] | 42.60 (1.90) [8] | 36.60 (1.32) [13] | 46.33 (1.86) [5] | 61.27 (2.61) [2] | 38.00 (2.53) [12] | 27.47 (3.03) [19] | 46.27 (1.81) [6] | 27.07 (5.43) [20] | 32.73 (2.24) [16] | 38.93 (2.16) [11] | 39.67 (2.18) [10] | 44.27 (2.02) [7] | 27.93 (10.26) [18] | 35.27 (1.93) [14] | 32.80 (2.58) [15] | 66.20 (0.49) [1] |
| MNIST-C spatter | 65.47 (1.39) [3] | 53.80 (1.29) [11] | 63.07 (1.67) [5] | 46.60 (1.51) [16] | 55.60 (1.29) [9] | 45.93 (1.02) [17] | 61.80 (1.36) [6] | 71.27 (1.18) [2] | 53.73 (1.69) [12] | 41.67 (3.39) [18] | 63.87 (1.13) [4] | 28.47 (7.82) [19] | 47.73 (1.32) [14] | 57.13 (2.08) [7] | 55.87 (1.50) [9] | 56.07 (1.00) [8] | 24.40 (12.23) [20] | 50.87 (1.81) [13] | 47.53 (1.67) [15] | 75.44 (0.80) [1] |
| MNIST-C stripe | 95.80 (0.50) [4] | 94.33 (0.42) [11] | 95.47 (0.16) [5] | 93.07 (0.39) [16] | 95.40 (0.57) [6] | 94.53 (0.86) [8] | 94.67 (0.56) [7] | 98.40 (0.80) [1] | 93.80 (0.27) [13] | 88.07 (1.84) [18] | 94.47 (0.81) [9] | 44.67 (22.65) [19] | 93.20 (0.65) [15] | 96.67 (0.47) [3] | 94.40 (0.49) [10] | 95.40 (0.77) [14] | 93.40 (0.77) [14] | 97.40 (0.18) [2] |
| MNIST-C translate | 37.53 (1.20) [2] | 22.67 (1.23) [13] | 32.13 (1.33) [5] | 18.47 (1.00) [19] | 24.20 (1.53) [10] | 22.87 (1.51) [11] | 30.87 (1.53) [6] | 37.47 (0.62) [3] | 22.87 (0.88) [11] | 14.20 (0.69) [20] | 30.67 (1.52) [7] | 20.07 (2.31) [15] | 18.80 (0.78) [17] | 33.07 (1.22) [4] | 25.73 (0.83) [9] | 30.47 (0.88) [8] | 19.53 (4.44) [16] | 22.00 (0.76) [14] | 18.53 (1.36) [18] | 52.48 (0.85) [1] |
| MNIST-C zigzag | 69.07 (1.04) [3] | 53.93 (1.29) [11] | 61.80 (1.20) [5] | 46.40 (1.47) [15] | 56.40 (1.02) [8] | 45.40 (1.22) [17] | 61.67 (1.32) [6] | 70.20 (0.96) [2] | 53.80 (1.87) [12] | 38.07 (3.05) [18] | 63.07 (0.44) [4] | 29.80 (5.66) [19] | 47.67 (1.74) [14] | 60.20 (1.80) [7] | 55.00 (1.59) [10] | 55.07 (1.65) [9] | 35.87 (9.54) [20] | 52.00 (0.92) [13] | 46.13 (2.01) [16] | 81.04 (0.22) [1] |
| MVTec-AD bottle | 94.11 (1.94) [3] | 91.53 (0.57) [14] | 91.67 (0.58) [12] | 89.18 (1.45) [17] | 94.07 (1.63) [4] | 92.62 (2.12) [8] | 94.25 (1.43) [2] | 93.91 (1.76) [5] | 92.15 (0.38) [10] | 89.95 (1.77) [16] | 46.28 (6.17) [20] | 81.51 (8.43) [19] | 90.09 (1.35) [15] | 92.16 (1.29) [9] | 93.79 (1.96) [6] | 92.96 (3.04) [7] | 92.08 (2.54) [11] | 91.61 (0.95) [13] | 88.23 (2.44) [18] | 94.31 (1.54) [1] |
| MVTec-AD cable | 79.32 (2.68) [2] | 59.84 (3.34) [13] | 59.25 (3.37) [14] | 54.37 (2.68) [17] | 74.12 (3.69) [8] | 78.21 (2.23) [5] | 72.43 (4.35) [9] | 60.46 (2.45) [13] | 55.60 (3.44) [16] | 47.15 (2.50) [19] | 42.60 (10.17) [20] | 56.66 (2.36) [15] | 77.51 (1.52) [6] | 78.38 (2.15) [4] | 77.12 (1.20) [7] | 70.47 (3.98) [10] | 60.95 (2.62) [14] | 51.58 (3.40) [18] | 79.09 (2.50) [3] |
| MVTec-AD capsule | 80.88 (1.81) [6] | 62.94 (1.60) [11] | 62.25 (3.35) [12] | 56.07 (2.57) [18] | 83.40 (2.54) [3] | 82.82 (2.07) [5] | 71.18 (2.66) [8] | 60.96 (2.45) [13] | 57.06 (1.98) [16] | 55.28 (1.79) [19] | 52.85 (2.30) [20] | 57.28 (2.17) [15] | 83.27 (2.21) [4] | 79.45 (1.69) [7] | 83.84 (3.13) [2] | 70.84 (1.17) [9] | 60.95 (2.62) [14] | 56.17 (1.92) [17] | 84.58 (1.89) [1] |
| MVTec-AD carpet | 74.95 (1.59) [5] | 60.32 (2.46) [12] | 60.43 (3.03) [11] | 58.08 (4.47) [17] | 76.08 (1.54) [2] | 74.85 (2.01) [6] | 75.50 (1.16) [3] | 67.87 (4.30) [9] | 59.28 (3.07) [14] | 59.01 (2.97) [15] | 51.04 (4.67) [19] | 49.48 (11.46) [20] | 58.36 (4.97) [16] | 75.11 (3.20) [4] | 74.55 (1.88) [7] | 70.53 (3.98) [8] | 67.50 (1.97) [10] | 59.67 (4.24) [13] | 53.23 (3.18) [18] | 77.11 (0.96) [1] |
| MVTec-AD grid | 76.11 (3.18) [3] | 53.76 (1.34) [13] | 53.19 (1.22) [14] | 50.86 (1.59) [16] | 74.18 (2.91) [6] | 74.40 (3.17) [5] | 75.10 (3.73) [4] | 65.30 (6.70) [9] | 53.55 (1.61) [13] | 49.17 (3.15) [18] | 34.07 (5.74) [20] | 42.36 (8.78) [19] | 51.82 (1.55) [15] | 76.30 (2.88) [2] | 72.64 (4.40) [7] | 67.23 (3.93) [8] | 49.36 (2.02) [17] | 77.39 (2.63) [1] |
| MVTec-AD hazelnut | 66.83 (3.48) [5] | 52.73 (3.07) [14] | 53.95 (1.56) [11] | 50.00 (2.97) [17] | 66.43 (3.91) [6] | 67.21 (2.58) [4] | 64.77 (4.20) [8] | 62.13 (3.37) [9] | 52.56 (1.93) [15] | 52.75 (4.24) [13] | 44.31 (5.45) [19] | 35.96 (8.91) [20] | 50.48 (3.16) [16] | 70.58 (5.72) [1] | 67.60 (5.79) [3] | 36.50 (5.66) [7] | 54.27 (2.69) [10] | 53.11 (2.88) [12] | 48.84 (2.37) [18] | 67.73 (4.39) [2] |
| MVTec-AD leather | 97.13 (1.54) [4] | 95.47 (2.18) [13] | 94.31 (2.15) [17] | 95.07 (2.67) [15] | 96.89 (1.09) [6] | 95.85 (1.77) [9] | 97.13 (1.91) [4] | 97.16 (0.50) [2] | 95.72 (1.95) [10] | 94.94 (2.80) [16] | 76.74 (5.37) [19] | 49.55 (16.69) [20] | 95.20 (2.44) [14] | 96.89 (1.19) [6] | 97.14 (1.26) [3] | 96.89 (0.02) [6] | 95.60 (1.96) [11] | 95.53 (1.94) [12] | 92.37 (1.36) [18] | 97.27 (1.10) [1] |
| MVTec-AD metal nut | 84.17 (2.64) [2] | 61.43 (3.31) [14] | 61.65 (1.91) [12] | 59.16 (2.35) [17] | 82.87 (1.96) [4] | 81.83 (2.32) [5] | 84.07 (2.89) [3] | 72.42 (5.34) [9] | 62.46 (2.42) [11] | 61.50 (1.78) [13] | 53.40 (1.75) [19] | 56.20 (3.94) [20] | 59.63 (3.24) [16] | 79.34 (1.95) [6] | 78.87 (2.92) [7] | 77.93 (3.58) [8] | 67.32 (2.26) [10] | 57.19 (3.15) [18] | 84.62 (2.49) [1] |
| MVTec-AD pill | 77.71 (2.31) [7] | 65.52 (1.96) [11] | 65.14 (3.49) [12] | 63.16 (1.93) [16] | 79.82 (2.49) [5] | 78.62 (3.08) [6] | 82.20 (2.10) [2] | 69.59 (4.29) [10] | 64.41 (2.42) [13] | 62.56 (2.65) [17] | 56.77 (2.06) [18] | 56.20 (3.94) [20] | 63.55 (2.48) [15] | 80.19 (1.21) [4] | 81.37 (1.65) [3] | 72.48 (0.94) [8] | 69.60 (2.26) [9] | 64.08 (2.89) [14] | 56.37 (2.15) [19] | 82.74 (0.95) [1] |
| MVTec-AD screw | 72.09 (3.92) [6] | 51.12 (4.67) [12] | 51.95 (4.45) [11] | 45.55 (2.66) [17] | 73.62 (2.82) [5] | 74.51 (1.88) [2] | 73.74 (3.60) [4] | 64.06 (0.94) [9] | 51.06 (3.51) [13] | 45.26 (6.22) [18] | 43.33 (5.35) [20] | 47.14 (3.04) [16] | 75.96 (3.40) [1] | 69.91 (2.47) [7] | 64.25 (3.10) [8] | 58.02 (4.24) [10] | 50.01 (3.82) [14] | 44.00 (3.77) [19] | 74.30 (2.70) [3] |
| MVTec-AD tile | 84.63 (1.54) [6] | 74.54 (1.66) [11] | 73.97 (2.29) [14] | 73.13 (2.66) [15] | 85.06 (1.64) [4] | 81.05 (1.97) [9] | 83.56 (3.26) [7] | 81.32 (3.50) [8] | 74.37 (2.75) [12] | 72.93 (2.01) [16] | 57.94 (3.62) [19] | 46.00 (13.53) [20] | 71.27 (4.42) [17] | 84.77 (2.28) [5] | 85.30 (1.60) [2] | 85.27 (3.03) [3] | 78.47 (0.94) [10] | 74.26 (2.85) [13] | 66.31 (1.20) [18] | 85.58 (1.93) [1] |
| MVTec-AD toothbrush | 98.56 (0.91) [8] | 89.20 (4.33) [11] | 83.40 (5.20) [13] | 70.23 (2.72) [17] | 99.41 (0.75) [2] | 99.41 (0.75) [2] | 99.41 (0.75) [2] | 98.57 (1.22) [7] | 80.41 (3.58) [15] | 85.04 (6.00) [12] | 68.50 (7.80) [18] | 48.81 (15.02) [20] | 71.36 (1.31) [16] | 99.18 (1.16) [5] | 94.09 (5.90) [9] | 93.33 (2.97) [10] | 81.58 (3.60) [14] | 62.61 (1.61) [19] | 99.43 (0.74) [1] |
| MVTec-AD transistor | 76.14 (5.12) [2] | 46.36 (6.04) [14] | 45.47 (5.40) [16] | 47.20 (3.84) [12] | 76.40 (3.87) [1] | 75.24 (3.81) [4] | 73.30 (2.23) [6] | 68.41 (6.35) [9] | 45.20 (2.04) [17] | 48.47 (8.50) [11] | 21.11 (11.63) [20] | 38.80 (13.25) [19] | 46.36 (3.65) [14] | 72.73 (3.06) [7] | 74.61 (4.09) [5] | 71.50 (5.12) [8] | 54.31 (3.83) [10] | 46.89 (2.58) [13] | 45.12 (3.53) [18] | 75.75 (3.02) [3] |
| MVTec-AD wood | 82.71 (3.13) [2] | 61.15 (2.62) [14] | 67.09 (1.95) [11] | 59.03 (3.91) [18] | 82.70 (2.72) [3] | 82.32 (2.33) [5] | 83.84 (3.37) [1] | 74.46 (3.16) [13] | 62.55 (1.81) [12] | 60.95 (2.05) [15] | 49.29 (2.54) [20] | 54.02 (9.36) [19] | 59.39 (4.06) [17] | 82.13 (2.81) [6] | 81.37 (4.81) [7] | 78.12 (3.07) [8] | 67.84 (4.64) [10] | 61.67 (3.34) [13] | 60.83 (2.85) [16] | 82.59 (1.80) [4] |
| MVTec-AD zipper | 92.52 (1.68) [2] | 77.83 (3.81) [14] | 79.17 (3.69) [11] | 72.77 (2.78) [17] | 91.43 (2.16) [3] | 89.60 (3.68) [6] | 90.22 (2.89) [5] | 86.22 (4.17) [9] | 77.84 (3.16) [13] | 76.54 (3.23) [15] | 69.67 (6.96) [18] | 54.85 (7.62) [20] | 74.03 (2.88) [16] | 89.13 (1.74) [7] | 90.80 (1.89) [4] | 88.67 (2.31) [8] | 82.64 (4.65) [10] | 78.26 (2.52) [12] | 60.51 (4.22) [19] | 93.26 (1.06) [1] |
| SVHN 0 | 18.08 (1.18) [7] | 16.28 (1.84) [12] | 17.95 (0.91) [8] | 14.62 (0.63) [17] | 16.67 (1.86) [10] | 18.33 (1.65) [6] | 19.23 (1.46) [5] | 21.54 (1.65) [3] | 15.77 (1.84) [13] | 13.46 (1.46) [19] | 19.49 (1.93) [4] | 13.85 (2.38) [18] | 15.26 (0.94) [16] | 22.18 (1.26) [1] | 16.54 (2.04) [11] | 17.44 (2.12) [9] | 10.64 (6.80) [20] | 15.77 (1.84) [13] | 22.00 (0.38) [2] |
| SVHN 1 | 20.27 (1.67) [7] | 18.07 (1.22) [11] | 14.80 (1.00) [17] | 18.67 (1.01) [9] | 18.87 (1.28) [7] | 14.07 (1.51) [19] | 19.07 (1.47) [5] | 23.73 (0.88) [1] | 17.47 (1.13) [15] | 17.87 (1.81) [13] | 19.67 (1.12) [4] | 14.63 (1.04) [18] | 19.93 (1.91) [6] | 14.18 (1.44) [19] | 17.91 (1.36) [15] | 19.10 (1.36) [13] | 19.85 (1.34) [7] | 21.93 (1.91) [9] | 20.73 (1.42) [16] | 21.70 (0.18) [2] |
| SVHN 2 | 22.73 (1.83) [4] | 21.20 (2.12) [13] | 22.00 (1.70) [7] | 21.60 (1.81) [10] | 21.53 (2.14) [12] | 19.87 (1.20) [17] | 22.93 (2.06) [3] | 23.00 (1.28) [2] | 22.00 (1.92) [7] | 19.27 (2.15) [18] | 21.60 (1.61) [10] | 15.13 (0.96) [19] | 21.13 (1.60) [14] | 20.87 (1.85) [15] | 22.73 (1.90) [4] | 22.53 (1.92) [6] | 13.00 (4.98) [20] | 21.93 (1.91) [9] | 20.73 (1.42) [16] | 27.20 (0.33) [1] |
| SVHN 3 | 22.24 (1.07) [1] | 19.25 (1.45) [12] | 17.31 (1.50) [17] | 19.55 (1.59) [8] | 18.06 (1.31) [14] | 21.19 (0.98) [4] | 19.33 (1.32) [11] | 19.48 (1.40) [9] | 19.48 (1.55) [9] | 14.63 (1.93) [18] | 14.18 (1.44) [19] | 17.91 (1.36) [15] | 19.10 (1.36) [13] | 19.85 (1.34) [7] | 20.30 (1.07) [5] | 13.60 (5.29) [20] | 17.87 (0.86) [13] | 18.87 (1.24) [7] | 21.70 (0.18) [2] |
| SVHN 4 | 19.83 (0.92) [6] | 20.00 (0.47) [4] | 18.64 (1.30) [15] | 18.98 (0.21) [14] | 20.00 (0.54) [4] | 16.09 (1.36) [18] | 19.66 (0.50) [10] | 21.62 (1.77) [3] | 19.74 (0.34) [8] | 19.83 (1.22) [6] | 22.21 (0.83) [1] | 12.26 (1.81) [19] | 19.23 (0.32) [13] | 16.26 (2.00) [17] | 19.74 (1.03) [8] | 18.17 (0.97) [16] | 1.82 (1.84) [20] | 19.40 (0.51) [11] | 19.40 (0.70) [11] | 22.04 (0.31) [2] |
| SVHN 5 | 19.24 (1.19) [7] | 20.18 (1.47) [12] | 22.86 (1.19) [9] | 19.72 (1.53) [15] | 20.92 (0.99) [10] | 18.89 (1.54) [17] | 23.50 (1.20) [3] | 19.72 (1.58) [15] | 21.29 (1.16) [9] | 15.39 (0.85) [18] | 22.76 (0.95) [6] | 14.65 (0.74) [19] | 19.91 (1.44) [13] | 33.78 (1.11) [2] | 21.75 (1.68) [8] | 23.13 (1.50) [4] | 9.95 (5.66) [20] | 20.92 (1.26) [10] | 19.82 (1.84) [14] | 24.03 (0.49) [1] |
| SVHN 6 | 19.34 (1.10) [5] | 17.68 (1.64) [14] | 18.56 (1.14) [7] | 17.13 (1.56) [15] | 18.78 (1.44) [6] | 16.69 (1.07) [16] | 19.67 (1.14) [3] | 19.45 (1.46) [4] | 18.23 (1.68) [9] | 14.03 (1.08) [18] | 20.00 (2.11) [2] | 12.93 (0.75) [19] | 17.35 (1.66) [12] | 16.57 (1.75) [17] | 18.56 (1.42) [7] | 17.35 (1.45) [12] | 8.18 (2.87) [20] | 18.23 (1.52) [9] | 17.35 (2.06) [12] | 20.07 (0.27) [1] |
| SVHN 7 | 23.18 (2.79) [6] | 22.95 (2.37) [7] | 21.36 (1.37) [17] | 23.18 (2.20) [4] | 22.84 (2.53) [9] | 23.41 (1.81) [3] | 22.16 (2.62) [14] | 24.32 (1.81) [1] | 23.52 (2.26) [2] | 22.27 (1.85) [13] | 22.73 (2.10) [10] | 15.23 (0.84) [19] | 23.18 (2.11) [4] | 21.82 (1.78) [16] | 19.89 (2.27) [18] | 2.50 (5.00) [20] | 22.73 (2.19) [10] | 21.93 (1.71) [15] | 22.93 (1.07) [8] |
| SVHN 8 | 20.00 (1.84) [5] | 17.36 (1.30) [13] | 20.63 (1.84) [3] | 16.35 (1.54) [17] | 17.86 (1.72) [12] | 18.11 (1.34) [11] | 19.62 (1.75) [7] | 19.25 (2.16) [8] | 18.62 (2.16) [9] | 14.47 (2.42) [18] | 21.13 (2.24) [2] | 12.08 (2.33) [19] | 16.48 (1.34) [16] | 20.50 (2.74) [4] | 18.49 (1.67) [10] | 21.89 (1.84) [1] | 2.52 (5.03) [20] | 17.76 (1.62) [13] | 16.86 (1.01) [15] | 19.92 (0.97) [6] |
| SVHN 9 | 17.14 (2.83) [6] | 16.05 (2.53) [10] | 17.01 (2.82) [7] | 15.37 (2.49) [16] | 15.92 (3.12) [12] | 18.10 (2.81) [1] | 16.46 (2.49) [8] | 15.10 (2.52) [17] | 12.93 (3.07) [18] | 14.47 (2.42) [18] | 17.69 (2.89) [2] | 12.24 (1.61) [19] | 15.65 (3.19) [14] | 17.55 (2.41) [3] | 16.05 (2.53) [10] | 16.19 (3.51) [9] | 5.03 (6.27) [20] | 15.65 (2.62) [14] | 15.92 (3.67) [12] | 17.47 (0.54) [4] |
| agnews 0 | 26.13 (0.86) [2] | 16.80 (1.26) [10] | 20.73 (0.68) [3] | 9.67 (1.38) [19] | 17.33 (1.28) [8] | 14.47 (1.28) [14] | 20.47 (1.42) [4] | 17.80 (1.29) [7] | 16.13 (0.81) [11] | 10.73 (1.24) [16] | 19.27 (1.12) [5] | 2.73 (5.47) [20] | 10.53 (1.54) [17] | 15.13 (1.05) [12] | 14.20 (0.88) [15] | 18.73 (0.65) [10] | 17.93 (0.88) [13] | 18.53 (1.29) [6] | 11.20 (1.56) [17] | 28.76 (0.92) [1] |
| agnews 1 | 34.00 (1.15) [3] | 16.67 (0.87) [14] | 36.67 (1.52) [2] | 10.87 (1.42) [18] | 18.40 (0.39) [9] | 18.07 (0.98) [11] | 29.00 (1.21) [4] | 21.00 (0.84) [6] | 18.07 (1.16) [11] | 10.00 (0.76) [19] | 22.93 (0.49) [5] | 0.00 (0.00) [20] | 11.40 (0.68) [16] | 19.20 (1.07) [7] | 12.40 (1.29) [15] | 18.27 (0.65) [10] | 17.93 (0.88) [13] | 18.53 (1.29) [6] | 11.20 (1.56) [17] | 37.00 (1.88) [1] |
| agnews 2 | 38.20 (1.13) [2] | 26.47 (0.78) [12] | 36.73 (0.85) [3] | 16.73 (0.85) [18] | 27.40 (0.80) [10] | 25.47 (0.72) [13] | 34.80 (0.65) [5] | 28.47 (0.69) [8] | 26.47 (0.69) [11] | 18.07 (1.06) [15] | 33.47 (1.75) [4] | 10.07 (5.30) [20] | 18.40 (1.06) [14] | 17.07 (1.45) [17] | 17.00 (1.49) [13] | 28.27 (0.39) [9] | 31.13 (0.34) [6] | 12.60 (1.10) [17] | 40.84 (0.74) [1] |
| amazon | 11.13 (2.15) [10] | 11.60 (1.20) [7] | 9.40 (1.24) [17] | 11.07 (1.91) [11] | 12.00 (1.17) [5] | 12.47 (1.36) [2] | 9.33 (1.86) [18] | 10.60 (0.83) [13] | 11.33 (1.51) [9] | 13.47 (3.74) [14] | 14.13 (1.77) [1] | 0.00 (0.00) [20] | 12.40 (1.89) [3] | 9.07 (0.82) [19] | 10.50 (0.61) [2] | 5.80 (1.33) [11] | 4.67 (0.56) [20] | 5.67 (0.87) [13] | 11.44 (0.50) [8] |
| imdb | 5.53 (0.45) [15] | 5.00 (0.56) [17] | 6.47 (0.78) [6] | 5.73 (0.85) [12] | 5.00 (0.56) [17] | 6.87 (0.91) [5] | 6.13 (0.45) [10] | 4.93 (0.77) [19] | 5.33 (0.76) [16] | 6.20 (0.86) [9] | 7.27 (0.77) [3] | 6.40 (5.28) [7] | 5.60 (0.39) [14] | 10.60 (0.61) [2] | 5.80 (1.33) [11] | 4.67 (0.56) [20] | 5.67 (0.87) [13] | 6.27 (0.98) [8] | 6.88 (0.72) [4] |
| yelp | 21.20 (2.26) [2] | 18.87 (1.72) [9] | 19.80 (1.76) [4] | 17.27 (1.37) [12] | 18.60 (1.73) [10] | 16.07 (1.32) [16] | 19.53 (2.07) [5] | 20.07 (1.68) [3] | 18.93 (1.95) [8] | 16.40 (1.57) [15] | 19.27 (0.39) [6] | 9.27 (5.01) [18] | 16.67 (1.53) [14] | 8.20 (1.00) [19] | 6.87 (0.27) [20] | 16.93 (1.51) [13] | 19.13 (1.02) [7] | 18.33 (1.56) [11] | 15.73 (1.91) [17] | 21.36 (1.02) [1] |
| 20news 0 | 38.91 (0.81) [1] | 24.78 (0.81) [13] | 34.78 (2.17) [3] | 14.73 (2.17) [17] | 27.83 (0.87) [8] | 25.00 (1.94) [12] | 27.83 (0.53) [8] | 28.70 (0.53) [6] | 29.78 (1.77) [5] | 14.35 (2.11) [17] | 31.09 (1.47) [4] | 16.09 (5.12) [16] | 13.91 (1.06) [18] | 16.52 (1.74) [13] | 28.48 (2.52) [7] | 26.30 (0.81) [11] | 27.19 (1.09) [10] | 23.43 (1.44) [19] | 38.44 (0.76) [2] |
| 20news 1 | 18.40 (3.31) [2] | 9.60 (1.55) [15] | 10.13 (2.17) [12] | 6.40 (2.13) [19] | 12.00 (1.89) [8] | 13.33 (3.60) [7] | 14.40 (2.29) [5] | 11.73 (1.96) [10] | 14.13 (2.61) [6] | 6.93 (2.59) [18] | 17.07 (3.09) [4] | 9.60 (6.39) [15] | 6.40 (1.17) [19] | 18.13 (2.99) [3] | 12.00 (1.19) [8] | 10.40 (2.29) [11] | 9.87 (1.81) [14] | 10.13 (1.36) [12] | 7.47 (2.73) [17] | 19.04 (1.38) [1] |
| 20news 2 | 13.78 (2.62) [2] | 8.38 (1.32) [15] | 9.73 (1.58) [8] | 8.11 (2.42) [16] | 9.46 (0.85) [9] | 11.35 (2.51) [4] | 12.97 (2.02) [3] | 10.27 (2.02) [6] | 9.11 (1.83) [14] | 10.05 (1.68) [7] | 8.92 (1.08) [11] | 9.46 (2.96) [9] | 8.65 (1.83) [14] | 10.00 (1.43) [7] | 7.30 (1.28) [9] | 7.57 (1.63) [18] | 7.57 (1.63) [18] | 8.92 (1.83) [11] | 16.29 (0.94) [1] |
| 20news 3 | 43.66 (9.19) [3] | 26.99 (9.50) [8] | 23.69 (10.24) [9] | 16.83 (10.75) [16] | 46.65 (9.20) [1] | 40.02 (5.75) [5] | 46.02 (9.98) [2] | 36.78 (7.23) [6] | 23.16 (6.66) [11] | 17.60 (10.48) [14] | 1.48 (2.96) [20] | 14.57 (6.71) [18] | 18.28 (11.84) [12] | 17.28 (4.88) [15] | 16.38 (11.02) [17] | 23.69 (8.67) [9] | 35.23 (11.75) [7] | 18.01 (6.37) [13] | 13.99 (7.26) [19] | 41.21 (8.13) [4] |
| 20news 4 | 14.69 (4.16) [14] | 14.49 (4.16) [14] | 14.78 (2.13) [6] | 6.96 (2.88) [18] | 16.09 (2.61) [4] | 12.17 (2.22) [14] | 17.39 (3.07) [3] | 14.35 (1.74) [8] | 13.04 (1.37) [12] | 6.09 (2.54) [20] | 13.91 (1.74) [9] | 12.17 (2.22) [14] | 14.78 (4.64) [6] | 13.91 (4.03) [9] | 16.09 (2.22) [4] | 12.61 (1.63) [13] | 8.26 (4.43) [17] | 18.05 (1.42) [2] |
| 20news 5 | 19.57 (3.07) [2] | 12.17 (1.06) [14] | 14.78 (2.13) [6] | 6.96 (2.88) [18] | 16.09 (2.61) [4] | 12.17 (2.22) [14] | 17.39 (3.07) [3] | 14.35 (1.74) [8] | 13.04 (1.37) [12] | 6.09 (2.54) [20] | 13.91 (1.74) [9] | 12.17 (2.22) [14] | 14.78 (4.64) [6] | 13.91 (4.03) [9] | 16.09 (2.22) [4] | 12.61 (1.63) [13] | 8.26 (4.43) [17] | 21.05 (0.83) [1] |
| Average | 53.18 (29.33) [3] | 47.85 (28.73) [10] | 48.42 (26.22) [9] | 41.85 (27.36) [14] | 52.50 (31.10) [4] | 49.44 (30.67) [7] | 49.70 (28.65) [6] | 54.24 (30.14) [2] | 46.35 (27.68) [12] | 39.83 (28.37) [18] | 41.07 (27.23) [15] | 29.03 (21.07) [20] | 40.87 (27.46) [16] | 50.42 (30.20) [5] | 47.30 (30.63) [11] | 48.60 (27.76) [8] | 37.89 (30.62) [19] | 45.58 (26.79) [13] | 40.25 (26.26) [17] | 56.38 (29.50) [1] |

*Table 10.* Full results, including mean, standard deviation and rank F1-score performance of all methods for each ADBench dataset and the overall average. The best performance is highlighted in **bold**, the second best highlighted with underline.

| Model | NCSBAD | KNN | LOF | PCA | KPCA | KDE | GMM | LUNAR | AE | EIF | ABOD | DROCC | GOAD | ICL | SLAD | DTE-C | DTENonParametric | MCM | DDPM | DenoiserAD (Ours) |
|---|---|---|---|---|---|---|---|---|---|---|---|---|---|---|---|---|---|---|---|---|
| ALOI | 2.44 (0.27) [13] | 2.74 (0.50) [10] | 4.88 (0.77) [2] | 4.65 (0.49) [3] | 2.74 (0.61) [10] | 2.74 (0.42) [10] | 3.78 (0.41) [7] | 5.34 (0.38) [1] | 1.71 (0.40) [16] | 2.87 (0.60) [8] | 3.94 (0.45) [5] | -3.24 (0.00) [20] | 1.41 (0.92) [18] | 1.73 (0.54) [15] | 2.37 (0.48) [14] | 1.09 (0.46) [19] | 2.83 (0.45) [9] | 1.62 (0.27) [17] | 3.92 (0.59) [6] | 4.44 (0.11) [4] |
| annthyroid | 39.40 (1.41) [19] | 58.48 (1.42) [5] | 45.51 (1.62) [14] | 45.37 (1.57) [15] | 54.96 (0.98) [8] | 50.26 (1.48) [11] | 45.71 (1.22) [13] | 42.65 (1.06) [17] | 47.95 (2.61) [12] | 50.40 (1.41) [10] | 54.61 (1.17) [9] | 55.84 (4.89) [6] | 55.16 (7.00) [7] | 40.01 (4.34) [18] | 63.17 (1.45) [2] | 74.39 (3.17) [1] | 58.90 (1.01) [4] | 62.09 (1.00) [3] | 38.58 (3.20) [20] | 43.89 (1.81) [16] |
| backdoor | 88.13 (1.08) [1] | 51.09 (2.11) [12] | 71.21 (1.90) [10] | 6.04 (1.01) [18] | 86.86 (1.31) [4] | 43.77 (1.89) [13] | 85.23 (0.96) [6] | 87.94 (1.17) [2] | 10.89 (0.83) [15] | 5.71 (3.96) [19] | 82.16 (0.76) [8] | 85.17 (0.91) [7] | 10.56 (14.26) [16] | 87.11 (1.34) [3] | 79.14 (2.09) [9] | 61.32 (1.62) [11] | 15.17 (8.69) [14] | 9.65 (0.92) [17] | 86.61 (0.88) [5] |
| breastw | 91.71 (1.49) [9] | 90.96 (1.02) [10] | 72.24 (6.29) [17] | 90.55 (0.93) [11] | 93.72 (1.08) [4] | 94.10 (1.22) [3] | 90.37 (1.22) [13] | 92.13 (1.18) [6] | 88.89 (0.62) [15] | 92.37 (1.45) [5] | 68.46 (4.91) [19] | -8.16 (57.94) [20] | 89.71 (0.76) [14] | 91.87 (1.28) [8] | 91.92 (1.33) [7] | 72.09 (6.72) [18] | 95.53 (4.27) [1] | 90.51 (1.26) [12] | 80.45 (4.73) [16] | 94.41 (0.38) [2] |
| campaign | 32.68 (0.45) [17] | 42.92 (0.32) [5] | 33.74 (0.37) [16] | 41.08 (0.32) [10] | 42.66 (0.36) [6] | 40.26 (0.33) [12] | 37.17 (0.61) [14] | 41.60 (0.32) [9] | 38.37 (0.75) [13] | 41.84 (3.74) [14] | 40.73 (0.55) [11] | -14.54 (0.00) [20] | 4.92 (8.17) [19] | 43.91 (0.71) [4] | 42.23 (0.36) [7] | 46.20 (0.68) [2] | 42.19 (0.29) [8] | 16.55 (0.23) [15] | 29.75 (1.45) [18] |
| Cardiotocography | 16.05 (2.06) [17] | 23.82 (1.51) [8] | 26.90 (2.03) [5] | 46.62 (1.15) [1] | 20.33 (1.44) [10] | 20.33 (1.30) [10] | 23.92 (1.19) [7] | 25.31 (1.09) [6] | 27.90 (1.39) [4] | 39.65 (4.71) [3] | 18.84 (1.30) [13] | 2.60 (11.45) [20] | 41.24 (1.09) [2] | 20.33 (6.25) [10] | 5.79 (2.03) [19] | 10.87 (1.91) [18] | 20.63 (1.32) [9] | 16.54 (1.26) [15] | 16.54 (1.07) [15] | 17.34 (2.94) [14] |
| cardio | 62.21 (1.55) [7] | 57.35 (3.03) [13] | 57.77 (2.21) [11] | 72.13 (2.96) [2] | 60.09 (1.82) [9] | 64.95 (1.23) [5] | 57.35 (2.88) [13] | 67.27 (2.31) [4] | 58.41 (2.55) [10] | 62.84 (3.92) [6] | 67.90 (0.85) [3] | 41.73 (23.48) [20] | 70.02 (2.17) [2] | 50.17 (4.08) [18] | 57.77 (3.89) [11] | 55.24 (2.88) [16] | 50.80 (3.93) [17] | 61.79 (2.86) [8] | 48.06 (5.15) [19] | 56.89 (0.47) [15] |

Continued on next page

| Model | NCSBAD | KNN | LOF | PCA | KPCA | KDE | GMM | LUNAR | AE | EIF | ABOD | DROCC | GOAD | ICL | SLAD | DTE-C | DTENonParametric | MCM | DDPM | DenoiserAD (Ours) |
|---|---|---|---|---|---|---|---|---|---|---|---|---|---|---|---|---|---|---|---|---|
| celeba | 1.62 (1.27) [18] | 15.27 (1.04) [6] | -0.48 (0.26) [19] | 25.26 (1.81) [1] | 13.92 (1.18) [10] | 12.27 (1.29) [12] | 19.57 (1.16) [3] | 7.38 (0.99) [15] | 13.60 (0.91) [11] | 18.75 (1.52) [4] | -1.24 (1.34) [20] | 5.91 (0.92) [16] | 2.10 (4.37) [17] | 11.68 (0.50) [13] | 11.64 (1.40) [14] | 16.83 (2.43) [5] | 14.13 (1.03) [9] | 22.08 (4.28) [2] | 14.98 (1.26) [7] | 14.42 (4.57) [8] |
| census | 13.76 (0.85) [11] | 17.07 (0.74) [3] | 6.75 (0.85) [19] | 15.14 (0.73) [8] | 16.66 (0.63) [5] | 15.74 (0.97) [7] | 14.61 (0.86) [9] | 8.95 (0.48) [16] | 17.37 (0.82) [2] | 7.10 (1.62) [17] | 12.00 (0.80) [13] | 9.52 (1.50) [15] | 3.12 (5.00) [20] | 18.53 (0.95) [1] | 7.04 (11.53) [18] | 12.00 (0.71) [13] | 16.89 (0.58) [4] | 14.21 (1.66) [10] | 13.75 (0.91) [12] | 15.94 (0.67) [6] |
| cover | 80.05 (1.62) [2] | 65.95 (3.11) [9] | 82.28 (2.93) [1] | 15.73 (1.79) [16] | 63.49 (2.85) [10] | 42.61 (2.84) [12] | 17.59 (0.56) [15] | 77.64 (1.97) [4] | 69.30 (6.85) [8] | 10.50 (1.47) [17] | 75.34 (1.27) [5] | 40.26 (5.18) [13] | -1.00 (0.00) [20] | 42.66 (16.33) [11] | 8.92 (4.46) [19] | 73.17 (5.21) [6] | 69.67 (3.49) [7] | 38.02 (10.28) [14] | 9.24 (1.49) [18] | 78.48 (1.91) [3] |
| donors | 40.81 (1.34) [14] | 94.49 (0.91) [6] | 73.48 (1.55) [8] | 33.54 (1.22) [18] | 98.04 (0.49) [2] | 71.20 (0.74) [9] | 40.46 (0.46) [15] | 99.55 (0.17) [1] | 61.89 (4.84) [11] | 45.34 (2.09) [13] | 69.85 (1.64) [10] | 34.30 (21.39) [17] | 4.22 (18.17) [20] | 96.58 (1.77) [4] | 53.31 (8.82) [12] | 85.40 (1.28) [7] | 96.96 (0.57) [3] | 95.31 (3.17) [5] | 26.41 (2.73) [19] | 37.11 (1.31) [16] |
| fault | 3.99 (0.94) [14] | 3.89 (1.12) [15] | -6.14 (1.23) [20] | 4.74 (2.07) [13] | 2.52 (1.35) [18] | 5.37 (1.85) [11] | 10.22 (1.13) [4] | 3.15 (1.62) [17] | 12.02 (2.20) [3] | 3.57 (3.15) [16] | 5.26 (2.37) [12] | 9.48 (6.05) [5] | 6.42 (2.18) [7] | 9.28 (4.45) [6] | 14.13 (1.98) [1] | 5.79 (1.96) [10] | 5.90 (1.66) [9] | 12.54 (1.72) [2] | 6.42 (3.14) [7] | 0.70 (0.51) [19] |
| fraud | 57.67 (12.39) [5] | 48.60 (4.91) [11] | 63.06 (5.62) [2] | 36.49 (4.46) [15] | 46.72 (11.06) [13] | 43.86 (8.79) [14] | 62.48 (7.90) [3] | 56.74 (8.62) [7] | 52.72 (4.90) [9] | 30.66 (5.31) [17] | 19.43 (9.77) [18] | -0.17 (0.00) [19] | 31.29 (20.29) [16] | 57.04 (9.44) [6] | 50.25 (1.97) [10] | 62.25 (4.50) [4] | -0.17 (0.00) [19] | 47.65 (7.70) [12] | 75.02 (4.47) [1] | 53.72 (6.52) [8] |
| glass | 79.10 (3.85) [3] | 29.64 (15.81) [11] | 24.46 (11.61) [12] | 14.86 (9.17) [19] | 76.04 (0.50) [4] | 41.45 (12.58) [6] | 15.91 (8.36) [18] | 55.71 (13.75) [5] | 37.41 (11.57) [8] | 17.21 (10.83) [16] | -4.44 (0.00) [20] | 21.81 (17.18) [15] | 16.95 (12.89) [17] | 85.23 (9.11) [1] | 31.52 (11.23) [10] | 35.77 (9.44) [9] | 38.66 (11.75) [7] | 24.45 (18.46) [13] | 22.31 (6.12) [14] | 83.95 (8.25) [2] |
| Hepatitis | 99.51 (0.98) [2] | 77.27 (3.52) [11] | 25.81 (10.08) [19] | 50.46 (10.50) [16] | 99.51 (0.98) [2] | 99.51 (0.98) [2] | 72.99 (5.17) [13] | 99.51 (0.98) [2] | 75.46 (4.95) [12] | 77.98 (3.95) [10] | 60.29 (10.30) [15] | 7.94 (20.32) [20] | 48.61 (9.04) [17] | 99.51 (0.98) [2] | 99.51 (0.98) [2] | 92.94 (4.68) [8] | 92.39 (4.26) [9] | 68.03 (7.47) [14] | 42.13 (8.86) [18] | 99.56 (0.88) [1] |
| http | 82.59 (22.75) [13] | 100.00 (0.00) [1] | 96.33 (2.43) [7] | 92.83 (0.98) [9] | 99.44 (0.46) [4] | 99.63 (0.45) [3] | 92.31 (1.08) [10] | 99.83 (0.34) [2] | 99.44 (0.75) [4] | 25.94 (16.49) [18] | -0.38 (0.00) [19] | -0.38 (0.00) [19] | 54.92 (18.13) [15] | 80.33 (38.46) [14] | 89.28 (8.34) [12] | 28.67 (11.39) [17] | 90.50 (2.09) [11] | 47.10 (38.94) [16] | 98.71 (1.76) [6] | 93.35 (12.22) [8] |
| InternetAds | 59.55 (0.94) [1] | 36.16 (1.27) [12] | 41.21 (1.96) [11] | 28.98 (2.15) [17] | 49.45 (1.97) [5] | 48.27 (1.62) [7] | 54.85 (2.05) [3] | 53.79 (2.25) [4] | 85.61 (4.32) [6] | 28.75 (7.62) [18] | 47.68 (1.66) [8] | 18.05 (4.50) [20] | 29.22 (2.22) [16] | 42.63 (2.31) [10] | 44.74 (1.29) [9] | 49.33 (2.15) [6] | 31.69 (7.50) [14] | 32.51 (0.94) [13] | 27.58 (1.68) [19] | 58.70 (1.07) [2] |
| Ionosphere | 86.88 (1.06) [5] | 80.16 (4.81) [11] | 72.33 (5.16) [14] | 54.60 (6.01) [19] | 89.82 (2.01) [1] | 89.43 (3.41) [2] | 84.52 (2.94) [9] | 85.61 (4.32) [6] | 85.61 (2.32) [7] | 73.34 (2.18) [13] | 55.64 (6.53) [18] | 23.48 (27.58) [20] | 68.78 (10.36) [17] | 87.25 (2.05) [4] | 85.39 (1.41) [8] | 78.22 (2.67) [12] | 82.00 (2.81) [10] | 69.49 (8.90) [16] | 70.03 (3.18) [15] | 88.47 (2.67) [3] |
| landsat | 30.96 (0.69) [7] | 33.57 (1.11) [5] | 36.48 (1.07) [1] | 10.12 (0.72) [17] | 29.54 (1.27) [8] | 22.37 (1.39) [12] | 5.68 (0.47) [20] | 34.38 (1.05) [4] | 18.74 (0.36) [15] | 23.31 (1.07) [11] | 33.20 (0.93) [6] | 18.85 (4.90) [14] | 10.25 (1.51) [16] | 36.34 (2.00) [2] | 27.95 (0.68) [10] | 9.57 (1.29) [18] | 34.38 (1.16) [3] | 20.44 (1.11) [13] | 9.57 (1.14) [18] | 28.00 (0.86) [9] |
| letter | -2.50 (2.42) [6] | -6.43 (0.88) [15] | 3.93 (2.08) [2] | -6.43 (0.88) [15] | -6.43 (0.88) [15] | -5.36 (1.60) [11] | -6.07 (0.88) [13] | -7.14 (0.00) [18] | -7.14 (0.00) [18] | -4.64 (1.43) [10] | -1.43 (3.07) [14] | 4.64 (7.28) [1] | -6.07 (0.88) [13] | -0.71 (1.43) [3] | -5.36 (1.60) [11] | -3.93 (1.33) [9] | -7.14 (0.00) [18] | -3.21 (2.08) [8] | -2.86 (1.43) [7] | -2.21 (0.86) [5] |
| Lymphography | 96.51 (4.28) [9] | 94.00 (5.63) [12] | 75.40 (5.00) [17] | 93.26 (3.07) [13] | 100.00 (0.00) [1] | 100.00 (0.00) [1] | 99.05 (1.90) [8] | 100.00 (0.00) [1] | 92.99 (6.94) [14] | 95.77 (5.26) [10] | -4.42 (0.00) [20] | 30.89 (29.06) [19] | 95.73 (4.41) [11] | 100.00 (0.00) [1] | 99.09 (1.82) [7] | 81.63 (10.10) [16] | 68.83 (12.66) [18] | 100.00 (0.00) [1] | 91.48 (5.77) [15] | 100.00 (0.00) [1] |
| magic.gamma | 57.56 (0.36) [2] | 48.11 (0.26) [9] | 47.68 (0.33) [10] | 24.05 (0.95) [19] | 47.10 (0.34) [11] | 31.96 (0.51) [16] | 41.29 (0.79) [12] | 52.66 (0.58) [5] | 49.61 (0.52) [7] | 35.25 (2.78) [15] | 50.68 (0.43) [6] | 40.11 (1.50) [13] | 18.39 (2.25) [20] | 36.58 (1.62) [14] | 25.74 (0.47) [17] | 56.12 (0.35) [3] | 49.36 (0.31) [8] | 60.14 (0.50) [1] | 24.56 (1.73) [18] | 53.72 (0.43) [4] |
| mammography | 25.67 (4.10) [16] | 38.14 (2.61) [8] | 36.96 (2.66) [11] | 41.95 (2.44) [4] | 43.53 (3.50) [2] | 41.82 (2.33) [5] | 42.21 (1.71) [3] | 44.18 (1.25) [1] | 35.12 (1.27) [13] | 40.51 (3.44) [6] | 37.75 (2.29) [9] | 28.82 (3.01) [15] | 32.36 (4.59) [14] | 18.97 (2.84) [18] | 17.00 (2.51) [19] | 36.83 (2.09) [12] | 39.19 (2.10) [7] | 23.04 (2.09) [17] | 12.53 (1.92) [20] | 37.59 (1.26) [10] |
| mnist | 67.83 (1.99) [8] | 69.43 (0.76) [4] | 68.68 (1.38) [6] | 60.57 (1.16) [19] | 70.54 (0.94) [2] | 71.44 (0.81) [1] | 65.66 (1.09) [10] | 68.79 (1.29) [5] | 66.45 (1.32) [9] | 52.09 (5.21) [18] | 68.26 (0.65) [7] | 55.06 (3.59) [17] | 61.95 (2.92) [14] | 60.79 (2.63) [15] | 65.24 (0.54) [13] | 51.62 (1.07) [19] | 70.27 (0.81) [3] | 65.39 (1.56) [11] | 46.48 (0.95) [20] | 65.25 (2.24) [12] |
| musk | 100.00 (0.00) [1] | 100.00 (0.00) [1] | 100.00 (0.00) [1] | 100.00 (0.00) [1] | 100.00 (0.00) [1] | 100.00 (0.00) [1] | 91.45 (3.46) [16] | 100.00 (0.00) [1] | 100.00 (0.00) [1] | 57.22 (21.15) [18] | 96.08 (1.33) [15] | 13.02 (18.23) [19] | 100.00 (0.00) [1] | 81.46 (4.45) [17] | 100.00 (0.00) [1] | 100.00 (0.00) [1] | 6.95 (8.16) [20] | 97.86 (1.75) [14] | 46.48 (0.95) [20] | 100.00 (0.00) [1] |
| optdigits | 56.95 (3.59) [2] | 17.79 (6.16) [10] | 49.16 (1.17) [4] | -2.82 (0.46) [20] | 6.34 (2.65) [14] | 39.31 (4.10) [7] | -0.53 (1.33) [18] | 85.12 (2.29) [1] | 6.34 (2.44) [14] | 4.51 (3.12) [16] | 41.37 (1.68) [6] | 17.56 (10.81) [11] | -2.59 (0.56) [19] | 48.02 (8.65) [5] | 37.94 (3.72) [8] | 9.08 (2.36) [12] | 29.47 (4.32) [9] | 7.03 (2.84) [13] | 3.36 (2.34) [17] | 54.66 (2.88) [3] |
| PageBlocks | 57.09 (1.83) [5] | 54.82 (1.12) [11] | 61.54 (0.78) [2] | 40.30 (1.36) [19] | 52.34 (1.23) [13] | 53.80 (1.05) [12] | 51.10 (1.26) [14] | 55.26 (2.90) [8] | 55.26 (1.14) [8] | 37.96 (1.33) [20] | 57.09 (1.15) [5] | 64.17 (2.77) [1] | 45.77 (1.12) [17] | 59.42 (1.14) [3] | 46.58 (1.12) [16] | 56.70 (1.40) [7] | 41.61 (5.59) [18] | 57.74 (1.34) [4] | 48.33 (0.85) [15] | 54.94 (1.98) [10] |
| pendigits | 90.85 (1.77) [5] | 90.42 (1.87) [6] | 74.30 (2.02) [9] | 41.62 (2.80) [15] | 85.84 (1.19) [7] | 92.16 (1.74) [3] | 16.14 (3.00) [19] | 95.42 (0.43) [1] | 64.50 (1.77) [10] | 56.00 (6.03) [13] | 80.39 (1.54) [8] | 15.27 (7.00) [20] | 37.48 (6.18) [17] | 63.19 (7.65) [11] | 39.88 (1.87) [16] | 54.26 (3.31) [14] | 93.03 (0.87) [2] | 58.61 (3.16) [12] | 21.36 (2.33) [18] | 91.34 (0.77) [4] |
| Pima | 42.05 (4.71) [5] | 39.23 (4.74) [8] | 28.45 (6.17) [14] | 37.25 (2.47) [9] | 52.63 (2.28) [1] | 39.85 (1.08) [7] | 36.71 (5.10) [10] | 44.50 (4.45) [2] | 28.44 (5.48) [15] | 36.36 (4.15) [11] | 29.58 (1.66) [12] | -15.93 (27.34) [20] | 26.79 (9.04) [16] | 42.05 (4.38) [5] | 13.83 (7.91) [18] | 28.70 (8.37) [13] | 42.29 (3.82) [4] | 26.51 (4.48) [17] | 10.24 (5.91) [19] | 44.36 (5.72) [3] |
| satellite | 48.51 (1.80) [5] | 47.32 (1.07) [10] | 49.15 (1.11) [4] | 30.13 (1.54) [20] | 43.27 (1.57) [15] | 43.76 (1.25) [14] | 46.44 (1.37) [12] | 47.96 (1.36) [7] | 47.93 (1.26) [8] | 37.78 (2.27) [18] | 49.79 (1.12) [3] | 40.38 (6.21) [16] | 33.09 (2.05) [19] | 51.19 (0.86) [2] | 59.32 (1.53) [1] | 46.65 (1.47) [11] | 48.18 (1.42) [6] | 37.90 (1.67) [17] | 44.11 (1.85) [13] | 47.37 (0.83) [9] |
| satimage-2 | 89.64 (1.88) [7] | 91.05 (0.94) [4] | 83.05 (1.76) [14] | 88.22 (0.00) [10] | 91.52 (1.15) [2] | 90.11 (1.76) [5] | 79.75 (2.83) [16] | 89.17 (1.16) [8] | 92.46 (0.94) [1] | 82.11 (2.40) [15] | 75.04 (17.21) [17] | 91.52 (1.15) [2] | 89.17 (2.82) [8] | 87.75 (3.12) [11] | 74.57 (1.76) [18] | 84.93 (2.40) [12] | 43.96 (7.51) [20] | 67.04 (4.71) [19] | 83.46 (1.46) [13] | 86.62 (2.00) [6] |
| skin | 87.58 (0.83) [5] | 95.58 (0.60) [2] | 60.02 (2.62) [15] | 15.54 (0.66) [19] | 94.94 (0.36) [3] | 66.54 (0.49) [12] | 70.26 (0.55) [11] | 91.24 (0.65) [4] | 52.64 (4.24) [17] | 71.74 (0.98) [9] | 75.50 (0.99) [8] | 70.27 (2.02) [10] | 37.12 (4.29) [18] | -34.78 (1.51) [20] | 65.32 (3.27) [13] | 75.51 (0.70) [7] | 96.02 (0.62) [1] | 65.13 (11.91) [14] | 56.57 (1.11) [16] | 86.62 (2.00) [6] |
| smtp | 51.47 (15.10) [13] | 76.17 (8.32) [1] | 74.17 (8.36) [9] | 76.17 (8.32) [1] | 76.17 (8.32) [1] | 76.17 (8.32) [1] | 42.54 (9.31) [16] | 76.17 (8.32) [1] | 76.17 (8.32) [1] | -0.04 (0.00) [19] | -0.04 (0.00) [19] | 14.97 (30.01) [18] | 50.05 (6.94) [14] | 15.67 (16.40) [17] | 76.17 (8.32) [1] | 76.17 (8.32) [1] | 66.50 (6.34) [11] | 55.66 (28.44) [12] | 47.25 (19.16) [15] | 69.58 (3.95) [10] |
| SpamBase | 42.98 (0.35) [16] | 42.05 (1.53) [9] | 23.59 (1.58) [20] | 36.07 (1.21) [15] | 45.30 (1.07) [2] | 45.06 (1.61) [3] | 34.06 (0.93) [17] | 37.61 (3.43) [13] | 39.02 (1.34) [11] | 44.06 (3.44) [4] | 43.28 (0.29) [5] | 26.61 (4.21) [19] | 36.71 (1.96) [14] | 38.26 (1.81) [12] | 46.86 (1.31) [1] | 41.51 (1.30) [10] | 42.81 (1.14) [8] | 36.07 (1.10) [15] | 28.98 (0.96) [18] | 42.02 (0.65) [7] |
| speech | 2.69 (1.35) [6] | 0.49 (1.10) [14] | 2.69 (1.35) [6] | 2.69 (1.35) [6] | 2.69 (1.35) [6] | 7.09 (2.86) [1] | 6.54 (2.46) [2] | 4.89 (2.20) [3] | -0.06 (1.35) [16] | 1.04 (3.48) [13] | -1.16 (1.10) [19] | 2.69 (4.47) [6] | 1.59 (2.60) [12] | 2.69 (2.80) [6] | 3.79 (1.74) [4] | 3.79 (1.74) [4] | -1.71 (0.00) [20] | -0.06 (1.35) [16] | -0.06 (2.20) [16] | -0.04 (1.49) [15] |
| Stamps | 84.04 (3.60) [5] | 77.04 (9.02) [7] | 64.30 (11.47) [11] | 60.58 (7.20) [15] | 93.94 (3.83) [1] | 90.11 (5.88) [2] | 62.71 (11.15) [13] | 86.69 (5.09) [4] | 74.94 (3.43) [8] | 64.07 (6.44) [12] | -2.11 (6.31) [20] | 16.53 (27.92) [19] | 53.62 (16.96) [16] | 68.64 (5.59) [9] | 40.33 (10.10) [18] | 60.59 (6.97) [14] | 83.82 (3.34) [6] | 66.90 (12.78) [10] | 43.76 (6.19) [17] | 90.05 (2.71) [3] |
| thyroid | 64.46 (0.89) [15] | 78.45 (2.74) [5] | 52.00 (4.40) [20] | 74.72 (1.37) [6] | 76.18 (2.01) [4] | 74.72 (2.69) [6] | 72.89 (3.36) [10] | 71.79 (3.77) [11] | 54.57 (3.15) [18] | 78.38 (3.91) [1] | 58.23 (2.93) [17] | 70.69 (3.07) [12] | 74.72 (2.14) [6] | 53.10 (6.41) [19] | 76.19 (3.07) [3] | 78.38 (2.13) [1] | 62.26 (0.90) [16] | 74.72 (0.73) [6] | 69.59 (7.01) [13] | 69.55 (0.88) [14] |
| vertebral | 39.23 (2.68) [4] | 7.76 (5.44) [11] | 17.97 (3.59) [8] | -1.52 (2.75) [17] | 65.20 (3.00) [2] | -4.80 (3.71) [19] | -3.83 (5.18) [18] | 30.65 (12.43) [5] | 13.39 (3.36) [10] | 1.85 (5.05) [15] | -15.13 (1.67) [20] | 0.99 (20.86) [16] | 2.10 (3.70) [14] | 49.42 (9.94) [3] | 3.85 (10.08) [13] | 23.67 (5.88) [7] | 25.11 (8.36) [6] | 14.05 (7.48) [9] | 7.62 (3.17) [12] | 66.49 (6.02) [1] |
| vowels | 40.55 (4.03) [3] | 28.80 (1.69) [12] | 35.72 (7.76) [6] | 11.52 (1.70) [18] | 30.87 (2.19) [9] | 27.42 (0.00) [13] | 43.32 (4.69) [1] | 41.94 (2.59) [2] | 39.86 (1.69) [5] | 14.29 (5.95) [17] | 29.49 (2.77) [10] | 2.53 (7.70) [19] | 17.74 (6.70) [16] | 29.49 (6.77) [10] | 32.26 (3.53) [8] | 40.55 (2.59) [3] | 2.53 (8.29) [19] | 24.26 (2.58) [14] | 22.58 (5.17) [15] | 34.47 (2.51) [7] |
| Waveform | 8.60 (2.53) [15] | 24.06 (2.53) [4] | 25.44 (2.57) [2] | 7.22 (1.54) [16] | 20.97 (2.88) [9] | 23.03 (2.52) [6] | 6.88 (1.29) [17] | 25.44 (2.06) [2] | 11.35 (0.84) [12] | 8.94 (4.21) [14] | 16.50 (1.37) [10] | 21.32 (7.71) [8] | 6.54 (2.79) [19] | 24.06 (3.83) [4] | 0.35 (1.09) [20] | 10.32 (2.53) [13] | 22.69 (2.87) [7] | 45.02 (1.54) [1] | 6.88 (3.15) [17] | 11.76 (1.91) [11] |
| WBC | 91.03 (9.67) [6] | 85.75 (4.36) [11] | 15.43 (8.62) [19] | 89.40 (1.74) [9] | 97.37 (5.25) [1] | 94.45 (5.49) [2] | 77.54 (7.01) [13] | 89.91 (7.35) [7] | 72.80 (8.36) [14] | 91.55 (2.56) [5] | -5.02 (0.00) [20] | 24.96 (27.83) [18] | 89.40 (1.74) [9] | 93.48 (6.24) [4] | 89.56 (6.27) [8] | 29.91 (3.66) [17] | 50.49 (16.05) [16] | 83.06 (7.15) [12] | 70.06 (7.79) [15] | 93.63 (4.38) [3] |
| WDBC | 97.78 (2.78) [1] | 85.31 (4.85) [12] | 90.88 (6.54) [8] | 84.02 (6.49) [14] | 93.46 (4.09) [2] | 91.23 (5.34) [5] | 91.05 (4.04) [6] | 89.59 (7.68) [9] | 88.20 (6.41) [11] | 90.96 (4.97) [7] | -2.99 (0.00) [20] | 7.31 (20.60) [19] | 85.31 (4.85) [12] | 92.60 (4.78) [3] | 88.47 (5.46) [10] | 69.78 (8.36) [16] | 40.27 (20.01) [18] | 72.12 (14.72) [15] | 51.02 (7.80) [17] | 91.88 (5.09) [4] |
| Wilt | 22.10 (1.53) [4] | -3.35 (0.67) [15] | 11.37 (1.70) [7] | -4.04 (0.67) [17] | -1.84 (0.44) [12] | -5.01 (0.34) [20] | 26.70 (0.93) [3] | -3.77 (1.10) [16] | -2.30 (0.52) [13] | -4.87 (0.34) [19] | 13.30 (1.80) [6] | -4.73 (1.26) [18] | 8.48 (2.85) [9] | 29.81 (5.89) [2] | 1.87 (1.26) [11] | 8.62 (2.01) [8] | -2.53 (0.62) [14] | 40.14 (0.97) [1] | 2.56 (2.94) [10] | 14.32 (1.64) [5] |
| wine | 98.76 (2.48) [5] | 85.87 (2.95) [13] | 81.43 (7.91) [14] | 63.22 (11.89) [16] | 99.59 (0.83) [2] | 99.59 (0.83) [2] | 89.70 (7.72) [11] | 97.93 (4.14) [7] | 89.87 (4.05) [10] | 77.90 (4.89) [15] | -8.31 (1.70) [20] | 7.10 (13.75) [19] | 62.41 (11.48) [17] | 99.13 (1.07) [4] | 100.00 (0.00) [1] | 96.13 (5.63) [8] | 91.72 (2.73) [9] | 86.12 (7.79) [12] | 53.88 (5.22) [18] | 98.01 (3.99) [6] |
| WPBC | 84.96 (4.65) [5] | 28.92 (3.35) [11] | 18.44 (5.56) [14] | 7.06 (5.59) [18] | 86.23 (4.75) [1] | 86.02 (5.89) [2] | 21.23 (5.39) [13] | 77.83 (4.52) [7] | 22.66 (8.30) [12] | 22.66 (4.58) [12] | -12.27 (11.24) [20] | 5.17 (5.93) [19] | 9.57 (6.92) [17] | 86.02 (4.61) [2] | 82.26 (6.18) [6] | 44.82 (4.17) [9] | 52.00 (4.95) [8] | 31.01 (4.92) [10] | 12.25 (3.68) [16] | 85.34 (3.94) [4] |
| yeast | -5.30 (3.48) [7] | -11.32 (1.53) [11] | -8.45 (1.65) [10] | -18.44 (0.67) [20] | -13.51 (2.66) [14] | -13.51 (3.03) [14] | -11.74 (1.20) [12] | -14.46 (0.69) [17] | -8.04 (1.14) [9] | -14.47 (1.76) [18] | -13.23 (1.70) [13] | -0.79 (6.57) [1] | -1.33 (8.77) [2] | -1.88 (3.64) [3] | -5.17 (2.27) [6] | -3.25 (0.97) [4] | -13.51 (1.88) [14] | -14.87 (1.37) [19] | -4.62 (3.60) [5] | -6.27 (0.84) [8] |
| CIFAR10 0 | 27.05 (2.25) [4] | 25.31 (1.81) [10] | 28.25 (1.82) [1] | 25.58 (1.82) [8] | 27.05 (2.13) [4] | 22.11 (1.77) [16] | 27.58 (2.26) [2] | 25.58 (1.56) [8] | 25.71 (1.95) [7] | 21.84 (1.53) [17] | 27.58 (1.77) [2] | 11.28 (1.86) [19] | 25.18 (1.89) [13] | 19.16 (1.40) [18] | 26.11 (1.61) [6] | 25.04 (1.76) [14] | 0.73 (12.23) [20] | 25.31 (2.00) [10] | 23.31 (2.88) [15] | 25.19 (0.83) [12] |
| CIFAR10 1 | 16.76 (2.59) [9] | 14.09 (2.62) [14] | 25.85 (2.43) [1] | 13.95 (2.82) [15] | 16.36 (2.58) [10] | 12.08 (2.18) [16] | 19.43 (2.69) [3] | 14.36 (2.17) [13] | 19.70 (2.13) [2] | 4.07 (2.18) [19] | 17.16 (3.05) [8] | 6.87 (2.76) [18] | 15.02 (2.72) [12] | 18.77 (2.52) [5] | 19.30 (2.00) [4] | 17.96 (1.36) [7] | -4.75 (1.60) [20] | 18.10 (2.41) [6] | 10.75 (2.01) [17] | 15.40 (0.60) [11] |
| CIFAR10 2 | 13.02 (2.17) [7] | 10.75 (2.26) [16] | 15.69 (2.17) [1] | 10.48 (2.64) [17] | 11.68 (2.65) [12] | 11.82 (1.74) [10] | 13.29 (1.96) [5] | 13.96 (1.96) [3] | 12.48 (2.39) [9] | 9.41 (2.22) [18] | 12.49 (1.12) [8] | 8.08 (2.63) [19] | 11.15 (2.60) [14] | 11.28 (1.29) [13] | 13.15 (2.67) [6] | 13.95 (2.85) [4] | 1.00 (3.81) [20] | 11.82 (2.99) [10] | 10.88 (2.79) [15] | 14.03 (0.69) [2] |
| CIFAR10 3 | 12.08 (2.37) [8] | 11.95 (2.21) [10] | 14.89 (1.82) [1] | 12.08 (2.41) [8] | 12.62 (2.33) [7] | 10.21 (1.77) [16] | 13.55 (2.55) [2] | 13.42 (2.30) [3] | 13.02 (2.40) [5] | 6.74 (0.80) [19] | 11.95 (1.96) [10] | 7.27 (2.44) [18] | 11.68 (2.75) [13] | 9.41 (1.92) [17] | 13.15 (2.24) [4] | 11.95 (1.96) [10] | 2.06 (7.11) [20] | 13.02 (2.55) [5] | 10.65 (2.13) [15] | 10.66 (0.54) [14] |
| CIFAR10 4 | 35.87 (2.03) [1] | 31.32 (1.71) [12] | 31.46 (2.45) [10] | 31.72 (1.66) [9] | 32.26 (1.96) [7] | 29.72 (2.33) [17] | 33.73 (2.13) [3] | 33.60 (1.50) [4] | 31.06 (1.76) [13] | 30.26 (1.24) [14] | 32.53 (2.71) [5] | 13.82 (4.43) [19] | 32.13 (1.62) [8] | 26.25 (2.56) [18] | 32.39 (1.54) [6] | 30.25 (1.50) [15] | 0.33 (11.43) [20] | 31.46 (2.09) [10] | 30.12 (2.92) [16] | 35.54 (0.83) [2] |
| CIFAR10 5 | 11.02 (1.55) [6] | 8.88 (1.87) [15] | 15.16 (1.74) [1] | 9.28 (1.60) [14] | 10.61 (1.37) [9] | 8.74 (1.31) [17] | 12.35 (2.13) [3] | 10.75 (1.09) [7] | 11.95 (2.25) [4] | 5.00 (0.78) [18] | 10.75 (1.31) [7] | 4.87 (2.01) [19] | 9.55 (0.68) [13] | 8.88 (3.00) [15] | 11.55 (2.26) [5] | 13.82 (4.72) [2] | 3.67 (8.56) [20] | 9.68 (1.36) [12] | 10.18 (0.47) [11] |
| CIFAR10 6 | 22.91 (1.62) [2] | 20.90 (1.09) [11] | 23.84 (2.90) [1] | 19.03 (1.07) [15] | 22.11 (1.82) [6] | 17.83 (2.43) [16] | 22.51 (1.40) [5] | 21.97 (1.23) [8] | 20.50 (1.12) [12] | 15.96 (1.15) [17] | 22.11 (1.38) [6] | 12.75 (1.82) [19] | 10.84 (1.52) [13] | 14.76 (2.09) [18] | 22.77 (1.38) [4] | 22.91 (2.26) [2] | -3.28 (3.89) [20] | 21.57 (1.31) [10] | 19.70 (1.43) [14] | 21.90 (0.75) [9] |
| CIFAR10 7 | 20.77 (2.30) [7] | 20.64 (2.13) [9] | 24.64 (2.29) [1] | 21.30 (1.71) [6] | 20.50 (2.03) [10] | 14.22 (1.24) [18] | 22.11 (1.77) [3] | 20.77 (2.88) [7] | 22.37 (1.76) [2] | 19.97 (0.50) [7] | 18.63 (4.07) [15] | 10.61 (5.67) [19] | 21.70 (1.66) [5] | 14.62 (1.48) [17] | 20.10 (1.44) [11] | 19.03 (1.08) [13] | 0.19 (6.98) [20] | 22.11 (1.77) [3] | 18.76 (2.73) [14] | 26.56 (0.44) [4] |
| CIFAR10 8 | 20.50 (1.12) [4] | 18.36 (0.27) [11] | 24.51 (0.73) [1] | 16.36 (0.50) [17] | 19.30 (0.50) [9] | 17.03 (1.76) [14] | 20.10 (0.53) [6] | 23.04 (2.00) [2] | 19.97 (0.50) [7] | 14.36 (1.86) [18] | 19.57 (1.24) [8] | 13.29 (3.44) [19] | 16.89 (1.87) [15] | 18.50 (1.46) [10] | 21.04 (1.36) [3] | 16.89 (1.08) [15] | 0.33 (7.40) [20] | 20.37 (0.27) [5] | 17.96 (1.66) [12] | 17.72 (0.76) [13] |
| CIFAR10 9 | 24.91 (1.44) [7] | 23.71 (1.86) [9] | 28.52 (1.98) [1] | 21.57 (1.72) [13] | 23.71 (1.36) [9] | 19.17 (2.11) [17] | 26.92 (1.72) [2] | 26.92 (1.54) [2] | 23.71 (1.39) [9] | 23.71 (1.29) [9] | 25.18 (2.00) [4] | 21.30 (6.08) [14] | 18.69 (1.80) [18] | 24.51 (2.53) [6] | 26.91 (1.61) [3] | 24.51 (1.29) [6] | 0.19 (11.49) [20] | 23.31 (1.29) [12] | 21.04 (1.36) [15] | 26.56 (0.44) [5] |
| FashionMNIST 0 | 57.22 (2.84) [2] | 51.31 (3.45) [10] | 56.33 (3.59) [3] | 45.16 (3.08) [17] | 52.09 (2.65) [7] | 45.61 (2.41) [15] | 56.22 (3.43) [4] | 61.02 (1.39) [1] | 51.53 (3.37) [9] | 35.00 (2.77) [18] | 50.86 (2.76) [11] | 22.16 (4.00) [19] | 46.50 (2.98) [14] | 54.10 (3.78) [6] | 52.09 (2.90) [7] | 47.96 (2.63) [13] | 2.95 (12.37) [20] | 48.74 (3.30) [12] | 45.50 (2.46) [16] | 55.10 (0.76) [5] |
| FashionMNIST 1 | 78.89 (1.78) [3] | 75.43 (1.84) [9] | 76.33 (1.68) [8] | 72.30 (1.48) [16] | 76.43 (1.88) [7] | 74.54 (1.71) [12] | 78.22 (2.09) [5] | 81.24 (0.76) [2] | 74.98 (1.92) [10] | 67.61 (2.23) [18] | 78.67 (1.55) [4] | 41.48 (15.95) [19] | 72.97 (1.98) [15] | 77.22 (0.65) [6] | 74.98 (1.85) [10] | 72.08 (2.15) [17] | 18.81 (10.25) [20] | 74.20 (1.56) [13] | 73.53 (1.44) [14] | 81.37 (0.45) [1] |
| FashionMNIST 2 | 51.75 (1.49) [9] | 56.60 (1.76) [12] | 52.20 (2.41) [3] | 42.49 (2.52) [16] | 47.85 (2.49) [8] | 41.14 (1.92) [17] | 51.19 (2.01) [6] | 51.86 (1.74) [4] | 46.62 (1.89) [11] | 47.73 (2.36) [9] | 24.28 (13.11) [19] | 43.26 (2.51) [14] | 53.76 (2.41) [2] | 47.29 (1.82) [10] | 49.30 (2.41) [7] | 0.15 (6.91) [20] | 46.17 (2.30) [13] | 43.15 (3.45) [15] | 56.45 (0.60) [1] |
| FashionMNIST 3 | 58.79 (0.42) [4] | 54.10 (0.74) [10] | 59.01 (1.48) [3] | 52.31 (0.76) [14] | 55.55 (0.76) [8] | 49.85 (0.42) [17] | 57.90 (0.45) [5] | 60.91 (1.37) [2] | 54.10 (0.96) [10] | 42.37 (4.34) [18] | 56.56 (1.67) [7] | 26.85 (4.80) [19] | 51.87 (0.74) [15] | 57.56 (1.28) [6] | 54.32 (1.08) [9] | 52.65 (1.24) [13] | 5.41 (15.10) [20] | 53.87 (1.30) [12] | 50.97 (1.30) [16] | 61.20 (0.58) [1] |
| FashionMNIST 4 | 53.43 (1.35) [4] | 48.85 (1.04) [11] | 54.21 (0.79) [3] | 45.05 (1.15) [17] | 50.52 (1.44) [9] | 46.51 (0.82) [14] | 53.43 (1.52) [4] | 52.87 (2.17) [6] | 49.18 (1.42) [10] | 30.09 (2.81) [18] | 50.75 (0.96) [8] | 20.70 (10.69) [20] | 45.17 (1.14) [16] | 57.00 (0.94) [2] | 47.73 (0.97) [13] | 51.08 (1.10) [7] | 23.72 (13.89) [19] | 48.07 (1.27) [12] | 45.13 (3.45) [15] | 58.85 (0.86) [1] |
| FashionMNIST 5 | 78.67 (1.29) [9] | 78.78 (1.27) [7] | 77.89 (1.56) [14] | 78.67 (1.47) [9] | 79.34 (1.17) [3] | 79.78 (1.19) [2] | 78.22 (1.37) [13] | 79.12 (1.30) [5] | 79.00 (1.09) [6] | 73.42 (1.30) [18] | 80.01 (1.14) [1] | 43.82 (21.26) [19] | 78.33 (1.29) [12] | 77.33 (1.60) [15] | 78.78 (1.17) [7] | 76.65 (1.61) [17] | 14.78 (18.00) [20] | 79.23 (1.14) [4] | 78.45 (1.25) [11] | 77.08 (0.40) [16] |
| FashionMNIST 6 | 45.05 (2.02) [2] | 35.00 (1.95) [11] | 42.93 (2.38) [6] | 25.95 (1.30) [16] | 37.12 (1.26) [10] | 32.88 (2.22) [14] | 44.83 (1.39) [3] | 44.72 (1.00) [4] | 34.78 (2.07) [12] | 12.89 (4.07) [18] | 37.23 (1.72) [9] | 8.98 (7.92) [19] | 29.19 (1.71) [15] | 44.72 (1.97) [4] | 38.46 (1.78) [8] | 40.58 (2.56) [7] | 0.94 (8.62) [20] | 33.55 (1.41) [13] | 25.62 (1.30) [17] | 50.81 (0.45) [1] |
| FashionMNIST 7 | 82.47 (1.64) [9] | 82.24 (1.71) [10] | 82.02 (1.43) [14] | 81.91 (1.68) [15] | 82.58 (1.63) [8] | 82.69 (1.83) [6] | 83.02 (1.89) [3] | 83.36 (1.85) [2] | 82.69 (1.45) [6] | 81.24 (1.64) [17] | 83.47 (1.68) [1] | 41.14 (22.52) [19] | 81.91 (1.68) [15] | 82.91 (1.48) [4] | 82.80 (1.59) [5] | 80.79 (1.68) [18] | 19.37 (16.94) [20] | 82.24 (1.81) [10] | 82.13 (1.83) [12] | 82.24 (0.00) [10] |
| FashionMNIST 8 | 37.01 (1.78) [4] | 26.85 (1.84) [11] | 37.23 (1.35) [3] | 19.47 (2.07) [18] | 28.19 (2.28) [10] | 25.95 (1.10) [12] | 35.11 (1.29) [6] | 41.59 (1.52) [2] | 25.40 (1.20) [13] | 19.81 (3.49) [17] | 31.54 (1.75) [7] | 14.56 (2.78) [19] | 20.93 (2.11) [16] | 36.57 (2.44) [5] | 29.97 (1.60) [9] | 30.42 (2.08) [8] | 13.22 (11.04) [20] | 23.27 (2.02) [14] | 21.26 (1.54) [15] | 44.05 (0.88) [1] |
| FashionMNIST 9 | 77.00 (1.14) [3] | 75.54 (1.14) [10] | 76.88 (1.44) [4] | 74.54 (1.75) [15] | 76.43 (1.51) [7] | 75.54 (1.78) [10] | 75.88 (1.52) [9] | 80.01 (2.48) [1] | 76.33 (2.33) [18] | 77.55 (1.08) [2] | 38.68 (19.05) [19] | 74.31 (1.37) [16] | 76.77 (1.15) [6] | 75.09 (1.35) [13] | 73.08 (1.47) [17] | 19.25 (16.25) [20] | 75.32 (1.34) [12] | 74.98 (1.39) [14] | 76.88 (0.56) [4] |
| MNIST-C brightness | 47.22 (2.23) [3] | 26.39 (0.81) [12] | 36.03 (1.81) [7] | 19.50 (0.72) [18] | 28.71 (1.01) [11] | 28.79 (1.92) [10] | 36.25 (1.76) [6] | 50.60 (1.13) [2] | 24.21 (1.01) [13] | 14.64 (1.63) [20] | 38.43 (1.28) [5] | 18.30 (2.62) [19] | 19.99 (0.36) [17] | 44.27 (1.58) [4] | 31.53 (1.42) [9] | 33.85 (1.52) [8] | 21.89 (4.22) [15] | 22.87 (0.96) [14] | 21.68 (1.49) [16] | 70.53 (0.87) [1] |
| MNIST-C canny edges | 49.26 (1.43) [3] | 27.66 (1.40) [12] | 48.06 (1.03) [4] | 19.49 (1.03) [18] | 28.50 (1.61) [10] | 28.36 (1.82) [11] | 36.03 (1.60) [8] | 54.33 (1.45) [2] | 27.59 (1.80) [13] | 16.68 (3.85) [19] | 39.06 (0.41) [6] | 23.93 (9.68) [15] | 20.41 (1.97) [16] | 40.75 (2.07) [5] | 33.29 (1.50) [9] | 36.67 (1.71) [7] | 16.40 (10.74) [20] | 24.92 (1.65) [14] | 20.06 (1.95) [17] | 68.44 (0.88) [1] |
| MNIST-C dotted line | 57.37 (1.38) [3] | 41.52 (0.98) [10] | 49.47 (1.31) [5] | 33.78 (1.17) [16] | 43.70 (1.26) [8] | 35.75 (1.08) [14] | 48.00 (1.57) [6] | 62.49 (1.19) [2] | 40.31 (1.50) [12] | 22.03 (3.48) [18] | 50.32 (1.56) [4] | 24.91 (4.44) [19] | 35.47 (1.76) [15] | 46.03 (1.53) [7] | 42.16 (1.36) [9] | 41.38 (1.73) [11] | 17.24 (12.40) [20] | 37.72 (1.26) [13] | 32.73 (2.18) [17] | 72.00 (0.22) [1] |
| MNIST-C fog | 78.04 (1.64) [3] | 58.34 (1.64) [11] | 71.01 (1.31) [5] | 52.29 (1.25) [17] | 59.33 (1.61) [10] | 56.23 (1.84) [14] | 72.13 (1.67) [4] | 88.50 (0.68) [2] | 57.71 (1.91) [12] | 38.36 (2.97) [19] | 70.37 (1.19) [6] | 23.23 (9.80) [20] | 52.64 (1.52) [16] | 67.28 (0.94) [7] | 62.49 (2.03) [8] | 60.94 (2.05) [9] | 26.25 (16.46) [19] | 57.07 (1.26) [13] | 53.69 (1.31) [14] | 89.74 (0.22) [1] |
| MNIST-C glass blur | 83.82 (0.80) [3] | 72.84 (0.85) [10] | 77.20 (0.95) [6] | 67.77 (1.36) [17] | 74.81 (1.19) [8] | 68.05 (1.16) [16] | 80.15 (1.12) [4] | 88.60 (1.13) [2] | 72.20 (0.97) [12] | 59.33 (4.96) [18] | 79.17 (0.87) [5] | 36.31 (16.17) [19] | 68.54 (1.21) [15] | 75.93 (0.69) [7] | 72.76 (0.72) [11] | 73.33 (1.25) [9] | 31.60 (13.05) [20] | 70.23 (1.27) [13] | 63.08 (1.31) [14] | 89.74 (0.22) [1] |
| MNIST-C identity | 3.24 (0.99) [15] | 3.94 (0.63) [10] | 3.52 (0.81) [11] | 3.38 (1.64) [14] | 3.52 (0.78) [11] | 3.45 (0.36) [13] | 4.58 (0.93) [2] | 4.44 (1.05) [3] | 2.40 (0.65) [18] | 4.86 (0.76) [1] | 4.16 (1.50) [8] | 3.24 (1.20) [15] | 2.26 (1.34) [19] | 4.02 (1.23) [9] | 4.09 (0.85) [8] | 3.10 (1.75) [17] | 4.44 (1.15) [3] | 4.29 (0.77) [5] | 2.00 (0.52) [20] |
| MNIST-C impulse noise | 97.96 (0.41) [5] | 97.11 (0.26) [13] | 97.04 (0.36) [15] | 97.11 (0.26) [13] | 96.06 (0.34) [16] | 98.10 (0.26) [4] | 97.38 (0.35) [8] | 99.30 (0.32) [1] | 97.18 (0.22) [11] | 91.56 (2.23) [18] | 97.82 (0.26) [6] | 64.18 (06.05) [19] | 97.18 (0.22) [11] | 98.59 (0.45) [3] | 98.74 (0.26) [2] | 91.98 (0.52) [17] | 33.95 (13.95) [20] | 97.75 (0.26) [9] | 97.47 (0.47) [10] | 97.76 (0.10) [8] |
| MNIST-C motion blur | 71.57 (1.60) [3] | 54.12 (1.15) [12] | 63.12 (1.17) [6] | 47.29 (1.92) [17] | 56.23 (0.93) [9] | 55.67 (1.20) [10] | 60.81 (1.12) [7] | 74.67 (0.13) [2] | 52.71 (1.61) [13] | 35.33 (2.59) [18] | 65.38 (1.03) [5] | 29.00 (10.61) [19] | 48.00 (1.19) [16] | 67.91 (1.64) [4] | 55.24 (1.31) [11] | 58.98 (1.52) [8] | 22.52 (9.18) [20] | 51.37 (1.93) [14] | 48.35 (1.36) [15] | 81.55 (0.47) [1] |
| MNIST-C rotate | 15.21 (1.05) [3] | 9.08 (0.61) [14] | 10.49 (0.93) [8] | 7.04 (0.79) [18] | 9.57 (0.74) [10] | 9.43 (0.91) [11] | 12.11 (0.61) [5] | 17.39 (1.05) [2] | 9.15 (0.61) [13] | 6.06 (1.16) [19] | 11.26 (0.61) [6] | 9.43 (1.45) [11] | 7.53 (0.98) [17] | 13.44 (0.89) [4] | 9.93 (1.05) [9] | 10.63 (0.97) [7] | 4.76 (0.73) [16] | 10.63 (0.96) [7] | 9.02 (0.50) [15] | 20.92 (0.43) [1] |
| MNIST-C scale | 42.51 (1.27) [3] | 29.56 (1.55) [12] | 37.58 (1.71) [5] | 23.23 (1.03) [17] | 31.88 (1.66) [9] | 24.63 (1.36) [15] | 38.56 (1.29) [4] | 45.89 (1.12) [2] | 30.05 (1.60) [11] | 14.15 (4.51) [20] | 32.72 (2.01) [7] | 16.82 (8.40) [19] | 24.70 (1.39) [14] | 32.66 (1.08) [8] | 30.76 (1.55) [10] | 35.54 (0.90) [6] | 19.92 (8.04) [18] | 29.21 (1.55) [13] | 24.49 (1.58) [16] | 51.95 (1.39) [1] |
| MNIST-C shear | 30.76 (1.58) [3] | 22.10 (1.19) [12] | 25.13 (1.63) [8] | 18.44 (1.21) [17] | 23.02 (1.10) [10] | 19.07 (1.11) [14] | 26.11 (0.83) [5] | 31.53 (2.02) [2] | 21.24 (1.03) [13] | 16.82 (1.80) [18] | 24.56 (1.76) [7] | 10.56 (3.86) [20] | 18.58 (1.67) [16] | 24.35 (0.97) [8] | 23.72 (1.40) [9] | 27.52 (1.65) [4] | 14.08 (5.66) [19] | 21.39 (1.25) [13] | 19.07 (1.26) [14] | 39.05 (0.85) [1] |
| MNIST-C shot noise | 50.39 (2.26) [3] | 36.81 (2.49) [9] | 43.99 (1.61) [4] | 27.94 (1.79) [17] | 39.41 (2.01) [8] | 33.08 (1.40) [13] | 43.35 (1.97) [5] | 59.12 (2.76) [2] | 34.55 (2.67) [12] | 23.44 (3.20) [19] | 43.28 (1.91) [6] | 23.02 (5.73) [20] | 29.00 (2.36) [16] | 35.54 (2.29) [11] | 36.31 (2.30) [10] | 41.17 (2.13) [7] | 23.93 (10.83) [18] | 31.67 (2.02) [14] | 27.22 (2.72) [15] | 64.32 (0.52) [1] |
| MNIST-C spatter | 63.55 (1.47) [3] | 51.23 (1.37) [11] | 61.02 (1.76) [5] | 43.63 (1.60) [18] | 53.13 (1.36) [10] | 42.93 (1.08) [17] | 59.68 (1.43) [6] | 69.67 (1.25) [2] | 51.16 (1.78) [12] | 38.43 (3.57) [18] | 61.86 (1.19) [4] | 24.49 (8.25) [19] | 44.83 (1.40) [14] | 54.75 (2.20) [7] | 53.41 (1.59) [9] | 53.63 (1.05) [8] | 20.20 (12.90) [20] | 48.14 (1.91) [13] | 44.62 (1.76) [15] | 74.08 (0.85) [1] |
| MNIST-C stripe | 95.57 (0.52) [4] | 94.02 (0.45) [13] | 95.21 (0.18) [5] | 92.68 (0.41) [16] | 95.14 (0.60) [6] | 94.23 (0.91) [8] | 94.37 (0.59) [7] | 98.31 (0.84) [1] | 93.46 (0.28) [13] | 87.40 (1.94) [18] | 94.16 (0.85) [9] | 41.59 (23.91) [19] | 92.82 (0.69) [15] | 96.48 (0.50) [3] | 94.09 (0.52) [10] | 91.20 (0.74) [17] | 30.61 (14.23) [20] | 93.53 (0.17) [13] | 93.03 (0.82) [14] | 97.26 (0.19) [2] |
| MNIST-C translate | 34.06 (1.27) [2] | 18.37 (1.30) [13] | 28.36 (1.40) [5] | 13.94 (1.06) [19] | 19.99 (1.61) [10] | 18.58 (1.60) [11] | 27.03 (1.62) [6] | 33.90 (0.65) [3] | 18.58 (0.93) [11] | 9.43 (0.72) [20] | 26.82 (1.60) [7] | 15.63 (2.44) [15] | 14.29 (0.82) [17] | 29.35 (1.29) [4] | 21.61 (0.87) [9] | 26.60 (0.93) [8] | 15.06 (4.68) [16] | 17.67 (0.80) [14] | 14.01 (1.43) [18] | 49.84 (0.90) [1] |
| MNIST-C zigzag | 67.35 (1.10) [3] | 51.37 (1.36) [11] | 59.68 (1.27) [5] | 43.42 (1.55) [15] | 53.98 (1.08) [8] | 42.37 (1.28) [17] | 59.54 (1.39) [6] | 68.54 (1.01) [2] | 51.16 (1.76) [12] | 34.63 (3.22) [18] | 61.01 (0.47) [4] | 25.90 (5.97) [20] | 44.76 (1.83) [14] | 57.99 (1.90) [7] | 52.50 (1.68) [10] | 52.57 (1.74) [9] | 33.30 (10.07) [19] | 49.33 (0.97) [13] | 40.81 (1.86) [16] | 72.99 (0.34) [1] |
| MVTec-AD bottle | 91.95 (2.65) [3] | 88.42 (0.78) [14] | 88.62 (0.80) [12] | 85.21 (1.99) [17] | 91.88 (2.24) [4] | 89.91 (2.90) [8] | 92.14 (1.96) [2] | 91.67 (2.40) [5] | 89.27 (0.53) [9] | 86.25 (2.42) [16] | 26.54 (8.43) [20] | 74.72 (11.53) [19] | 86.46 (1.84) [15] | 89.27 (1.76) [9] | 91.50 (2.69) [6] | 90.37 (1.43) [7] | 89.17 (3.47) [11] | 88.53 (1.30) [13] | 83.91 (3.34) [18] | 92.22 (2.11) [1] |
| MVTec-AD cable | 69.71 (3.93) [2] | 41.18 (4.89) [13] | 40.33 (4.93) [14] | 33.91 (3.93) [17] | 71.13 (4.49) [1] | 62.10 (5.40) [8] | 68.09 (3.27) [5] | 59.62 (6.66) [9] | 42.39 (2.34) [12] | 34.97 (5.03) [16] | 22.60 (3.66) [19] | 15.94 (14.89) [20] | 36.53 (3.49) [15] | 67.06 (2.22) [6] | 68.34 (3.15) [4] | 66.50 (3.19) [7] | 56.76 (5.83) [10] | 42.84 (5.66) [11] | 29.09 (4.88) [18] | 69.37 (3.67) [3] |

| Model | NCSBAD | KNN | LOF | PCA | KPCA | KDE | GMM | LUNAR | AE | EIF | ABOD | DROCC | GOAD | ICL | SLAD | DTE-C | DTENonParametric | MCM | DDPM | DenoiserAD (Ours) |
|---|---|---|---|---|---|---|---|---|---|---|---|---|---|---|---|---|---|---|---|---|
| MVTec-AD capsule | 65.92 (3.22) [6] | 33.96 (2.85) [11] | 32.73 (5.97) [12] | 21.72 (4.57) [18] | 70.58 (4.52) [3] | 69.38 (3.69) [5] | 70.98 (4.93) [2] | 48.64 (6.53) [8] | 30.42 (4.37) [13] | 23.48 (3.52) [16] | 20.31 (3.20) [19] | 15.97 (4.09) [20] | 23.87 (3.88) [15] | 70.19 (3.95) [4] | 63.39 (3.01) [7] | 44.53 (3.96) [10] | 48.04 (2.08) [9] | 30.41 (4.67) [14] | 21.89 (3.42) [17] | **72.51 (3.37) [1]** |
| MVTec-AD carpet | 63.97 (2.29) [5] | 42.93 (3.54) [12] | 43.08 (4.36) [11] | 39.70 (6.43) [17] | 65.60 (2.21) [2] | 63.82 (2.89) [6] | 64.76 (1.96) [3] | 53.78 (6.17) [9] | 41.42 (4.41) [14] | 41.04 (4.28) [15] | 29.57 (6.71) [19] | 27.34 (16.49) [20] | 40.09 (7.15) [16] | 64.20 (4.60) [4] | 63.39 (2.71) [7] | 57.61 (5.73) [8] | 53.26 (2.84) [10] | 42.00 (6.11) [13] | 32.72 (4.37) [18] | **67.08 (1.37) [1]** |
| MVTec-AD grid | 69.86 (4.01) [3] | 41.67 (1.43) [11] | 40.94 (1.54) [14] | 38.01 (2.01) [16] | 67.42 (3.68) [6] | 67.70 (4.01) [5] | 68.59 (4.70) [4] | 56.22 (8.45) [9] | 41.40 (2.03) [13] | 35.86 (3.98) [18] | 16.81 (7.24) [20] | 27.27 (11.08) [19] | 39.21 (1.95) [15] | 70.09 (3.63) [2] | 65.48 (5.55) [7] | 58.65 (4.96) [8] | 48.28 (2.37) [10] | 41.52 (2.12) [12] | 36.11 (2.56) [17] | **71.46 (3.31) [1]** |
| MVTec-AD hazelnut | 60.38 (4.16) [5] | 43.54 (3.66) [14] | 45.00 (1.86) [11] | 40.27 (3.55) [17] | 59.90 (4.67) [6] | 60.83 (3.08) [4] | 57.91 (5.02) [8] | 54.76 (4.02) [9] | 43.33 (2.30) [15] | 43.56 (5.06) [13] | 33.40 (6.51) [19] | 23.51 (10.65) [20] | 40.85 (3.77) [16] | **64.86 (6.83) [1]** | 61.30 (6.91) [3] | 59.75 (6.76) [7] | 45.38 (3.22) [10] | 43.99 (3.44) [12] | 38.89 (2.83) [18] | 61.45 (5.25) [2] |
| MVTec-AD leather | 95.60 (2.36) [4] | 93.04 (3.34) [13] | 91.27 (3.30) [17] | 92.42 (4.10) [15] | 95.22 (1.67) [6] | 93.62 (2.72) [9] | 95.60 (2.93) [4] | 95.46 (0.77) [2] | 93.43 (2.99) [10] | 92.23 (4.30) [16] | 64.30 (8.24) [19] | 22.57 (25.61) [20] | 92.63 (3.75) [14] | 95.22 (1.56) [6] | 95.62 (1.94) [3] | 93.25 (3.01) [11] | 93.13 (2.98) [12] | 88.28 (2.10) [18] | **95.81 (1.69) [1]** |
| MVTec-AD metal nut | 73.91 (4.36) [2] | 36.44 (3.80) [14] | 36.81 (3.15) [12] | 32.71 (3.87) [17] | 71.78 (3.23) [4] | 70.05 (3.82) [5] | 73.75 (4.73) [3] | 54.56 (8.80) [9] | 38.14 (3.99) [11] | 36.56 (2.93) [13] | 23.20 (2.89) [19] | 15.14 (9.41) [20] | 33.48 (5.34) [16] | 65.95 (3.21) [6] | 65.18 (4.82) [7] | 63.64 (5.89) [8] | 46.15 (3.72) [10] | 34.77 (4.97) [15] | 29.46 (5.19) [18] | **74.66 (4.11) [1]** |
| MVTec-AD pill | 57.26 (4.43) [7] | 33.90 (3.76) [11] | 33.17 (6.68) [12] | 29.37 (3.70) [16] | 61.32 (4.78) [5] | 59.01 (5.90) [6] | 65.87 (4.03) [2] | 41.70 (8.22) [10] | 31.57 (3.89) [13] | 28.22 (5.08) [17] | 17.13 (3.95) [18] | 16.02 (7.55) [20] | 30.12 (4.75) [15] | 62.02 (2.31) [4] | 64.27 (3.16) [3] | 47.25 (1.81) [8] | 41.72 (4.34) [9] | 31.14 (5.53) [14] | 16.35 (4.23) [19] | **66.90 (1.83) [1]** |
| MVTec-AD screw | 59.21 (5.73) [6] | 28.56 (6.83) [12] | 29.77 (6.50) [11] | 20.43 (3.89) [17] | 61.45 (4.13) [5] | 62.74 (2.75) [2] | 61.63 (5.26) [4] | 47.48 (5.76) [9] | 28.48 (5.12) [13] | 19.99 (9.09) [18] | 17.18 (7.82) [20] | 23.78 (8.23) [15] | 22.74 (4.44) [16] | **64.86 (4.97) [1]** | 56.02 (3.61) [7] | 47.76 (4.54) [8] | 35.72 (6.20) [10] | 26.94 (5.58) [14] | 18.16 (5.52) [19] | 62.44 (3.94) [3] |
| MVTec-AD tile | 77.61 (2.24) [6] | 62.92 (2.43) [11] | 62.09 (3.33) [14] | 60.87 (3.87) [15] | 78.24 (2.38) [4] | 72.40 (2.87) [9] | 76.05 (4.74) [7] | 72.79 (5.10) [8] | 62.68 (4.00) [12] | 60.58 (2.93) [16] | 38.74 (5.27) [19] | 21.35 (19.70) [20] | 58.16 (6.43) [17] | 77.82 (3.33) [5] | 78.59 (2.33) [2] | 78.55 (4.42) [3] | 68.64 (1.36) [10] | 62.52 (4.15) [13] | 50.93 (1.75) [18] | **79.01 (2.82) [1]** |
| MVTec-AD toothbrush | 97.65 (1.48) [8] | 82.29 (7.01) [11] | 72.79 (8.53) [13] | 51.19 (4.47) [17] | 99.03 (1.24) [3] | 99.03 (1.24) [3] | 99.03 (1.24) [3] | 97.66 (2.00) [7] | 67.89 (5.88) [15] | 75.48 (9.83) [12] | 48.36 (12.80) [18] | 53.06 (2.15) [16] | 98.65 (1.90) [5] | 98.65 (1.90) [5] | 90.31 (9.68) [9] | 89.06 (4.87) [10] | 70.04 (4.27) [14] | 38.71 (2.64) [19] | **99.07 (1.21) [1]** |
| MVTec-AD transistor | 72.17 (5.97) [2] | 37.44 (5.91) [14] | 36.40 (6.29) [16] | 38.42 (4.48) [12] | **72.48 (4.51) [1]** | 71.12 (4.44) [4] | 68.86 (2.60) [6] | 63.16 (7.41) [9] | 36.09 (2.36) [17] | 39.90 (9.91) [11] | 7.99 (13.56) [20] | 28.62 (15.45) [19] | 37.44 (4.26) [14] | 68.20 (3.56) [7] | 70.39 (4.77) [5] | 66.76 (5.97) [8] | 46.71 (4.46) [10] | 38.06 (3.01) [13] | 35.99 (4.11) [18] | 71.72 (3.52) [3] |
| MVTec-AD wood | 77.43 (4.08) [2] | 49.27 (3.42) [14] | 57.03 (2.54) [11] | 46.50 (5.10) [18] | 77.41 (3.56) [3] | 76.91 (3.04) [5] | **77.85 (4.40) [1]** | 67.16 (1.95) [9] | 51.09 (2.36) [12] | 49.00 (2.68) [15] | 33.77 (3.33) [20] | 39.96 (12.23) [19] | 46.97 (5.30) [17] | 76.67 (6.29) [7] | 75.67 (6.29) [8] | 71.43 (4.01) [8] | 58.00 (6.06) [10] | 49.95 (4.37) [13] | 48.85 (3.72) [16] | 77.27 (2.45) [4] |
| MVTec-AD zipper | 86.26 (3.09) [2] | 59.24 (7.00) [14] | 61.71 (6.79) [11] | 49.93 (5.11) [17] | 84.25 (3.98) [3] | 80.89 (6.77) [6] | 82.02 (5.30) [5] | 74.66 (7.66) [9] | 59.27 (5.81) [13] | 53.38 (5.92) [15] | 44.25 (9.11) [18] | 17.01 (14.00) [20] | 52.25 (5.30) [16] | 80.01 (3.20) [7] | 83.08 (3.48) [4] | 79.16 (4.25) [8] | 68.08 (8.54) [10] | 60.02 (4.62) [12] | 27.40 (7.77) [19] | **87.61 (1.94) [1]** |
| SVHN 0 | 13.53 (1.24) [7] | 11.64 (1.94) [12] | 13.40 (0.96) [8] | 9.88 (0.66) [17] | 12.04 (1.96) [10] | 13.80 (1.74) [6] | 12.66 (1.76) [9] | 17.19 (1.75) [3] | 11.10 (1.94) [13] | 8.66 (1.54) [19] | 15.02 (2.03) [4] | 10.56 (1.09) [16] | 13.87 (1.10) [10] | **17.87 (1.33) [1]** | 11.91 (2.16) [11] | 12.85 (2.24) [9] | 5.69 (7.18) [20] | 13.10 (1.94) [13] | 10.96 (2.04) [15] | 17.68 (0.40) [2] |
| SVHN 1 | 15.84 (1.76) [3] | 13.52 (1.29) [11] | 10.07 (1.06) [17] | 14.15 (1.07) [9] | 14.36 (1.34) [7] | 9.29 (1.59) [19] | 14.57 (1.55) [5] | **19.49 (0.93) [1]** | 12.88 (1.19) [15] | 13.30 (1.91) [14] | 15.21 (1.18) [4] | 9.78 (2.73) [18] | 11.05 (1.19) [16] | 13.44 (1.20) [12] | 14.57 (1.53) [5] | 0.57 (6.65) [20] | 13.31 (0.91) [13] | 14.36 (1.31) [7] | 18.81 (0.39) [2] | |
| SVHN 2 | 18.44 (1.93) [4] | 16.82 (2.24) [13] | 17.67 (1.80) [7] | 17.24 (1.90) [10] | 17.17 (2.26) [12] | 15.41 (1.27) [17] | 18.65 (2.18) [3] | 18.72 (1.35) [2] | 17.67 (2.03) [7] | 14.78 (2.27) [18] | 17.24 (1.70) [10] | 10.42 (1.01) [19] | 16.75 (1.69) [14] | 16.47 (1.95) [15] | 18.44 (2.01) [4] | 18.23 (2.02) [6] | 8.17 (5.25) [20] | 17.60 (2.02) [9] | 16.33 (1.50) [16] | **23.16 (0.35) [1]** |
| SVHN 3 | **17.92 (1.13) [1]** | 14.77 (1.95) [12] | 16.90 (0.90) [3] | 12.72 (1.59) [17] | 15.08 (1.68) [8] | 13.51 (1.38) [14] | 16.82 (1.04) [4] | 99.03 (1.24) [2] | 15.29 (1.25) [9] | 15.26 (1.67) [16] | 16.92 (1.21) [9] | 10.69 (0.90) [18] | 13.35 (1.43) [15] | 14.61 (1.44) [13] | 15.40 (1.42) [7] | 15.87 (1.13) [5] | 0.51 (7.11) [20] | 15.00 (1.62) [9] | 13.04 (1.48) [16] | 17.45 (0.19) [2] |
| SVHN 4 | 15.38 (0.97) [6] | 15.56 (0.49) [4] | 14.12 (1.37) [15] | 14.48 (0.22) [14] | 15.56 (0.57) [4] | 11.43 (1.43) [18] | 15.20 (0.52) [10] | 17.27 (1.87) [3] | 15.29 (0.36) [8] | 15.38 (1.29) [6] | **17.90 (0.87) [1]** | 7.39 (1.91) [19] | 14.75 (0.34) [13] | 11.61 (2.11) [17] | 15.29 (1.09) [8] | 13.32 (1.02) [16] | -4.47 (1.94) [20] | 14.93 (0.54) [11] | 14.93 (0.83) [11] | 17.71 (0.32) [2] |
| SVHN 5 | 17.60 (1.25) [7] | 15.75 (1.55) [12] | 18.57 (1.25) [5] | 15.27 (1.61) [15] | 16.53 (1.05) [10] | 14.39 (1.63) [17] | 19.25 (1.27) [3] | 15.26 (1.67) [16] | 16.92 (1.21) [9] | 10.69 (0.90) [18] | 18.48 (1.00) [6] | 9.92 (0.78) [19] | 15.46 (1.52) [13] | 19.55 (1.18) [2] | 17.41 (1.75) [8] | 18.87 (1.58) [4] | 4.95 (5.98) [20] | 16.53 (1.33) [10] | 15.36 (1.95) [14] | **19.92 (0.52) [1]** |
| SVHN 6 | 14.86 (1.17) [5] | 13.11 (1.73) [11] | 14.04 (1.20) [7] | 12.53 (1.65) [15] | 14.28 (1.52) [6] | 12.06 (1.13) [16] | 15.21 (1.20) [3] | 14.98 (1.54) [4] | 13.70 (1.77) [9] | 9.26 (1.14) [18] | 15.56 (2.33) [2] | 8.10 (0.79) [19] | 12.76 (1.76) [13] | 11.94 (1.84) [17] | 14.04 (1.50) [7] | 12.77 (1.50) [12] | 3.08 (3.03) [20] | 13.69 (1.61) [10] | 12.76 (2.17) [13] | **15.65 (0.28) [1]** |
| SVHN 7 | 18.92 (2.95) [4] | 18.68 (2.50) [7] | 17.00 (1.45) [17] | 18.92 (2.32) [4] | 18.56 (2.66) [9] | 19.16 (1.91) [3] | 17.84 (2.76) [14] | **20.12 (1.91) [1]** | 19.28 (2.39) [2] | 17.96 (1.95) [13] | 18.44 (2.21) [10] | 10.53 (0.88) [19] | 18.92 (2.22) [4] | 17.48 (1.88) [16] | 15.44 (2.40) [18] | 18.08 (1.72) [12] | -2.91 (5.28) [20] | 18.44 (2.31) [10] | 15.48 (1.02) [6] | |
| SVHN 8 | 15.56 (1.94) [5] | 12.78 (1.37) [13] | 16.23 (1.94) [3] | 11.71 (1.63) [17] | 13.31 (1.81) [12] | 13.57 (1.42) [11] | 15.17 (1.85) [7] | 14.77 (2.28) [8] | 14.10 (2.28) [9] | 9.72 (2.55) [18] | 16.76 (2.36) [2] | 7.20 (2.46) [19] | 11.85 (1.42) [16] | 16.10 (2.90) [4] | 13.97 (1.76) [10] | 17.56 (1.94) [1] | -2.89 (5.31) [20] | 12.78 (1.71) [13] | 12.24 (1.06) [15] | 15.48 (1.02) [6] |
| SVHN 9 | 12.55 (2.99) [6] | 11.40 (2.67) [10] | 12.40 (2.98) [7] | 10.68 (2.63) [16] | 12.83 (3.03) [5] | 11.26 (3.29) [12] | 13.55 (2.96) [1] | 11.83 (2.62) [8] | 10.39 (2.66) [17] | 8.09 (3.25) [18] | 13.12 (3.05) [2] | 7.37 (1.70) [19] | 10.97 (3.37) [14] | 12.98 (2.54) [3] | 11.40 (2.67) [10] | 11.54 (3.70) [9] | -0.24 (6.62) [20] | 10.97 (2.76) [14] | 11.25 (3.87) [13] | 12.80 (0.57) [4] |
| agnews 0 | 22.03 (0.91) [2] | 12.18 (1.35) [10] | 16.33 (0.72) [3] | 4.65 (1.46) [19] | 12.74 (1.35) [8] | 9.72 (1.35) [14] | 16.05 (1.50) [4] | 13.23 (1.36) [7] | 14.78 (1.19) [5] | -2.67 (5.77) [20] | 5.56 (1.63) [17] | 10.42 (1.10) [12] | 9.43 (0.93) [15] | 14.22 (1.52) [6] | 12.53 (1.04) [9] | 10.28 (1.26) [13] | 5.42 (1.84) [18] | **24.00 (0.95) [1]** | | |
| agnews 1 | 30.33 (1.22) [3] | 12.04 (0.92) [14] | 33.15 (1.60) [2] | 5.91 (1.50) [18] | 13.87 (0.41) [9] | 13.52 (1.03) [11] | 25.05 (1.28) [4] | 16.61 (0.89) [6] | 13.52 (1.22) [11] | 5.00 (0.80) [19] | 18.65 (0.52) [5] | -5.56 (0.00) [20] | 6.48 (0.72) [16] | 14.71 (1.13) [7] | 7.53 (1.36) [15] | 13.73 (0.68) [10] | 13.37 (0.93) [13] | 14.01 (1.36) [8] | 6.27 (1.64) [17] | **33.50 (1.14) [1]** |
| agnews 2 | 34.77 (1.19) [2] | 22.38 (0.82) [12] | 29.35 (1.63) [5] | 12.11 (0.90) [18] | 23.37 (0.84) [10] | 21.33 (0.76) [13] | 31.18 (0.69) [3] | 24.49 (0.73) [8] | 26.53 (0.87) [7] | 13.51 (1.12) [15] | 29.77 (1.84) [4] | 5.07 (5.60) [20] | 13.87 (1.12) [14] | 12.46 (1.53) [17] | 7.75 (0.14) [19] | 24.28 (0.41) [9] | 22.94 (0.45) [11] | 27.31 (0.36) [6] | 12.88 (1.57) [16] | **37.55 (0.78) [1]** |
| agnews 3 | 34.13 (2.01) [2] | 16.68 (0.61) [12] | 27.59 (1.29) [4] | 7.32 (0.57) [19] | 19.71 (0.61) [8] | 14.99 (1.55) [13] | 28.65 (1.48) [3] | 18.02 (1.00) [9] | 17.10 (0.73) [11] | 7.67 (1.08) [18] | 23.16 (1.39) [5] | -0.84 (5.79) [20] | 8.03 (0.65) [16] | 12.46 (1.80) [14] | 8.59 (1.51) [15] | 20.34 (1.82) [7] | 17.17 (0.65) [10] | 20.97 (0.96) [6] | 7.74 (1.17) [17] | **35.53 (0.76) [1]** |
| amazon | 6.20 (2.27) [10] | 6.69 (1.27) [7] | 4.37 (1.31) [17] | 6.13 (2.02) [11] | 7.11 (1.24) [5] | 7.61 (1.45) [2] | 4.30 (1.96) [18] | 5.63 (1.81) [15] | 5.28 (1.81) [15] | 6.41 (1.59) [9] | **9.36 (1.87) [1]** | -5.56 (0.00) [20] | 7.53 (2.00) [3] | 4.02 (1.03) [19] | 5.56 (0.53) [14] | 7.32 (1.19) [4] | 7.04 (1.45) [6] | 5.84 (1.96) [12] | 6.52 (0.52) [8] | |
| imdb | 0.29 (0.48) [15] | -0.28 (0.59) [17] | 1.27 (0.82) [6] | 0.50 (0.90) [12] | -0.28 (0.59) [17] | 1.69 (0.96) [5] | 0.92 (0.48) [10] | -0.35 (0.81) [19] | 0.07 (0.80) [16] | 0.99 (0.91) [9] | 2.11 (0.81) [3] | 1.20 (5.57) [7] | 0.36 (0.41) [14] | **5.71 (0.97) [1]** | 5.63 (0.64) [2] | 0.57 (1.40) [11] | -0.63 (0.59) [20] | 0.43 (0.92) [13] | 1.06 (1.03) [8] | 1.71 (0.29) [4] |
| yelp | 16.82 (2.40) [2] | 14.36 (1.82) [9] | 15.34 (1.88) [4] | 12.67 (1.45) [12] | 14.08 (1.83) [10] | 11.41 (1.40) [16] | 15.06 (2.19) [5] | 15.63 (1.77) [3] | 14.43 (2.05) [8] | 11.76 (1.65) [15] | 14.78 (0.41) [6] | 4.23 (5.29) [18] | 12.04 (1.42) [14] | 3.10 (1.06) [19] | 1.69 (0.26) [20] | 12.32 (1.59) [13] | 14.64 (2.02) [7] | 13.70 (1.65) [11] | 11.05 (2.02) [17] | **16.99 (1.08) [1]** |
| 20news 0 | 35.53 (0.86) [1] | 20.62 (0.89) [5] | 31.17 (2.29) [3] | 7.54 (1.56) [20] | 23.83 (0.92) [8] | 20.85 (2.65) [12] | 24.75 (0.56) [6] | 23.83 (0.56) [7] | 25.39 (1.86) [5] | 9.61 (2.22) [17] | 27.27 (1.56) [4] | 11.44 (5.40) [16] | 9.15 (1.13) [18] | 18.55 (1.62) [14] | 11.90 (1.34) [15] | 24.52 (2.66) [7] | 22.22 (0.86) [11] | 23.14 (1.26) [10] | 8.00 (1.52) [19] | **35.03 (0.80) [2]** |
| 20news 1 | 13.89 (3.49) [2] | 4.61 (1.64) [15] | 5.17 (2.28) [12] | 1.23 (2.25) [19] | 7.14 (1.99) [8] | 8.55 (3.48) [7] | 9.67 (2.42) [5] | 6.86 (2.07) [10] | 9.39 (2.76) [6] | 1.80 (2.73) [18] | 12.49 (3.26) [4] | 4.61 (6.74) [15] | 1.23 (1.87) [19] | 13.61 (3.16) [3] | 7.14 (1.26) [8] | 5.45 (2.42) [11] | 4.89 (1.91) [14] | 5.17 (1.44) [12] | 2.36 (2.90) [17] | **14.57 (1.45) [1]** |
| 20news 2 | 9.03 (2.77) [2] | 3.33 (1.40) [15] | 4.75 (1.66) [8] | 3.04 (2.55) [16] | 4.47 (0.90) [9] | 6.46 (2.64) [4] | 5.32 (2.13) [6] | 8.17 (2.13) [3] | 3.04 (0.90) [16] | 2.47 (1.93) [18] | 3.90 (1.14) [11] | 4.47 (3.12) [9] | 3.61 (1.93) [14] | 5.04 (1.93) [7] | 2.19 (1.14) [20] | 5.61 (2.45) [5] | 3.90 (1.45) [11] | 2.47 (1.93) [18] | 3.90 (1.93) [11] | **11.68 (0.99) [1]** |
| 20news 3 | 40.93 (9.64) [3] | 23.45 (9.97) [8] | 19.99 (10.74) [9] | 12.80 (11.27) [16] | **44.06 (9.65) [1]** | 37.11 (6.02) [5] | 43.40 (10.47) [2] | 33.71 (7.59) [6] | 19.44 (6.98) [11] | 13.60 (10.98) [14] | -3.29 (3.11) [20] | 10.42 (7.03) [18] | 14.32 (12.42) [12] | 13.27 (5.12) [15] | 12.33 (11.55) [17] | 19.99 (9.09) [9] | 32.00 (12.32) [7] | 14.04 (5.63) [13] | 9.82 (7.61) [19] | 38.36 (8.53) [4] |
| 20news 4 | 12.59 (3.50) [2] | 7.34 (1.12) [14] | 10.09 (2.25) [6] | 8.72 (2.20) [9] | 6.99 (3.45) [14] | 5.27 (3.05) [20] | 10.44 (2.92) [3] | 7.85 (2.86) [12] | 6.99 (3.95) [14] | 6.99 (3.95) [14] | 7.85 (4.18) [12] | 5.70 (3.70) [18] | 8.72 (2.20) [9] | 10.01 (2.11) [5] | 9.58 (2.36) [7] | 10.44 (2.92) [3] | 6.99 (4.39) [14] | 9.15 (1.61) [8] | 8.28 (4.22) [11] | **13.54 (1.50) [1]** |
| 20news 5 | 15.14 (3.24) [2] | 7.34 (1.12) [14] | 10.09 (2.25) [6] | 1.83 (3.05) [18] | 11.46 (2.76) [5] | 7.34 (2.34) [14] | 12.84 (3.25) [3] | 9.63 (1.84) [8] | 8.25 (1.45) [12] | 0.92 (2.68) [20] | 9.17 (1.84) [9] | 7.34 (2.34) [14] | 1.83 (3.05) [18] | 10.09 (4.90) [6] | 9.17 (4.25) [9] | 11.47 (2.34) [4] | 9.17 (1.84) [9] | 7.80 (1.53) [13] | 3.21 (4.68) [17] | **16.70 (0.88) [1]** |
| Average | 48.24 (30.55) [2] | 40.78 (29.70) [11] | 40.95 (27.24) [10] | 33.63 (28.29) [14] | 46.82 (32.41) [4] | 43.29 (31.80) [7] | 43.55 (29.63) [6] | 48.16 (31.71) [3] | 39.03 (28.32) [12] | 32.07 (28.11) [17] | 32.58 (27.67) [16] | 18.73 (18.97) [20] | 32.76 (27.47) [15] | 44.33 (31.49) [5] | 41.53 (31.48) [9] | 42.06 (28.29) [8] | 30.73 (30.13) [19] | 38.21 (27.27) [13] | 31.71 (26.43) [18] | **50.94 (31.14) [1]** |

*Table 11.* Full results, including mean, standard deviation and rank ADJ F1-score performance of all methods for each ADBench dataset and the overall average. The best performance is highlighted in **bold**, the second best highlighted with underline.

