# OpenReview forum: "Denoising without Diffusion: Fixed-Noise Denoiser Anomaly Detection in Tabular Data"
_ICML.cc/2026/Conference — ICML 2026 regular_

### Official Review · Reviewer_zraT · 2026-03-10

**Soundness:** 4
**Presentation:** 4
**Significance:** 3
**Originality:** 3
**Overall Recommendation:** 5
**Confidence:** 3

**Summary:**

DenoiserAD is a fixed-noise denoising-based anomaly detection method for tabular data that replaces the multi-step diffusion process with a single noise scale and scores anomalies by the instability of the denoiser's output under repeated stochastic perturbations.

**Compliance With Llm Reviewing Policy:**

Affirmed.

**Final Justification:**

The rebuttal has addressed all my comments.

**Key Questions For Authors:**

1. The anomaly score A(y) measures deviation relative to the original input y, why is this preferable to measuring deviation relative to the denoised mean, given that y itself may be noisy at test time?
2. The training set is assumed clean, but in practice anomaly-free datasets are rare, how does the method compare against methods explicitly designed for unsupervised settings without this assumption such as isolation forest?
3. The method is validated on tabular data, is it suitable for time-series or multivariate sensor data, which share similar one-class detection requirements?

**Limitations:**

Yes

**Strengths And Weaknesses:**

Strengths:
1. Simplifying the diffusion pipeline while achieving state-of-the-art on a 57-dataset benchmark is practically valuable.
2. Core intuition is clearly conveyed, especially Figure 1 and the noise sensitivity analysis.
3.  The combination of fixed-noise denoising, input-referenced stability scoring, and reference-channel parameterization is clean and well-motivated.

Weaknesses:
1. The absolute gain over the best baseline is modest. Some baselines are not the latest, some classical baselines should be included such as Isolation Forest and its variants.
2. The framework seems to combine some existing component, should provide a clear comparison between the proposed methods and other diffusion based methods such DTE.
3. The figure 1 is a good demo, but should give more explanation on it, it looks that the performance of reconstruction-based method is also good.

---

> ### Author Rebuttal · Authors · 2026-03-26
>
> ## General
>
> Thank you for the careful and constructive review. We especially appreciate that you highlight the practical value of simplifying the diffusion pipeline and recognize the combination of fixed-noise denoising, input-referenced stability scoring, and the reference-channel parameterization as a clean and well-motivated design.
>
> We would like to clarify that several of the concerns you raise are already addressed in the current manuscript.
>
> First, regarding classical baselines: isolation-based methods are already included. In particular, the main benchmark compares against EIF (Extended Isolation Forest) with stronger performance than classic IF, alongside stronger classical baselines such as KPCA, GMM and kNN. We deliberately focused on a well-defined set of strong baselines established in the most relevant prior work in order to keep the empirical evaluation clear and interpretable.
>
> Second, the comparison to diffusion-based methods such as DTE is also already included explicitly. In the main benchmark we compare against DDPM, DTE-C, DTE-NP, and NCSBAD. In addition, Table 3 goes beyond a standard baseline comparison and tests whether the gains come from the backbone or from the proposed formulation itself. There, we apply our fixed-noise stability formulation to representative diffusion-based frameworks, including DTE, and observe consistent improvements across architectures.
>
> Third, concerning Figure 1 and the role of reconstruction, we appreciate your remark that Figure 1 is “a good demo,” because this already indicates that the figure succeeds in conveying the intended intuition clearly. The fact that reconstruction-based behavior can also look plausible in that visualization is not a counterpoint to the paper, but part of its motivation: if reconstruction can appear locally reasonable, then one should not rely on visual intuition or reconstruction error alone. This is exactly why the manuscript goes beyond Figure 1 and tests the distinction explicitly in Table 2, where the proposed input-referenced stability score outperforms plain reconstruction, augmented reconstruction / DAE+TTA, and mean-centered variance.
>
> ## Response to the key questions
>
> **1. Why is the anomaly score A(y) measured relative to the original input \(y\), rather than relative to the denoised mean?**
> This is already addressed empirically in Table 2. The manuscript compares our input-referenced score \( S_2(y)=\mathbb{E}_n \|f_\theta(y+n)-y\|^2 \) directly against the mean-centered variance score S_3, which measures deviation relative to the denoised mean. The result is clear: S_3 performs substantially worse, while S_2 gives the best overall performance. Intuitively, this is because we are not merely interested in output variability, but in whether perturbations of a test point are pulled back consistently toward the same local anchor. Appendix D.1 also provides a very clear visualization of a real example.
>
> **2. The training set is assumed clean, but anomaly-free datasets are rare in practice. How does the method compare against methods designed for unsupervised settings such as Isolation Forest?**
> Both parts of this question are already addressed in the submission. First, EIF  (significantly better than IF in previous publications) is already included as a baseline in the main benchmark. Second, the paper explicitly discusses the clean-training assumption and evaluates the contaminated fully unsupervised setting in Appendix F. There, the manuscript shows that the vanilla one-class formulation is sensitive to contamination, while a simple trimmed-loss variant improves robustness substantially and becomes competitive with strong baselines such as GMM and EIF.
>
> **3. The method is validated on tabular data; is it suitable for time-series or multivariate sensor data?**
> The current paper is intentionally scoped to tabular one-class anomaly detection, and we do not claim validation on raw time-series or sensor sequences. That said, the core score only requires a differentiable denoiser operating on a continuous feature representation, so representations derived from time-series or multivariate sensor data could in principle be compatible with the formulation. However, whether the same design choices and hyperparameter regime transfer directly is outside the scope of the current submission and is already acknowledged in the limitations.
>
> ## Conclusion
>
> Overall, we are very grateful for this thoughtful review. We especially appreciate that your comments engage directly with the method’s central design choices, and we hope the clarifications above make clear that the comparisons to isolation-based baselines, diffusion-based baselines such as DTE, and the distinction between reconstruction and input-referenced stability are already addressed in the current manuscript.
>
> We sincerely thank you again for your careful and constructive feedback. We very much appreciate the time and attention you devoted to the paper.

---

> > ### Author Rebuttal · Reviewer_zraT · 2026-04-03
> >
> > Thanks for the rebuttal, I have updated my score.

---

> > > ### Author Response · Authors · 2026-04-03
> > >
> > > Thank you very much for your positive and thoughtful feedback. We are very glad that our response was helpful in resolving your concerns, and we sincerely appreciate your careful engagement with the paper.

---

### Official Review · Reviewer_DmUz · 2026-03-10

**Soundness:** 3
**Presentation:** 4
**Significance:** 3
**Originality:** 2
**Overall Recommendation:** 4
**Confidence:** 4

**Summary:**

This paper presents DenoiserAD, a diffusion-based approach to tabular anomaly detection that eliminates the multi-step noise schedules and time conditioning. The authors propose a stability-based anomaly score that measures input-referenced consistency under repeated stochastic perturbations, rather than single-shot reconstruction error. They demonstrate that probing local manifold stability at a single noise scale is sufficient for anomaly detection, achieving state-of-the-art performance on the ADBench benchmark while maintaining computational efficiency.

**Compliance With Llm Reviewing Policy:**

Affirmed.

**Key Questions For Authors:**

- How does DenoiserAD handle categorical features? Does the method require one-hot encoding, embedding, or is it restricted to purely numerical tabular data?
- The noise scale σ² appears robust within [0.12, 0.35] on ADBench: is this range expected to work well across diverse datasets, or is it dependent on data standardization and distribution characteristics?
- How robust is the method when the training set contains some fraction of anomalies (label contamination)? At what contamination rate does performance degrade significantly?

**Limitations:**

yes

**Strengths And Weaknesses:**

Strengths
  - Clear writing and well-motivated approach that identifies and isolates the key mechanism (denoising stability) from complex diffusion pipelines, resulting in a simpler and more efficient method.
  - Comprehensive experimental validation on ADBench with strong ablation studies demonstrating the contribution of each component.
  - The method achieves state-of-the-art performance while being faster than diffusion-based alternatives.

Weaknesses
  - No discussion of how categorical features are handled. Tabular data often contains mixed feature types, and the method appears designed for continuous features only.
  - While the noise scale shows stability within a range, practical guidance for selecting σ² on new datasets is missing; it's unclear whether the [0.12, 0.35] range generalizes across individual datasets within and beyond ADBench.
  - The one-class setting assumes clean training data; no analysis of robustness to label noise or contaminated training sets, which is common in real-world anomaly detection scenarios.

---

> ### Author Rebuttal · Authors · 2026-03-26
>
> ## General
>
> Thank you for the careful and very positive review. We especially appreciate that you highlight the core mechanism of the paper—denoising stability under repeated perturbations—as both well motivated and practically effective.
>
> We would like to clarify that two of the three concerns you raise are already addressed in the current manuscript. In particular, the practical choice of the noise scale is analyzed explicitly in Sec. 5, and robustness under contaminated training is discussed both in the main paper and in Appendix F. The main point on which we agree more directly is that the manuscript could be more explicit about the scope of the method with respect to categorical / mixed-type tabular features.
>
> Regarding categorical features: DenoiserAD is formulated for real-valued tabular vectors with additive Gaussian perturbations. It is therefore directly applicable to numerical tabular data, or more generally to tabular data once all features are represented in a real-valued space. The paper does not propose a dedicated categorical-feature mechanism such as a special encoding or discrete diffusion process. This regime aligns with the properties of ADBench, in which the datasets are also purely real-valued numericals. The handling of categorical-feature was therefore already performed by the authors of ADBench during its development and thus requires no further explicit handling of categorical-feature on our part within the method. We will highlight this in the camara ready if we get accepted.
>
>
> Regarding the noise scale, Sec. 5 already studies sensitivity to $\sigma^2$ through a logarithmic sweep on ADBench and shows a broad plateau of strong performance for approximately $\sigma^2 \in [0.12, 0.35]$. Importantly, this range should be interpreted relative to the benchmark preprocessing: all inputs are standardized using the training samples, and $\sigma^2 = 0.2$ is fixed *a priori* for all experiments rather than tuned per dataset. Thus, our claim is not that $[0.12, 0.35]$ is a universal constant for arbitrary raw data, but that on standardized tabular inputs the method operates in a broad and practically robust regime.
>
> Regarding contamination, this is also already analyzed in the submission. The main paper explicitly states that the method is designed for the strict one-class setting with uncontaminated training data, and Appendix F evaluates the fully unsupervised regime where the training data contains anomalies. There, the paper shows that the vanilla method is sensitive to contamination and degrades relative to specialized robust baselines, while a simple trimmed-loss variant substantially improves robustness and becomes competitive with strong robust baselines.
>
> ## Response to the key questions
>
> **1. How does DenoiserAD handle categorical features?**
> In its current formulation, DenoiserAD operates on real-valued tabular feature vectors with additive Gaussian perturbations. Accordingly, it is directly applicable to numerical tabular data and, more generally, to tabular data once features are represented in a continuous space. The paper does not propose a dedicated categorical-feature mechanism such as a special encoding or discrete diffusion process. This is also consistent with ADBench, where all datasets are provided in a purely real-valued numerical representation.
>
> **2. Is the robust range $[0.12, 0.35]$ expected to generalize across datasets?**
> The paper’s claim is specifically tied to standardized tabular inputs. Because the benchmark applies standardization based on the training samples, $\sigma^2$ has a consistent meaning across datasets, and Sec. 5 shows a broad plateau of strong performance in that setting. We therefore interpret this as evidence for a stable operating regime on standardized tabular data, not as a claim that exactly the same interval must transfer unchanged to arbitrary non-standardized domains.
>
> **3. How robust is the method under label contamination?**
> This is already evaluated in Appendix F. The current submission shows that the vanilla one-class version is sensitive to contamination, while a trimmed-loss variant substantially improves robustness in the fully unsupervised contaminated regime and makes the method competitive with strong robust baselines. What we do not claim in the current paper is a precise universal contamination threshold at which performance drops sharply.
>
> ## Conclusion
>
> Overall, we are grateful for this very careful review. We especially appreciate that your comments engage directly with the intended scope and practical use of the method, and we hope the clarifications above make clear that the questions on noise-scale selection and contaminated training are already addressed in the current manuscript, while the categorical-feature point is best understood as a scope clarification of the present formulation.
>
> Thank you again for this thoughtful review. We truly appreciate the care with which you engaged with the paper.

---

> > ### Author Rebuttal · Reviewer_DmUz · 2026-04-03
> >
> > Thanks for the detailed response. I'll keep my score.

---

> > > ### Author Response · Authors · 2026-04-03
> > >
> > > Thank you very much for the positive feedback. We are glad that the clarifications, helped address your concerns. We appreciate your careful reading and constructive engagement.

---

### Official Review · Reviewer_aYRN · 2026-03-13

**Soundness:** 3
**Presentation:** 2
**Significance:** 3
**Originality:** 3
**Overall Recommendation:** 4
**Confidence:** 3

**Summary:**

This paper argues that multi-step diffusion generation is unnecessary for tabular one-class anomaly detection and proposes a simpler fixed-noise denoising framework instead. The method trains a denoising predictor with an explicit linear reference channel and detects anomalies using the expected deviation under repeated perturbations, which is theoretically motivated as a local stability proxy rather than mere distance to the data manifold. Experiments on ADBench show that this single-step, stability-based approach is both more efficient and more effective than existing methods, improving AUCROC by 1.22% and AUCPR by 1.13%.

**Compliance With Llm Reviewing Policy:**

Affirmed.

**Final Justification:**

The response has addressed my main concerns. Therefore, my final score is 4 (weak accept).

**Key Questions For Authors:**

1. The paper claims that diffusion-style sequential denoising is unnecessary, but it does not convincingly explain why repeated independent perturbations should be preferable to sequential perturbations. Could the authors clarify this design choice more explicitly?
2. How should an appropriate value of K be selected, especially across different datasets?

**Limitations:**

Please see Weakneses.

**Strengths And Weaknesses:**

Strengths:
1. The paper makes a compelling case that full diffusion-style multi-step generative modeling is unnecessary for tabular one-class anomaly detection, and replaces it with a much simpler fixed-noise denoising formulation.
2. The first-order expansion provides an interpretable bias–stability decomposition, linking the score to both denoising bias and local Jacobian sensitivity.
3. The method shows consistent gains on ADBench over both diffusion-based and classical baselines.

Weaknesses:
1. Although simpler than diffusion, the anomaly score depends on repeated stochastic perturbations at test time, which introduces nontrivial computational overhead compared with single-pass reconstruction-based methods.
2. While the fixed-noise simplification is valuable, the method remains close to denoising autoencoder-style anomaly detection.
3. The method depends on the noise scale and the number of perturbations, yet these choices may affect both performance and efficiency.
4. The paper argues convincingly for tabular anomaly detection, but it remains unclear whether the same conclusions would hold in other modals.

---

> ### Author Rebuttal · Authors · 2026-03-26
>
> ## General
>
> Thank you for the careful and constructive review, and for recognizing the paper’s main technical contribution: the full multi-step diffusion machinery is unnecessary, and a fixed-noise denoising formulation together with a stability-based score is both simpler and more effective. We are encouraged that you find the paper technically solid and highlight the simplicity/performance trade-off and the bias–stability interpretation as strengths.
>
> We would like to clarify that the two main concerns you raise are already addressed in the current manuscript.
>
> First, the paper does not aim to approximate a generative reverse trajectory. Instead, the method probes the local behavior of a learned denoising map around a test point, and the anomaly score is defined as an expectation over repeated perturbations of that same point. In Sec. 3, the first-order analysis motivates this score as a bias–stability quantity, i.e., a measure of how consistently the denoiser responds in a local neighborhood of the input, rather than a simulation of a path-dependent reverse diffusion chain. From this perspective, repeated independent perturbations are not a surrogate for sequential denoising. This interpretation is also supported empirically. In the fixed-noise versus schedule ablation, fixed noise performs better than scheduled and timestep-conditioned variants. Thus, the claim is not only conceptual, but already directly supported by the experiments.
>
> Second, regarding the choice of K: this is also already analyzed in the paper. Appendix D.2 studies the effect of K directly and shows that performance improves in the small-K regime. The same appendix reports that runtime remains nearly constant up to this range because the repeated perturbations can be evaluated in parallel, and only grows more noticeably for larger K. It is not an arbitrary choice, but the practical near-saturation point identified by the study already included in the submission.
>
> We would also like to clarify the concern that the method is  close to denoising-autoencoder-style anomaly detection. Its key contribution is the input-referenced stability score under repeated perturbations. In the ablation that disentangles reconstruction from stability, the proposed score outperforms plain reconstruction, mean-centered variance, and augmented reconstruction / DAE+TTA. This shows that the gains do not come from denoising reconstruction, but from the specific stability-based scoring rule introduced in the paper.
>
> Finally, on the scope beyond tabular data: the manuscript already states that the paper is intentionally focused on tabular one-class anomaly detection. We do not claim that the same conclusions automatically extend to other modalities, and this is already discussed in the limitations. While we are not aware of any conceptual reason that would restrict the idea to tabular data, we do not wish to overstate applicability beyond the settings we evaluated.
>
> ## Response to key questions
>
> **1. Why are repeated independent perturbations preferable here to sequential perturbations?**
>
> The much faster repeated independent perturbations are not intended as a surrogate for the slow sequential reverse-diffusion trajectory. Our objective is not generation from heavy noise, but the estimation of a local stability quantity around a test point. Accordingly, Sec. 3 defines the anomaly score as an expectation over \(K\) independent perturbations of the same input, i.e., repeated probes of the denoiser’s response in a local neighborhood of that point. This is also why the paper explicitly avoids a noise schedule, timestep conditioning, and a multi-step forward/reverse process. It is further supported empirically by the fixed-noise vs. schedule ablation in Sec. 5.
>
> **2. How should K be selected, especially across datasets?**
>
> This is already analyzed in Appendix D.2. Inference cost remains nearly constant up to about \(K \approx 15\) because the perturbations can be evaluated in parallel. Based on this efficiency frontier, we fix \(K = 15\) in all experiments. Thus, K is not chosen arbitrarily or tuned per dataset; rather, the paper already identifies a robust near-saturation operating point that works well while keeping test-time cost moderate.
>
> ## Conclusion
>
> Overall, we believe the current manuscript already provides both the conceptual and empirical justification for (i) repeated independent perturbations instead of sequential denoising, and (ii) the practical choice of K. We are grateful for your questions because they highlight exactly the core design choices of the paper, but we would submit that these points are already contained in the present version rather than requiring new analysis.
>
> We would like to thank you once more for the thoughtful and technically engaged review. We greatly appreciate both your positive assessment of the paper’s core contribution and the care with which you raised these important questions.

---

> > ### Author Rebuttal · Reviewer_aYRN · 2026-04-02
> >
> > Thank you very much for the reviewer's response. It has resolved my doubts and I will maintain my original score (Weak accept).

---

> > > ### Author Response · Authors · 2026-04-03
> > >
> > > Thank you very much for your kind feedback. We are glad that our response was able to resolve your concerns. We also sincerely appreciate your careful reading and constructive engagement with our work.

---

### Official Review · Reviewer_S6Gr · 2026-03-21

**Soundness:** 3
**Presentation:** 3
**Significance:** 3
**Originality:** 3
**Overall Recommendation:** 5
**Confidence:** 4

**Summary:**

The authors propose an approach to tabular anomaly detection based on diffusion models, but rather than running an entire diffusion process, their loss considers only one noising and denoising step. The loss is used to train a denoiser, i.e. a neural network that takes in a noised sample and predicts the clean sample.
Instead of scoring anomalies based on the same reconstruction loss, they derive a stability based objective from the first-order approximation of expected denoising error under noise.

In an empirical study, the method is compared to diffusion based approaches to AD, old deep learning based approaches to AD, and shallow methods.

**Compliance With Llm Reviewing Policy:**

Affirmed.

**Final Justification:**

My main issues with the paper during review were the framing as being derived from diffusion based AD methods instead of being related to self-supervised AD methods. Typically in self-supervised AD the training loss is also used as the the anomaly loss and this is this paper's contribution in my eyes: To instead suggest that the anomaly loss should be stability based. In my opinion, the resulting method is still self-supervised because the training loss involves transforming the data in some way (in this case adding Gaussian noise) and then solving an auxiliary task.

The main reason that I am updating my score is that the rebuttal convinced me that my understanding of which self-supervised AD methods are strong baselines for tabular AD was outdated. I therefore agree that the baselines are adequate.

As the authors mention in their rebuttal, I strongly encourage them to:  "(i) better connect the paper to reconstruction-aware and self-supervised anomaly detection, including the references you suggested; (ii) make the breadth of the current baseline suite more explicit."

**Key Questions For Authors:**

- in what ways does your approach resemble self-supervised anomaly detection methods

- does your approach as is work better than the self-supervised approach here:
Qiu C, Pfrommer T, Kloft M, Mandt S, Rudolph M. Neural transformation learning for deep anomaly detection beyond images. InInternational conference on machine learning 2021 Jul 1 (pp. 8703-8714). PMLR
Or other self-supervised approaches mentioned in this benchmark on ADBench:
Alvarez M, Verdier JC, Nkashama DJ, Frappier M, Tardif PM, Kabanza F. A revealing large-scale evaluation of unsupervised anomaly detection algorithms.

- if you framed your approach as a self-supervised anomaly detection approach instead, could you increase the significance by showing how to derive a stability based-anomaly score not just from a reconstruction loss, but also by taking other self-supervised objectives, that are known to work better than the reconstruction loss on , e.g. Qiu C, Pfrommer T, Kloft M, Mandt S, Rudolph M. Neural transformation learning for deep anomaly detection beyond images. InInternational conference on machine learning 2021 Jul 1 (pp. 8703-8714). PMLR

- Why didn't you evaluate your method on image data. Did the proposed stability based anomaly scoring rule not work as well there?

**Limitations:**

yes

**Strengths And Weaknesses:**

Soundness: It makes sense to not use the full generative capabilities of diffusion models. It is well know from the autoencoder literature that reconstruction-based losses are not well suited for anomaly detection. I think the proposed anomaly score is interesting and could be useful. Using only a single denoising step, the paper is much closer to self-supervised approaches to deep anomaly detection than to diffusion based approaches. Often the self-supervised training objective is also directly used as the anomaly score. The idea to use a stability based score instead is interesting and supported by the experimental results.

However, I am lowering my soundness score based on how this work is placed in relation to previous work. First, there is a significant amount of work that made similar arguments about reconstruction based losses (and generative models) not being suitable for anomaly detection. Some of these papers should be cited:
Eric Nalisnick, Akihiro Matsukawa, Yee Whye Teh, Dilan Gorur, and Balaji Lakshminarayanan. Do
deep generative models know what they don’t know? In International Conference on Learning
Representations, 2019.
Marcella Astrid, Muhammad Zaigham Zaheer, Jae-Yeong Lee, and Seung-Ik Lee. Learning not to
reconstruct anomalies. arXiv preprint arXiv:2110.09742, 2021.
Marcella Astrid, Muhammad Zaigham Zaheer, Djamila Aouada, and Seung-Ik Lee. Exploiting
autoencoder’s weakness to generate pseudo anomalies. Neural Computing and Applications, pp.
1–17, 2024.
Nicholas Merrill and Azim Eskandarian. Modified autoencoder training and scoring for robust
unsupervised anomaly detection in deep learning. IEEE Access, 8:101824–101833, 2020.
Zhen Cheng, Siwei Wang, Pei Zhang, Siqi Wang, Xinwang Liu, and En Zhu. Improved autoencoder
for unsupervised anomaly detection. International Journal of Intelligent Systems, 36(12):7103–
7125, 2021.
Dong Gong, Lingqiao Liu, Vuong Le, Budhaditya Saha, Moussa Reda Mansour, Svetha Venkatesh,
and Anton van den Hengel. Memorizing normality to detect anomaly: Memory-augmented deep
autoencoder for unsupervised anomaly detection. In Proceedings of the IEEE/CVF international
conference on computer vision, pp. 1705–1714, 2019

(there are many more citations in that direction...)

While these papers have suggested improvements to autoencoders the answer of the deep AD community has been to instead explore other self-supervised losses. This paper, imo sits squarely in that literature. The soundness of the experiments could have strengthened by comparing to stronger self-supervised baselines, e.g. Qiu C, Pfrommer T, Kloft M, Mandt S, Rudolph M. Neural transformation learning for deep anomaly detection beyond images. InInternational conference on machine learning 2021 Jul 1 (pp. 8703-8714). PMLR
Or other self-supervised approaches mentioned in this benchmark on ADBench:
Alvarez M, Verdier JC, Nkashama DJ, Frappier M, Tardif PM, Kabanza F. A revealing large-scale evaluation of unsupervised anomaly detection algorithms.

(though there might have been additional improvements since 2021)

Presentation: Good

Significance: I think the stability based anomaly score can be useful. However, the significance of this work could be improved if the proposed method was also evaluated on image data. (Or is there a reason it only works well on tabular data?)
Also, by instead framing this as an improvement of diffusion-based AD, framing this as an improvement over self-supervised appraoches to AD, one could increase the significance of this work: Namely, one could derive a stability based-anomaly score not just from a reconstruction loss, but also by taking other self-supervised objectives, that are known to work better than the reconstruction loss , e.g. Qiu C, Pfrommer T, Kloft M, Mandt S, Rudolph M. Neural transformation learning for deep anomaly detection beyond images. ICML 2021.

---

> ### Author Rebuttal · Authors · 2026-03-26
>
> ## General
>
> Thank you for the careful and constructive review. We especially appreciate that you find the stability-based anomaly score interesting and potentially useful.
>
> Our intended claim is not that anomaly detection benefits from a stronger generative diffusion model; rather, it is the opposite. The core message is that for one-class tabular anomaly detection, the multi-step diffusion machinery is unnecessary, and that a single fixed-noise denoiser together with an input-referenced stability score is sufficient and more effective in this setting. In that sense, we agree that the method is close in spirit to self-supervised / denoising-based anomaly detection: the model is trained on normal data only, but the anomaly score is not the training loss itself. Instead, it probes local instability of the learned denoising map under repeated perturbations.
>
> This distinction is central to our contribution. The proposed score is motivated in Sec. 3 via the bias–stability decomposition and is designed to measure input-referenced local sensitivity rather than single-shot reconstruction error. In Sec. 5, the ablation disentangling reconstruction from stability shows that the best performance comes from the combination of (i) the explicit reference-channel parameterization and (ii) the input-referenced stability score, while plain reconstruction, augmented reconstruction / DAE+TTA, and mean-centered variance all perform worse.
>
> We thank you for the suggested references and will incorporate them in the camera-ready (if we get accepted), together with a clearer discussion of how our contribution differs from this line of work in the appendix.
>
> On image data: the present paper is intentionally scoped to tabular anomaly detection. Our ADBench evaluation includes image and NLP groups only as embeddings in tabular shape, not as raw-image anomaly detection models. Thus, the paper should not be read as evidence that the method fails on images; rather, we deliberately focus on tabular / feature-space anomaly detection. While we are not aware of any conceptual reason that would restrict the idea to tabular data, we prefer not to make stronger claims beyond the settings we evaluated.
>
> ## Response to the key questions
>
> **1. In what ways does our approach resemble self-supervised anomaly detection methods?**
> DenoiserAD is closely related in spirit to self-supervised / one-class anomaly detection: it is trained using only normal data and learns a task-defined representation of normality. The key difference is that the anomaly score is not the pretext/training loss itself; instead, inference probes the local stability of the learned denoising map under repeated perturbations via an input-referenced score derived from the bias–stability decomposition.
>
> **2. Does the approach work better than Qiu et al. / other stronger self-supervised baselines?**
> We agree this is an interesting comparison axis. We did not include the exact Qiu et al. (2021) method (performs poorly in the similar comparison, thimonier2023beyond: NeuTraLAD [Qiu] < GOAD << kNN < ICL), so we do not want to overclaim beyond the experiments we report. At the same time, the current benchmark already includes strong deep one-class / self-supervised baselines such as LUNAR, DROCC, GOAD, and ICL, in addition to classical methods (e.g., EIF, LOF, kNN, KPCA, GMM, ABOD, SLAD) and diffusion-based baselines (DDPM, DTE-C, DTE-NP, NCSBAD). We deliberately focused on a well-defined set of strong baselines from the most closely related prior work to keep the empirical comparison readable and interpretable.
>
> **3. Could the significance be increased by framing the method as self-supervised AD and generalizing the score beyond reconstruction losses?**
> Conceptually, the core idea is broader than improving diffusion-based AD: it is to separate training on normal-only data from an inference-time score that measures local stability under perturbations. In the current paper, we instantiate this principle for fixed-noise denoising and show that the resulting score is not reducible to plain reconstruction, mean-centered variance, or denoising with test-time augmentation.
>
> **4. Why didn’t we evaluate on image data? Did the proposed stability score not work there?**
> The current paper is intentionally scoped to tabular anomaly detection. We do not evaluate raw-image anomaly detection models in this submission, so the paper should not be read as evidence that the method fails there. Rather, transfer to other domains remains an interesting direction outside the scope of the present study.
>
> ## Conclusion
>
> In the camera-ready, we will therefore (i) better connect the paper to reconstruction-aware and self-supervised anomaly detection, including the references you suggested; (ii) make the breadth of the current baseline suite more explicit.
>
> We thank you again for this very helpful feedback. We believe your comments will significantly improve the positioning and clarity of the paper.

---

> > ### Author Rebuttal · Reviewer_S6Gr · 2026-04-02
> >
> > I thank the reviewers for the additional information, especially the justification of the baseline methods considered. I will adjust my score accordingly.

---

> > > ### Author Response · Authors · 2026-04-02
> > >
> > > Thank you very much for the encouraging feedback. We are glad that the additional clarifications, especially on the baseline selection, helped address your concerns. We appreciate your careful reading and constructive engagement.

---

### Decision · Program_Chairs · 2026-04-30

**Decision:**

Accept (regular)

**Comment:**

The paper addresses the timely problem of anomaly detection in tabular data. After the rebuttal, all reviewers were positive about the paper and noted that the core idea is sound and that the method is valuable. The empirical evaluation using ADBench is valid and standard in this domain. The method demonstrated performance gains over diffusion-based and classical baselines, and ablation studies supported the contributions of the proposed scoring rule and reference-channel design. Some limitations mentioned by reviewers include that the method does not handle categorical features, that it is evaluated on other data types, and that it could also be evaluated with contaminated training data to make the setting more realistic. However, the authors properly addressed these points, and overall, the paper’s merits remain valid. Finally, I believe the paper makes a well-written and useful contribution that will be of interest to the anomaly detection and tabular learning communities, and I recommend acceptance.